# A MEAN FIELD THEORY OF BATCH NORMALIZATION

**Greg Yang**[†,*]**, Jeffrey Pennington**[°]**, Vinay Rao**[°]**, Jascha Sohl-Dickstein**[°]**, & Samuel S. Schoenholz**[°]

† Microsoft Research AI, ○ Google Brain
`gregyang@microsoft.com`, `{jpennin,vinaysrao,jaschasd,schsam}@google.com`

## ABSTRACT

We develop a mean field theory for batch normalization in fully-connected feed-forward neural networks. In so doing, we provide a precise characterization of signal propagation and gradient backpropagation in wide batch-normalized networks at initialization. Our theory shows that gradient signals grow exponentially in depth and that these exploding gradients cannot be eliminated by tuning the initial weight variances or by adjusting the nonlinear activation function. Indeed, batch normalization itself is the cause of gradient explosion. As a result, vanilla batch-normalized networks without skip connections are not trainable at large depths for common initialization schemes, a prediction that we verify with a variety of empirical simulations. While gradient explosion cannot be eliminated, it can be reduced by tuning the network close to the linear regime, which improves the trainability of deep batch-normalized networks without residual connections. Finally, we investigate the learning dynamics of batch-normalized networks and observe that after a single step of optimization the networks achieve a relatively stable equilibrium in which gradients have dramatically smaller dynamic range. Our theory leverages Laplace, Fourier, and Gegenbauer transforms and we derive new identities that may be of independent interest.

## 1 INTRODUCTION

Deep neural networks have been enormously successful across a broad range of disciplines. These successes are often driven by architectural innovations. For example, the combination of convolutions (LeCun et al., 1990), residual connections (He et al., 2015), and batch normalization (Ioffe & Szegedy, 2015) has allowed for the training of very deep networks and these components have become essential parts of models in vision (Zoph et al.), language (Chen & Wu, 2017), and reinforcement learning (Silver et al., 2017). However, a fundamental problem that has accompanied this rapid progress is a lack of theoretical clarity. An important consequence of this gap between theory and experiment is that two important issues become conflated. In particular, it is generally unclear whether novel neural network components improve generalization or whether they merely increase the fraction of hyperparameter configurations where good generalization can be achieved. Resolving this confusion has the promise of allowing researchers to more effectively and deliberately design neural networks.

Recently, progress has been made (Poole et al., 2016; Schoenholz et al., 2016; Daniely et al., 2016; Pennington et al., 2017; Hanin & Rolnick, 2018; Yang, 2019) in this direction by considering neural networks at initialization, before any training has occurred. In this case, the parameters of the network are random variables which induces a distribution of the activations of the network as well as the gradients. Studying these distributions is equivalent to understanding the prior over functions that these random neural networks compute. Picking hyperparameters that correspond to well-conditioned priors ensures that the neural network will be trainable and this fact has been extensively verified experimentally. However, to fulfill its promise of making neural network design less of a black box, these techniques must be applied to neural network architectures that are used in practice. Over the past year, this gap has closed significantly and theory for networks with skip connections (Yang & Schoenholz, 2017; 2018), convolutional networks (Xiao et al., 2018), and gated

---

*Please see `https://arxiv.org/abs/1902.08129` for the full and most current version of this paper

recurrent networks (Chen et al., 2018; Gilboa et al., 2019) have been developed. More recently, Yang (2019) devised a formalism that extends this approach to include an even wider range of architectures.

Before state-of-the-art models can be analyzed in this framework, a slowly-decreasing number of architectural innovations must be studied. One particularly important component that has thus-far remained elusive is batch normalization.

**Our Contributions.** In this paper, we develop a theory of fully-connected networks with batch normalization whose weights and biases are randomly distributed. A significant complication in the case of batch normalization (compared to e.g. layer normalization or weight normalization) is that the statistics of the network depend non-locally on the entire batch. Thus, our first main result is to recast the theory for random fully-connected networks so that it can be applied to batches of data. We then extend the theory to include batch normalization explicitly and validate this theory against Monte-Carlo simulations. We show that as in previous cases we can leverage our theory to predict valid hyperparameter configurations.

In the process of our investigation, we identify a number of previously unknown properties of batch normalization that make training unstable. In particular, for most nonlinearities used in practice, batchnorm in a deep, randomly initialized network induces high degree of symmetry in the embeddings of the samples in the batch (Thm 3.4). Whenever this symmetry takes hold, we show that for any choice of nonlinearity, gradients of fully-connected networks with batch normalization explode exponentially in the depth of the network (Thm 3.9). This imposes strong limits on the maximum trainable depth of batch normalized networks. This limit can be lifted partially but not completely by pushing activation functions to be more linear at initialization. It might seem that such gradient explosion ought to lead to learning dynamics that are unfavorable. However, we show that networks with batch normalization causes the scale of the gradients to naturally equilibrate after a single step of gradient descent (provided the initial gradients are not so large as to cause numerical instabilities). For shallower networks, this equilibrating effect is sufficient to allow adequate training.

Finally, we note that there is a related vein of research that has emerged that leverages the prior over functions induced by random networks to perform exact Bayesian inference (Lee et al., 2017; de G. Matthews et al., 2018; Novak et al., 2019; Garriga-Alonso et al., 2019; Yang, 2019). One of the natural consequences of this work is that the prior for networks with batch normalization can be computed exactly in the wide network limit. As such, it is now possible to perform exact Bayesian inference in the case of wide neural networks with batch normalization.

## 2   RELATED WORK

Batch normalization has rapidly become an essential part of the deep learning toolkit. Since then, a number of similar modifications have been proposed including layer normalization (Ba et al., 2016) and weight normalization (Salimans & Kingma, 2016). Comparisons of performance between these different schemes have been challenging and inconclusive (Gitman & Ginsburg, 2017). The original introduction of batchnorm in Ioffe & Szegedy (2015) proposed that batchnorm prevents "internal covariate shift" as an explanation for its effectiveness. Since then, several papers have approached batchnorm from a theoretical angle, especially following Ali Rahimi's catalyzing call to action at NIPS 2017. Balduzzi et al. (2017) found that batchnorm in resnets allow deep gradient signal propagation in contrast to the case without batchnorm. Santurkar et al. (2018) found that batchnorm does not help covariate shift but helps by smoothing loss landscape. Bjorck et al. (2018) reached the opposite conclusion as our paper for residual networks with batchnorm, that batchnorm works in this setting because it induces beneficial gradient dynamics and thus allows a much bigger learning rate. Luo et al. (2018) explores similar ideas that batchnorm allows large learning rates and likewise uses random matrix theory to support their claims. Kohler et al. (2018) identified situations in which batchnorm can provably induce acceleration in training. Of the above that mathematically analyze batchnorm, all but Santurkar et al. (2018) make simplifying assumptions on the form of batchnorm and typically do not have gradients flowing through the batch variance. Even Santurkar et al. (2018) only analyzes a vanilla network which gets added a single batchnorm at a single moment in training. Our analysis here on the other hand works for networks with arbitrarily many batchnorm layers with very general activation functions, and indeed, this deep stacking of batchnorm is precisely what

leads to gradient explosion. It is an initialization time analysis for networks of infinite width, but we show experimentally that the resulting insights predict both training and test time behavior.

We remark that Philipp & Carbonell (2018) has also empirically noted gradient explosion happens in deep batchnorm networks with various nonlinearities. In this work, in contrast, we develop a precise theoretical characterization of gradient statistics and, as a result, we are able to make significantly stronger conclusions.

## 3 THEORY

We begin with a brief recapitulation of mean field theory in the fully-connected setting. In addition to recounting earlier results, we rephrase the formalism developed previously to compute statistics of neural networks over a batch of data. Later, we will extend the theory to include batch normalization. We consider a fully-connected network of depth $L$ whose layers have width $N_l$, activation function[1] $\phi$, weights $W^l \in \mathbb{R}^{N_{l-1} \times N_l}$, and biases $b^l \in \mathbb{R}^{N_l}$. Given a batch of $B$ inputs[2] $\{x_i : x_i \in \mathbb{R}^{N_0}\}_{i=1,\cdots,B}$, the pre-activations of the network are defined by the recurrence relation,

$$\boldsymbol{h}_i^1 = W^1 \boldsymbol{x}_i + b^1 \qquad \text{and} \qquad \boldsymbol{h}_i^l = W^l \phi(\boldsymbol{h}_i^{l-1}) + b^l \quad \forall\, l > 1. \tag{1}$$

At initialization, we choose the weights and biases to be i.i.d. as $W_{\alpha\beta}^l \sim \mathcal{N}(0, \sigma_w^2/N_{l-1})$ and $b_\alpha^l \sim \mathcal{N}(0, \sigma_b^2)$. In the following, we use $\alpha, \beta, \ldots$ for the neuron index, and $i, j, \ldots$ for the batch index. We will be concerned with understanding the statistics of the pre-activations and the gradients induced by the randomness in the weights and biases. For ease of exposition we will typically take the network to have constant width $N_l = N$.

In the mean field approximation, we iteratively replace the pre-activations in Eq. (2) by Gaussian random variables with matching first and second moments. In the infinite width limit this approximation becomes exact (Lee et al., 2017; de G. Matthews et al., 2018; Yang, 2019). Since the weights are i.i.d. with zero mean it follows that the mean of each pre-activation is zero and the covariance between distinct neurons are zero. The pre-activation statistics are therefore given by $(h_{\alpha_1 1}^l, \cdots, h_{\alpha_B B}^l) \xrightarrow{N_{l-1} \to \infty} \mathcal{N}(0, \Sigma^l \delta_{\alpha_1 \cdots \alpha_B})$ where $\Sigma^l$ are $B \times B$ covariance matrices and $\delta_{\alpha_1, \cdots \alpha_B}$ is the Kronecker-$\delta$ that is one if $\alpha_1 = \alpha_2 = \cdots = \alpha_B$ and zero otherwise.

**Definition 3.1.** Let V be the operator on functions $\phi$ such that $V_\phi(\Sigma) = \mathbb{E}[\phi(h)\phi(h)^T : h \sim \mathcal{N}(0, \Sigma)]$ computes the matrix of uncentered second moments of $\phi(h)$ for $h \sim \mathcal{N}(0, \Sigma)$.

Using the above notation, we can express the covariance matrices by the recurrence relation,

$$\Sigma^l = \sigma_w^2 V_\phi(\Sigma^{l-1}) + \sigma_b^2 \mathbf{1}\mathbf{1}^T \tag{2}$$

At first Eq. (2) may seem challenging since the expectation involves a Gaussian integral in $\mathbb{R}^B$. However, each term in the expectation of $V_\phi$ involves at most a pair of pre-activations and so the expectation may be reduced to the evaluation of $\mathcal{O}(B^2)$ two-dimensional integrals. These integrals can either be performed analytically (Cho & Saul, 2009; Williams, 1997) or efficiently approximated numerically (Lee et al., 2017), and so Eq. (2) defines a computationally efficient method for computing the statistics of neural networks after random initialization. This theme of dimensionality reduction will play a prominent role in the forthcoming discussion on batch normalization.

Eq. (2) defines a dynamical system over the space of covariance matrices. Studying the statistics of random feed-forward networks therefore amounts to investigating this dynamical system and is an enormous simplification compared with studying the pre-activations of the network directly. As is common in the dynamical systems literature, a significant amount of insight can be gained by investigating the behavior of Eq. (2) in the vicinity of its fixed points. For most common activation functions, Eq. (2) has a fixed point at some $\Sigma^*$. Moreover, when the inputs are non-degenerate, this fixed point generally has a simple structure with $\Sigma^* = q^*[(1-c^*)I + c^*\mathbf{1}\mathbf{1}^T]$ owing to permutation symmetry among elements of the batch. We refer to fixed points with such symmetry as BSB1 (*Batch Symmetry Breaking 1* or *1 Block Symmetry Breaking*) fixed points. As we will discuss later, in the context of batch normalization other fixed points with fewer symmetries ("BSBk fixed points")

---

[1]The activation function may be layer dependent, but for ease of exposition we assume that it is not.

[2]Throughout the text, we assume that all elements of the batch are unique.

may become preferred. In the fully-connected setting fixed points may efficiently be computed by solving the fixed point equation induced by Eq. (2) in the special case $B = 2$. The structure of this fixed point implies that in asymptotically deep feed-forward neural networks all inputs yield pre-activations of identical norm with identical angle between them. Neural networks that are deep enough so that their pre-activation statistics lie in this regime have been shown to be untrainable (Schoenholz et al., 2016).

**Notation**    As we often talk about matrices and also linear operators over matrices, we write $\mathcal{T}\{\Sigma\}$ for an operator $\mathcal{T}$ applied to a matrix $\Sigma$, and matrix multiplication is still written as juxtaposition. Composition of matrix operators are denoted with $\mathcal{T}_1 \circ \mathcal{T}_2$.

**Local Convergence to BSB1 Fixed Point.**    To understand the behavior of Eq. (2) near its BSB1 fixed point we can consider the Taylor series in the deviation from the fixed point, $\Delta\Sigma^l = \Sigma^l - \Sigma^*$. To lowest order we generically find,

$$\Delta\Sigma^l = J\{\Delta\Sigma^{l-1}\} \tag{3}$$

where $J = \frac{\mathrm{d}V_\phi}{\mathrm{d}\Sigma}\big|_{\Sigma=\Sigma^*}$ is the $B^2 \times B^2$ Jacobian of $V_\phi$. In most prior work where $\phi$ was a pointwise non-linearity, Eq. (3) reduces to the case $B = 2$ which naturally gave rise to linearized dynamics in $q^l = \mathbb{E}[(h_i^l)^2]$ and $c^l = \mathbb{E}[h_i^l z_j^l]/q^l$. However, in the case of batch normalization we will see that one must consider the evolution of Eq. (3) as a whole. This is qualitatively reminiscent of the case of convolutional networks studied in Xiao et al. (2018) where the evolution of the entire pixel $\times$ pixel covariance matrix had to be evaluated.

The dynamics induced by Eq. (3) will be controlled by the eigenvalues of $J$: Suppose $J$ has eigenvalues $\lambda_i$ — ordered such that $\lambda_1 \geq \lambda_2 \geq \cdots \geq \lambda_{B^2}$ — with associated eigen"vectors" $e_i$ (note that the $e_i$ will themselves be $B \times B$ matrices). It follows that if $\Delta\Sigma^0 = \sum_i c_i e_i$ for some choice of constants $c_i$ then $\Delta\Sigma^l = \sum_i c_i \lambda_i^l e_i$. Thus, if $\lambda_i < 1$ for all $i$, $\Delta\Sigma^l$ will approach zero exponentially and the fixed-point will be stable. The number of layers over which $\Sigma$ will approach $\Sigma^*$ will be given by $-1/\log(\lambda_1)$. By contrast if $\lambda_i > 1$ for any $i$ then the fixed point will be unstable. In this case, there is typically a different, stable, fixed point that must be identified. It follows that if the eigenvalues of $J$ can be computed then the dynamics will follow immediately.

While $J$ may appear to be a complicated object at first, a moment's thought shows that each diagonal element $J\{\Delta\Sigma\}_{ii}$ is only affected by the corresponding diagonal element $\Delta\Sigma_{ii}$, and each off-diagonal element $J\{\Delta\Sigma\}_{ij}$ is only affected by $\Delta\Sigma_{ii}, \Delta\Sigma_{jj}$, and $\Delta\Sigma_{ij}$. Such a *Diagonal-Off-diagonal Semidirect* (*DOS*) operator has a simple eigendecomposition with two eigenspaces corresponding to changes in the off-diagonal and changes in the diagonal (Thm E.72). The associated eigenvalues are precisely those calculated by Schoenholz et al. (2016) in a simplified analysis. DOS operators are a particularly simple form of a more general operator possessing an abundance of symmetries called *ultrasymmetric operators*, which will play a prominent role in the analysis of batchnorm below.

**Gradient Dynamics.**    Similar arguments allow us to develop a theory for the statistics of gradients. The backpropagation algorithm gives an efficient method of propagating gradients from the end of the network to the earlier layers as,

$$\frac{\mathrm{d}\mathcal{L}}{\mathrm{d}W^l} = \sum_i \boldsymbol{\delta}_i^l \phi(\boldsymbol{h}_i^{l-1})^T \qquad \delta_{\alpha i}^l = \phi'(h_{\alpha i}^l) \sum_\beta W_{\beta\alpha}^{l+1} \delta_{\beta i}^{l+1}. \tag{4}$$

Here $\mathcal{L}$ is the loss function and $\boldsymbol{\delta}_i^l = \nabla_{\boldsymbol{h}_i^l}\mathcal{L}$ are $N_l$-dimensional vectors that describe the error signal from neurons in the $l$'th layer due to the $i$'th element of the batch. The preceding discussion gave a precise characterization of the statistics of the $\boldsymbol{h}_i^l$ that we can leverage to understand the statistics of $\boldsymbol{\delta}_i^l$. Assuming *gradient independence*, that is, that an iid set of weights are used during backpropagation (see Appendix B for more discussions), it is easy to see that $\mathbb{E}[\delta_{\alpha i}^l] = 0$ and $\mathbb{E}[\delta_{\alpha i}^l \delta_{\beta j}^l] = \Pi_{ij}^l$ if $\alpha = \beta$ and 0 otherwise, where $\Pi^l$ is a covariance matrix and we may once again drop the neuron index. We can construct a recurrence relation to compute $\Pi^l$,

$$\Pi^l = \sigma_w^2 V_{\phi'}(\Sigma^l) \odot \Pi^{l+1}. \tag{5}$$

Typically, we will be interested in understanding the dynamics of $\Pi^l$ when $\Sigma^l$ has converged exponentially towards its fixed point. Thus, we study the approximation,

$$\Pi^l \approx \sigma_w^2 \mathrm{V}_{\phi'}(\Sigma^*) \odot \Pi^{l+1}. \tag{6}$$

Since these dynamics are linear (and in fact componentwise), explosion and vanishing of gradients will be controlled by $\mathrm{V}_{\phi'}(\Sigma^*)$.

## 3.1 Batch Normalization

We now extend the mean field formalism to include batch normalization. Here, the definition for the neural network is modified to be the coupled equations,

$$\boldsymbol{h}_i^l = W^l \phi(\tilde{\boldsymbol{h}}_i^{l-1}) + b^l \qquad \tilde{h}_{\alpha i}^l = \gamma_\alpha \frac{h_{\alpha i}^l - \mu_\alpha}{\sigma_\alpha} + \beta_\alpha \tag{7}$$

where $\gamma_\alpha$ and $\beta_\alpha$ are parameters, and $\mu_\alpha = \frac{1}{N_l} \sum_i h_{\alpha i}$ and $\sigma_\alpha^2 = \sqrt{\frac{1}{N_l} \sum_i (h_{\alpha i} - \mu_\alpha)^2 + \epsilon}$ are the per-neuron batch statistics. In practice $\epsilon \approx 10^{-5}$ or so to prevent division by zero, but in this paper, unless stated otherwise (in the last few sections), $\epsilon$ is assumed to be 0. Unlike in the case of vanilla fully-connected networks, here the pre-activations are invariant to $\sigma_w^2$ and $\sigma_b^2$. Without a loss of generality, we therefore set $\sigma_w^2 = 1$ and $\sigma_b^2 = 0$ for the remainder of the text. In principal, batch normalization additionally yields a pair of hyperparameters $\gamma$ and $\beta$ which are set to be constants. However, these may be incorporated into the nonlinearity and so without a loss of generality we set $\gamma = 1$ and $\beta = 0$. In order to avoid degenerate results, we assume $B \geq 4$ unless stated otherwise; we shall discuss the small $B$ regime in Appendix J.

If one treats batchnorm as a "batchwise nonlinearity", then the arguments from the previous section can proceed identically and we conclude that as the width of the network grows, the pre-activations will be jointly Gaussian with identically distributed neurons. Thus, we arrive at an analogous expression to Eq. (2),

$$\Sigma^l = \mathrm{V}_{\mathcal{B}_\phi}(\Sigma^{l-1}) \qquad \text{where} \qquad \mathcal{B}_\phi : \mathbb{R}^B \to \mathbb{R}^B, \ \mathcal{B}_\phi(h) = \phi\left(\frac{\sqrt{B}Gh}{||Gh||}\right). \tag{8}$$

Here we have introduced the projection operator $G = I - \frac{1}{B}\mathbf{1}\mathbf{1}^T$ which is defined such that $Gx = x - \mu\mathbf{1}$ with $\mu = \sum_i x_i/B$. Unlike $\phi$, $\mathcal{B}_\phi$ does not act component-wise on $h$. It is therefore not obvious whether $\mathrm{V}_{\mathcal{B}_\phi}$ can be evaluated without performing a $B$-dimensional Gaussian integral.

**Theoretical tools.** In this paper, we present several ways to analyze high dimensional integrals like the above: 1. the Laplace method 2. the Fourier method 3. spherical integration 4. and the Gegenbauer method. The former two use the Laplace and Fourier transforms to simplify expressions like the above, where the Laplace method requires that $\phi$ be positive homogeneous. Often the Laplace method will give clean, closed form answers for such $\phi$. Because batchnorm can be thought of as a linear projection ($G$) followed by projection to the sphere of radius $\sqrt{B}$, spherical integration techniques are often very useful and in fact is typically the most straightforward way of numerically evaluating quantities. Lastly, the Gegenbauer method expresses objects in terms of the Gegenbauer coefficients of $\phi$. Briefly, Gegenbauer polynomials $\{C_l^{(\alpha)}(x)\}_{l=0}^\infty$ are orthogonal polynomials with respect to the measure $(1 - x^2)^{\alpha - \frac{1}{2}}$ on $[-1, 1]$. They are intimately related to spherical harmonics (a natural basis for functions on a sphere), which explains their appearance in this context. The Gegenbauer method is the most illuminating amongst them all, and is what allows us to conclude that gradient explosion happens regardless of nonlinearity under general conditions. See Appendix D for a more in-depth discussion of these techniques. In what follows, we will mostly present results by the Laplace method for $\phi = \mathrm{relu}$ and by the Gegenbauer method for general $\phi$, but give pointers to the appendix for others.

Back to the topic of Eq. (8), we present a pair of results that expresses Eq. (8) in a more manageable form. From previous work (Poole et al., 2016), $\mathrm{V}_\phi$ can be expressed in terms of a two-dimensional Gaussian integrals independent of $B$. When $\phi$ is degree-$\alpha$ positive homogeneous (e.g. rectified linear activations) we can relate $\mathrm{V}_\phi$ and $\mathrm{V}_{\mathcal{B}_\phi}$ by the Laplace transform. (see Thm E.5).

**Theorem 3.2.** *Suppose $\phi : \mathbb{R} \to \mathbb{R}$ is degree-$\alpha$ positive homogeneous. For any positive semi-definite matrix $\Sigma$ define the projection $\Sigma^G = G\Sigma G$. Then*

$$\mathrm{V}_{\mathcal{B}_\phi}(\Sigma) = \frac{B^\alpha}{\Gamma(\alpha)} \int_0^\infty \mathrm{d}s \; s^{\alpha-1} \frac{\mathrm{V}_\phi(\Sigma^G(I + 2s\Sigma^G)^{-1})}{\sqrt{\det(I + 2s\Sigma^G)}}. \tag{9}$$

*whenever the integral exists.*

Using this parameterization, when $\mathrm{V}_\phi$ has a closed form solution (like ReLU), $\mathrm{V}_{\mathcal{B}_\phi}$ involves only a single integral. A similar expression can be derived for general $\phi$ by applying Fourier transform; see Appendix E.2. Next we express Eq. (8) as a spherical integral (see Proposition E.34)

**Theorem 3.3.** *Let $\mathbb{e} \in \mathbb{R}^{B \times B-1}$ have as columns a set of orthonormal basis vectors of $\{x \in \mathbb{R}^B : Gx = x\}$. Then with $S^{B-2} \subseteq \mathbb{R}^{B-1}$ denoting the $(B-2)$-dimensional sphere,*

$$\mathrm{V}_{\mathcal{B}_\phi}(\Sigma) = \mathop{\mathbb{E}}_{v \sim S^{B-2}} \frac{\phi(\sqrt{B}\mathbb{e}v)^{\otimes 2}}{\sqrt{\det \mathbb{e}^T \Sigma \mathbb{e}} (v^T(\mathbb{e}^T \Sigma \mathbb{e})^{-1}v)^{\frac{B-1}{2}}}. \tag{10}$$

Together these theorems provide analytic recurrence relations for random neural networks with batch normalization over a wide range of activation functions. By analogy to the fully-connected case we would like to study the dynamical system over covariance matrices induced by these equations.

We begin by investigating the fixed point structure of Eq. (8). As in the case of feed-forward networks, permutation symmetry implies that there exist BSB1 fixed points $\Sigma^* = q^*[(1-c^*)I+c^*\mathbf{1}\mathbf{1}^T]$. We will see that this fixed point is in fact unique, and a clean expression of $q^*$ and $c^*$ can be obtained in terms of Gegenbauer basis (see Thm F.13).

**Theorem 3.4** (Gegenbauer expansions of BSB1 fixed point)**.** *If $\phi(\sqrt{B-1}x)$ has Gegenbauer expansion $\sum_{l=0}^\infty a_l \frac{1}{c_{B-1,l}} C_l^{(\frac{B-3}{2})}(x)$ where $c_{B-1,l} = \frac{B-3}{B-3+2l}$, then*

$$q^* = \sum_{l=0}^\infty a_l^2 \frac{1}{c_{B-1,l}} C_l^{(\frac{B-3}{2})}(1), \qquad q^* c^* = \sum_{l=0}^\infty a_l^2 \frac{1}{c_{B-1,l}} C_l^{(\frac{B-3}{2})}\left(\frac{-1}{B-1}\right).$$

Thus the entries of the BSB1 fixed point are diagonal quadratic forms of the Gegenbauer coefficients of $\phi(\sqrt{B-1}x)$. Even more concise closed forms are available when the activation functions are degree $\alpha$ positive homogeneous (see Thm F.8). In particular, for ReLU we arrive at the following

**Theorem 3.5** (BSB1 fixed point for ReLU)**.** *When $\phi = \mathrm{relu}$, then*

$$q^* = \frac{1}{2}, \qquad c^* = \mathrm{J}_1\left(\frac{-1}{B-1}\right) = \frac{1}{\pi} - \frac{1}{2(B-1)} + O\left(\frac{1}{(B-1)^2}\right) \tag{11}$$

*where $\mathrm{J}_1(c) = \frac{1}{\pi}(\sqrt{1-c^2} + (\pi - \arccos(c))c)$ is the arccosine kernel (Cho & Saul, 2009).*

In the appendix, Thm F.5 also describes a trigonometric integral formula for general nonlinearities.

In the presence of batch normalization, when the activation function grows quickly, a winner-take-all phenomenon can occur where a subset of samples in the batch have much bigger activations than others. This causes the covariance matrix to form blocks of differing magnitude, breaking the BSB1 symmetry. One notices this, for example, as the degree $\alpha$ of $\alpha$-ReLU (i.e. $\alpha$th power of ReLU) increases past a point $\alpha_{\mathrm{transition}}(B)$ depending on the batch size $B$ (see Fig. 1). However, we observe, through simulations, that by far most of the nonlinearities used in practice, like ReLU, leaky ReLU, tanh, sigmoid, etc, all lead to BSB1 fixed points, and we can prove this rigorously for $\phi = \mathrm{id}$ (see Corollary F.3). We discuss symmetry-broken fixed points (BSB2 fixed points) in Appendices J and K, but in the main text, from here on,

*Assumption* 1. Unless stated otherwise, we assume that any nonlinearity $\phi$ mentioned induces $\Sigma^l$ to converge to a BSB1 fixed point under the dynamics of Eq. (8).

### 3.1.1 LINEARIZED DYNAMICS

With the fixed point structure for batch normalized networks having been described, we now investigate the linearized dynamics of Eq. (8) in the vicinity of these fixed points.

To determine the eigenvalues of $\frac{\mathrm{dV}_{\mathcal{B}_\phi}}{\mathrm{d}\Sigma}\big|_{\Sigma=\Sigma^*}$ it is helpful to consider the action of batch normalization in more detail. In particular, we notice that $\mathcal{B}_\phi$ can be decomposed into the composition of three separate operations, $\mathcal{B}_\phi = \phi \circ \mathsf{n} \circ G$. As discussed above, $Gh$ subtracts the mean from $h$ and we introduce the new function $\mathsf{n}(h) = \sqrt{B}h/||h||$ which normalizes a centered $h$ by its standard deviation. Applying the chain rule, we can rewrite the Jacobian as,

$$\frac{\mathrm{dV}_{\mathcal{B}_\phi}(\Sigma)}{\mathrm{d}\Sigma} = \frac{\mathrm{dV}_{[\phi\circ\mathsf{n}]}(\Sigma^G)}{\mathrm{d}\Sigma^G} \circ G^{\otimes 2} \tag{12}$$

where $\circ$ denotes composition and $G^{\otimes 2}$ is the natural extension of $G$ to act on matrices as $G^{\otimes 2}\{\Sigma\} = G\Sigma G = \Sigma^G$. It ends up being advantageous to study $G^{\otimes 2} \circ \frac{\mathrm{dV}_{[\phi\circ\mathsf{n}]}}{\mathrm{d}\Sigma}\big|_{\Sigma=\Sigma^*} \circ G^{\otimes 2} =: G^{\otimes 2} \circ \hat{J} \circ G^{\otimes 2}$ and to note that the nonzero eigenvalues of this object are identical to the nonzero eigenvalues of the Jacobian (see Lemma F.17).

At face value, this is a complicated object since it simultaneously has large dimension and possesses an intricate block structure. However, the permutation symmetry of the BSB1 $\Sigma^*$ induces strong symmetries in $\hat{J}$ that significantly simplify the analysis (see Appendix F.3). In particular while $\hat{J}_{ijkl}$ is a four-index object, we have $\hat{J}_{ijkl} = \hat{J}_{\pi(i)\pi(j)\pi(k)\pi(l)}$ for all permutations $\pi$ on $B$ and $\hat{J}_{ijkl} = \hat{J}_{jilk}$. We call linear operators possessing such symmetries *ultrasymmetric* (Defn E.53) and show that all ultrasymmetric operators conjugated by $G^{\otimes 2}$ admit an eigendecomposition that contains three distinct eigenspaces with associated eigenvalues (see Thm E.62).

**Theorem 3.6.** *Let $\mathcal{T}$ be an ultrasymmetric matrix operator. Then on the space of symmetric matrices, $G^{\otimes 2} \circ \mathcal{T} \circ G^{\otimes 2}$ has the following orthogonal (under trace inner product) eigendecomposition,*

1. *an eigenspace $\{\Sigma : \Sigma^G = 0\}$ with eigenvalue 0.*

2. *a 1-dimensional eigenspace $\mathbb{R}G$ with eigenvalue $\lambda_G^{G,\mathcal{T}}$.*

3. *a $(B-1)$-dimensional eigenspace $\mathbb{L} := \{D^G : D \text{ diagonal}, \operatorname{tr} D = 0\}$, with eigenvalue $\lambda_{\mathbb{L}}^{G,\mathcal{T}}$.*

4. *a $\frac{B(B-3)}{2}$-dimensional eigenspace $\mathbb{M} := \{\Sigma : \Sigma^G = \Sigma, \operatorname{Diag}\Sigma = 0\}$, with eigenvalue $\lambda_{\mathbb{M}}^{G,\mathcal{T}}$.*

The specific forms of the eigenvalues can be obtained as linear functions of the entries of $\mathcal{T}$; see Thm E.62 for details. Note that, as the eigenspaces are orthogonal, this implies that $G^{\otimes 2} \circ \mathcal{T} \circ G^{\otimes 2}$ is self-adjoint (even when $\mathcal{T}$ is not).

In our context with $\mathcal{T} = \hat{J}$, the eigenspaces can be roughly interpreted as follows: The deviation $\Delta\Sigma = \Sigma - \Sigma^*$ from the fixed point decomposes as a linear combination of components in each of the eigenspaces. The $\mathbb{R}G$-component captures the average norm of elements of the batch (the trace of $\Delta\Sigma$), the $\mathbb{L}$-component captures the fluctuation of such norms, and the $\mathbb{M}$-component captures the covariances between elements of the batch.

Because of the explicit normalization of batchnorm, one sees immediately that the $\mathbb{R}G$-component goes to 0 after 1 step. For positive homogeneous $\phi$, we can use the Laplace method to obtain closed form expressions for the other eigenvalues (see Thm F.33). The below theorem shows that, as the batch size becomes larger, a deep ReLU-batchnorm network takes more layers to converge to a BSB1 fixed point.

**Theorem 3.7.** *Let $\phi = \operatorname{relu}$ and $B > 3$. The eigenvalues of $G^{\otimes 2} \circ \hat{J} \circ G^{\otimes 2}$ for $\mathbb{L}$ and $\mathbb{M}$ are*

$$\lambda_{\mathbb{L}}^{\uparrow} = \frac{1}{2(B+1)\mu^*}\left((B-2)\left[1 - \mathrm{J}_1\left(\frac{-1}{B-1}\right)\right] + \frac{B}{B-1}\mathrm{J}_1'\left(\frac{-1}{B-1}\right)\right) \nearrow 1 \tag{13}$$

$$\lambda_{\mathbb{M}}^{\uparrow} = \frac{B}{2(B+1)\mu^*}\mathrm{J}_1'\left(\frac{-1}{B-1}\right) \nearrow \frac{\pi}{2(\pi-1)} \approx 0.733 \tag{14}$$

*where $\mu^* = q^*(1-c^*) = \frac{1}{2}\left(1 - \mathrm{J}_1\left(\frac{-1}{B-1}\right)\right)$, and $\nearrow$ denotes increasing limit as $B \to \infty$.*

More generally, we can evaluate them for general nonlinearity using spherical integration (Appendix F.3.1) and, more enlightening, using the Gegenbauer method, is the following

**Theorem 3.8.** *If $\phi(\sqrt{B-1}x)$ has Gegenbauer expansion $\sum_{l=0}^{\infty} a_l \frac{1}{c_{B-1,l}} C_l^{\left(\frac{B-3}{2}\right)}(x)$, then*

$$\lambda_{\mathbb{L}}^{\uparrow} = \frac{\sum_{l=0}^{\infty} a_l^2 w_{B-1,l} + a_l a_{l+2} u_{B-1,l}}{\sum_{l=0}^{\infty} a_l^2 v_{B-1,l}}, \qquad \lambda_{\mathbb{M}}^{\uparrow} = \frac{\sum_{l=0}^{\infty} a_l^2 \tilde{w}_{B-1,l} + a_l a_{l+2} \tilde{u}_{B-1,l}}{\sum_{l=0}^{\infty} a_l^2 v_{B-1,l}} \qquad (15)$$

*where the coefficients $w_{B-1,l}, u_{B-1,l}, \tilde{w}_{B-1,l}, \tilde{u}_{B-1,l}, v_{B-1,l}$ are given in Thms F.22 and F.24.*

A BSB1 fixed point is not locally attracting if $\lambda_{\mathbb{L}}^{\uparrow} > 1$ or $\lambda_{\mathbb{M}}^{\uparrow} > 1$. Thus Thm 3.8 yields insight on the stability of the BSB1 fixed point, which we can interpret heuristically as follows. The specific forms of the coefficients $w_{B-1,l}, u_{B-1,l}, \tilde{w}_{B-1,l}, \tilde{u}_{B-1,l}, v_{B-1,l}$ show that $\lambda_{\mathbb{M}}^{\uparrow}$ is typically much smaller than $\lambda_{\mathbb{L}}^{\uparrow}$ (but there are exceptions like $\phi = \sin$), and $w_{B-1,1} < v_{B-1,1}$ but $w_{B-1,l} \geq v_{B-1,l}$ for all $l \geq 2$. Thus one expects that, the larger $a_1$ is, i.e. the "more linear" and less explosive $\phi$ is, the smaller $\lambda_{\mathbb{L}}^{\uparrow}$ is and the more likely that Eq. (8) converges to a BSB1 fixed point. This is consistent with the "winner-take-all" intuition for the emergence of BSB2 fixed point explained above. See Appendix F.3.2 for more discussion.

### 3.1.2 Gradient Backpropagation

With a mean field theory of the pre-activations of feed-forward networks with batch normalization having been developed, we turn our attention to the backpropagation of gradients. In contrast to the case of networks without batch normalization, we will see that exploding gradients at initialization are a severe problem here. To this end, one of the main results from this section will be to show that fully-connected networks with batch normalization feature exploding gradients for *any* choice of nonlinearity such that $\Sigma^l$ converges to a BSB1 fixed point. Below, by *rate of gradient explosion* we mean the $\beta$ such that the gradient norm squared grows as $\beta^{L+o(L)}$ with depth $L$. As before, all computations below assumes *gradient independence* (see Appendix B for a discussion).

As a starting point we seek an analog of Eq. (6) in the case of batch normalization. However, because the activation functions no longer act point-wise on the pre-activations, the backpropagation equation becomes,

$$\delta_{\alpha i}^l = \sum_{\beta j} \frac{\partial \mathcal{B}_\phi(\boldsymbol{h}_\alpha^l)_j}{\partial h_{\alpha i}^l} W_{\beta \alpha}^{l+1} \delta_{\beta j}^{l+1} \qquad (16)$$

where $\boldsymbol{h}_\alpha^l = (h_{\alpha 1}^l, \ldots, h_{\alpha B}^l)$ and we observe the additional sum over the batch. Computing the resulting covariance matrix $\Pi^l$, we arrive at the recurrence relation,

$$\Pi^l = \mathbb{E}\left[ \left( \frac{\mathrm{d}\mathcal{B}_\phi(h)}{\mathrm{d}h} \right)^T \Pi^{l+1} \frac{\mathrm{d}\mathcal{B}_\phi(h)}{\mathrm{d}h} : h \sim \mathcal{N}(0, \Sigma^l) \right] =: \mathrm{V}_{\mathcal{B}_\phi'}(\Sigma^l)^\dagger \{\Pi^{l+1}\} \qquad (17)$$

where we have defined the linear operator $\mathrm{V}_F(\Sigma)^\dagger\{\cdot\}$ such that $\mathrm{V}_F(\Sigma)^\dagger\{\Pi\} = \mathbb{E}[F_h^T \Pi F_h : h \sim \mathcal{N}(0, \Sigma)]$ for any vector-indexed linear operator $F_h$. As in the case of vanilla feed-forward networks, here we will be concerned with the behavior of gradients when $\Sigma^l$ is close to its fixed point. We therefore study the asymptotic approximation to Eq. (17) given by $\Pi^l = \mathrm{V}_{\mathcal{B}_\phi'}(\Sigma^*)^\dagger\{\Pi^{l+1}\}$. In this case the dynamics of $\Pi$ are linear and are therefore naturally determined by the eigenvalues of $\mathrm{V}_{\mathcal{B}_\phi'}(\Sigma^*)^\dagger$.

As in the forward case, batch normalization is the composition of three operations $\mathcal{B}_\phi = \phi \circ \mathsf{n} \circ G$. Applying the chain rule, Eq. (17) can be rewritten as,

$$\mathrm{V}_{\mathcal{B}_\phi'}(\Sigma)^\dagger = G^{\otimes 2} \circ \mathbb{E}\left[ \left( \frac{\mathrm{d}(\phi \circ \mathsf{n})(z)}{\mathrm{d}z} \bigg|_{z=Gh} \right)^{\otimes 2} : h \sim \mathcal{N}(0, \Sigma) \right] =: G^{\otimes 2} \circ F(\Sigma) \qquad (18)$$

with $F(\Sigma)$ appropriately defined. Note that since $G^{\otimes 2}$ is an idempotent operator, $(\mathrm{V}_{\mathcal{B}_\phi'}(\Sigma)^\dagger)^n = (G^{\otimes 2} \circ F(\Sigma))^n = (G^{\otimes 2} \circ F(\Sigma) \circ G^{\otimes 2})^{n-1} \circ G^{\otimes 2} \circ F(\Sigma)$, so that it suffices to study the eigendecomposition of $G^{\otimes 2} \circ F(\Sigma^*) \circ G^{\otimes 2}$. Due to the symmetry of $\Sigma^*$, $F(\Sigma^*)$ is ultrasymmetric, so that

$G^{\otimes 2} \circ F(\Sigma^*) \circ G^{\otimes 2}$ has eigenspaces $\mathbb{R}G, \mathbb{L}, \mathbb{M}$ and we can compute its eigenvalues via Thm 3.6. More illuminating, however, is the Gegenbauer expansion (see Thm G.5). It requires a new identity Thm E.47 involving Gegenbauer polynomials integrated over a sphere, which may be of independent interest.

**Theorem 3.9** (Batchnorm causes gradient explosion). *Suppose* $\phi(\sqrt{B-1}x), \phi'(\sqrt{B-1}x) \in L^2((1-x^2)^{\frac{B-3}{2}})$. *If* $\phi(\sqrt{B-1}x)$ *has Gegenbauer expansion* $\sum_{l=0}^{\infty} a_l \frac{1}{c_{B-1,l}} C_l^{(\frac{B-3}{2})}(x)$, *then gradients explode at the rate of*

$$\lambda_G^{\downarrow} = \frac{\sum_{l=0}^{\infty} \left(\frac{l+B-3}{B-3} \cdot l\right) a_l^2 r_l}{\sum_{l=0}^{\infty} a_l^2 r_l}$$

*where* $r_l = c_{B-1,l}^{-1}\left(C_l^{(\frac{B-3}{2})}(1) - C_l^{(\frac{B-3}{2})}\left(\frac{-1}{B-1}\right)\right) > 0$. *Consequently, for any non-constant* $\phi$ *(i.e. there is a* $j > 0$ *such that* $a_j \neq 0$*),* $\lambda_G^{\downarrow} > 1$*;* $\phi$ *minimizes* $\lambda_G^{\downarrow}$ *iff it is linear (i.e.* $a_i = 0, \forall i \geq 2$*), in which case gradients explode at the rate of* $\frac{B-2}{B-3}$.

This contrasts starkly with the case of non-normalized fully-connected networks, which can use the weight and bias variances to control its mean field network dynamics (Poole et al., 2016; Schoenholz et al., 2016). As a corollary, we disprove the conjecture of the original batchnorm paper (Ioffe & Szegedy, 2015) that "Batch Normalization may lead the layer Jacobians to have singular values close to 1" in the initialization setting, and in fact prove the exact opposite, that *batchnorm forces the layer Jacobian singular values away from 1*.

Appendix G.1 discusses the numerical evaluation of all eigenvalues, and as usual, the Laplace method yields closed forms for positive homogeneous $\phi$ (Thm G.12). We highlight the result for ReLU.

**Theorem 3.10.** *In a ReLU-batchnorm network, the gradient norm explodes exponentially at the rate of*

$$\frac{1}{B-3}\left(\frac{B-1+J_1'\left(\frac{-1}{B-1}\right)}{1-J_1\left(\frac{-1}{B-1}\right)} - 1\right) \tag{19}$$

*which decreases to* $\frac{\pi}{\pi-1} \approx 1.467$ *as* $B \to \infty$. *In contrast, for a linear batchnorm network, the gradient norm explodes exponentially at the rate of* $\frac{B-2}{B-3}$, *which goes to 1 as* $B \to \infty$.

Fig. 1 shows theory and simulation for ReLU gradient dynamics.

**Weight Gradient**    While all of the above only study the gradient with respect to the hidden preactivations, Appendix L shows that the weight gradient norms at layer $l$ is just $\langle \Pi^l, \mu^* G \rangle = \mu^* \operatorname{tr} \Pi^l$, and thus by Thm 3.9, the weight gradients explode as well at the same rate $\lambda_G^{\downarrow}$.

**Effect of $\epsilon$ as a hyperparameter**    In practice, $\epsilon$ is usually treated as small constant and is not regarded as a hyperparameter to be tuned. Nevertheless, we can investigate its effect on gradient explosion. A straightforward generalization of the analysis presented above to the case of $\epsilon > 0$ suggests somewhat larger $\epsilon$ values than typically used can ameliorate (but not eliminate) gradient explosion problems. See Fig. 5(c,d).

## 3.2 CROSS-BATCH DYNAMICS

**Forward**    In addition to analyzing the correlation between preactivations of samples in a batch, we also study the correlation between those of different batches. The dynamics Eq. (8) can be generalized to simultaneous propagation of $k$ batches (see Eq. (54) and Appendix H).

$$\widetilde{\Sigma}^l = V_{\mathcal{B}_\phi^{\oplus k}}(\widetilde{\Sigma}^{l-1}) = \underset{(h_1,\ldots,h_k)\sim\mathcal{N}(0,\widetilde{\Sigma}^{l-1})}{\mathbb{E}}(\mathcal{B}_\phi(h_1),\ldots,\mathcal{B}_\phi(h_k))^{\otimes 2}, \quad \text{with } \widetilde{\Sigma}^l \in \mathbb{R}^{kB \times kB}. \tag{20}$$

Here the domain of the dynamics is the space of block matrices, with diagonal blocks and off-diagonal blocks resp. representing within-batch and cross-batch covariance. We observe empirically

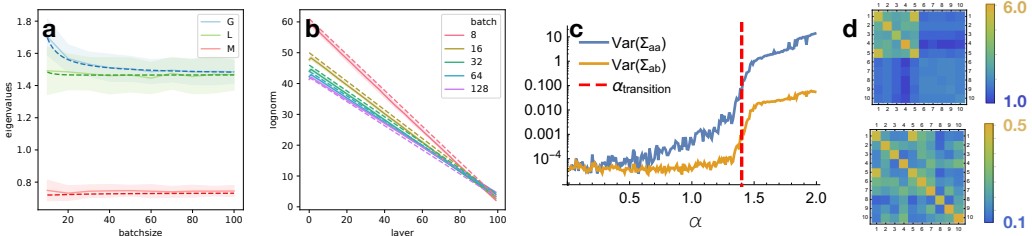

Figure 1: **Numerical confirmation of theoretical predictions.** (a,b) Comparison between theoretical prediction (dashed lines) and Monte Carlo simulations (solid lines) for the eigenvalues of the backwards Jacobian (see Thms 3.10 and G.12) as a function of batch size and the magnitude of gradients as a function of depth respectively for rectified linear networks. In each case Monte Carlo simulations are averaged over 200 sample networks of width 1000 and shaded regions denote 1 standard deviation. Dashed lines are shifted slightly for easier comparison. (c,d) Demonstration of the existence of a BSB1 to BSB2 symmetry breaking transition as a function of $\alpha$ for $\alpha$-ReLU (i.e. the $\alpha$th power of ReLU) activations. In (c) we plot the empirical variance of the diagonal and off-diagonal entries of the covariance matrix which clearly shows a jump at the transition. In (d) we plot representative covariance matrices for the two phases (BSB1 bottom, BSB2 top).

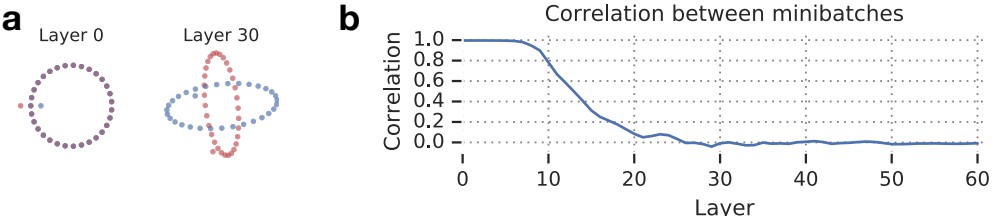

Figure 2: **Batch norm leads to a chaotic input-output map with increasing depth.** A linear network with batch norm is shown acting on two minibatches of size 64 after random orthogonal initialization. The datapoints in the minibatch are chosen to form a 2d circle in input space, except for one datapoint that is perturbed separately in each minibatch (leftmost datapoint at input layer 0). Because the network is linear, for a given minibatch it performs an affine transformation on its inputs – a circle in input space remains an ellipse throughout the network. However, due to batch norm the coefficients of that affine transformation change nonlinearly as the datapoints in the minibatch are changed. *(a)* Each pane shows a scatterplot of activations at a given layer for all datapoints in the minibatch, projected onto the top two PCA directions. PCA directions are computed using the concatenation of the two minibatches. Due to the batch norm nonlinearity, minibatches that are nearly identical in input space grow increasingly dissimilar with depth. Intuitively, this chaotic input-output map can be understood as the source of exploding gradients when batch norm is applied to very deep networks, since very small changes in an input correspond to very large movements in network outputs. *(b)* The correlation between the two minibatches, as a function of layer, for the same network. Despite having a correlation near one at the input layer, the two minibatches rapidly decorrelate with depth. See Appendix H for a theoretical treatment.

that for most nonlinearities used in practice like ReLU, or even for the identity function, ie no pointwise nonlinearity, the global fixed point of this dynamics is *cross-batch BSB1*, with diagonal BSB1 blocks, and off-diagonal entries all equal to the same constant $c_{\text{CB}}^*$:

$$\widetilde{\Sigma}^* = \begin{pmatrix} \Sigma^* & c_{\text{CB}}^* & c_{\text{CB}}^* & \cdots \\ c_{\text{CB}}^* & \Sigma^* & c_{\text{CB}}^* & \cdots \\ c_{\text{CB}}^* & c_{\text{CB}}^* & \Sigma^* & \cdots \\ \vdots & \vdots & \vdots & \ddots \end{pmatrix}. \tag{21}$$

To interpret this phenomenon, note that, after mean centering of the preactivations, this covariance matrix becomes a multiple of identity. Thus, deep embedding of batches loses the mutual information between them in the input space. Qualitatively, this implies that two batches that are similar at the input to the network will become increasingly dissimilar — i.e. *chaotic* — as the signal propagates deep into the network. This loss in fact happens exponentially fast, as illustrated in Fig. 2, and as shown theoretically by Thm H.13, at a rate of $\lambda_{\widetilde{\mathbb{M}}}^{\uparrow} < 1$. Formally, the linearized dynamics of Eq. (20) has an eigenspace $\widetilde{\mathbb{M}}$ given by all block matrices with zero diagonal blocks. The associated eigenvalue is $\lambda_{\widetilde{\mathbb{M}}}^{\uparrow}$. We shall return to $\lambda_{\widetilde{\mathbb{M}}}^{\downarrow}$ shortly and give its Gegenbauer expansion (Thm 3.11).

**Backward**   The gradients of two batches of input are correlated throughout the course of backpropagation. We hence also study the generalization of Eq. (17) to the $k$-batch setting (see Appendix I):

$$\tilde{\Pi}^l = V_{\mathcal{B}_\phi^{\oplus 2'}}(\widetilde{\Sigma}^l)^\dagger \{\tilde{\Pi}^{l+1}\} \tag{22}$$

$$V_{\mathcal{B}_\phi^{\oplus 2'}}(\widetilde{\Sigma}^*)^\dagger \left\{ \begin{pmatrix} \Pi_1 & \Xi \\ \Xi^T & \Pi_2 \end{pmatrix} \right\} = \mathop{\mathbb{E}}_{(x,y)\sim\mathcal{N}(0,\widetilde{\Sigma}^*)} \left[ \begin{pmatrix} \mathcal{B}_\phi'(x)^T\Pi_1\mathcal{B}_\phi'(x) & \mathcal{B}_\phi'(x)^T\Xi\mathcal{B}_\phi'(y) \\ \mathcal{B}_\phi'(y)^T\Xi^T\mathcal{B}_\phi'(x) & \mathcal{B}_\phi'(y)^T\Pi_2\mathcal{B}_\phi'(y) \end{pmatrix} \right] \tag{23}$$

where for simplicity of exposition and without loss of generality, we take $k = 2$. Like in Eq. (17), we have assumed that $\widetilde{\Sigma}^l$ has converged to its limit $\widetilde{\Sigma}^*$. It is not hard to see that each block of $\tilde{\Pi}^l$ evolves independently, with the diagonal blocks in particular evolving according to Eq. (17). Analyzing the off-diagonal blocks might seem unwieldy at first, but it turns out that its dynamics is given simply by scalar multiplication by a constant $\lambda_{\widetilde{\mathbb{M}}}^{\downarrow}$ (Thm I.4). Even more surprisingly,

**Theorem 3.11** (Batchnorm causes information loss). *The convergence rate of cross-batch preactivation covariance $\lambda_{\widetilde{\mathbb{M}}}^{\uparrow}$ is equal to the decay rate of the cross-batch gradient covariance $\lambda_{\widetilde{\mathbb{M}}}^{\downarrow}$. If $\phi(\sqrt{B-1}x)$ has Gegenbauer expansion $\sum_{l=0}^{\infty} a_l \frac{1}{c_{B-1,l}} C_l^{(\frac{B-3}{2})}(x)$, then both are given by*

$$\lambda_{\mathbb{M}}^{\uparrow} = \lambda_{\mathbb{M}}^{\downarrow} = \frac{\frac{1}{2\pi}B(B-1)a_1^2 \operatorname{Beta}\left(\frac{B}{2},\frac{1}{2}\right)^2}{\sum_{l=0}^{\infty} a_l^2 \frac{1}{c_{B-1,l}} \left( C_l^{(\frac{B-3}{2})}(1) - C_l^{(\frac{B-3}{2})}\left(\frac{-1}{B-1}\right) \right)}. \tag{24}$$

*This quantity is always $< 1$, and for any fixed $B$, is maximized by $\phi = \mathrm{id}$. Furthermore, for $\phi = \mathrm{id}$, $\lambda_{\mathbb{M}}^{\uparrow} = \lambda_{\mathbb{M}}^{\downarrow} = \frac{B-1}{2}\mathrm{P}\left(\frac{B}{2},\frac{1}{2}\right)^{-2}$ and increase to 1 from below as $B \to \infty$.*

Thus, a deep batchnorm network loses correlation information between two input batches, exponentially fast in depth, no matter what nonlinearity (that induces fixed point of the form given by Eq. (21)). This again contrasts with the case for vanilla networks which can control this rate of information loss by tweaking the initialization variances for weights and biases Poole et al. (2016); Schoenholz et al. (2016). The absence of coordinatewise nonlinearity, i.e. $\phi = \mathrm{id}$, maximally suppresses both this loss of information as well as the gradient explosion, and in the ideal, infinite batch scenario, can cure both problems.

## 4   EXPERIMENTS

Having developed a theory for neural networks with batch normalization at initialization, we now explore the relationship between the properties of these random networks and their learning dynamics. We will see that the trainability of networks with batch normalization is controlled by gradient explosion. We quantify the depth scale over which gradients explode by $\xi = 1/\log \lambda_G^{\downarrow}$ where, as above, $\lambda_G^{\downarrow}$ is the largest eigenvalue of the jacobian. Across many different experiments we will see strong agreement between $\xi$ and the maximum trainable depth.

We first investigate the relationship between trainability and initialization for rectified linear networks as a function of batch size. The results of these experiments are shown in Fig. 3 where in each case we plot the test accuracy after training as a function of the depth and the batch size and overlay $16\xi$ in white dashed lines. In Fig. 3 (a) we consider networks trained using SGD on MNIST where we observe that networks deeper than about 50 layers are untrainable regardless of batch size. In (b) we compare standard batch normalization with a modified version in which the batch size is

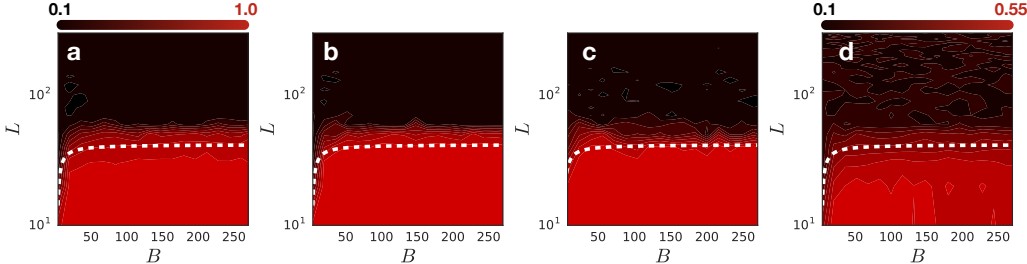

Figure 3: **Batch normalization strongly limits the maximum trainable depth.** Colors show test accuracy for rectified linear networks with batch normalization and $\gamma = 1$, $\beta = 0$, $\epsilon = 10^{-3}$, $N = 384$, and $\eta = 10^{-5}B$. (a) trained on MNIST for 10 epochs (b) trained with fixed batch size 1000 and batch statistics computed over sub batches of size $B$. (c) trained using RMSProp. (d) Trained on CIFAR10 for 50 epochs. In each case, White dashed line indicates a theoretical prediction of trainable depth, as discussed in the text.

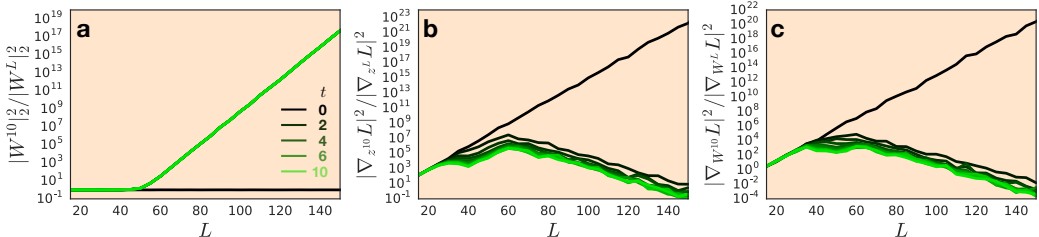

Figure 4: **Gradients in networks with batch normalization quickly achieve dynamical equilibrium.** Plots of the relative magnitudes of (a) the weights (b) the gradients of the loss with respect to the pre-activations and (c) the gradients of the loss with respect to the weights for rectified linear networks of varying depths during the first 10 steps of training. Colors show step number from 0 (black) to 10 (green).

held fixed but batch statistics are computed over subsets of size $B$. This removes subtle gradient fluctuation effects noted in Smith & Le (2018). In (c) we do the same experiment with RMSProp and in (d) we train the networks on CIFAR10. In all cases we observe a nearly identical trainable region.

It is counter intuitive that training can occur at intermediate depths, from 10 to 50 layers, where there is significant gradient explosion. To gain insight into the behavior of the network during learning we record the magnitudes of the weights, the gradients with respect to the pre-activations, and the gradients with respect to the weights for the first 10 steps of training for networks of different depths. The result of this experiment is shown in Fig. 4. Here we see that before learning, as expected, the norm of the weights is constant and independent of layer while the gradients feature exponential explosion. However, we observe that two related phenomena occur after a single step of learning: the weights grow exponentially in the depth and the magnitude of the gradients are stable up to some threshold after which they vanish exponentially in the depth. This is as the result of the scaling property of batchnorm, where $\mathcal{B}'_\phi(\alpha h) = \alpha^{-1} \mathcal{B}'_\phi(h)$: The first-step gradients dominate the weights due to gradient explosion, hence the exponential growth in weight norms, and thereafter, the gradients are scaled down commensurately. Thus, it seems that although the gradients of batch normalized networks at initialization are ill-conditioned, the gradients appear to quickly reach a stable dynamical equilibrium. While this appears to be beneficial for shallower networks, in deeper ones, the relative gradient vanishing can in fact be so severe as to cause lower layers to mostly stay constant during training. Aside from numerical issues, this seems to be the primary mechanism through which gradient explosion causes training problems for networks deeper than 50 layers.

As discussed in the theoretical exposition above, batch normalization necessarily features exploding gradients for any nonlinearity that converges to a BSB1 fixed point. We performed a number of experiments exploring different ways of ameliorating this gradient explosion. These experiments

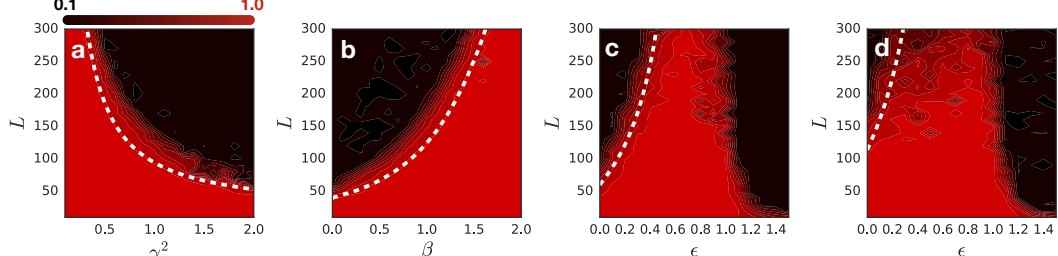

Figure 5: **Three techniques for counteracting gradient explosion.** Test accuracy on MNIST as a function of different hyperparameters along with theoretical predictions (white dashed line) for the maximum trainable depth. (a) tanh network changing the overall scale of the pre-activations, here $\gamma \to 0$ corresponds to the linear regime. (b) Rectified linear network changing the mean of the pre-activations, here $\beta \to \infty$ corresponds to the linear regime. (c,d) tanh and rectified linear networks respectively as a function of $\epsilon$, here we observe a well defined phase transition near $\epsilon \sim 1$. Note that in the case of rectified linear activations we use $\beta = 2$ so that the function is locally linear about 0. We also find initializing $\beta$ and/or setting $\epsilon > 0$ having positive effect on VGG19 with batchnorm. See Figs. 8 and 9.

are shown in Fig. 5 with theoretical predictions for the maximum trainable depth overlaid; in all cases we see exceptional agreement. In Fig. 5 (a,b) we explore two different ways of tuning the degree to which activation functions in a network are nonlinear. In Fig. 5 (a) we tune $\gamma \in [0, 2]$ for networks with tanh-activations and note that in the $\gamma \to 0$ limit the function is linear. In Fig. 5 (b) we tune $\beta \in [0, 2]$ for networks with rectified linear activations and we note, similarly, that in the $\beta \to \infty$ limit the function is linear. As expected, we see the maximum trainable depth increase significantly with decreasing $\gamma$ and increasing $\beta$. In Fig. 5 (c,d) we vary $\epsilon$ for tanh and rectified linear networks respectively. In both cases, we observe a critical point at large $\epsilon$ where gradients do not explode and very deep networks are trainable.

## 5 CONCLUSION

In this work we have presented a theory for neural networks with batch normalization at initialization. In the process of doing so, we have uncovered a number of counterintuitive aspects of batch normalization and – in particular – the fact that at initialization it unavoidably causes gradients to explode with depth. We have introduced several methods to reduce the degree of gradient explosion, enabling the training of significantly deeper networks in the presence of batch normalization. Finally, this work paves the way for future work on more advanced, state-of-the-art, network architectures and topologies.

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

Barret Zoph, Vijay Vasudevan, Jonathon Shlens, and Quoc V Le. Learning transferable architectures for scalable image recognition.

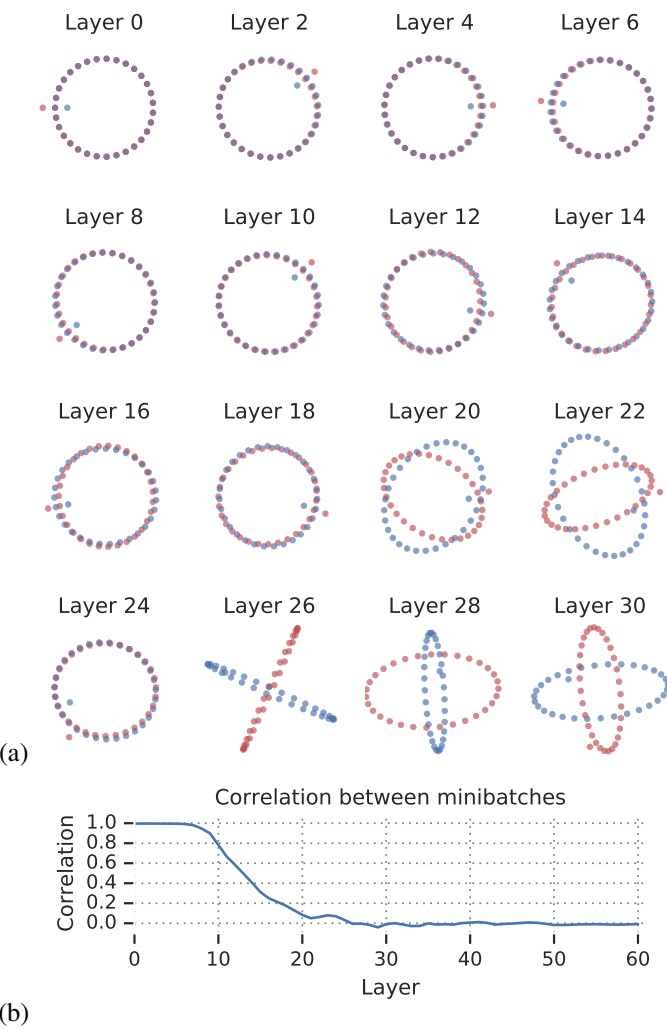

(a)

(b)

Figure 6: **Batch norm leads to a chaotic input-output map with increasing depth.** A linear network with batch norm is shown acting on two minibatches of size 64 after random orthogonal initialization. The datapoints in the minibatch are chosen to form a 2d circle in input space, except for one datapoint that is perturbed separately in each minibatch (leftmost datapoint at input layer 0). Because the network is linear, for a given minibatch it performs an affine transformation on its inputs – a circle in input space remains an ellipse throughout the network. However, due to batch norm the coefficients of that affine transformation change nonlinearly as the datapoints in the minibatch are changed. *(a)* Each pane shows a scatterplot of activations at a given layer for all datapoints in the minibatch, projected onto the top two PCA directions. PCA directions are computed using the concatenation of the two minibatches. Due to the batch norm nonlinearity, minibatches that are nearly identical in input space grow increasingly dissimilar with depth. Intuitively, this chaotic input-output map can be understood as the source of exploding gradients when batch norm is applied to very deep networks, since very small changes in an input correspond to very large movements in network outputs. *(b)* The correlation between the two minibatches, as a function of layer, for the same network. Despite having a correlation near one at the input layer, the two minibatches rapidly decorrelate with depth. See Appendix H for a theoretical treatment.

## A    VGG19 WITH BATCHNORM ON CIFAR100

Even though at initialization time batchnorm causes gradient explosion, after the first few epochs, the relative gradient norms $\|\nabla_\theta L\|/\|\theta\|$ for weight parameters $\theta = W$ or BN scale parameter $\theta = \gamma$, equilibrate to about the same magnitude. See Fig. 7.

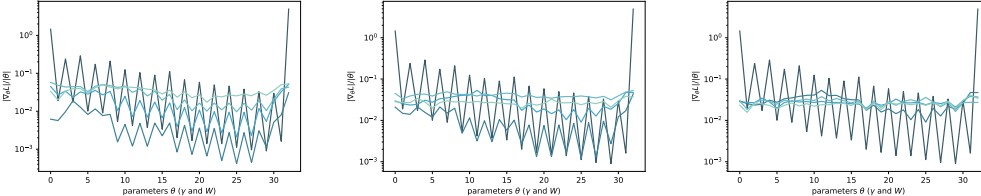

Figure 7: relative gradient norms of different parameters in layer order (input to output from left to right), with $\gamma$ and $W$ interleaving. From dark to light blue, each curve is separated by (a) 3, (b) 5, or (c) 10 epochs. We see that after 10 epochs, the relative gradient norms of both $\gamma$ and $W$ for all layers become approximately equal despite gradient explosion initially.

We find acceleration effects, especially in initial training, due to setting $\epsilon > 0$ and/or initializing $\beta > 0$. See Figs. 8 and 9.

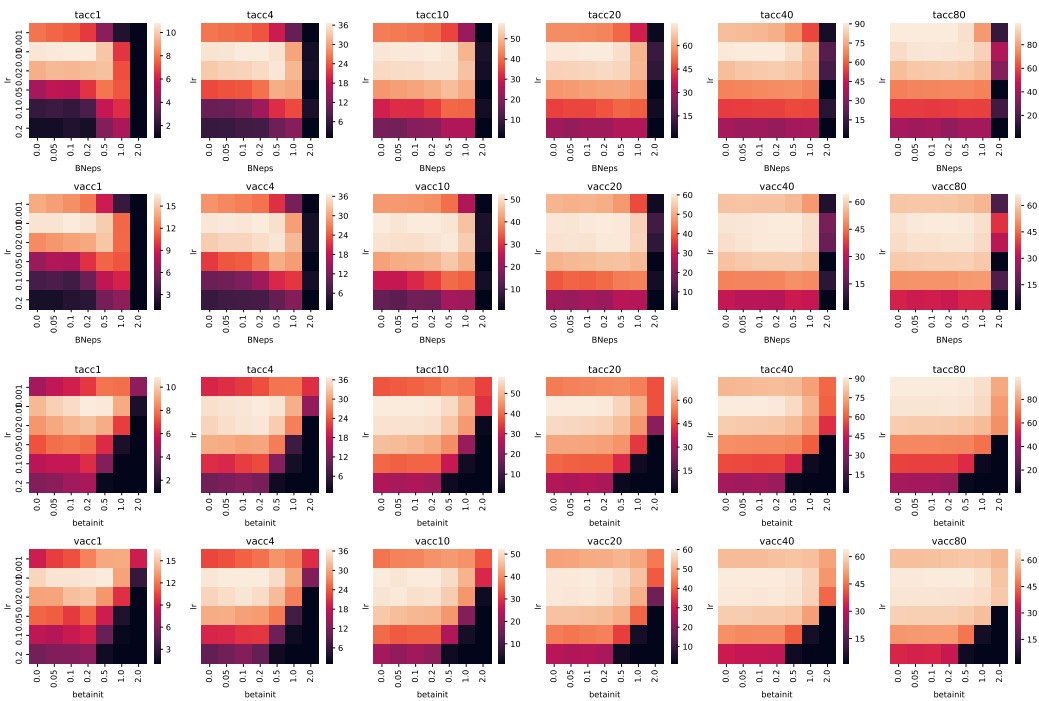

Figure 8: We sweep over different values of learning rate, $\beta$ initialization, and $\epsilon$, in training VGG19 with batchnorm on CIFAR100 with data augmentation. We use 8 random seeds for each combination, and assign to each combination the median training/validation accuracy over all runs. We then aggregate these scores here. In the first row we look at training accuracy with different learning rate vs $\beta$ initialization at different epochs of training, presenting the max over $\epsilon$. In the second row we do the same for validation accuracy. In the third row, we look at the matrix of training accuracy for learning rate vs $\epsilon$, taking max over $\beta$. In the fourth row, we do the same for validation accuracy.

## B GRADIENT INDEPENDENCE ASSUMPTION

Following prior literature Schoenholz et al. (2016); Yang & Schoenholz (2017); Xiao et al. (2018), in this paper, in regards to computations involving backprop, we assume

*Assumption 2.* During backpropagation, whenever we multiply by $W^T$ for some weight matrix $W$, we multiply by an iid copy instead.

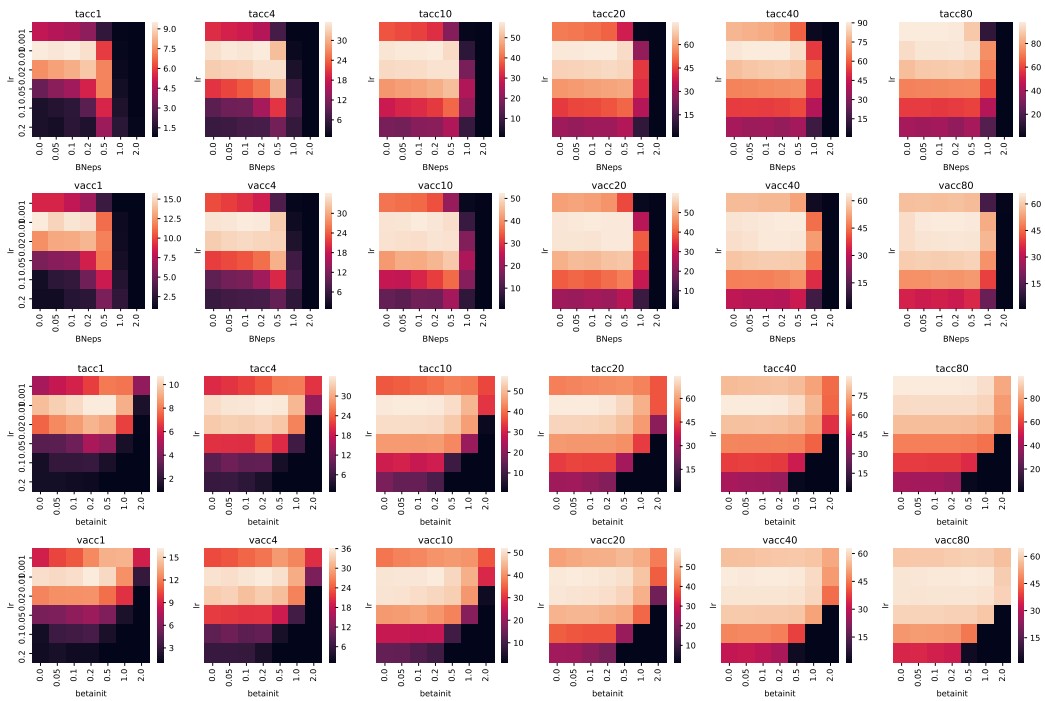

Figure 9: In the same setting as Fig. 8, except we don't take the max over the unseen hyperparameter but rather set it to 0 (the default value).

As in these previous works, we find excellent agreement between computations made under this assumption and the simulations (see Fig. 1). Yang (2019) in fact recently rigorously justified this assumption as used in the computation of moments, for a wide variety of architectures, like multilayer perceptron (Schoenholz et al., 2016), residual networks (Yang & Schoenholz, 2017), and convolutional networks (Xiao et al., 2018) studied previously, but without batchnorm. The reason that their argument does not extend to batchnorm is because of the singularity in its Jacobian at 0. However, as observed in our experiments, we expect that a proof can be found to extend Yang & Schoenholz (2017)'s work to batchnorm.

## C  NOTATIONS

**Definition C.1.** Let $\mathcal{S}_B$ be the space of PSD matrices of size $B \times B$. Given a measurable function $\Phi : \mathbb{R}^B \to \mathbb{R}^A$, define the integral transform $V_\Phi : \mathcal{S}_B \to \mathcal{S}_A$ by $V_\Phi(\Sigma) = \mathbb{E}[\Phi(h)^{\otimes 2} : h \sim \mathcal{N}(0, \Sigma)]$.[3] When $\phi : \mathbb{R} \to \mathbb{R}$ and $B$ is clear from context, we also write $V_\phi$ for $V$ applied to the function acting coordinatewise by $\phi$.

**Definition C.2.** For any $\phi : \mathbb{R} \to \mathbb{R}$, let $\mathcal{B}_\phi : \mathbb{R}^B \to \mathbb{R}^B$ be batchnorm (applied to a batch of neuronal activations) followed by coordinatewise applications of $\phi$, $\mathcal{B}_\phi(x)_j = \phi\left( (x_j - \mathrm{Avg}\,x) / \sqrt{\frac{1}{B} \sum_{i=1}^B (x_i - \mathrm{Avg}\,x)^2} \right)$ (here $\mathrm{Avg}\,x = \frac{1}{B} \sum_{i=1}^B x_i$). When $\phi = \mathrm{id}$ we will also write $\mathcal{B} = \mathcal{B}_{\mathrm{id}}$.

**Definition C.3.** Define the matrix $G^B = I - \frac{1}{B}\mathbf{1}\mathbf{1}^T$. Let $\mathcal{S}_B^G$ be the space of PSD matrices of size $B \times B$ with zero mean across rows and columns, $\mathcal{S}_B^G := \{\Sigma \in \mathcal{S}_B : G^B \Sigma G^B = \Sigma\} = \{\Sigma \in \mathcal{S}_B : \Sigma\mathbf{1} = 0\}$.

---

[3]This definition of V absorbs the previous definitions of V and W in Yang & Schoenholz (2017) for the scalar case

When $B$ is clear from context, we will suppress the subscript/superscript $B$. In short, for $h \in \mathbb{R}^B$, $Gh$ zeros the sample mean of $h$. $G$ is a projection matrix to the subspace of vectors $h \in \mathbb{R}^B$ of zero coordinate sum. With the above definitions, we then have $\mathcal{B}_\phi(h) = \phi(\sqrt{B}Gh/\|Gh\|)$.

We use $\Gamma$ to denote the Gamma function, $P(a, b) := \Gamma(a+b)/\Gamma(a)$ to denote the Pochhammer symbol, and $\text{Beta}(a, b) = \frac{\Gamma(a)\Gamma(b)}{\Gamma(a+b)}$ to denote the Beta function. We use $\langle \cdot, \cdot \rangle$ to denote the dot product for vectors and trace inner product for matrices. Sometimes when it makes for simpler notation, we will also use $\cdot$ for dot product.

We adopt the matrix convention that, for a multivariate function $f : \mathbb{R}^n \to \mathbb{R}^m$, the Jacobian is

$$\frac{\mathrm{d}f}{\mathrm{d}x} = \begin{pmatrix} \frac{\partial f_1}{\partial x_1} & \frac{\partial f_1}{\partial x_2} & \cdots & \frac{\partial f_1}{\partial x_n} \\ \frac{\partial f_2}{\partial x_1} & \frac{\partial f_2}{\partial x_2} & \cdots & \frac{\partial f_2}{\partial x_n} \\ \vdots & \vdots & \ddots & \vdots \\ \frac{\partial f_m}{\partial x_1} & \frac{\partial f_m}{\partial x_2} & \cdots & \frac{\partial f_m}{\partial x_n} \end{pmatrix},$$

so that $f(x + \Delta x) = \frac{\mathrm{d}f}{\mathrm{d}x}\Delta x + o(\Delta x)$, where $x$ and $\Delta x$ are treated as column vectors. In what follows we further abbreviate $\mathcal{B}'_\phi(x) = \frac{\mathrm{d}\mathcal{B}_\phi(z)}{\mathrm{d}z}\big|_{z=x}, \mathcal{B}'_\phi(x)\Delta x = \frac{\mathrm{d}\mathcal{B}_\phi(z)}{\mathrm{d}z}\big|_{z=x}\Delta x$.

## D    A GUIDE TO THE REST OF THE APPENDIX

As discussed in the main text, we are interested in several closely related dynamics induced by batchnorm in a fully-connected network with random weights. One is the forward propagation equation Eq. (43)

$$\Sigma^l = \mathrm{V}_{\mathcal{B}_\phi}(\Sigma^{l-1})$$

studied in Appendix F. Another is the backward propagation equation Eq. (51)

$$\Pi^l = \mathrm{V}_{\mathcal{B}'_\phi}(\Sigma^*)^\dagger\{\Pi^{l+1}\}$$

studied in Appendix G. We will also study their generalizations to the simultaneous propagation of multiple batches in Appendices H and I.

In general, we discover several ways to go about such analyses, each with their own benefits and drawbacks:

- *The Laplace Method* (Appendix E.1): If the nonlinearity $\phi$ is positive homogeneous, we can use Schwinger's parametrization to turn $\mathrm{V}_{\mathcal{B}_\phi}$ and $\mathrm{V}_{\mathcal{B}'_\phi}$ into nicer forms involving only 1 or 2 integrals, as mentioned in the main text. All relevant quantities, such as eigenvalues of the backward dynamics, can be further simplified to closed forms. Lemma E.2 gives the *master equation* for the Laplace method. Because the Laplace method requires $\phi$ to be positive homogeneous to apply, we review relevant results of such functions in Appendices E.1.1 and E.1.2.

- *The Fourier Method* (Appendix E.2): For polynomially bounded and continuous nonlinearity $\phi$ (for the most general set of conditions, see Thm E.25), we can obtain similar simplifications using the Fourier expansion of the delta function, with the penalty of an additional complex integral.

- *Spherical Integration* (Appendix E.3): Batchnorm can naturally be interpreted as a linear projection followed by projection to a sphere of radius $\sqrt{B}$. Assuming that the forward dynamics converges to a BSB1 fixed point, one can express $\mathrm{V}_{\mathcal{B}_\phi}$ and $\mathrm{V}_{\mathcal{B}'_\phi}$ as spherical integrals, for all measurable $\phi$. One can reduce this $(B-1)$-dimensional integral, in spherical angular coordinates, into 1 or 2 dimensions by noting that the integrand depending on $\phi$ only depends on 1 or 2 angles. Thus this method is very suitable for numerical evaluation of quantities of interest for general $\phi$.

- *The Gegenbauer Method* (Appendix E.4): Gegenbauer polynomials are orthogonal polynomials under the weight $(1 - x^2)^\alpha$ for some $\alpha$. They correspond to a special type of

spherical harmonics called *zonal harmonics* (Defn E.44) that has the very useful *reproducing property* (Fact E.46). By expressing the nonlinearity $\phi$ in this basis, we can see easily that the relevant eigenvalues of the forward and backward dynamics are ratios of two quadratic forms of $\phi$'s Gegenbauer coefficients. In particular, for the largest eigenvalue of the backward dynamics, the two quadratic forms are both diagonal, and in a way that makes apparent the necessity of gradient explosion (under the BSB1 fixed point assumption). In regards to numerical computations, the Gegenbauer method has the benefit of only requiring 1-dimensional integrals to obtain the coefficients, but if $\phi$ has slowly-decaying coefficients, then a large number of such integrals may be required to get an accurate answer.

In each of the following sections, we will attempt to conduct an analysis with each method when it is feasible.

When studying the convergence rate of the forward dynamics and the gradient explosion of the backward dynamics, we encounter linear operators on matrices that satisfy an abundance of symmetries, called *ultrasymmetric* operators. We study these operators in Appendix E.5 and contribute several structural results Thms E.61, E.62, E.72 and E.72 on their eigendecomposition that are crucial to deriving the asymptotics of the dynamics.

All of the above will be explained in full in the technique section Appendix E. We then proceed to study the forward and backward dynamics in detail (Appendices F to I), now armed with tools we need. Up to this point, we only consider the dynamics assuming Eq. (43) converges to a BSB1 fixed point, and $B \geq 4$. In Appendix J we discuss what happens outside this regime, and in Appendix K we show our current understanding of BSB2 fixed point dynamics.

Note that the backward dynamics studied in these sections is that of the gradient with respect to the hidden preactivations. Nevertheless, we can compute the moments of the weight gradients from this, which is of course what is eventually used in gradient descent. We do so in the final section Appendix L.

**Main Technical Results**   Corollary F.3 establishes the global convergence of Eq. (43) to BSB1 fixed point for $\phi = \mathrm{id}$ (as long as the initial covariance is nondegenerate), but we are not able to prove such a result for more general $\phi$. We thus resort to studying the local convergence properties of BSB1 fixed points.

First, Thms F.5, F.8 and F.13 respectively compute the BSB1 fixed point from the perspectives of spherical integration, the Laplace method, and the Gegenbauer method. Thms F.22 and F.24 give the Gegenbauer expansion of the BSB1 local convergence rate, and Thm F.33 gives a more succinct form of the same for positive homogeneous $\phi$. Appendix F.3.1 discusses how to do this computation via spherical integration.

Next, we study the gradient dynamics, and Thms G.5 and G.12 yield the gradient explosion rates with the Gegenbauer method and the Laplace method (for positive homogeneous $\phi$), respectively. Appendix G.1.1 discusses how to compute this for general $\phi$ using spherical integration.

We then turn to the dynamics of cross batch covariances. Thm H.3 gives the form of the cross batch fixed point via spherical integration and the Gegenbauer method, and Thm H.6 yields a more specific form for positive homogeneous $\phi$ via the Laplace method. The local convergence rate to this fixed point is given by Thms H.13 and H.14, respectively using the Gegenbauer method and the Laplace method. Finally, the correlation between gradients of the two batches is shown to decrease exponentially in Corollary I.6 using the Gegenbauer method, and the decay rate is given succinctly for positive homogeneous $\phi$ in Thm I.3.

# E   TECHNIQUES

We introduce the key techniques and tools in our analysis in this section.

### E.1 LAPLACE METHOD

As discussed above, the Laplace method is useful for deriving closed form expressions for positive homogeneous $\phi$. The key insight here is to apply Schwinger parametrization to deal with normalization.

**Lemma E.1** (Schwinger parametrization). *For $z > 0$ and $c > 0$,*

$$c^{-z} = \Gamma(z)^{-1} \int_0^\infty x^{z-1} e^{-cx} \, \mathrm{d}x$$

The following is the key lemma in the Laplace method.

**Lemma E.2** (The Laplace Method Master Equation). *For $A, A' \in \mathbb{N}$, let $f : \mathbb{R}^A \to \mathbb{R}^{A'}$ and let $k \geq 0$. Suppose $\|f(y)\| \leq h(\|y\|)$ for some nondecreasing function $h : \mathbb{R}^{\geq 0} \to \mathbb{R}^{\geq 0}$ such that $\mathbb{E}[h(r\|y\|) : y \in \mathcal{N}(0, I_A)]$ exists for every $r \geq 0$. Define $\wp(\Sigma) := \mathbb{E}[\|y\|^{-2k} f(y) : y \sim \mathcal{N}(0, \Sigma)]$. Then on $\{\Sigma \in \mathcal{S}_A : \mathrm{rank}\,\Sigma > 2k\}$, $\wp(\Sigma)$ is well-defined and continuous, and furthermore satisfies*

$$\wp(\Sigma) = \Gamma(k)^{-1} \int_0^\infty \mathrm{d}s \, s^{k-1} \det(I + 2s\Sigma)^{-1/2} \, \mathbb{E}[f(y) : y \sim \mathcal{N}(0, \Sigma(I + 2s\Sigma)^{-1})] \qquad (25)$$

*Proof.* $\wp(\Sigma)$ is well-defined for full rank $\Sigma$ because the $\|y\|^{-2k}$ singularity at $y = 0$ is Lebesgue-integrable in a neighborhood of 0 in dimension $A > 2k$.

We prove Eq. (25) in the case when $\Sigma$ is full rank and then apply a continuity argument.

*Proof of Eq. (25) for full rank $\Sigma$.* First, we will show that we can exchange the order of integration

$$\int_0^\infty \mathrm{d}s \int_{\mathbb{R}^A} \mathrm{d}y \, s^{k-1} f(y) e^{-\frac{1}{2} y^T (\Sigma^{-1} + 2sI) y} = \int_{\mathbb{R}^A} \mathrm{d}y \int_0^\infty \mathrm{d}s \, s^{k-1} f(y) e^{-\frac{1}{2} y^T (\Sigma^{-1} + 2sI) y}$$

by Fubini-Tonelli's theorem. Observe that

$$(2\pi)^{-A/2} \det \Sigma^{-1/2} \int_0^\infty \mathrm{d}s \, s^{k-1} \int_{\mathbb{R}^A} \mathrm{d}y \, \|f(y)\| e^{-\frac{1}{2} y^T (\Sigma^{-1} + 2sI) y}$$

$$= \int_0^\infty \mathrm{d}s \, s^{k-1} \det(\Sigma(\Sigma^{-1} + 2sI))^{-1/2} \, \mathbb{E}[\|f(y)\| : y \sim \mathcal{N}(0, (\Sigma^{-1} + 2sI)^{-1})]$$

$$= \int_0^\infty \mathrm{d}s \, s^{k-1} \det(I + 2s\Sigma)^{-1/2} \, \mathbb{E}[\|f(y)\| : y \sim \mathcal{N}(0, \Sigma(I + 2s\Sigma)^{-1})]$$

For $\lambda = \|\Sigma\|_2$,

$$\|f(\sqrt{\Sigma(I + 2s\Sigma)^{-1}} y)\| \leq h(\|\sqrt{\Sigma(I + 2s\Sigma)^{-1}} y\|) \leq h\left(\left\|\sqrt{\frac{\lambda}{1 + 2s\lambda}} y\right\|\right) \leq h(\sqrt{\lambda} \|y\|)$$

Because $\mathbb{E}[h(\sqrt{\lambda} \|y\|) : y \in \mathcal{N}(0, I)]$ exists by assumption,

$$\mathbb{E}[\|f(y)\| : y \sim \mathcal{N}(0, \Sigma(I + 2s\Sigma)^{-1})]$$
$$= \mathbb{E}[\|f(\sqrt{\Sigma(I + 2s\Sigma)^{-1}} y)\| : y \sim \mathcal{N}(0, I)]$$
$$\to \mathbb{E}[\|f(0)\| : y \sim \mathcal{N}(0, I)] = \|f(0)\|$$

as $s \to \infty$, by dominated convergence with dominating function $h(\sqrt{\lambda} \|y\|) e^{-\frac{1}{2} \|y\|^2} (2\pi)^{-A/2}$. By the same reasoning, the function $s \mapsto \mathbb{E}[\|f(y)\| : y \sim \mathcal{N}(0, \Sigma(I + 2s\Sigma)^{-1})]$ is continuous. In particular this implies that $\sup_{0 \leq s \leq \infty} \mathbb{E}[\|f(y)\| : y \sim \mathcal{N}(0, \Sigma(I + 2s\Sigma)^{-1})] < \infty$. Combined with the fact that $\det(I + 2s\Sigma)^{-1/2} = \Theta(s^{-A/2})$ as $s \to \infty$,

$$\int_0^\infty \mathrm{d}s \, s^{k-1} \det(I + 2s\Sigma)^{-1/2} \, \mathbb{E}[\|f(y)\| : y \sim \mathcal{N}(0, \Sigma(I + 2s\Sigma)^{-1})]$$
$$\leq \int_0^\infty \mathrm{d}s \, \Theta(s^{k-1-A/2})$$

which is bounded by our assumption that $A/2 > k$. This shows that we can apply Fubini-Tonelli's theorem to allow exchanging order of integration.

Thus,

$$\mathbb{E}[\|y\|^{-2k} f(y) : y \sim \mathcal{N}(0, \Sigma)]$$

$$= \mathbb{E}[f(y) \int_0^\infty ds\, \Gamma(k)^{-1} s^{k-1} e^{-\|y\|^2 s} : y \sim \mathcal{N}(0, \Sigma)]$$

$$= (2\pi)^{-A/2} \det \Sigma^{-1/2} \int dy\, e^{-\frac{1}{2} y^T \Sigma^{-1} y} f(y) \int_0^\infty ds\, \Gamma(k)^{-1} s^{k-1} e^{-\|y\|^2 s}$$

$$= (2\pi)^{-A/2} \Gamma(k)^{-1} \det \Sigma^{-1/2} \int_0^\infty ds\, s^{k-1} \int dy\, f(y) e^{-\frac{1}{2} y^T (\Sigma^{-1} + 2sI) y}$$

$$\text{(by Fubini-Tonelli)}$$

$$= \Gamma(k)^{-1} \int_0^\infty ds\, s^{k-1} \det(\Sigma(\Sigma^{-1} + 2sI))^{-1/2} \mathbb{E}[f(y) : y \sim \mathcal{N}(0, (\Sigma^{-1} + 2sI)^{-1})]$$

$$= \Gamma(k)^{-1} \int_0^\infty ds\, s^{k-1} \det(I + 2s\Sigma)^{-1/2} \mathbb{E}[f(y) : y \sim \mathcal{N}(0, \Sigma(I + 2s\Sigma)^{-1})]$$

*Domain and continuity of $\wp(\Sigma)$.* The LHS of Eq. (25), $\wp(\Sigma)$, is defined and continuous on $\operatorname{rank}\Sigma/2 > k$. Indeed, if $\Sigma = M I_C M^T$, where $M$ is a full rank $A \times C$ matrix with $\operatorname{rank}\Sigma = C \leq A$, then

$$\mathbb{E}[\|y\|^{-2k} \|f(y)\| : y \sim \mathcal{N}(0, \Sigma)]$$

$$= \mathbb{E}[\|Mz\|^{-2k} \|f(Mz)\| : z \sim \mathcal{N}(0, I_C)].$$

This is integrable in a neighborhood of 0 iff $C > 2k$, while it's always integrable outside a ball around 0 because $\|f\|$ by itself already is. So $\wp(\Sigma)$ is defined whenever $\operatorname{rank}\Sigma > 2k$. Its continuity can be established by dominated convergence.

*Proof of Eq. (25) for $\operatorname{rank}\Sigma > 2k$.* Observe that $\det(I + 2s\Sigma)^{-1/2}$ is continuous in $\Sigma$ and, by an application of dominated convergence as in the above, $\mathbb{E}[f(y) : y \sim \mathcal{N}(0, \Sigma(I + 2s\Sigma)^{-1})]$ is continuous in $\Sigma$. So the RHS of Eq. (25) is continuous in $\Sigma$ whenever the integral exists. By the reasoning above, $\mathbb{E}[f(y) : y \sim \mathcal{N}(0, \Sigma(I + 2s\Sigma)^{-1})]$ is bounded in $s$ and $\det(I + 2s\Sigma)^{-1/2} = \Theta(s^{-\operatorname{rank}\Sigma/2})$, so that the integral exists iff $\operatorname{rank}\Sigma/2 > k$.

To summarize, we have proved that both sides of Eq. (25) are defined and continous for $\operatorname{rank}\Sigma > 2k$. Because the full rank matrices are dense in this set, by continuity Eq. (25) holds for all $\operatorname{rank}\Sigma > 2k$. $\qquad\square$

If $\phi$ is degree-$\alpha$ positive homogeneous, i.e. $\phi(ru) = r^\alpha \phi(u)$ for any $u \in \mathbb{R}, r \in \mathbb{R}^+$, we can apply Lemma E.2 with $k = \alpha$,

$$V_{\mathcal{B}_\phi}(\Sigma) = \mathbb{E}[\phi(\sqrt{B}Gh/\|Gh\|)^{\otimes 2} : h \sim \mathcal{N}(0, \Sigma)] = B^\alpha \mathbb{E}[\phi(Gh)^{\otimes 2}/\|Gh\|^{2\alpha} : h \sim \mathcal{N}(0, \Sigma)]$$

$$= B^\alpha \mathbb{E}[\phi(z)^{\otimes 2}/\|z\|^{2\alpha} : z \sim \mathcal{N}(0, G\Sigma G)]$$

$$= B^\alpha \Gamma(\alpha)^{-1} \int_0^\infty ds\, s^{\alpha-1} \det(I + 2s\Sigma G)^{-1/2} V_\phi(G(I + 2s\Sigma G)^{-1}\Sigma G).$$

If $\phi = \text{relu}$, then

$$V_\phi(\Sigma)_{ij} = \begin{cases} \frac{1}{2}\Sigma_{ii} & \text{if } i = j \\ \frac{1}{2} J_1(\Sigma_{ij}/\sqrt{\Sigma_{ii}\Sigma_{jj}})\sqrt{\Sigma_{ii}\Sigma_{jj}} & \text{otherwise} \end{cases}$$

and, more succinctly, $V_\phi(\Sigma) = D^{1/2} V_\phi(D^{-1/2}\Sigma D^{-1/2}) D^{1/2}$ where $D = \text{Diag}(\Sigma)$. Here $J_1(c) := \frac{1}{\pi}(\sqrt{1 - c^2} + (\pi - \arccos(c))c)$ (Cho & Saul, 2009).

**Matrix simplification.** We can simplify the expression $G(I + 2s\Sigma G)^{-1}\Sigma G$, leveraging the fact that $G$ is a projection matrix.

**Definition E.3.** Let $\mathbb{e}$ be an $B \times (B-1)$ matrix whose columns form an orthonormal basis of $\operatorname{im} G := \{Gv : v \in \mathbb{R}^B\} = \{w \in \mathbb{R}^B : \sum_i w_i = 0\}$. Then the $B \times B$ matrix $\tilde{\mathbb{e}} = (\mathbb{e}|B^{-1/2}\mathbf{1})$ is an orthogonal matrix. For much of this paper $\mathbb{e}$ can be any such basis, but at certain sections we will consider specific realizations of $\mathbb{e}$ for explicit computation.

From easy computations it can be seen that $G = \tilde{\mathbb{e}} \begin{pmatrix} I_{B-1} & 0 \\ 0 & 0 \end{pmatrix} \tilde{\mathbb{e}}^T$. Suppose $\Sigma = \tilde{\mathbb{e}} \begin{pmatrix} \Sigma_\square & v \\ v^T & a \end{pmatrix} \tilde{\mathbb{e}}^T$ where $\Sigma_\square$ is $(B-1) \times (B-1)$, $v$ is a column vector and $a$ is a scalar. Then $\Sigma_\square = \mathbb{e}^T \Sigma \mathbb{e}$ and $\Sigma G = \tilde{\mathbb{e}} \begin{pmatrix} \Sigma_\square & 0 \\ v^T & 0 \end{pmatrix} \tilde{\mathbb{e}}^T$ is block lower triangular, and

$$(I + 2s\Sigma G)^{-1} = \tilde{\mathbb{e}} \begin{pmatrix} (I + 2s\Sigma_\square)^{-1} & 0 \\ * & 1 \end{pmatrix} \tilde{\mathbb{e}}^T$$

$$(I + 2s\Sigma G)^{-1}\Sigma G = \tilde{\mathbb{e}} \begin{pmatrix} (I + 2s\Sigma_\square)^{-1}\Sigma_\square & 0 \\ * & 0 \end{pmatrix} \tilde{\mathbb{e}}^T$$

$$G(I + 2s\Sigma G)^{-1}\Sigma G = \tilde{\mathbb{e}} \begin{pmatrix} (I + 2s\Sigma_\square)^{-1}\Sigma_\square & 0 \\ 0 & 0 \end{pmatrix} \tilde{\mathbb{e}}^T$$

$$= \mathbb{e}(I + 2s\Sigma_\square)^{-1}\Sigma_\square \mathbb{e}^T$$

$$= G\Sigma G(I + 2sG\Sigma G)^{-1}$$

$$=: \Sigma^G(I + 2s\Sigma^G)^{-1}$$

where

**Definition E.4.** For any matrix $\Sigma$, write $\Sigma^G := G\Sigma G$.

Similarly, $\det(I_B + 2s\Sigma G) = \det(I_{B-1} + 2s\Sigma_\square) = \det(I_B + 2s\Sigma^G)$. So, altogether, we have

**Theorem E.5.** *Suppose $\phi : \mathbb{R} \to \mathbb{R}$ is degree-$\alpha$ positive homogeneous. Then for any $B \times (B-1)$ matrix $\mathbb{e}$ whose columns form an orthonormal basis of $\operatorname{im} G := \{Gv : v \in \mathbb{R}^B\} = \{w \in \mathbb{R}^B : \sum_i w_i = 0\}$, with $\Sigma_\square = \mathbb{e}^T \Sigma \mathbb{e}$,*

$$V_{\mathcal{B}_\phi}(\Sigma) = B^\alpha \Gamma(\alpha)^{-1} \int_0^\infty \mathrm{d}s\, s^{\alpha-1} \det(I + 2s\Sigma_\square)^{-1/2} V_\phi(\mathbb{e}(I + 2s\Sigma_\square)^{-1}\Sigma_\square^{-1}\mathbb{e}^T) \quad (26)$$

$$= B^\alpha \Gamma(\alpha)^{-1} \int_0^\infty \mathrm{d}s\, s^{\alpha-1} \det(I + 2s\Sigma^G)^{-1/2} V_\phi((I + 2s\Sigma^G)^{-1}\Sigma^G)^{-1}) \quad (27)$$

Now, in order to use Laplace's method, we require $\phi$ to be positive homogeneous. What do these functions look like? The most familiar example to a machine learning audience is most likely ReLU. It turns out that 1-dimensional positive homogeneous functions can always be described as a linear combination of powers of ReLU and its reflection across the y-axis. Below, we review known facts about these functions and their integral transforms, starting with the powers of ReLU in Appendix E.1.1 and then onto the general case in Appendix E.1.2.

### E.1.1 $\alpha$-ReLU

Recall that $\alpha$-ReLUs (Yang & Schoenholz, 2017) are, roughly speaking, the $\alpha$th power of ReLU.

**Definition E.6.** The $\alpha$-ReLU function $\rho_\alpha : \mathbb{R} \to \mathbb{R}$ sends $x \mapsto x^\alpha$ when $x > 0$ and $x \mapsto 0$ otherwise.

This is a continuous function for $\alpha > 0$ but discontinuous at 0 for all other $\alpha$.

We briefly review what is currently known about the V and W transforms of $\rho_\alpha$ (Cho & Saul, 2009; Yang & Schoenholz, 2017).

**Definition E.7.** For any $\alpha > -\frac{1}{2}$, define $\mathsf{c}_\alpha = \frac{1}{\sqrt{\pi}} 2^{\alpha-1}\Gamma(\alpha + \frac{1}{2})$.

When considering only 1-dimensional Gaussians, $V_{\rho_\alpha}$ is very simple.

**Proposition E.8.** *If $\alpha > -\frac{1}{2}$, then for any $q \in \mathcal{S}_1 = \mathbb{R}^{\geq 0}$, $V_{\rho_\alpha}(q) = c_\alpha q^\alpha$*

To express results of $V_{\rho_\alpha}$ on $\mathcal{S}_B$ for higher $B$, we first need the following

**Definition E.9.** Define

$$J_\alpha(\theta) := \frac{1}{2\pi c_\alpha}(\sin\theta)^{2\alpha+1}\Gamma(\alpha+1)\int_0^{\pi/2}\frac{\mathrm{d}\eta\cos^\alpha\eta}{(1-\cos\theta\cos\eta)^{1+\alpha}}$$

and $\mathrm{J}_\alpha(c) = J_\alpha(\arccos c)$ for $\alpha > -1/2$.

Then

**Proposition E.10.** *For any $\Sigma \in \mathcal{S}_B$, let $D$ be the diagonal matrix with the same diagonal as $\Sigma$. Then*

$$V_{\rho_\alpha}(\Sigma) = c_\alpha D^{\alpha/2}\mathrm{J}_\alpha(D^{-1/2}\Sigma D^{-1/2})D^{\alpha/2}$$

*where $\mathrm{J}_\alpha$ is applied entrywise.*

For example, $J_\alpha$ and $\mathrm{J}_\alpha$ for the first few integral $\alpha$ are

$$J_0(\theta) = \frac{\pi-\theta}{\pi}$$

$$J_1(\theta) = \frac{\sin\theta + (\pi-\theta)\cos\theta}{\pi}$$

$$J_2(\theta) = \frac{3\sin\theta\cos\theta + (\pi-\theta)(1+2\cos^2\theta)}{3\pi}$$

$$\mathrm{J}_0(c) = \frac{\pi-\arccos c}{\pi}$$

$$\mathrm{J}_1(c) = \frac{\sqrt{1-c^2} + (\pi-\arccos c)c}{\pi}$$

$$\mathrm{J}_2(c) = \frac{3c\sqrt{1-c^2} + (\pi-\arccos c)(1+2c^2)}{3\pi}$$

One can observe very easily that (Daniely et al., 2016; Yang & Schoenholz, 2017)

**Proposition E.11.** *For each $\alpha > -1/2$, $\mathrm{J}_\alpha(c)$ is an increasing and convex function on $c \in [0,1]$, and is continous on $c \in [0,1]$ and smooth on $c \in (0,1)$. $\mathrm{J}_\alpha(1) = 1$, $\mathrm{J}_\alpha(0) = \frac{1}{2\sqrt{\pi}}\frac{\Gamma(\frac{\alpha}{2}+\frac{1}{2})^2}{\Gamma(\alpha+\frac{1}{2})}$, and $\mathrm{J}_\alpha(-1) = 0$.*

Yang & Schoenholz (2017) also showed the following fixed point structure

**Theorem E.12.** *For $\alpha \in [1/2, 1)$, $\mathrm{J}_\alpha(c) = c$ has two solutions: an unstable solution at 1 ("unstable" meaning $\mathrm{J}'_\alpha(1) > 1$) and a stable solution in $c^* \in (0,1)$ ("stable" meaning $\mathrm{J}'_\alpha(c^*) < 1$).*

The $\alpha$-ReLUs satisfy very interesting relations amongst themselves. For example,

**Lemma E.13.** *Suppose $\alpha > 1$. Then*

$$J_\alpha(\theta) = \cos\theta J_{\alpha-1}(\theta) + (\alpha-1)^2(2\alpha-1)^{-1}(2\alpha-3)^{-1}\sin^2\theta J_{\alpha-2}(\theta)$$

$$\mathrm{J}_\alpha(c) = c\mathrm{J}_{\alpha-1}(c) + (\alpha-1)^2(2\alpha-1)^{-1}(2\alpha-3)^{-1}(1-c^2)\mathrm{J}_{\alpha-2}(c)$$

In additional, surprisingly, one can use differentiation to go from $\alpha$ to $\alpha+1$ *and* from $\alpha$ to $\alpha-1$!

**Proposition E.14** (Yang & Schoenholz (2017))**.** *Suppose $\alpha > 1/2$. Then*

$$J'_\alpha(\theta) = -\alpha^2(2\alpha-1)^{-1}J_{\alpha-1}(\theta)\sin\theta$$

$$\mathrm{J}'_\alpha(c) = \alpha^2(2\alpha-1)^{-1}\mathrm{J}_{\alpha-1}(c)$$

*so that*

$$\mathrm{J}_{\alpha-n}(c) = \left[\prod_{\beta=\alpha-n+1}^{\alpha}\beta^{-2}(2\beta-1)\right](\partial/\partial c)^n\mathrm{J}_\alpha(c)$$

We have the following from Cho & Saul (2009)

**Proposition E.15** (Cho & Saul (2009)). *For all $\alpha \geq 0$ and integer $n \geq 1$*

$$J_{n+\alpha}(c) = \frac{c_\alpha}{c_{n+\alpha}}(1-c^2)^{n+\alpha+\frac{1}{2}}(\partial/\partial c)^n(J_\alpha(c)/(1-c^2)^{\alpha+\frac{1}{2}})$$

$$= \left[\prod_{\beta=\alpha}^{\alpha+n-1}(2\beta+1)\right]^{-1}(1-c^2)^{n+\alpha+\frac{1}{2}}(\partial/\partial c)^n(J_\alpha(c)/(1-c^2)^{\alpha+\frac{1}{2}})$$

This implies in particular that we can obtain $J'_\alpha$ from $J_\alpha$ and $J_{\alpha+1}$.

**Proposition E.16.** *For all $\alpha \geq 0$,*

$$J'_\alpha(c) = (2\alpha+1)(1-c^2)^{-1}(J_{1+\alpha}(c) - cJ_\alpha(c)))$$

*Proof.*

$$J_{1+\alpha}(c) = \frac{c_\alpha}{c_{1+\alpha}}(1-c^2)^{1+\alpha+\frac{1}{2}}(\partial/\partial c)(J_\alpha(c)/(1-c^2)^{\alpha+\frac{1}{2}})$$

$$= (2\alpha+1)^{-1}(1-c^2)^{\alpha+3/2}(J'_\alpha(c)/(1-c^2)^{\alpha+1/2} + 2c(\alpha+1/2)J_\alpha(c)/(1-c^2)^{\alpha+3/2})$$

$$= (2\alpha+1)^{-1}J'_\alpha(c)(1-c^2) + cJ_\alpha(c)$$

$$J'_\alpha(c) = (2\alpha+1)(1-c^2)^{-1}(J_{1+\alpha}(c) - cJ_\alpha(c))$$

$\square$

Note that we can also obtain this via Lemma E.13 and Proposition E.14.

### E.1.2 POSITIVE-HOMOGENEOUS FUNCTIONS IN 1 DIMENSION

Suppose for some $\alpha \in \mathbb{R}$, $\phi : \mathbb{R} \to \mathbb{R}$ is degree $\alpha$ positive-homogeneous, i.e. $\phi(rx) = r^\alpha\phi(x)$ for any $x \in R$, $r > 0$. The following simple lemma says that we can always express $\phi$ as linear combination of powers of $\alpha$-ReLUs.

**Proposition E.17.** *Any degree $\alpha$ positive-homogeneous function $\phi : \mathbb{R} \to \mathbb{R}$ with $\phi(0) = 0$ can be written as $x \mapsto a\rho_\alpha(x) - b\rho_\alpha(-x)$.*

*Proof.* Take $a = \phi(1)$ and $b = \phi(-1)$. Then positive-homogeneity determines the value of $\phi$ on $\mathbb{R} \setminus \{0\}$ and it coincides with $x \mapsto a\rho_\alpha(x) - b\rho_\alpha(-x)$. $\square$

As a result we can express the V and W transforms of any positive-homogeneous function in terms of those of $\alpha$-ReLUs.

**Proposition E.18.** *Suppose $\phi : \mathbb{R} \to \mathbb{R}$ is degree $\alpha$ positive-homogeneous. By Proposition E.17, $\phi$ restricted to $\mathbb{R} \setminus \{0\}$ can be written as $x \mapsto a\rho_\alpha(x) - b\rho_\alpha(-x)$ for some $a$ and $b$. Then for any PSD $2 \times 2$ matrix $M$,*

$$V_\phi(M)_{11} = (a^2+b^2)V_{\rho_\alpha}(M)_{11} = c_\alpha(a^2+b^2)M_{11}^\alpha$$

$$V_\phi(M)_{22} = (a^2+b^2)V_{\rho_\alpha}(M)_{22} = c_\alpha(a^2+b^2)M_{22}^\alpha$$

$$V_\phi(M)_{12} = V_\phi(M)_{21} = (a^2+b^2)V_{\rho_\alpha}(M)_{12} - 2abV_{\rho_\alpha}(M')_{12}$$

$$= c_\alpha(M_{11}M_{22})^{\alpha/2}((a^2+b^2)J_\alpha(M_{12}/\sqrt{M_{11}M_{22}}) - 2abJ_\alpha(-M_{12}/\sqrt{M_{11}M_{22}}))$$

*where $M' := \begin{pmatrix} 1 & 0 \\ 0 & -1 \end{pmatrix} M \begin{pmatrix} 1 & 0 \\ 0 & -1 \end{pmatrix}$.*

*Proof.* We directly compute, using the expansion of $\phi$ into $\rho_\alpha$s:

$$V_\phi(M)_{11} = \mathbb{E}[\phi(x)^2 : x \sim \mathcal{N}(0, M_{11})]$$

$$= \mathbb{E}[a^2\rho_\alpha(x)^2 + b^2\rho_\alpha(-x)^2 + 2ab\rho_\alpha(x)\rho_\alpha(-x)]$$

$$= a^2\,\mathbb{E}[\rho_\alpha(x)^2] + b^2\,\mathbb{E}[\rho_\alpha(-x)^2]$$

$$= (a^2+b^2)\,\mathbb{E}[\rho_\alpha(x)^2 : x \sim \mathcal{N}(0, M_{11})] \tag{28}$$

$$= c_\alpha(a^2+b^2)M_{11}^\alpha$$

where in Eq. (28) we used negation symmetry of centered Gaussians. The case of $\mathrm{V}_\phi(M)_{22}$ is similar.

$$
\begin{aligned}
\mathrm{V}_\phi(M)_{12} &= \mathbb{E}[\phi(x)\phi(y) : (x,y) \sim \mathcal{N}(0, M)] \\
&= \mathbb{E}[a^2 \rho_\alpha(x)\rho_\alpha(y) + b^2 \rho_\alpha(-x)\rho_\alpha(-y) - ab\rho_\alpha(x)\rho_\alpha(-y) - ab\rho_\alpha(-x)\rho_\alpha(y)] \\
&= (a^2 + b^2)\mathrm{V}_{\rho_\alpha}(M)_{12} - 2ab\mathrm{V}_{\rho_\alpha}(M')_{12} \\
&= \mathsf{c}_\alpha(M_{11}M_{22})^{\alpha/2}((a^2 + b^2)\mathrm{J}_\alpha(M_{12}/\sqrt{M_{11}M_{22}}) - 2ab\mathrm{J}_\alpha(-M_{12}/\sqrt{M_{11}M_{22}}))
\end{aligned}
$$

where in the last equation we have applied Proposition E.10. $\qquad\square$

This then easily generalizes to PSD matrices of arbitrary dimension:

**Corollary E.19.** *Suppose $\phi : \mathbb{R} \to \mathbb{R}$ is degree $\alpha$ positive-homogeneous. By Proposition E.17, $\phi$ restricted to $\mathbb{R} \setminus \{0\}$ can be written as $x \mapsto a\rho_\alpha(x) - b\rho_\alpha(-x)$ for some $a$ and $b$. Let $\Sigma \in \mathcal{S}_B$. Then*

$$
\mathrm{V}_\phi(\Sigma) = \mathsf{c}_\alpha D^{\alpha/2} \mathrm{J}_\phi(D^{-1/2}\Sigma D^{-1/2})D^{\alpha/2}
$$

*where $\mathrm{J}_\phi$ is defined below and is applied entrywise. Explicitly, this means that for all $i$,*

$$
\mathrm{V}_\phi(\Sigma)_{ii} = \mathsf{c}_\alpha \Sigma_{ii}^\alpha \mathrm{J}_\phi(1)
$$
$$
\mathrm{V}_\phi(\Sigma)_{ij} = \mathsf{c}_\alpha \mathrm{J}_\phi(\Sigma_{ij}/\sqrt{\Sigma_{ii}\Sigma_{jj}})\Sigma_{ii}^{\alpha/2}\Sigma_{jj}^{\alpha/2}
$$

**Definition E.20.** Suppose $\phi : \mathbb{R} \to \mathbb{R}$ is degree $\alpha$ positive-homogeneous. By Proposition E.17, $\phi$ restricted to $\mathbb{R} \setminus \{0\}$ can be written as $x \mapsto a\rho_\alpha(x) - b\rho_\alpha(-x)$ for some $a$ and $b$. Define $\mathrm{J}_\phi(c) := (a^2 + b^2)\mathrm{J}_\alpha(c) - 2ab\mathrm{J}_\alpha(-c)$.

Let us immediately make the following easy but important observations.

**Proposition E.21.** $\mathrm{J}_\phi(1) = a^2 + b^2$ *and* $\mathrm{J}_\phi(0) = (a^2 + b^2 - 2ab)\mathrm{J}_\alpha(0) = (a - b)^2 \frac{1}{2\sqrt{\pi}}\frac{\Gamma(\frac{\alpha}{2} + \frac{1}{2})^2}{\Gamma(\alpha + \frac{1}{2})}$,

*Proof.* Use the fact that $\mathrm{J}_\alpha(-1) = 0$ and $\mathrm{J}_\alpha(0) = \frac{1}{2\sqrt{\pi}}\frac{\Gamma(\frac{\alpha}{2} + \frac{1}{2})^2}{\Gamma(\alpha + \frac{1}{2})}$ by Proposition E.11. $\qquad\square$

As a sanity check, we can easily compute that $\mathrm{J}_{\mathrm{id}}(c) = 2\mathrm{J}_1(c) - 2\mathrm{J}_1(-c) = 2c$ because $\mathrm{id}(x) = \mathrm{relu}(x) - \mathrm{relu}(-x)$. By Corollary E.19 and $\mathsf{c}_1 = \frac{1}{2}$, this recovers the obvious fact that $\mathrm{V}_{(\mathrm{id})}(\Sigma) = \Sigma$.

We record the partial derivatives of $\mathrm{V}_\phi$.

**Proposition E.22.** *Let $\phi$ be positive homogeneous of degree $\alpha$. Then for all $i$ with $\Sigma_{ii} \neq 0$,*

$$
\frac{\partial \mathrm{V}_\phi(\Sigma)_{ii}}{\partial \Sigma_{ii}} = \mathsf{c}_\alpha \alpha \Sigma_{ii}^{\alpha-1}\mathrm{J}_\phi(1)
$$

*For all $i \neq j$ with $\Sigma_{ii}, \Sigma_{jj} \neq 0$,*

$$
\frac{\partial \mathrm{V}_\phi(\Sigma)_{ii}}{\partial \Sigma_{ij}} = 0
$$
$$
\frac{\partial \mathrm{V}_\phi(\Sigma)_{ij}}{\partial \Sigma_{ij}} = \mathsf{c}_\alpha \Sigma_{ii}^{(\alpha-1)/2}\Sigma_{jj}^{(\alpha-1)/2}\mathrm{J}'_\phi(c_{ij})
$$
$$
\frac{\partial \mathrm{V}_\phi(\Sigma)_{ij}}{\partial \Sigma_{ii}} = \frac{1}{2}\mathsf{c}_\alpha \Sigma_{ii}^{\frac{1}{2}(\alpha-2)}\Sigma_{jj}^{\frac{1}{2}\alpha}(\alpha\mathrm{J}_\phi(c_{ij}) - c_{ij}\mathrm{J}'_\phi(c_{ij}))
$$

*where $c_{ij} = \Sigma_{ij}/\sqrt{\Sigma_{ii}\Sigma_{jj}}$ and $\mathrm{J}'_\phi$ denotes its derivative.*

**Proposition E.23.** *If $\phi(c) = a\rho_\alpha(c) - b\rho_\alpha(-c)$ with $\alpha > 1/2$ on $\mathbb{R} \setminus \{0\}$, then*

$$
\begin{aligned}
\mathrm{J}'_\phi(c) &= (2\alpha - 1)^{-1}\mathrm{J}_{\phi'}(c) \\
&= (2\alpha + 1)(1 - c^2)^{-1}((a^2 + b^2)\mathrm{J}_{\alpha+1}(c) + 2ab\mathrm{J}_{\alpha+1}(-c)) - c\mathrm{J}_\phi(c) \\
\mathrm{J}'_\phi(0) &= (a + b)^2 \frac{1}{\sqrt{\pi}}\frac{\Gamma(\frac{\alpha}{2} + 1)^2}{\Gamma(\alpha + \frac{1}{2})}
\end{aligned}
$$

*Proof.* We have $\phi'(c) = a\alpha\rho_{\alpha-1}(c) + b\alpha\rho_{\alpha-1}(-c)$. On the other hand, $\mathrm{J}'_\phi = (a^2 + b^2)\mathrm{J}'_\alpha(c) + 2ab\mathrm{J}'_\alpha(-c)$ which by Proposition E.14 is $\alpha^2(2\alpha - 1)^{-1}((a^2 + b^2)\mathrm{J}_{\alpha-1}(c) + 2ab\mathrm{J}_{\alpha-1}(-c)) = (2\alpha - 1)^{-1}\mathrm{J}_{\phi'}(c)$. This proves the first equation.

With Proposition E.16,

$$\begin{aligned}
\mathrm{J}'_\phi(c) &= (a^2 + b^2)\mathrm{J}'_\alpha(c) + 2ab\mathrm{J}'_\alpha(-c) \\
&= (a^2 + b^2)(2\alpha + 1)(1 - c^2)^{-1}(\mathrm{J}_{\alpha+1}(c) - c\mathrm{J}_\alpha(c)) \\
&\quad + 2ab(2\alpha + 1)(1 - c^2)^{-1}(\mathrm{J}_{\alpha+1}(-c) + c\mathrm{J}_\alpha(-c)) \\
&= (2\alpha + 1)(1 - c^2)^{-1}((a^2 + b^2)\mathrm{J}_{\alpha+1}(c) + 2ab\mathrm{J}_{\alpha+1}(-c)) - c\mathrm{J}_\phi(c)
\end{aligned}$$

This gives the second equation.

Expanding $\mathrm{J}_{\alpha+1}(0)$ With Proposition E.11, we get

$$\begin{aligned}
\mathrm{J}'_\phi(0) &= (2\alpha + 1)((a^2 + b^2)\mathrm{J}_{\alpha+1}(0) + 2ab\mathrm{J}_{\alpha+1}(0)) \\
&= (2\alpha + 1)(a + b)^2 \frac{1}{2\sqrt{\pi}} \frac{\Gamma(\frac{\alpha}{2} + 1)^2}{\Gamma(\alpha + \frac{3}{2})}
\end{aligned}$$

Unpacking the definition of $\Gamma(\alpha + \frac{3}{2})$ then yields the third equation. $\qquad\square$

In general, we can factor diagonal matrices out of $\mathrm{V}_\phi$.

**Proposition E.24.** *For any $\Sigma \in \mathcal{S}_B$, $D$ any diagonal matrix, and $\phi$ positive-homogeneous with degree $\alpha$,*

$$\mathrm{V}_\phi(D\Sigma D) = D^\alpha \mathrm{V}_\phi(\Sigma)D^\alpha$$

*Proof.* For any $i$,

$$\begin{aligned}
\mathrm{V}_\phi(D\Sigma D)_{ii} &= \mathbb{E}[\phi(x)^2 : x \sim \mathcal{N}(0, D_{ii}^2\Sigma_{ii})] \\
&= \mathbb{E}[\phi(D_{ii}x)^2 : x \sim \mathcal{N}(0, \Sigma_{ii})] \\
&= \mathbb{E}[D_{ii}^{2\alpha}\phi(x)^2 : x \sim \mathcal{N}(0, \Sigma_{ii})] \\
&= D_{ii}^{2\alpha}\mathrm{V}_\phi(\Sigma)_{ii}
\end{aligned}$$

For any $i \neq j$,

$$\begin{aligned}
\mathrm{V}_\phi(D\Sigma D)_{ij} &= \mathbb{E}[\phi(x)\phi(y) : (x, y) \sim \mathcal{N}(0, \begin{pmatrix} D_{ii}^2\Sigma_{ii} & D_{ii}D_{jj}\Sigma_{ij} \\ D_{ii}D_{jj}\Sigma_{ji} & D_{jj}^2\Sigma_{jj} \end{pmatrix})) \\
&= \mathbb{E}[\phi(D_{ii}x)\phi(D_{jj}y) : (x, y) \sim \mathcal{N}(0, \begin{pmatrix} \Sigma_{ii} & \Sigma_{ij} \\ \Sigma_{ji} & \Sigma_{jj} \end{pmatrix}))] \\
&= D_{ii}^\alpha D_{jj}^\alpha \mathbb{E}[\phi(x)\phi(y) : (x, y) \sim \mathcal{N}(0, \begin{pmatrix} \Sigma_{ii} & \Sigma_{ij} \\ \Sigma_{ji} & \Sigma_{jj} \end{pmatrix}))] \\
&= D_{ii}^\alpha D_{jj}^\alpha \mathrm{V}_\phi(\Sigma)_{ij}
\end{aligned}$$

$\qquad\square$

## E.2 FOURIER METHOD

The Laplace method crucially used the fact that we can pull out the norm factor $\|Gh\|$ out of $\phi$, so that we can apply Schwinger parametrization. For general $\phi$ this is not possible, but we can apply

some wishful thinking and proceed as follows

$$V_{\mathcal{B}_\phi}(\Sigma) = \mathbb{E}[\phi(\sqrt{B}Gh/\|Gh\|)^{\otimes 2} : h \sim \mathcal{N}(0, \Sigma)]$$

$$= \mathbb{E}[\phi(Gh/r)^{\otimes 2} \int_0^\infty d(r^2)\delta(r^2 - \|Gh\|^2/B) : h \sim \mathcal{N}(0, \Sigma)]$$

$$= \mathbb{E}[\phi(Gh/r)^{\otimes 2} \int_0^\infty d(r^2) \int_{-\infty}^\infty \frac{d\lambda}{2\pi} e^{i\lambda(r^2 - \|Gh\|^2/B)} : h \sim \mathcal{N}(0, \Sigma)]$$

$$\text{``} = \text{''} \int_{\mathbb{R}^B} \frac{dh}{\sqrt{\det(2\pi\Sigma)}} e^{\frac{-1}{2}h^T\Sigma^{-1}h} \phi(Gh/r)^{\otimes 2} \int_0^\infty d(r^2) \int_{-\infty}^\infty \frac{d\lambda}{2\pi} e^{i\lambda(r^2 - \|Gh\|^2/B)}$$

$$\text{``} = \text{''} \int_0^\infty d(r^2) \int_{-\infty}^\infty \frac{d\lambda}{2\pi} \int_{\mathbb{R}^B} \frac{dh}{\sqrt{\det(2\pi\Sigma)}} e^{\frac{-1}{2}h^T\Sigma^{-1}h} \phi(Gh/r)^{\otimes 2} e^{i\lambda(r^2 - \|Gh\|^2/B)}$$

$$\tag{29}$$

$$= \frac{1}{\sqrt{\det(2\pi\Sigma)}} \int_0^\infty d(r^2) \int_{-\infty}^\infty \frac{d\lambda}{2\pi} e^{i\lambda r^2} \int_{\mathbb{R}^B} dh \phi(Gh/r)^{\otimes 2} e^{\frac{-1}{2}(h^T\Sigma^{-1}h + 2i\lambda\|Gh\|^2/B)}$$

$$\tag{30}$$

$$= \frac{1}{\sqrt{\det(2\pi\Sigma)}} \int_0^\infty d(r^2) \int_{-\infty}^\infty \frac{d\lambda}{2\pi} e^{i\lambda r^2} \int_{\mathbb{R}^B} dh \phi(Gh/r)^{\otimes 2} e^{\frac{-1}{2}(h^T(\Sigma^{-1} + 2i\lambda G/B)h)}$$

$$\text{``} = \text{''} \frac{1}{\sqrt{\det(2\pi\Sigma)}} \int_0^\infty d(r^2) \int_{-\infty}^\infty \frac{d\lambda}{2\pi} e^{i\lambda r^2}$$

$$\sqrt{\det(2\pi(\Sigma^{-1} + 2i\lambda G/B)^{-1})} \mathbb{E}[\phi(Gh/r)^{\otimes 2} : h \sim \mathcal{N}(0, \Sigma^{-1} + 2i\lambda G/B)]$$

$$= \int_0^\infty d(r^2) \int_{-\infty}^\infty \frac{d\lambda}{2\pi} e^{i\lambda r^2} \det(I + 2i\lambda G\Sigma/B)^{-1/2} \mathbb{E}_{h\sim\mathcal{N}(0, G(\Sigma^{-1} + 2i\lambda G/B)Gr^{-2}))} \phi(h)^{\otimes 2}$$

$$\tag{31}$$

$$= \int_0^\infty d(r^2) \int_{-\infty}^\infty \frac{d\lambda}{2\pi} e^{i\lambda r^2} \det(I + 2i\lambda G\Sigma/B)^{-1/2} V_\phi(G(\Sigma^{-1} + 2i\lambda G/B)Gr^{-2})$$

Here all steps are legal except for possibly Eq. (29), Eq. (30), and Eq. (31). In Eq. (29), we "Fourier expanded" the delta function — this could possibly be justified as a principal value integral, but this cannot work in combination with Eq. (30) where we have switched the order of integration, integrating over $h$ first. Fubini's theorem would not apply here because $e^{i\lambda(r^2 - \|Gh\|^2/B)}$ has norm 1 and is not integrable. Finally, in Eq. (31), we need to extend the definition of Gaussian to complex covariance matrices, via complex Gaussian integration. This last point is no problem, and we will just define

$$\mathbb{E}_{h\sim\mathcal{N}(0,\Sigma)} f(h) := \det(2\pi\Sigma)^{-1/2} \int_{\mathbb{R}^B} dh\, f(h) e^{-\frac{1}{2}h^T\Sigma^{-1}h}$$

for general complex $\Sigma$, whenever $\Sigma$ is nonsingular and this integral is exists, and

$$\mathbb{E}_{h\sim\mathcal{N}(0,G\Sigma G)} f(h) := \det(2\pi(\mathbb{e}^T\Sigma\mathbb{e}))^{-1/2} \int_{\mathbb{R}^{B-1}} dz\, f(\mathbb{e}z) e^{-\frac{1}{2}z^T(\mathbb{e}^T\Sigma\mathbb{e})^{-1}z}$$

if $\mathbb{e}^T\Sigma\mathbb{e}$ is nonsingular (as in the case above). The definition of $V_\phi$ stays the same given the above extension of the definition of Gaussian expectation.

Nevertheless, the above derivation is not correct mathematically due to the other points. However, its end result can be justified rigorously by carefully expressing the delta function as the limit of mollifiers.

**Theorem E.25.** *Let $\Sigma$ be a positive-definite $D \times D$ matrix. Let $\phi : \mathbb{R} \to \mathbb{R}$. Suppose for any* b > a > 0,

1. *$\int_a^b d(r^2)|\phi(z_a/r)\phi(z_b/r)|$ exists and is finite for any $z_a, z_b \in \mathbb{R}$ (i.e. $\phi(z_a/r)\phi(z_b/r)$ is locally integrable in $r$).*

2. *there exists an $\epsilon > 0$ such that any $\gamma \in [-\epsilon, \epsilon]$ and for each $a, b \in [D]$,*

$$\int_{\|z\|^2 \in [\mathsf{a},\mathsf{b}]} \mathrm{d}z \; e^{-\frac{1}{2}z^T\Sigma^{-1}z} |\phi(z_a/\sqrt{\|z\|^2 + \gamma})\phi(z_b/\sqrt{\|z\|^2 + \gamma})|$$

*exists and is uniformly bounded by some number possibly depending on $\mathsf{a}$ and $\mathsf{b}$.*

3. *for each $a, b \in [D]$, $\int_{\mathbb{R}^D} \mathrm{d}z \int_{\mathsf{a}}^{\mathsf{b}} \mathrm{d}(r^2) e^{-\frac{1}{2}z^T\Sigma^{-1}z} |\phi(z_a/r)\phi(z_b/r)|$ exists and is finite.*

4. *$\int_{-\infty}^{\infty} \mathrm{d}\lambda \left| \det(I + 2i\lambda\Sigma)^{-1/2} \mathrm{V}_\phi((\Sigma^{-1} + 2i\lambda I)^{-1}r^{-2}) \right|$ exists and is finite for each $r > 0$.*

*If $e^{-\frac{1}{2}z^T\Sigma^{-1}z}\phi(z/\|z\|)^{\otimes 2}$ is integrable over $z \in \mathbb{R}^D$, then*

$$(2\pi)^{-D/2} \det\Sigma^{-1/2} \int \mathrm{d}z \; e^{-\frac{1}{2}z^T\Sigma^{-1}z}\phi(z/\|z\|)^{\otimes 2}$$

$$= \frac{1}{2\pi} \lim_{\mathsf{a}\searrow 0,\mathsf{b}\nearrow\infty} \int_{\mathsf{a}}^{\mathsf{b}} \mathrm{d}s \int_{-\infty}^{\infty} \mathrm{d}\lambda \; e^{i\lambda s} \det(I + 2i\lambda\Sigma)^{-1/2}\mathrm{V}_\phi((\Sigma^{-1} + 2i\lambda I)^{-1}s^{-1}).$$

*Similarly,*

$$(2\pi)^{-D/2} \det\Sigma^{-1/2} \int \mathrm{d}z \; e^{-\frac{1}{2}z^T\Sigma^{-1}z}\phi(\sqrt{D}z/\|z\|)^{\otimes 2}$$

$$= \frac{1}{2\pi} \lim_{\mathsf{a}\searrow 0,\mathsf{b}\nearrow\infty} \int_{\mathsf{a}}^{\mathsf{b}} \mathrm{d}s \int_{-\infty}^{\infty} \mathrm{d}\lambda \; e^{i\lambda s} \det(I + 2i\lambda\Sigma)^{-1/2}\mathrm{V}_\phi((\Sigma^{-1} + 2i\lambda I/D)^{-1}s^{-1}).$$

*If $G$ is the mean-centering projection matrix and $\Sigma^G = G\Sigma G$, then by the same reasoning as in Thm E.5,*

$$\mathrm{V}_{\mathcal{B}_\phi}(\Sigma) = (2\pi)^{-D/2} \det\Sigma^{-1/2} \int \mathrm{d}z \; e^{-\frac{1}{2}z^T\Sigma^{-1}z}\phi(\sqrt{D}Gz/\|Gz\|)^{\otimes 2}$$

$$= \frac{1}{2\pi} \lim_{\mathsf{a}\searrow 0,\mathsf{b}\nearrow\infty} \int_{\mathsf{a}}^{\mathsf{b}} \mathrm{d}s \int_{-\infty}^{\infty} \mathrm{d}\lambda \; e^{i\lambda s} \det(I + 2i\lambda\Sigma^G/D)^{-1/2}\mathrm{V}_\phi(\Sigma^G(I + 2i\lambda\Sigma^G/D)^{-1}s^{-1}).$$

$$(32)$$

Note that assumption (1) is satisfied if $\phi$ is continuous; assumption (4) is satisfied if $D \geq 3$ and for all $\Pi$, $\|\mathrm{V}_\phi(\Pi)\| \leq \|\Pi\|^\alpha$ for some $\alpha \geq 0$; this latter condition, as well as assumptions (2) and (3), will be satisfied if $\phi$ is polynomially bounded. Thus the common coordinatewise nonlinearities ReLU, identity, tanh, etc all satisfy these assumptions.

Warning: in general, we cannot swap the order of integration as $\int_{-\infty}^{\infty} \mathrm{d}\lambda \int_{-\infty}^{\infty} \mathrm{d}s$. For example, if $\phi = \mathrm{id}$, then

$$(2\pi)^{-D/2} \det\Sigma^{-1/2} \int \mathrm{d}z \; e^{-\frac{1}{2}z^T\Sigma^{-1}z}\phi(z/\|z\|)^{\otimes 2}$$

$$= \frac{1}{2\pi} \lim_{\mathsf{a}\searrow 0,\mathsf{b}\nearrow\infty} \int_{\mathsf{a}}^{\mathsf{b}} \mathrm{d}s \int_{-\infty}^{\infty} \mathrm{d}\lambda \; e^{i\lambda s} \det(I + 2i\lambda\Sigma)^{-1/2}(\Sigma^{-1} + 2i\lambda I)^{-1}s^{-1}$$

$$\neq \frac{1}{2\pi} \int_{-\infty}^{\infty} \mathrm{d}\lambda \int_{0}^{\infty} \mathrm{d}s \; e^{i\lambda s} \det(I + 2i\lambda\Sigma)^{-1/2}(\Sigma^{-1} + 2i\lambda I)^{-1}s^{-1}$$

because the $s$-integral in the latter diverges (in a neighborhood of 0).

*Proof.* We will only prove the first equation; the others follow similarly.

By dominated convergence,

$$\int \mathrm{d}z e^{-\frac{1}{2}z^T\Sigma^{-1}z}\phi(z/\|z\|)^{\otimes 2}$$

$$= \lim_{\mathsf{a}\searrow 0,\mathsf{b}\nearrow\infty} \int \mathrm{d}z e^{-\frac{1}{2}z^T\Sigma^{-1}z}\phi(z/\|z\|)^{\otimes 2}\mathbb{I}(\|z\|^2 \in [\mathsf{a},\mathsf{b}]).$$

Let $\eta : \mathbb{R} \to \mathbb{R}$ be a nonnegative bump function (i.e. compactly supported and smooth) with support $[-1, 1]$ and integral 1 such that $\eta'(0) = \eta''(0) = 0$. Then its Fourier transform $\hat{\eta}(t)$ decays like $O(t^{-2})$. Furthermore, $\eta_\epsilon(x) := \epsilon^{-1}\eta(x/\epsilon)$ is a mollifier, i.e. for all $f \in L^1(\mathbb{R})$, $f * \eta_\epsilon \to f$ in $L^1$ and pointwise almost everywhere.

Now, we will show that

$$
\int dz e^{-\frac{1}{2}z^T \Sigma^{-1} z} \phi(z_a/r)\phi(z_b/r)\mathbb{I}(\|z\|^2 \in [\mathsf{a}, \mathsf{b}])
$$

$$
= \lim_{\epsilon \searrow 0} \int dz e^{-\frac{1}{2}z^T \Sigma^{-1} z} \int_{\mathsf{a}}^{\mathsf{b}} d(r^2)\phi(z_a/r)\phi(z_b/r)\eta_\epsilon(\|z\|^2 - r^2)
$$

by dominated convergence. Pointwise convergence is immediate, because

$$
\int_{\mathsf{a}}^{\mathsf{b}} d(r^2)\phi(z_a/r)\phi(z_b/r)\eta_\epsilon(r^2 - \|z\|^2) = \int_{-\infty}^{\infty} d(r^2)\phi(z_a/r)\phi(z_b/r)\mathbb{I}(r^2 \in [\mathsf{a}, \mathsf{b}])\eta_\epsilon(\|z\|^2 - r^2)
$$

$$
= [(r^2 \mapsto \phi(z_a/r)\phi(z_b/r)\mathbb{I}(r^2 \in [\mathsf{a}, \mathsf{b}])) * \eta_\epsilon](\|z\|^2)
$$

$$
\to \phi(z_a/\|z\|)\phi(z_b/\|z\|)\mathbb{I}(\|z\|^2 \in [\mathsf{a}, \mathsf{b}])
$$

as $\epsilon \to 0$ (where we used assumption (1) that $\phi(z_a/r)\phi(z_b/r)\mathbb{I}(r^2 \in [\mathsf{a}, \mathsf{b}])$ is $L^1$).

Finally, we construct a dominating integrable function. Observe

$$
\int dz e^{-\frac{1}{2}z^T \Sigma^{-1} z} \int_{\mathsf{a}}^{\mathsf{b}} d(r^2)|\phi(z_a/r)\phi(z_b/r)|\eta_\epsilon(\|z\|^2 - r^2)
$$

$$
= \int dz e^{-\frac{1}{2}z^T \Sigma^{-1} z} \int_{\mathsf{a}}^{\mathsf{b}} d(r^2)|\phi(z_a/r)\phi(z_b/r)|\eta_\epsilon(\|z\|^2 - r^2)
$$

$$
= \int dz e^{-\frac{1}{2}z^T \Sigma^{-1} z} \int_{\mathsf{a}}^{\mathsf{b}} d(\|z\|^2 + \gamma)|\phi(z_a/\sqrt{\|z\|^2 + \gamma})\phi(z_b/\sqrt{\|z\|^2 + \gamma})|\eta_\epsilon(-\gamma)
$$

$$
= \int dz e^{-\frac{1}{2}z^T \Sigma^{-1} z} \int_{\mathsf{a}-\|z\|^2}^{\mathsf{b}-\|z\|^2} d\gamma\,|\phi(z_a/\sqrt{\|z\|^2 + \gamma})\phi(z_b/\sqrt{\|z\|^2 + \gamma})|\eta_\epsilon(-\gamma)
$$

$$
= \int_{\|z\|^2 \in [\mathsf{a}-\epsilon, \mathsf{b}+\epsilon]} dz e^{-\frac{1}{2}z^T \Sigma^{-1} z} \int_{\mathsf{a}-\|z\|^2}^{\mathsf{b}-\|z\|^2} d\gamma\,|\phi(z_a/\sqrt{\|z\|^2 + \gamma})\phi(z_b/\sqrt{\|z\|^2 + \gamma})|\eta_\epsilon(-\gamma)
$$

$$
\text{since } \operatorname{supp} \eta_\epsilon = [-\epsilon, \epsilon]
$$

$$
\leq \int_{\|z\|^2 \in [\mathsf{a}-\epsilon, \mathsf{b}+\epsilon]} dz e^{-\frac{1}{2}z^T \Sigma^{-1} z} \int_{-\epsilon}^{\epsilon} d\gamma\,|\phi(z_a/\sqrt{\|z\|^2 + \gamma})\phi(z_b/\sqrt{\|z\|^2 + \gamma})|\eta_\epsilon(-\gamma)
$$

$$
= \int_{-\epsilon}^{\epsilon} d\gamma\, \eta_\epsilon(-\gamma) \int_{\|z\|^2 \in [\mathsf{a}-\epsilon, \mathsf{b}+\epsilon]} dz e^{-\frac{1}{2}z^T \Sigma^{-1} z}|\phi(z_a/\sqrt{\|z\|^2 + \gamma})\phi(z_b/\sqrt{\|z\|^2 + \gamma})|
$$

For small enough $\epsilon$ then, this is integrable by assumption (2), and yields a dominating integrable function for our application of dominated convergence.

In summary, we have just proven that

$$
\int dz e^{-\frac{1}{2}z^T \Sigma^{-1} z} \phi(z/\|z\|)^{\otimes 2}
$$

$$
= \lim_{\mathsf{a}\searrow 0, \mathsf{b}\nearrow \infty} \lim_{\epsilon \searrow 0} \int dz e^{-\frac{1}{2}z^T \Sigma^{-1} z} \int_{\mathsf{a}}^{\mathsf{b}} d(r^2)\phi(z_a/r)\phi(z_b/r)\eta_\epsilon(\|z\|^2 - r^2)
$$

Now,

$$\int \mathrm{d}z e^{-\frac{1}{2}z^T\Sigma^{-1}z} \int_a^b \mathrm{d}(r^2)\phi(z_a/r)\phi(z_b/r)\eta_\epsilon(\|z\|^2 - r^2)$$

$$= \int \mathrm{d}z e^{-\frac{1}{2}z^T\Sigma^{-1}z} \int_a^b \mathrm{d}(r^2)\phi(z_a/r)\phi(z_b/r)\frac{1}{2\pi}\int_{-\infty}^\infty \mathrm{d}\lambda \, \hat{\eta}_\epsilon(\lambda)e^{-i\lambda(\|z\|^2-r^2)}$$

$$= \frac{1}{2\pi}\int \mathrm{d}z e^{-\frac{1}{2}z^T\Sigma^{-1}z} \int_a^b \mathrm{d}(r^2)\phi(z_a/r)\phi(z_b/r)\int_{-\infty}^\infty \mathrm{d}\lambda \, \hat{\eta}(\epsilon\lambda)e^{-i\lambda(\|z\|^2-r^2)}$$

Note that the absolute value of the integral is bounded above by

$$\frac{1}{2\pi}\int \mathrm{d}z e^{-\frac{1}{2}z^T\Sigma^{-1}z} \int_a^b \mathrm{d}(r^2)|\phi(z_a/r)\phi(z_b/r)|\int_{-\infty}^\infty \mathrm{d}\lambda \, |\hat{\eta}(\epsilon\lambda)e^{-i\lambda(\|z\|^2-r^2)}|$$

$$= \frac{1}{2\pi}\int \mathrm{d}z e^{-\frac{1}{2}z^T\Sigma^{-1}z} \int_a^b \mathrm{d}(r^2)|\phi(z_a/r)\phi(z_b/r)|\int_{-\infty}^\infty \mathrm{d}\lambda \, |\hat{\eta}(\epsilon\lambda)|$$

$$\leq C\int \mathrm{d}z e^{-\frac{1}{2}z^T\Sigma^{-1}z} \int_a^b \mathrm{d}(r^2)|\phi(z_a/r)\phi(z_b/r)|$$

for some $C$, by our construction of $\eta$ that $\hat{\eta}(t) = O(t^{-2})$ for large $|t|$. By assumption (3), this integral exists. Therefore we can apply the Fubini-Tonelli theorem and swap order of integration.

$$\frac{1}{2\pi}\int \mathrm{d}z e^{-\frac{1}{2}z^T\Sigma^{-1}z} \int_a^b \mathrm{d}(r^2)\phi(z_a/r)\phi(z_b/r)\int_{-\infty}^\infty \mathrm{d}\lambda \, \hat{\eta}(\epsilon\lambda)e^{-i\lambda(\|z\|^2-r^2)}$$

$$= \frac{1}{2\pi}\int_a^b \mathrm{d}(r^2)\int_{-\infty}^\infty \mathrm{d}\lambda \, \hat{\eta}(\epsilon\lambda)e^{-i\lambda(\|z\|^2-r^2)}\int \mathrm{d}z\phi(z_a/r)\phi(z_b/r)e^{-\frac{1}{2}z^T\Sigma^{-1}z}$$

$$= \frac{1}{2\pi}\int_a^b \mathrm{d}(r^2)\int_{-\infty}^\infty \mathrm{d}\lambda \, \hat{\eta}(\epsilon\lambda)e^{i\lambda r^2}\int \mathrm{d}z\phi(z_a/r)\phi(z_b/r)e^{-\frac{1}{2}z^T(\Sigma^{-1}+2i\lambda I)z}$$

$$= \frac{1}{2\pi}\int_a^b \mathrm{d}(r^2)r^D\int_{-\infty}^\infty \mathrm{d}\lambda \, \hat{\eta}(\epsilon\lambda)e^{i\lambda r^2}\int \mathrm{d}z\phi(z_a)\phi(z_b)e^{-\frac{1}{2}r^2 z^T(\Sigma^{-1}+2i\lambda I)z}$$

$$= (2\pi)^{D/2-1}\det\Sigma^{1/2}\int_a^b \mathrm{d}(r^2)\int_{-\infty}^\infty \mathrm{d}\lambda \, \hat{\eta}(\epsilon\lambda)e^{i\lambda r^2}\det(I + 2i\lambda\Sigma)^{-1/2}V_\phi((\Sigma^{-1} + 2i\lambda I)^{-1}r^{-2})$$

By assumption (4),

$$\int_a^b \mathrm{d}(r^2)\int_{-\infty}^\infty \mathrm{d}\lambda \left|\hat{\eta}(\epsilon\lambda)e^{i\lambda r^2}\det(I + 2i\lambda\Sigma)^{-1/2}V_\phi((\Sigma^{-1} + 2i\lambda I)^{-1}r^{-2})\right|$$

$$\leq C\int_a^b \mathrm{d}(r^2)\int_{-\infty}^\infty \mathrm{d}\lambda \left|\det(I + 2i\lambda\Sigma)^{-1/2}V_\phi((\Sigma^{-1} + 2i\lambda I)^{-1}r^{-2})\right|$$

is integrable (where we used the fact that $\hat{\eta}$ is bounded — which is true for all $\eta \in C_0^\infty(\mathbb{R})$), so by dominated convergence (applied to $\epsilon \searrow 0$),

$$\int \mathrm{d}z e^{-\frac{1}{2}z^T\Sigma^{-1}z}\phi(z/\|z\|)^{\otimes 2}$$

$$= \lim_{a\searrow 0, b\nearrow\infty}\lim_{\epsilon\searrow 0}\int \mathrm{d}z e^{-\frac{1}{2}z^T\Sigma^{-1}z}\int_a^b \mathrm{d}(r^2)\phi(z_a/r)\phi(z_b/r)\eta_\epsilon(\|z\|^2 - r^2)$$

$$= (2\pi)^{D/2-1}\det\Sigma^{1/2}\lim_{a\searrow 0, b\nearrow\infty}\int_a^b \mathrm{d}(r^2)\int_{-\infty}^\infty \mathrm{d}\lambda \, e^{i\lambda r^2}\det(I + 2i\lambda\Sigma)^{-1/2}V_\phi((\Sigma^{-1} + 2i\lambda I)^{-1}r^{-2})$$

where we used the fact that $\hat{\eta}(0) = \int_\mathbb{R}\eta(x)\,\mathrm{d}x = 1$ by construction. This gives us the desired result after putting back in some constants and changing $r^2 \mapsto s$. $\qquad\square$

### E.3 SPHERICAL INTEGRATION

As mentioned above, the scale invariance of batchnorm combined with the BSB1 fixed point assumption will naturally lead us to consider spherical integration. Here we review several basic facts.

**Definition E.26.** We define the spherical angular coordinates in $\mathbb{R}^B$, $(r, \theta_1, \ldots, \theta_{B-2}) \in [0, \infty) \times \mathsf{A} =: [0, \pi]^{B-3} \times [0, 2\pi]$, by

$$
\begin{aligned}
x_1 &= r \cos \theta_1 \\
x_2 &= r \sin \theta_1 \cos \theta_2 \\
x_3 &= r \sin \theta_1 \sin \theta_2 \cos \theta_3 \\
&\vdots \\
x_{B-2} &= r \sin \theta_1 \cdots \sin \theta_{B-3} \cos \theta_{B-2} \\
x_{B-1} &= r \sin \theta_1 \cdots \sin \theta_{B-3} \sin \theta_{B-2}.
\end{aligned}
$$

Then we set

$$
v := \begin{pmatrix}
\cos \theta_1 \\
\sin \theta_1 \cos \theta_2 \\
\sin \theta_1 \sin \theta_2 \cos \theta_3 \\
\vdots \\
\sin \theta_1 \cdots \sin \theta_{B-3} \cos \theta_{B-2} \\
\sin \theta_1 \cdots \sin \theta_{B-3} \sin \theta_{B-2}
\end{pmatrix}
$$

as the unit norm version of $x$. The integration element in this coordinate satisfies

$$
\mathrm{d}x_1 \ldots \mathrm{d}x_{B-1} = r^{B-2} \sin^{B-3} \theta_1 \sin^{B-4} \theta_2 \cdots \sin \theta_{B-3} \, \mathrm{d}r \, \mathrm{d}\theta_1 \cdots \mathrm{d}\theta_{B-2}.
$$

**Fact E.27.** *If $F : S^{B-2} \to \mathbb{R}$, then its expectation on the sphere can be written as*

$$
\mathop{\mathbb{E}}_{v \sim S^{B-2}} F(v) = \frac{1}{2} \pi^{\frac{1-B}{2}} \Gamma\left(\frac{B-1}{2}\right) \int_{\mathsf{A}} F(v) \sin^{B-3} \theta_1 \cdots \sin \theta_{B-3} \, \mathrm{d}\theta_1 \cdots \mathrm{d}\theta_{B-2}
$$

*where $\mathsf{A} = [0, \pi]^{B-3} \times [0, 2\pi]$ and $v$ are as in Defn E.26.*

The following integrals will be helpful for simplifying many expressions involving spherical integration using angular coordinates.

**Lemma E.28.** *For $j, k \geq 0$ and $0 \leq s \leq t \leq \pi/2$,*

$$
\int_s^t \sin^j \theta \cos^k \theta \, \mathrm{d}\theta = \frac{1}{2} \operatorname{Beta}\left(\cos^2 t, \cos^2 s; \frac{k+1}{2}, \frac{j+1}{2}\right).
$$

*By antisymmetry of $\cos$ with respect to $\theta \mapsto \pi - \theta$, if $\pi/2 \leq s \leq t \leq \pi$,*

$$
\int_s^t \sin^j \theta \cos^k \theta \, \mathrm{d}\theta = \frac{1}{2}(-1)^k \operatorname{Beta}\left(\cos^2 s, \cos^2 t; \frac{k+1}{2}, \frac{j+1}{2}\right).
$$

*Proof.* Set $x := \cos^2 \theta \implies \mathrm{d}x = -2 \cos \theta \sin \theta \, \mathrm{d}\theta$. So the integral in question is

$$
\frac{-1}{2} \int_{\cos^2 s}^{\cos^2 t} (1-x)^{\frac{j-1}{2}} x^{\frac{k-1}{2}} \, \mathrm{d}x
$$
$$
= \frac{1}{2}\left(\operatorname{Beta}(\cos^2 s; \frac{k+1}{2}, \frac{j+1}{2}) - \operatorname{Beta}(\cos^2 t; \frac{k+1}{2}, \frac{j+1}{2})\right)
$$

$\square$

As consequences,

**Lemma E.29.** *For $j, k \geq 0$,*

$$
\int_0^\pi \sin^j \theta \cos^k \theta \, \mathrm{d}\theta = \frac{1 + (-1)^k}{2} \operatorname{Beta}\left(\frac{j+1}{2}, \frac{k+1}{2}\right) = \frac{1 + (-1)^k}{2} \frac{\Gamma(\frac{1+j}{2})\Gamma(\frac{1+k}{2})}{\Gamma(\frac{2+j+k}{2})}.
$$

**Lemma E.30.**

$$\int_0^\pi \sin\theta_1 \, \mathrm{d}\theta_1 \int_0^\pi \sin^2\theta_2 \, \mathrm{d}\theta_2 \cdots \int_0^\pi \sin^k\theta_k \, \mathrm{d}\theta_k = \pi^{k/2}/\Gamma(\frac{k+2}{2}).$$

*Proof.* By Lemma E.29, $\int_0^\pi \sin^r\theta_r \, \mathrm{d}\theta_r = \mathrm{Beta}(\frac{r+1}{2}, \frac{1}{2}) = \frac{\Gamma(\frac{r+1}{2})\sqrt{\pi}}{\Gamma(\frac{r+2}{2})}$. Thus this product of integrals is equal to

$$\prod_{r=1}^k \sqrt{\pi}\frac{\Gamma(\frac{r+1}{2})}{\Gamma(\frac{r+2}{2})} = \pi^{k/2}\Gamma(1)/\Gamma(\frac{k+2}{2}) = \pi^{k/2}/\Gamma(\frac{k+2}{2}).$$

$\square$

**Lemma E.31.** *For* $B > 1, \mu > 0$, $\int_0^\infty r^B e^{-r^2/2\mu} \, \mathrm{d}r = 2^{\frac{B-1}{2}}\mu^{\frac{B+1}{2}}\Gamma(\frac{B+1}{2})$

*Proof.* Apply change of coordinates $z = r^2/2\mu$. $\square$

**Definition E.32.** Let $\mathcal{K}(v; \Sigma, \beta) := (\det \mathbb{e}^T\Sigma\mathbb{e})^{-1/2}(v^T(\mathbb{e}^T\Sigma\mathbb{e})^{-1}v)^{-\frac{B-1+\beta}{2}}$ and $\mathcal{K}(v; \Sigma) := \mathcal{K}(v; \Sigma, 0)$.

The following proposition allows us to express integrals involving batchnorm as expectations over the uniform spherical measure.

**Proposition E.33.** *If* $f : \mathbb{R}^{B-1} \to \mathbb{R}^{B'}$, $B > 1 - \beta$, *and* $\Sigma \in \mathbb{R}^{(B-1)\times(B-1)}$ *is nonsingular, then*

$$\mathop{\mathbb{E}}_{h\sim\mathcal{N}(0,\Sigma)} \|h\|^\beta f(h/\|h\|) = 2^{\beta/2}\mathrm{P}\left(\frac{B-1}{2}, \frac{\beta}{2}\right) \mathop{\mathbb{E}}_{v\sim S^{B-2}} f(v)(\det\Sigma)^{-1/2}(v^T\Sigma^{-1}v)^{-\frac{B-1+\beta}{2}}.$$

*If* $f : \mathbb{R}^B \to \mathbb{R}^{B'}$ *and* $\Sigma \in \mathbb{R}^{B\times B}$ *is such that* $\mathbb{e}^T\Sigma\mathbb{e}$ *is nonsingular (where* $\mathbb{e}$ *is as in Defn E.3) and* $G\Sigma G = \Sigma$,

$$\mathop{\mathbb{E}}_{h\sim\mathcal{N}(0,\Sigma)} \|Gh\|^\beta f(Gh/\|Gh\|) = 2^{\beta/2}\mathrm{P}\left(\frac{B-1}{2}, \frac{\beta}{2}\right) \mathop{\mathbb{E}}_{v\sim S^{B-2}} f(\mathbb{e}v)\mathcal{K}(v; \Sigma, \beta).$$

*In particular,*

$$\mathbb{E}[f(\mathcal{B}(h)) : h \sim \mathcal{N}(0, \Sigma)] = \mathop{\mathbb{E}}_{v\sim S^{B-2}} f(\mathbb{e}v)\mathcal{K}(v; \Sigma)$$

*Proof.* We will prove the second statement. The others follow trivially.

$$\mathbb{E}[\|Gh\|^{\beta}f(\mathcal{B}(h)) : h \sim \mathcal{N}(0,\Sigma)]$$

$$= \mathbb{E}[\|x\|^{\beta}f(\mathbb{e}x/\|x\|) : x \sim \mathcal{N}(0,\mathbb{e}^{T}\Sigma\mathbb{e})]$$

$$= (2\pi)^{\frac{1-B}{2}}(\det \mathbb{e}^{T}\Sigma\mathbb{e})^{-1/2}\int_{\mathbb{R}^{B-1}}\|x\|^{\beta}f(\mathbb{e}x/\|x\|)e^{-\frac{1}{2}x^{T}(\mathbb{e}^{T}\Sigma\mathbb{e})^{-1}x}\,\mathrm{d}x$$

$$= (2\pi)^{\frac{1-B}{2}}(\det \mathbb{e}^{T}\Sigma\mathbb{e})^{-1/2}\int_{A}f(\mathbb{e}v)\sin^{B-3}\theta_1\cdots\sin\theta_{B-3}\,\mathrm{d}r\,\mathrm{d}\theta_1\cdots\mathrm{d}\theta_{B-2}\times$$

$$\int_0^{\infty}r^{B-2+\beta}e^{-\frac{1}{2}r^2\cdot v^{T}(\mathbb{e}^{T}\Sigma\mathbb{e})^{-1}v}$$

(spherical coordinates Defn E.26)

$$= (2\pi)^{\frac{1-B}{2}}(\det \mathbb{e}^{T}\Sigma\mathbb{e})^{-1/2}\times 2^{\frac{B-3}{2}}\Gamma\left(\frac{B-1}{2}\right)\times\frac{2^{\frac{B-3+\beta}{2}}\Gamma(\frac{B-1+\beta}{2})}{2^{\frac{B-3}{2}}\Gamma\left(\frac{B-1}{2}\right)}$$

$$\times\int_{A}\frac{f(\mathbb{e}v)\sin^{B-3}\theta_1\cdots\sin\theta_{B-3}\,\mathrm{d}\theta_1\cdots\mathrm{d}\theta_{B-2}}{(v^{T}(\mathbb{e}^{T}\Sigma\mathbb{e})^{-1}v)^{\frac{B-1+\beta}{2}}}$$

$$= \frac{1}{2}\pi^{\frac{1-B}{2}}\Gamma\left(\frac{B-1}{2}\right)\times 2^{\beta/2}\Gamma\left(\frac{B-1+\beta}{2}\right)\Gamma\left(\frac{B-1}{2}\right)^{-1}$$

$$\times\int_{A}\frac{f(\mathbb{e}v)\sin^{B-3}\theta_1\cdots\sin\theta_{B-3}\,\mathrm{d}\theta_1\cdots\mathrm{d}\theta_{B-2}}{(v^{T}(\mathbb{e}^{T}\Sigma\mathbb{e})^{-1}v)^{\frac{B-1+\beta}{2}}}$$

$$= 2^{\beta/2}\Gamma\left(\frac{B-1+\beta}{2}\right)\Gamma\left(\frac{B-1}{2}\right)^{-1}\mathop{\mathbb{E}}_{v\sim S^{B-2}}\frac{f(\mathbb{e}v)}{(\det \mathbb{e}^{T}\Sigma\mathbb{e})^{1/2}(v^{T}(\mathbb{e}^{T}\Sigma\mathbb{e})^{-1}v)^{\frac{B-1+\beta}{2}}}$$

$$= 2^{\beta/2}\Gamma\left(\frac{B-1+\beta}{2}\right)\Gamma\left(\frac{B-1}{2}\right)^{-1}\mathop{\mathbb{E}}_{v\sim S^{B-2}}f(\mathbb{e}v)\mathcal{K}(v;\Sigma,\beta)$$

$\square$

As a straightforward consequence,

**Proposition E.34.** *If $\mathbb{e}^{T}\Sigma\mathbb{e}$ is nonsingular,*

$$\mathrm{V}_{\mathcal{B}_{\phi}}(\Sigma) = \mathop{\mathbb{E}}_{v\sim S^{B-2}}\phi(\sqrt{B}\mathbb{e}v)^{\otimes 2}\mathcal{K}(v;\Sigma)$$

For numerical calculations, it is useful to realize $\mathbb{e}$ as the matrix whose columns are $\mathbb{e}_{B-m} := (m(m+1))^{-1/2}(0,\ldots,0,-m,\overbrace{1,1,\ldots,1}^{m})^{T}$, for each $m = 1,\ldots,B-1$.

$$\mathbb{e}^{T} = \begin{pmatrix}\frac{1}{\sqrt{B(B-1)}} & & & & \\ & \frac{1}{\sqrt{(B-1)(B-2)}} & & & \\ & & \ddots & & \\ & & & 6^{-1/2} & \\ & & & & 2^{-1/2}\end{pmatrix}$$

$$\times\begin{pmatrix}-(B-1) & 1 & 1 & \cdots & 1 & 1 & 1 \\ 0 & -(B-2) & 1 & \cdots & 1 & 1 & 1 \\ \vdots & \vdots & \vdots & \ddots & \vdots & \vdots & \vdots \\ 0 & 0 & 0 & \cdots & -2 & 1 & 1 \\ 0 & 0 & 0 & \cdots & 0 & -1 & 1\end{pmatrix} \qquad (33)$$

We have

$$(\mathbb{e}v)_1 = -\sqrt{\frac{B-1}{B}}\cos\theta_1$$

$$(\mathbb{e}v)_2 = \cos\theta_1 - \sqrt{\frac{B-2}{B-1}}\sin\theta_1\cos\theta_2$$

$$(\mathbb{e}v)_3 = \cos\theta_1 + \sin\theta_1\cos\theta_2 - \sqrt{\frac{B-3}{B-2}}\sin\theta_1\sin\theta_2\cos\theta_3$$

$$(\mathbb{e}v)_4 = \cos\theta_1 + \sin\theta_1\cos\theta_2 + \sin\theta_1\sin\theta_2\cos\theta_3 - \sqrt{\frac{B-4}{B-3}}\sin\theta_1\sin\theta_2\sin\theta_3\cos\theta_4 \quad (34)$$

so in particular they only depend on $\theta_1,\ldots,\theta_4$.

**Angular coordinates** In many situations, the sparsity of this realization of $\mathbb{e}$ combined with sufficient symmetry allows us to simplify the high-dimensional spherical integral into only a few dimensions, using one of the following lemmas

**Lemma E.35.** *Let $f : \mathbb{R}^4 \to \mathbb{R}^A$ for some $A \in \mathbb{N}$. Suppose $B \geq 6$. Then for any $k < B - 1$*

$$\mathbb{E}[r^{-k}f(v_1,\ldots,v_4) : x \sim \mathcal{N}(0, I_{B-1})]$$
$$= \frac{\Gamma((B-1-k)/2)}{\Gamma((B-5)/2)}2^{-k/2}\pi^{-2}\int_0^\pi d\theta_1 \cdots \int_0^\pi d\theta_4 \ f(v_1^\theta,\ldots,v_4^\theta)\sin^{B-3}\theta_1\cdots\sin^{B-6}\theta_4$$

*where $v_i = x_i/\|x\|$ and $r = \|x\|$, and*

$$v_1^\theta = \cos\theta_1$$
$$v_2^\theta = \sin\theta_1\cos\theta_2$$
$$v_3^\theta = \sin\theta_1\sin\theta_2\cos\theta_3$$
$$v_4^\theta = \sin\theta_1\sin\theta_2\sin\theta_3\cos\theta_4.$$

**Lemma E.36.** *Let $f : \mathbb{R}^2 \to \mathbb{R}^A$ for some $A \in \mathbb{N}$. Suppose $B \geq 4$. Then for any $k < B - 1$*

$$\mathbb{E}[r^{-k}f(v_1, v_2) : x \sim \mathcal{N}(0, I_{B-1})]$$
$$= \frac{\Gamma((B-1-k)/2)}{\Gamma((B-3)/2)}2^{-k/2}\pi^{-1}\int_0^\pi d\theta_1 \int_0^\pi d\theta_2 \ f(v_1^\theta, v_2^\theta)\sin^{B-3}\theta_1\sin^{B-4}\theta_2$$

*where $v_i = x_i/\|x\|$ and $r = \|x\|$, and*

$$v_1^\theta = \cos\theta_1$$
$$v_2^\theta = \sin\theta_1\cos\theta_2$$

**Lemma E.37.** *Let $f : \mathbb{R} \to \mathbb{R}^A$ for some $A \in \mathbb{N}$. Suppose $B \geq 3$. Then for any $k < B - 1$*

$$\mathbb{E}[r^{-k}f(v) : x \sim \mathcal{N}(0, I_{B-1})]$$
$$= \frac{\Gamma((B-1-k)/2)}{\Gamma((B-2)/2)}2^{-k/2}\pi^{-1/2}\int_0^\pi d\theta \ f(\cos\theta)\sin^{B-3}\theta$$

*where $v_i = x_i/\|x\|$ and $r = \|x\|$.*

We will prove Lemma E.35; those of Lemma E.36 and Lemma E.37 are similar.

*Proof of Lemma E.35.*

$$\mathbb{E}[r^{-k}f(v_1,\ldots,v_4) : x \sim \mathcal{N}(0, I_{B-1})]$$

$$= (2\pi)^{-(B-1)/2} \int \mathrm{d}x \, r^{-k}f(v_1,\ldots,v_4)e^{-r^2/2}$$

$$= (2\pi)^{-(B-1)/2} \int \left( r^{B-2}\sin^{B-3}\theta_1\cdots\sin\theta_{B-3}\,\mathrm{d}r\,\mathrm{d}\theta_1\cdots\mathrm{d}\theta_{B-2} \right) r^{-k}f(v_1^\theta,\ldots,v_4^\theta)e^{-r^2/2}$$

$$= (2\pi)^{-(B-1)/2} \int f(v_1^\theta,\ldots,v_4^\theta)\sin^{B-3}\theta_1\cdots\sin\theta_{B-3}\,\mathrm{d}\theta_1\cdots\mathrm{d}\theta_{B-2}\int_0^\infty r^{B-2-k}e^{-r^2/2}\,\mathrm{d}r$$

$$= (2\pi)^{-(B-1)/2} \int f(v_1^\theta,\ldots,v_4^\theta)\sin^{B-3}\theta_1\cdots\sin\theta_{B-3}\,\mathrm{d}\theta_1\cdots\mathrm{d}\theta_{B-2}2^{(B-3-k)/2}\Gamma((B-1-k)/2)$$

(change of coordinate $r \leftarrow r^2$ and definition of $\Gamma$ function)

$$= \Gamma((B-1-k)/2)2^{-(2+k)/2}\pi^{-(B-1)/2}\int f(v_1^\theta,\ldots,v_4^\theta)\sin^{B-3}\theta_1\cdots\sin\theta_{B-3}\,\mathrm{d}\theta_1\cdots\mathrm{d}\theta_{B-2}$$

Because $v_1^\theta,\ldots,v_4^\theta$ only depends on $\theta_1,\ldots,\theta_4$, we can integrate out $\theta_5,\ldots,\theta_{B-2}$. By applying Lemma E.30 to $\theta_5,\ldots\theta_{B-3}$ and noting that $\int_0^{2\pi}\mathrm{d}\theta_{B-2} = 2\pi$, we get

$$\mathbb{E}[r^{-k}f(v_1^\theta,\ldots,v_4^\theta) : x \sim \mathcal{N}(0, I_{B-1})]$$

$$= \Gamma((B-1-k)/2)2^{-(2+k)/2}\pi^{-(B-1)/2}\int f(v_1^\theta,\ldots,v_4^\theta)\sin^{B-3}\theta_1\cdots\sin^{B-6}\theta_4\pi^{(B-7)/2}\Gamma((B-5)/2)^{-1}(2\pi)$$

$$= \frac{\Gamma((B-1-k)/2)}{\Gamma((B-5)/2)}2^{-k/2}\pi^{-2}\int f(v_1^\theta,\ldots,v_4^\theta)\sin^{B-3}\theta_1\cdots\sin^{B-6}\theta_4$$

□

**Cartesian coordinates.** We can often also simplify the high dimensional spherical integrals without trigonometry, as the next two lemmas show. Both are proved easily using change of coordinates.

**Lemma E.38.** *For any $u \in S^n$,*

$$\mathbb{E}_{v\sim S^n} f(u\cdot v) = \frac{\omega_{n-1}}{\omega_n}\int_{-1}^1 \mathrm{d}h\,(1-h^2)^{\frac{n-2}{2}}f(h)$$

*where $\omega_{n-1}$ is hypersurface of the $(n-1)$-dimensional unit hypersphere.*

**Lemma E.39.** *For any $u_1, u_2 \in S^n$,*

$$\mathbb{E}_{v\sim S^n} f(u_1\cdot v, u_2\cdot v) = \frac{\omega_{n-2}}{\omega_n}\int_{\substack{\|v\|\le 1\\ v\in\mathrm{span}(u_1,u_2)}} \mathrm{d}v\,(1-\|v\|^2)^{\frac{n-3}{2}}f(u_1\cdot v, u_2\cdot v)$$

*where $\omega_{n-1}$ is hypersurface of the $(n-1)$-dimensional unit hypersphere.*

### E.4 GEGENBAUER EXPANSION

**Definition E.40.** The Gegenbauer polynomials $\{C_l^{(\alpha)}(x)\}_{l=0}^\infty$, with $\deg C_l^{(\alpha)}(x) = l$, are the set of orthogonal polynomials with respect to the weight function $(1-x^2)^{\alpha-1/2}$ (i.e. $\int_{-1}^1(1-x^2)^{\alpha-1/2}C_l^{(\alpha)}(x)C_m^{(\alpha)}(x) = 0$ if $l \ne m$). By convention, they are normalized so that

$$\int_{-1}^1 \left[C_n^{(\alpha)}(x)\right]^2(1-x^2)^{\alpha-\frac{1}{2}}\,dx = \frac{\pi 2^{1-2\alpha}\Gamma(n+2\alpha)}{n!(n+\alpha)\Gamma(\alpha)^2}.$$

Here are several examples of low degree Gegenbauer polynomials

$$C_0^{(\alpha)}(x) = 1$$
$$C_1^{(\alpha)}(x) = 2\alpha x$$
$$C_2^{(\alpha)}(x) = -\alpha + 2\alpha(1+\alpha)x^2$$
$$C_3^{(\alpha)}(x) = -2\alpha(1+\alpha)x + \frac{4}{3}\alpha(1+\alpha)(2+\alpha)x^3.$$

They satisfy a few identities summarized below

**Fact E.41** (Suetin). $C_n^{(\alpha)}(\pm 1) = (\pm 1)^n \binom{n+2\alpha-1}{n}$

**Fact E.42** (Weisstein).

$$C_n^{(\lambda)\prime}(x) = 2\lambda C_{n-1}^{(\lambda+1)}(x) \tag{35}$$

$$xC_n^{(\lambda)\prime}(x) = nC_n^{(\lambda)}(x) + C_{n-1}^{(\lambda)}{}'(x) \tag{36}$$

$$C_{n+1}^{(\lambda)}{}'(x) = (n+2\lambda)C_n^{(\lambda)}(x) + xC_n^{(\lambda)\prime}(x) \tag{37}$$

$$nC_n^{(\lambda)}(x) = 2(n-1+\lambda)xC_{n-1}^{(\lambda)}(x) - (n-2+2\lambda)C_{n-2}^{(\lambda)}(x), \forall n \geq 2 \tag{38}$$

By repeated applications of Eq. (38),

**Proposition E.43.**

$$xC_n^{(\lambda)}(x) = \frac{1}{2(n+\lambda)}\left((n+1)C_{n+1}^{(\lambda)}(x) + (n-1+2\lambda)C_{n-1}^{(\lambda)}(x)\right), \forall n \geq 1$$

$$x^2 C_n^{(\lambda)}(x) = \kappa_2(n,\lambda)C_{n+2}^{(\lambda)}(x) + \kappa_0(n,\lambda)C_n^{(\lambda)}(x) + \kappa_{-2}(n,\lambda)C_{n-2}^{(\lambda)}(x), \forall n \geq 2$$

$$xC_0^{(\lambda)}(x) = x = \frac{1}{2\lambda}C_1^{(\lambda)}(x)$$

$$x^2 C_0^{(\lambda)}(x) = \frac{1}{4\lambda(1+\lambda)}\left(2C_2^{(\lambda)}(x) + 2\lambda C_0^{(\lambda)}(x)\right)$$

$$= \kappa_2(0,\lambda)C_2^{(\lambda)}(x) + \kappa_0(0,\lambda)C_0^{(\lambda)}(x)$$

$$x^2 C_1^{(\lambda)}(x) = \kappa_2(1,\lambda)C_3^{(\lambda)}(x) + \kappa_0(1,\lambda)C_1^{(\lambda)}(x)$$

where $\kappa_2(n,\lambda) := \frac{(n+1)(n+2)}{4(n+\lambda)(n+1+\lambda)}, \kappa_0(n,\lambda) := \frac{(n+1)(n+2\lambda)}{4(n+\lambda)(n+1+\lambda)} + \frac{(n-1+2\lambda)n}{4(n+\lambda)(n-1+\lambda)}, \kappa_{-2}(n,\lambda) := \frac{(n-1+2\lambda)(n-2+2\lambda)}{4(n+\lambda)(n-1+\lambda)}$.

This proposition is useful for the Gegenbauer expansion of the local convergence rate of Eq. (43).

**Zonal harmonics.**   The Gegenbauer polynomials are closely related to a special type of spherical harmonics called *zonal harmonics*, which are intuitively those harmonics that only depend on the "height" of the point along some fixed axis.

**Definition E.44.** Let $Z_u^{N-1,(l)}$ denote the degree $l$ zonal spherical harmonics on the $(N-1)$-dimensional sphere with axis $u \in S^{N-1}$, defined by $Z_u^{N-1,(l)}(v) = c_{N,l}^{-1}C_l^{(\frac{N-2}{2})}(u \cdot v)$ for any $u, v \in S^{N-1}$, where $c_{N,l} = \frac{N-2}{N+2l-2}$.

The first few zonal spherical harmonics are $Z_u^{N-1,(0)}(v) = 1, Z_u^{N-1,(1)}(v) = Nu \cdot v, \ldots$. We note a trivial lemma that is useful for simplying many expressions in Gegenbauer expansions.

**Lemma E.45.** $\frac{c_{B+1,l-1}}{c_{B-1,l}} = \frac{B-1}{B-3}$.

Let $\langle -, - \rangle_{S^N}$ denote the inner product induced by the uniform measure on $S^N$. One of the most important properties of zonal harmonics is their

**Fact E.46** (Reproducing property (Suetin)).

$$\langle Z_u^{N-1,(l)}, Z_v^{N-1,(l)} \rangle_{S^{N-1}} = Z_u^{N-1,(l)}(v) = c_{N,l}^{-1}C_l^{(\frac{N-2}{2})}(u \cdot v)$$

$$\langle Z_u^{N-1,(l)}, Z_v^{N-1,(m)} \rangle_{S^{N-1}} = 0, if\, l \neq m.$$

In our applications of the reproducing property, $u$ and $v$ will always be $\hat{\mathbb{e}}_{a,:} := \mathbb{e}_{a,:}/\|\mathbb{e}_{a,:}\| = \mathbb{e}_{a,:}\sqrt{\frac{B}{B-1}}$, where $\mathbb{e}_{a,:}$ is the $a$th row of $\mathbb{e}$.

Next, we present a new (to the best of our knowledge) identity showing that a certain quadratic form of $\phi$, reminiscent of Dirichlet forms, depending only on the derivative $\phi'$ through a spherical integral, is diagonalized in the Gegenbauer basis. This will be crucial to proving the necessity of gradient explosion under the BSB1 fixed point assumption.

**Theorem E.47** (Dirichlet-like form of $\phi$ diagonalizes over Gegenbauer basis). *Let $u_1, u_2 \in S^{B-2}$.*
*Suppose $\phi(\sqrt{B-1}x), \phi'(\sqrt{B-1}x) \in L^2((1-x^2)^{\frac{B-3}{2}})$ and $\phi(\sqrt{B-1}x)$ has Gegenbauer ex-*
*pansion $\sum_{l=0}^{\infty} a_l \frac{1}{c_{B-1,l}} C_l^{(\frac{B-3}{2})}(x)$. Then*

$$\mathop{\mathbb{E}}_{v \in S^{B-2}} (u_1 \cdot u_2 - (u_1 \cdot v)(u_2 \cdot v))\phi'(\sqrt{B-1}u_1 \cdot v)\phi'(\sqrt{B-1}u_2 \cdot v)$$

$$= \sum_{l=0}^{\infty} a_l^2 (l+B-3)l c_{B-1,l}^{-1} C_l^{(\frac{B-3}{2})}(u_1 \cdot u_2)$$

This is proved using the following lemmas.

**Lemma E.48.**

$$xC_l^{(\alpha)\prime}(x) = lC_l^{(\alpha)}(x) + \sum_{j \in [l-2]_2} 2(j+\alpha)C_j^{(\alpha)}(x)$$

$$C_l^{(\alpha)\prime}(x) = \sum_{j \in [l-1]_2} 2(j+\alpha)C_j^{(\alpha)}(x)$$

*where $[n]_2$ is the sequence $\{(n\%2), (n\%2)+1, \ldots, n-2, n\}$ and $n\%2$ is the remainder of $n/2$.*

*Proof.* Apply Eq. (36) and Eq. (37) alternatingly. $\square$

**Lemma E.49.**

$$lC_l^{(\alpha)}(t) + \sum_{j \in [l-2]_2} 2(j+\alpha)C_j^{(\alpha)}(t) - t \sum_{j \in [l-1]_2} 2(j+\alpha)C_j^{(\alpha)}(t) = 0$$

*Proof.* Apply Eq. (38) to $lC_l^{(\alpha)}(t), (l-2)C_{l-2}^{(\alpha)}(t), \ldots$. $\square$

*Proof of Thm E.47.* We have

$$\phi'(\sqrt{B-1}x) = \frac{1}{\sqrt{B-1}} \sum_{l=0}^{\infty} a_l \frac{1}{c_{B-1,l}} C_l^{(\frac{B-3}{2})\prime}(x)$$

$$= \frac{1}{\sqrt{B-1}} \sum_{l=0}^{\infty} a_l \frac{1}{c_{B-1,l}} \sum_{j \in [l-1]_2} (2j+B-3)C_j^{(\frac{B-3}{2})}(x)$$

$$= \frac{1}{\sqrt{B-1}} \sum_{l=0}^{\infty} \sum_{j \in [l-1]_2} a_l \frac{c_{B-1,j}}{c_{B-1,l}} (2j+B-3)c_{B-1,j}^{-1} C_j^{(\frac{B-3}{2})}(x)$$

$$= \frac{1}{\sqrt{B-1}} \sum_{j=0}^{\infty} c_{B-1,j}^{-1} C_j^{(\frac{B-3}{2})}(x) \sum_{l \in [j+1,\infty]_2} a_l \frac{c_{B-1,j}}{c_{B-1,l}} (2j+B-3)$$

where $[n, m]_2$ is the sequence $n, n+2, \ldots, m$. So for any $u_1, u_2 \in S^{B-2}$,

$$\mathop{\mathbb{E}}_{v \in S^{B-2}} \phi'(\sqrt{B-1}u_1 \cdot v)\phi'(\sqrt{B-1}u_2 \cdot v)$$

$$= \frac{1}{B-1} \sum_{j=0}^{\infty} c_{B-1,j}^{-1} C_j^{(\frac{B-3}{2})}(u_1 \cdot u_2) \left( \sum_{l \in [j+1,\infty]_2} a_l \frac{c_{B-1,j}}{c_{B-1,l}} (2j+B-3) \right)^2$$

$$= \frac{1}{B-1} \sum_{j=0}^{\infty} c_{B-1,j}(2j+B-3)^2 C_j^{(\frac{B-3}{2})}(u_1 \cdot u_2) \left( \sum_{l,m \in [j+1,\infty]_2} \frac{a_l}{c_{B-1,l}} \frac{a_m}{c_{B-1,m}} \right).$$

For $l < m$ and $m - l$ even, the coefficient of $a_l a_m$ is

$$\frac{2c_{B-1,l}^{-1}c_{B-1,m}^{-1}}{B-1} \sum_{j\in[l-1]_2} c_{B-1,j}(2j+B-3)^2 C_j^{(\frac{B-3}{2})}(u_1 \cdot u_2)$$

$$= \frac{2c_{B-1,l}^{-1}c_{B-1,m}^{-1}}{B-1} \sum_{j\in[l-1]_2} (B-3)(2j+B-3)C_j^{(\frac{B-3}{2})}(u_1 \cdot u_2). \tag{39}$$

For each $l$, the coefficient of $a_l^2$ is

$$\frac{c_{B-1,l}^{-2}}{B-1} \sum_{j\in[l-1]_2} c_{B-1,j}(2j+B-3)^2 C_j^{(\frac{B-3}{2})}(u_1 \cdot u_2)$$

$$= \frac{c_{B-1,l}^{-2}}{B-1} \sum_{j\in[l-1]_2} (B-3)(2j+B-3)C_j^{(\frac{B-3}{2})}(u_1 \cdot u_2). \tag{40}$$

Similarly, if $\phi'(x) \in L^2((1-x^2)^{\frac{B-3}{2}})$, we have

$$x\phi'(\sqrt{B-1}x) = \frac{1}{\sqrt{B-1}} \sum_{l=0}^{\infty} a_l \frac{1}{c_{B-1,l}} xC_l^{(\frac{B-3}{2})\prime}(x)$$

$$= \frac{1}{\sqrt{B-1}} \sum_{l=0}^{\infty} a_l \frac{1}{c_{B-1,l}} \left( lC_l^{(\frac{B-3}{2})}(x) + \sum_{j\in[l-2]_2} (2j+B-3)C_j^{(\frac{B-3}{2})}(x) \right)$$

$$= \frac{1}{\sqrt{B-1}} \sum_{j=0}^{\infty} c_{B-1,j}^{-1} C_j^{(\frac{B-3}{2})}(x) \left( a_j j + \sum_{l\in[j+2,\infty]_2} a_l \frac{c_{B-1,j}}{c_{B-1,l}}(2j+B-3) \right)$$

Then

$$\mathbb{E}_{v\in S^{B-2}}(u_1 \cdot v)(u_2 \cdot v)\phi'(\sqrt{B-1}u_1 \cdot v)\phi'(\sqrt{B-1}u_2 \cdot v)$$

$$= \frac{1}{B-1} \sum_{j=0}^{\infty} c_{B-1,j}^{-1} C_j^{(\frac{B-3}{2})}(u_1 \cdot u_2) \left( a_j j + \sum_{l\in[j+2,\infty]_2} a_l \frac{c_{B-1,j}}{c_{B-1,l}}(2j+B-3) \right)^2$$

For $l < m$ and $m - l$ even, the coefficient of $a_l a_m$ is

$$\frac{1}{B-1} \left( 2l(2l+B-3)\frac{c_{B-1,l}}{c_{B-1,m}}c_{B-1,l}^{-1}C_l^{(\frac{B-3}{2})}(u_1 \cdot u_2) \right.$$

$$\left. +2 \sum_{j\in[l-2]_2} \frac{c_{B-1,j}}{c_{B-1,l}}\frac{c_{B-1,j}}{c_{B-1,m}}(2j+B-3)^2 c_{B-1,j}^{-1}C_j^{(\frac{B-3}{2})}(u_1 \cdot u_2) \right)$$

$$= 2c_{B-1,l}^{-1}c_{B-1,m}^{-1}\frac{B-3}{B-1} \left( lC_l^{(\frac{B-3}{2})}(u_1 \cdot u_2) + \sum_{j\in[l-2]_2} (2j+B-3)C_j^{(\frac{B-3}{2})}(u_1 \cdot u_2) \right) \tag{41}$$

For each $l$, the coefficient of $a_l^2$ is

$$\frac{1}{B-1} \left( l^2 c_{B-1,l}^{-1}C_l^{(\frac{B-3}{2})}(u_1 \cdot u_2) + \sum_{j\in[l-2]_2} \left( \frac{c_{B-1,j}}{c_{B-1,l}} \right)^2 (2j+B-3)^2 c_{B-1,j}^{-1}C_j^{(\frac{B-3}{2})}(u_1 \cdot u_2) \right)$$

$$= c_{B-1,l}^{-2}\frac{B-3}{B-1} \left( \frac{l^2}{2l+B-3}C_l^{(\frac{B-3}{2})}(u_1 \cdot u_2) + \sum_{j\in[l-2]_2} (2j+B-3)C_j^{(\frac{B-3}{2})}(u_1 \cdot u_2) \right) \tag{42}$$

By Lemma E.49 and Eqs. (39) and (41), the coefficient of $a_l a_m$ with $m > l$ in $\mathbb{E}_{v\in S^{B-2}}(u_1 \cdot u_2 - (u_1 \cdot v)(u_2 \cdot v))\phi'(\sqrt{B-1}u_1 \cdot v)\phi'(\sqrt{B-1}u_2 \cdot v)$ is 0. Similarly, by Lemma E.49 and Eqs. (40) and (42), the coefficient of $a_l^2$ is $c_{B-1,l}^{-2}\frac{B-3}{B-1}\frac{l+B-3}{2l+B-3}lC_l^{(\frac{B-3}{2})}(u_1 \cdot u_2) = c_{B-1,l}^{-1}\frac{l+B-3}{B-1}lC_l^{(\frac{B-3}{2})}(u_1 \cdot u_2)$. $\quad\square$

### E.5 SYMMETRY AND ULTRASYMMETRY

As remarked before, all commonly used nonlinearities ReLU, tanh, and so on induce the dynamics Eq. (43) to converge to BSB1 fixed points, which we formally define as follows.

**Definition E.50.** We say a matrix $\Sigma \in \mathcal{S}_B$ is *BSB1* (short for "1-Block Symmetry Breaking") if $\Sigma$ has one common entry on the diagonal and one common entry on the off-diagonal, i.e.

$$\begin{pmatrix} a & b & b & \cdots \\ b & a & b & \cdots \\ b & b & a & \cdots \\ \vdots & \vdots & \vdots & \ddots \end{pmatrix}$$

We will denote such a matrix as $\mathrm{BSB1}(a, b)$. Note that $\mathrm{BSB1}(a, b)$ can be written as $(a-b)I + b\mathbf{1}\mathbf{1}^T$. Its spectrum is given below.

**Lemma E.51.** *A $B \times B$ matrix of the form $\mu I + \nu \mathbf{1}\mathbf{1}^T$ has two eigenvalues $\mu$ and $B\nu + \mu$, each with eigenspaces $\{x : \sum_i x_i = 0\}$ and $\{x : x_1 = \cdots = x_B\}$. Equivalently, if it has $a$ on the diagonal and $b$ on the off-diagonal, then the eigenvalues are $a - b$ and $(B - 1)b + a$.*

The following simple lemma will also be very useful

**Lemma E.52.** *Let $\Sigma := \mathrm{BSB1}_B(a, b)$. Then $G\Sigma G = (a - b)G$.*

*Proof.* $G$ and $\mathrm{BSB1}_B(a, b)$ can be simultaneously diagonalized by Lemma E.51. Note that $G$ zeros out the eigenspace $\mathbb{R}\mathbf{1}$, and is identity on its orthogonal complement. The result then follows from easy computations. □

The symmetry of BSB1 covariance matrices, especially the fact that they are *isotropic* in a codimension 1 subspace, is crucial to much of our analysis.

**Ultrasymmetry.** Indeed, as mentioned before, when investigating the asymptotics of the forward or backward dynamics, we encounter 4-tensor objects, such as $\frac{\mathrm{d}V_{\mathcal{B}_\phi}}{\mathrm{d}\Sigma}$ or $V_{\mathcal{B}'_\phi}$, that would seem daunting to analyze at first glance. However, under the BSB1 fixed point assumption, these 4-tensors contain considerable symmetry that somehow makes this possible. We formalize the notion of symmetry arising from such scenarios:

**Definition E.53.** Let $\mathcal{T} : \mathbb{R}^{B \times B} \to \mathbb{R}^{B \times B}$ be a linear operator. Let $\delta_i \in \mathbb{R}^B$ be the vector with 0 everywhere except 1 in coordinate $i$; then $\delta_i \delta_j^T$ is the matrix with 0 everywhere except 1 in position $(i, j)$. Write $[kl|ij] := \mathcal{T}(\delta_i \delta_j^T)_{kl}$. Suppose $\mathcal{T}$ has the property that for all $i, j, k, l \in [B] = \{1, \ldots, B\}$

- $[kl|ij] = [\pi(k)\pi(l)|\pi(i)\pi(j)]$ for all permutation $\pi$ on $[B]$, and

- $[ij|kl] = [ji|lk]$.

Then we say $\mathcal{T}$ is *ultrasymmetric*.

*Remark* E.54. In what follows, we will often "normalize" the representation "$[ij|kl]$" to the unique "$[i'j'|k'l']$" that is in the same equivalence class according to Defn E.53 and such that $i', j', k', l' \in [4]$ and $i' \le j', k' \le l'$ unless $i' = l', j' = k'$, in which case the normalization is $[12|21]$. Explicitly, we have the following equivalence classes and their normalized representations

We will study the eigendecomposition of an ultrasymmetric $\mathcal{T}$ as well as the projection $G^{\otimes 2} \circ \mathcal{T} \circ G^{\otimes 2} : \Sigma \mapsto G(\mathcal{T}\{G\Sigma G\})G$, respectively with the following domains

**Definition E.55.** Denote by $\mathcal{H}_B$ the space of symmetric matrices of dimension $B$. Also write $\mathcal{H}_B^G$ for the space of symmetric matrices $\Sigma$ of dimension $B$ such that $G\Sigma G = \Sigma$ (which is equivalent to saying rows of $\Sigma$ sum up to 0).

As in the case of $\mathcal{S}$, we omit subscript $B$ when it's clear from context.

The following subspaces of $\mathcal{H}_B$ and $\mathcal{H}_B^G$ will turn out to comprise the eigenspaces of $G^{\otimes 2} \circ \mathcal{T} \circ G^{\otimes 2}$.

| class | repr. |
|---|---|
| $i = j = k = l$ | $[11\vert11]$ |
| $i = j = k$ or $i = j = l$ | $[11\vert12]$ |
| $i = j$ and $k = l$ | $[11\vert22]$ |
| $i = j$ | $[11\vert23]$ |
| $i = k = l$ or $j = k = l$ | $[12\vert11]$ |
| $i = k$ and $j = l$ | $[12\vert12]$ |
| $i = k$ or $i = l$ | $[12\vert13]$ |
| $i = l$ and $j = k$ | $[12\vert21]$ |
| $k = l$ | $[12\vert33]$ |
| all different | $[12\vert34]$ |

**Definition E.56.** Let $\mathbb{L}_B := \{GDG : D \text{ diagonal}, \operatorname{tr} D = 0\} \subseteq \mathcal{H}_B^G$ and $\mathbb{M}_B := \{\Sigma \in \mathcal{H}_B^G : \operatorname{Diag}\Sigma = 0\}$. Note that $\dim \mathbb{L}_B = B - 1, \dim \mathbb{M}_B = \frac{B(B-3)}{2}$ and $\mathbb{R}G^B \oplus \mathbb{L}_B \oplus \mathbb{M}_B = \mathcal{H}_B^G$ is an orthogonal decomposition w.r.t Frobenius inner product.

For the eigenspaces of $\mathcal{T}$, we also need to define

**Definition E.57.** For any nonzero $a, b \in \mathbb{R}$, set

$$
\mathrm{L}_B(a,b) := \begin{pmatrix}
a & 0 & -b & -b & \cdots \\
0 & -a & b & b & \cdots \\
-b & b & 0 & 0 & \cdots \\
-b & b & 0 & 0 & \cdots \\
\vdots & \vdots & \vdots & \vdots & \ddots
\end{pmatrix} \in \mathcal{H}_B.
$$

In general, we say a matrix $M$ is $\mathrm{L}_B(a,b)$-shaped if $M = P\mathrm{L}_B(a,b)P^T$ for some permutation matrix $P$.

Note that

**Proposition E.58.** $\mathrm{L}(B-2,1)$-*shaped matrices span* $\mathbb{L}_B$.

**Proposition E.59.** $\mathrm{L}(B-2,1) = \frac{1}{B}[(-B+1,1,1,\ldots,1)^{\otimes 2} - (1,-B+1,1,\ldots,1)^{\otimes 2}] = BG[(1,0,0,\ldots,0)^{\otimes 2} - (0,1,0,\ldots,0)^{\otimes 2}]G$

*Proof.* Straightforward computation. $\square$

**Lemma E.60.** *Let* $L = \mathrm{L}_B(a,b)$. *Then* $G^{\otimes 2}\{L\} = GLG = \frac{a+2b}{B}\mathrm{L}_B(B-2,1)$.

*Proof.* $GLG$ can be written as the sum of outer products

$$
GLG = aG_1 \otimes G_1 - aG_2 \otimes G_2 + \sum_{i=3}^{B} -bG_1 \otimes G_i + bG_2 \otimes G_i - bG_i \otimes G_1 + bG_i \otimes G_2
$$

$$
= a\mathrm{L}_B\left(\left(\frac{B-1}{B}\right)^2 - \frac{1}{B^2}, \frac{B}{B^2}\right) + b\sum_{i=3}^{B}(-\delta_1 + \delta_2) \otimes G_i + G_i \otimes (-\delta_1 + \delta_2)
$$

$$
= \frac{a}{B}\mathrm{L}_B(B-2,1) + b((-\delta_1 + \delta_2) \otimes v + v \otimes (-\delta_1 + \delta_2))
$$

$$
\text{with } v = \left(-\frac{B-2}{B}, -\frac{B-2}{B}, \frac{2}{B}, \frac{2}{B}, \cdots\right)
$$

$$
= \frac{a}{B}\mathrm{L}_B(B-2,1) + \frac{2b}{B}\mathrm{L}_B(B-2,1)
$$

$$
= \frac{a+2b}{B}\mathrm{L}_B(B-2,1)
$$

$\square$

We are now ready to discuss the eigendecomposition of ultrasymmetric operators.

**Theorem E.61** (Eigendecomposition of an ultrasymmetric operator). *Let $\mathcal{T} : \mathbb{R}^{B \times B} \to \mathbb{R}^{B \times B}$ be an ultrasymmetric linear operator. Then $\mathcal{T}$ has the following eigendecomposition.*

1. *Two 1-dimensional eigenspace $\mathbb{R} \cdot \mathrm{BSB1}(\lambda^{\mathcal{T}}_{\mathrm{BSB1},i} - \alpha_{22}, \alpha_{21})$ with eigenvalue $\lambda^{\mathcal{T}}_{\mathrm{BSB1},i}$ for $i = 1, 2$, where*

$$\alpha_{11} = [11|11] + [11|22](B-1)$$
$$\alpha_{12} = 2(B-1)[11|12] + (B-2)(B-1)[11|23]$$
$$\alpha_{21} = 2[12|11] + (B-2)[12|33]$$
$$\alpha_{22} = [12|12] + 4(B-2)[12|13] + [12|21] + (B-2)(B-3)[12|34]$$

*and $\lambda^{\mathcal{T}}_{\mathrm{BSB1},1}$ and $\lambda^{\mathcal{T}}_{\mathrm{BSB1},2}$ are the roots to the quadratic*

$$x^2 - (\alpha_{11} + \alpha_{22})x + \alpha_{11}\alpha_{22} - \alpha_{12}\alpha_{21}.$$

2. *Two $(B-1)$-dimensional eigenspaces $\mathfrak{S}_B \cdot \mathrm{L}(\lambda^{\mathcal{T}}_{\mathbb{L},i} - \beta_{22}, \beta_{21})$ with eigenvalue $\lambda^{\mathcal{T}}_{\mathbb{L},i}$ for $i = 1, 2$. Here $\mathfrak{S}_B \cdot W$ denotes the linear span of the orbit of matrix $W$ under simultaneous permutation of its column and rows (by the same permutation), and*

$$\beta_{11} = [11|11] - [11|22]$$
$$\beta_{12} = 2(B-2)([11|23] - [11|12])$$
$$\beta_{21} = -[12|11] + [12|33]$$
$$\beta_{22} = [12|21] + [12|12] + 2(B-4)[12|13] - 2(B-3)[12|34].$$

*and $\lambda^{\mathcal{T}}_{\mathbb{L},1}$ and $\lambda^{\mathcal{T}}_{\mathbb{L},2}$ are the roots to the quadratic*

$$x^2 - (\beta_{11} + \beta_{22})x + \beta_{11}\beta_{22} - \beta_{12}\beta_{21}.$$

3. *Eigenspace $\mathbb{M}$ (dimension $B(B-3)/2$) with eigenvalue $\lambda^{\mathcal{T}}_{\mathbb{M}} := [12|12] + [12|21] - 4[12|13] + 2[12|34]$.*

The proof is by careful, but ultimately straightforward, computation.

*Proof.* We will use the bracket notation of Defn E.53 to denote entries of $\mathcal{T}$, and implicitly simplify it according to Remark E.54.

Item 1. Let $U \in \mathbb{R}^{B \times B}$ be the BSB1 matrix. By ultrasymmetry of $\mathcal{T}$ and BSB1 symmetry of $A$, $\mathcal{T}\{U\}$ is also BSB1. So we proceed to calculate the diagonal and off-diagonal entries of $\mathcal{T}\{G\}$.

We have

$$
\begin{aligned}
\mathcal{T}\{\mathrm{BSB1}(a,b)\}_{11} &= [11|11]a + 2(B-1)[11|12]b + [11|22](B-1)a + [11|23](B-2)(B-1)b \\
&= ([11|11] + [11|22](B-1))a + (2(B-1)[11|12] + (B-2)(B-1)[11|23])b \\
\mathcal{T}\{\mathrm{BSB1}(a,b)\}_{12} &= [12|12]b + 2[12|11]a + (B-2)[12|33]a + 2(B-2)[12|13]b \\
&\quad + [12|21]b + 2(B-2)[12|23]b + (B-2)(B-3)[12|34]b \\
&= (2[12|11] + (B-2)[12|33])a \\
&\quad + ([12|12] + 2(B-2)[12|13] + [12|21] + 2(B-2)[12|13] + (B-2)(B-3)[12|34])b \\
&= (2[12|11] + (B-2)[12|33])a \\
&\quad + ([12|12] + 4(B-2)[12|13] + [12|21] + (B-2)(B-3)[12|34])b
\end{aligned}
$$

Thus $\mathrm{BSB1}(\omega_1, \gamma_1)$ and $\mathrm{BSB1}(\omega_2, \gamma_2)$ are the eigenmatrices of $\mathcal{T}$, where $(\omega_1, \gamma_1)$ and $(\omega_2, \gamma_2)$ are the eigenvectors of the matrix

$$
\begin{pmatrix}
\alpha_{11} & \alpha_{12} \\
\alpha_{21} & \alpha_{22}
\end{pmatrix}
$$

with

$$\alpha_{11} = [11|11] + [11|22](B-1)$$
$$\alpha_{12} = 2(B-1)[11|12] + (B-2)(B-1)[11|23]$$
$$\alpha_{21} = 2[12|11] + (B-2)[12|33]$$
$$\alpha_{22} = [12|12] + 4(B-2)[12|13] + [12|21] + (B-2)(B-3)[12|34]$$

The eigenvalues are the two roots $\lambda^{\mathcal{T}}_{\mathrm{BSB1},1}, \lambda^{\mathcal{T}}_{\mathrm{BSB1},2}$ to the quadratic

$$x^2 - (\alpha_{11} + \alpha_{22})x + \alpha_{11}\alpha_{22} - \alpha_{12}\alpha_{21}$$

and the corresponding eigenvectors are

$$(\omega_1, \gamma_1) = (\lambda_1 - \alpha_{22}, \alpha_{21})$$
$$(\omega_2, \gamma_2) = (\lambda_2 - \alpha_{22}, \alpha_{21}).$$

Item 2. We will study the image of $\mathrm{L}_B(a,b)$ (Defn E.57) under $\mathcal{T}$. We have

$$
\begin{aligned}
\mathcal{T}\{\mathrm{L}(a,b)\}_{11} &= -\mathcal{T}\{\mathrm{L}(a,b)\}_{22} \\
&= [11|11]a + 2(B-2)[11|12](-b) + [11|22](-a) + 2(B-2)[11|23]b \\
&= ([11|11] - [11|22])a + 2(B-2)([11|23] - [11|12])b
\end{aligned}
$$

$$
\begin{aligned}
\mathcal{T}\{\mathrm{L}(a,b)\}_{12} &= \mathcal{T}\{\mathrm{L}(a,b)\}_{21} \\
&= [12|12]0 + [12|21]0 + [12|11](a-a) + [12|13](b-b) + [12|31](b-b) + [12|33]0 + [12|34]0 \\
&= 0
\end{aligned}
$$

$$
\begin{aligned}
\mathcal{T}\{\mathrm{L}(a,b)\}_{33} &= \mathcal{T}\{\mathrm{L}(a,b)\}_{ii}, \forall i \geq 3 \\
&= [11|11]0 + [11|12](2b-2b) + [11|22](a-a) + [11|23](2(B-3)b - 2(B-3)b) \\
&= 0
\end{aligned}
$$

$$
\begin{aligned}
\mathcal{T}\{\mathrm{L}(a,b)\}_{34} &= \mathcal{T}\{\mathrm{L}(a,b)\}_{ij}, \forall i \neq j \ \& \ i,j \geq 3 \\
&= [12|12]0 + [12|11]0 + [12|13](2b-2b) + [12|21]0 \\
&\quad + [12|31](2b-2b) + [12|33](a-a) + [12|34](2(B-4)b - 2(B-4)b) \\
&= 0
\end{aligned}
$$

$$
\begin{aligned}
\mathcal{T}\{\mathrm{L}(a,b)\}_{13} &= \mathcal{T}\{\mathrm{L}(a,b)\}_{1j}, \forall j \geq 3 \\
&= \mathcal{T}\{\mathrm{L}(a,b)\}_{j1}, \forall j \geq 3 \\
&= [12|12](-b) + [12|13](B-3-1)(-b) + [12|11]a + [12|21](-b) \\
&\quad + [12|31](B-3-1)(-b) + [12|33](-a) + [12|34]2(B-3)b \\
&= ([12|11] - [12|33])a + (2(B-3)[12|34] - 2(B-4)[12|13] - [12|12] - [12|21])b
\end{aligned}
$$

Thus $\mathrm{L}(a,b)$ transforms under $\mathcal{T}$ by the matrix

$$\mathrm{L}(a,b) \mapsto \mathrm{L}(\begin{pmatrix} \beta_{11} & \beta_{12} \\ \beta_{21} & \beta_{22} \end{pmatrix} \begin{pmatrix} a \\ b \end{pmatrix})$$

with

$$
\begin{aligned}
\beta_{11} &= [11|11] - [11|22] \\
\beta_{12} &= 2(B-2)([11|23] - [11|12]) \\
\beta_{21} &= -[12|11] + [12|33] \\
\beta_{22} &= [12|21] + [12|12] + 2(B-4)[12|13] - 2(B-3)[12|34].
\end{aligned}
$$

So if $\lambda^{\mathcal{T}}_{\mathbb{L},1}$ and $\lambda^{\mathcal{T}}_{\mathbb{L},2}$ are the roots of the equation

$$x^2 - (\beta_{11} + \beta_{22})x + \beta_{11}\beta_{22} - \beta_{12}\beta_{21}$$

then

$$\mathcal{T}\{\mathrm{L}(\lambda^{\mathcal{T}}_{\mathbb{L},1} - \beta_{22}, \beta_{21})\} = \lambda^{\mathcal{T}}_{\mathbb{L},1}\mathrm{L}(\lambda^{\mathcal{T}}_{\mathbb{L},1} - \beta_{22}, \beta_{21})$$
$$\mathcal{T}\{\mathrm{L}(\lambda^{\mathcal{T}}_{\mathbb{L},2} - \beta_{22}, \beta_{21})\} = \lambda^{\mathcal{T}}_{\mathbb{L},2}\mathrm{L}(\lambda^{\mathcal{T}}_{\mathbb{L},2} - \beta_{22}, \beta_{21})$$

Similarly, any image of these eigenvectors under simultaneous permutation of rows and columns remains eigenvectors with the same eigenvalue. This derives Item 2.

Item 3. Let $M \in \mathbb{M}$. We first show that $\mathcal{T}\{M\}$ has zero diagonal. We have

$$\mathcal{T}\{M\}_{11} = [11|12](\sum_{i=2}^{B} M_{1i} + M_{i1}) + [11|23](\sum_{i,j=1}^{B} M_{ij} - \left(\sum_{i=2}^{B} M_{1i} + M_{i1}\right))$$

$$= 0 + 0 = 0$$

which follows from $M\mathbf{1} = 0$ by definition of $\mathbb{M}$. Similarly $\mathcal{T}\{M\}_{ii} = 0$ for all $i$.

Now we show that $M$ is an eigenmatrix.

$$\mathcal{T}\{M\}_{12} = [12|12]M_{12} + [12|11]0 + [12|33]0 + [12|13]\left(\sum_{i=3}^{B} M_{1i} + \sum_{i=1}^{B} M_{i2}\right)$$

$$+ [12|21]M_{21} + [12|31]\left(\sum_{i=3}^{B} M_{i1} + \sum_{i=1}^{B} M_{2i}\right) + [12|34]\sum_{i\geq 3, j\geq 3} M_{ij}$$

$$= [12|12]M_{12} - 2M_{12}[12|13] + M_{21}[12|21] + [12|31](-2M_{12}) + [12|34](\sum_{i=3}^{B} -M_{i1} - M_{1i})$$

$$= M_{12}([12|12] - 2[12|13] + [12|21] - 2[12|31] + 2[12|34])$$

$$= M_{12}([12|12] + [12|21] - 4[12|13] + 2[12|34])$$

$$= \lambda_{\mathbb{M}}^{\mathcal{T}} M_{12}$$

Similarly $\mathcal{T}\{M\}_{ij} = \lambda_{\mathbb{M}}^{\mathcal{T}} M_{ij}$ for all $i \neq j$.

$\square$

Note that the eigenspaces described above are in general not orthogonal under trace inner product, so $\mathcal{T}$ is not self-adjoint (relative to the trace inner product) typically. However, as we see next, after projection by $G^{\otimes 2}$, it is self-adjoint.

**Theorem E.62** (Eigendecomposition of a projected ultrasymmetric operator). *Let $\mathcal{T} : \mathbb{R}^{B \times B} \to \mathbb{R}^{B \times B}$ be an ultrasymmetric linear operator. We write $G^{\otimes 2} \circ \mathcal{T} \upharpoonright \mathcal{H}_B^G$ for the operator $\mathcal{H}_B^G \to \mathcal{H}_B^G, \Sigma \mapsto \mathcal{T}\{\Sigma\} \mapsto G(\mathcal{T}\{\Sigma\})G$, restricted to $\Sigma \in \mathcal{H}_B^G$. Then $G^{\otimes 2} \circ \mathcal{T} \upharpoonright \mathcal{H}_B^G$ has the following eigendecomposition.*

1. *Eigenspace $\mathbb{R}G$ with eigenvalue $\lambda_G^{G,\mathcal{T}} := B^{-1}((B-1)(\alpha_{11} - \alpha_{21}) - (\alpha_{12} - \alpha_{22}))$, where as in Thm E.61,*

$$\alpha_{11} = [11|11] + [11|22](B-1)$$
$$\alpha_{12} = 2(B-1)[11|12] + (B-2)(B-1)[11|23]$$
$$\alpha_{21} = 2[12|11] + (B-2)[12|33]$$
$$\alpha_{22} = [12|12] + 4(B-2)[12|13] + [12|21] + (B-2)(B-3)[12|34]$$

2. *Eigenspace $\mathbb{L}$ with eigenvalue $\lambda_{\mathbb{L}}^{G,\mathcal{T}} := B^{-1}((B-2)\beta_{11} + \beta_{12} + 2(B-2)\beta_{21} + 2\beta_{22})$, where as in Thm E.61,*

$$\beta_{11} = [11|11] - [11|22]$$
$$\beta_{12} = 2(B-2)([11|23] - [11|12])$$
$$\beta_{21} = -[12|11] + [12|33]$$
$$\beta_{22} = [12|21] + [12|12] + 2(B-4)[12|13] - 2(B-3)[12|34].$$

3. *Eigenspace $\mathbb{M}$ with eigenvalue $\lambda_{\mathbb{M}}^{G,\mathcal{T}} := \lambda_{\mathbb{M}}^{\mathcal{T}} = [12|12] + [12|21] - 4[12|13] + 2[12|34]$.*

*Proof.* Item 1 As in the proof of Thm E.61, we find

$$\mathcal{T}\{\mathrm{BSB1}(a,b)\} = \mathrm{BSB1}\left(\begin{pmatrix} \alpha_{11} & \alpha_{12} \\ \alpha_{21} & \alpha_{22} \end{pmatrix}\begin{pmatrix} a \\ b \end{pmatrix}\right)$$

where

$$\begin{aligned}
\alpha_{11} &= [11|11] + [11|22](B-1) \\
\alpha_{12} &= 2(B-1)[11|12] + (B-2)(B-1)[11|23] \\
\alpha_{21} &= 2[12|11] + (B-2)[12|33] \\
\alpha_{22} &= [12|12] + 4(B-2)[12|13] + [12|21] + (B-2)(B-3)[12|34].
\end{aligned}$$

For $a = B - 1, b = -1$ so that $\mathrm{BSB1}(B-1,-1) = BG$, we get

$$\begin{aligned}
\mathcal{T}\{\mathrm{BSB1}(B-1,-1)\} &= \mathrm{BSB1}\left((B-1)\alpha_{11} - \alpha_{12}, (B-1)\alpha_{21} - \alpha_{22}\right) \\
G^{\otimes 2} \circ \mathcal{T}\{\mathrm{BSB1}(B-1,-1)\} &= G\, \mathrm{BSB1}\left((B-1)\alpha_{11} - \alpha_{12}, (B-1)\alpha_{21} - \alpha_{22}\right)\, G \\
&= \left((B-1)(\alpha_{11} - \alpha_{21}) - (\alpha_{12} - \alpha_{22})\right) G \\
&= B^{-1}\left((B-1)(\alpha_{11} - \alpha_{21}) - (\alpha_{12} - \alpha_{22})\right) \mathrm{BSB1}(B-1,-1) \\
&= \lambda_G^{G,\mathcal{T}} \mathrm{BSB1}(B-1,-1)
\end{aligned}$$

by Lemma E.52.

Item 2. It suffices to show that $\mathrm{L}(B-2,1)$ is an eigenmatrix with the eigenvalue $\lambda_{\mathbb{L}}^{G,\mathcal{T}}$.

As in the proof of Thm E.61, we find

$$\mathcal{T}\{\mathrm{L}(a,b)\} = \mathrm{L}\left(\begin{pmatrix} \beta_{11} & \beta_{12} \\ \beta_{21} & \beta_{22} \end{pmatrix}\begin{pmatrix} a \\ b \end{pmatrix}\right)$$

where

$$\begin{aligned}
\beta_{11} &= [11|11] - [11|22] \\
\beta_{12} &= 2(B-2)([11|23] - [11|12]) \\
\beta_{21} &= -[12|11] + [12|33] \\
\beta_{22} &= [12|21] + [12|12] + 2(B-4)[12|13] - 2(B-3)[12|34].
\end{aligned}$$

So with $a = B - 2, b = 1$, we have

$$\begin{aligned}
\mathcal{T}\{\mathrm{L}(B-2,1)\} &= \mathrm{L}((B-2)\beta_{11} + \beta_{12}, (B-2)\beta_{21} + \beta_{22}) \\
G^{\otimes 2} \circ \mathcal{T}\{\mathrm{L}(B-2,1)\} &= G\,\mathrm{L}((B-2)\beta_{11} + \beta_{12}, (B-2)\beta_{21} + \beta_{22})\, G \\
&= B^{-1}((B-2)\beta_{11} + \beta_{12} + 2((B-2)\beta_{21} + \beta_{22}))\mathrm{L}(B-2,1) \\
&= \lambda_{\mathbb{L}}^{G,\mathcal{T}} \mathrm{L}(B-2,1)
\end{aligned}$$

by Lemma E.60.

Item 3. The proof is exactly the same as in that of Thm E.61. $\qquad\square$

Noting that the eigenspaces of Thm E.62 are orthogonal, we have

**Proposition E.63.** $G^{\otimes 2} \circ \mathcal{T} \restriction \mathcal{H}^G$ *for any ultrasymmetric operator* $\mathcal{T}$ *is self-adjoint.*

**DOS Operators.**  In some cases, such as the original study of vanilla tanh networks by Schoenholz et al. (2016), the ultrasymmetric operator involved has a much simpler form:

**Definition E.64.** Let $\mathcal{T} : \mathcal{H}_B \to \mathcal{H}_B$ be such that for any $\Sigma \in \mathcal{H}_B$ and any $i \neq j \in [B]$,

$$\begin{aligned}
\mathcal{T}\{\Sigma\}_{ii} &= u\Sigma_{ii} \\
\mathcal{T}\{\Sigma\}_{ij} &= v\Sigma_{ii} + v\Sigma_{jj} + w\Sigma_{ij}
\end{aligned}$$

Then we say that $\mathcal{T}$ is *diagonal-off-diagonal semidirect*, or *DOS* for short. We write more specifically $\mathcal{T} = \mathrm{DOS}_B(u,v,w)$.

Thm E.62 and Thm E.61 still hold for DOS operators, but we can simplify the results and reason about them in a more direct way.

**Lemma E.65.** *Let $\mathcal{T} := \mathrm{DOS}_B(u, v, w)$. Then $\mathcal{T}\{\mathrm{L}_B(a, b)\} = \mathrm{L}_B(ua, wb - va)$.*

*Proof.* Let $L := \mathrm{L}_B(a, b)$. It suffices to verify the following, each of which is a direct computation.

1. $\mathcal{T}\{L\}_{3:B,3:B} = 0$.

2. $\mathcal{T}\{L\}_{1,2} = \mathcal{T}\{L\}_{2,1} = 0$

3. $\mathcal{T}\{L\}_{1,1} = -\mathcal{T}\{L\}_{2,2} = ua$

4. $\mathcal{T}\{L\}_{1,i} = -\mathcal{T}\{L\}_{2,i} = \mathcal{T}\{L\}_{i,1} = -\mathcal{T}\{L\}_{i,2} = va - wb$.

$\square$

**Lemma E.66.** *Let $\mathcal{T} := \mathrm{DOS}_B(u, v, w)$. Then $\mathrm{L}_B(w - u, v)$ and $\mathrm{L}_B(0, 1)$ are its eigenvectors:*

$$\mathcal{T}\{\mathrm{L}_B(w - u, v)\} = u\mathrm{L}_B(w - u, v)$$
$$\mathcal{T}\{\mathrm{L}_B(0, 1)\} = w\mathrm{L}_B(0, 1)$$

*Proof.* The map $(a, b) \mapsto (ua, wb - va)$ has eigenvalues $u$ and $w$ with corresponding eigenvectors $(w - u, v)$ and $(0, 1)$. By Lemma E.65, this implies the desired results. $\square$

**Lemma E.67.** *Let $\mathcal{T} := \mathrm{DOS}_B(u, v, w)$. Then for any $L \in \mathbb{L}_B$, $G^{\otimes 2} \circ \mathcal{T}\{L\} = \frac{(u-2v)(B-2)+2w}{B} L$.*

*Proof.* By Lemma E.60 and Lemma E.65, $G^{\otimes 2} \circ \mathcal{T}\{\mathrm{L}(B - 2, 1)\} = G^{\otimes 2}\{\mathrm{L}(u(B - 1), w - v(B - 1))\} = \frac{(u-2v)(B-2)+2w}{B}\mathrm{L}_B(B - 2, 1)$. By permutation symmetry, we also have the general result for any $\mathrm{L}_B(B - 2, 1)$-shaped matrix $L$. Since they span $\mathbb{L}_B$, this gives the conclusion we want. $\square$

**Lemma E.68.** *Let $\mathcal{T} := \mathrm{DOS}_B(u, v, w)$. Then $\mathcal{T}\{\mathrm{BSB1}_B(a, b)\} = \mathrm{BSB1}_B(ua, wb + 2va)$.*

*Proof.* Direct computation. $\square$

**Lemma E.69.** *Let $\mathcal{T} := \mathrm{DOS}_B(u, v, w)$. Then $\mathrm{BSB1}_B(w - u, wv)$ and $\mathrm{BSB1}_B(0, 1)$ are eigenvectors of $\mathcal{T}$.*

$$\mathcal{T}\{\mathrm{BSB1}_B(u - w, 2v)\} = u\mathrm{BSB1}_B(u - w, 2v)$$
$$\mathcal{T}\{\mathrm{BSB1}_B(0, 1)\} = w\mathrm{BSB1}_B(0, 1)$$

*Proof.* The linear map $(a, b) \mapsto (ua, wb + 2va)$ has eigenvalues $u$ and $w$ with corresponding eigenvectors $(u - w, 2v)$ and $(0, 1)$. The result then immediately follows from Lemma E.68. $\square$

**Lemma E.70.** *Let $\mathcal{T} := \mathrm{DOS}_B(u, v, w)$. Then $G^{\otimes 2} \circ \mathcal{T}\{G\} = \frac{(B-1)(u-2v)+w}{B}G$.*

*Proof.* Direct computation with Lemma E.68 and Lemma E.52 $\square$

**Definition E.71.** Define $\overline{\mathbb{M}}_B := \{\Sigma \in \mathcal{S}_B : \mathrm{Diag}\Sigma = 0\}$ (so compared to $\mathbb{M}$, matrices in $\overline{\mathbb{M}}$ do not need to have zero row sums). In addition, for any $a, b \in \mathbb{R}$, set

$$\overline{\mathrm{L}}_B(a, b) := \begin{pmatrix} a & -b & -b & \cdots \\ -b & 0 & 0 & \cdots \\ -b & 0 & 0 & \cdots \\ \vdots & \vdots & \vdots & \ddots \end{pmatrix} \in \mathcal{H}_B.$$

Define $\overline{\mathbb{L}}_B(a, b) := \mathrm{span}(P_{1i}^T \overline{\mathrm{L}}_B(a, b)P_{1i} : i \in [B])$ where $P_{1i}$ is the permutation matrix that swap the first entry with the $i$th entry, i.e. $\overline{\mathbb{L}}_B(a, b)$ is the span of the orbit of $\overline{\mathrm{L}}_B(a, b)$ under permuting rows and columns simultaneously.

Note that

$$L_B(a,b) = \overline{L}_B(a,b) - P_{12}^T \overline{L}_B(a,b)P_{12}$$

$$\text{BSB1}_B(a,-2b) = \sum_{i=1}^{B} P_{1i}^T \overline{L}_B(a,b)P_{1i}$$

So $L_B(a,b), \text{BSB1}_B(a,-2b) \in \overline{\mathbb{L}}_B(a,b)$.

**Theorem E.72.** *Let $\mathcal{T} := \text{DOS}_B(u,v,w)$. Suppose $w \neq u$. Then $\mathcal{T} \restriction \mathcal{H}_B$ has the following eigendecomposition:*

- *$\overline{\mathbb{M}}_B$ has eigenvalue $w$ ($\dim \overline{\mathbb{M}}_B = B(B-1)/2$)*

- *$\overline{\mathbb{L}}_B(w-u,v)$ has eigenvalue of $u$ ($\dim \overline{\mathbb{L}}_B(w-u,v) = B$)*

*If $w = u$, then $\overline{\mathbb{L}}_B(w-u,v) \subseteq \overline{\mathbb{M}}_B$.*

*Proof.* The case of $\overline{\mathbb{M}}_B$ is obvious. We will show that $\overline{\mathbb{L}}_B(w-u,v)$ is an eigenspace with eigenvalue $u$. Then by dimensionality consideration they are all of the eigenspaces of $\mathcal{T}$.

Let $L := \overline{\mathbb{L}}_B(w-u,v)$. Then it's not hard to see $\mathcal{T}\{L\} = \overline{\mathbb{L}}_B(a,b)$ for some $a,b \in \mathbb{R}$. It follows that $a$ has to be $u(w-u)$ and $b$ has to be $-(v(w-u)-wv) = uv$, which yields what we want. □

**Theorem E.73.** *Let $\mathcal{T} := \text{DOS}_B(u,v,w)$. Then $G^{\otimes 2} \circ \mathcal{T} \restriction \mathcal{H}_B^G : \mathcal{H}_B^G \to \mathcal{H}_B^G$ has the following eigendecomposition:*

- *$\mathbb{R}G$ has eigenvalue $\frac{(B-1)(u-2v)+w}{B}$ ($\dim \mathbb{R}G = 1$)*

- *$\mathbb{M}_B$ has eigenvalue $w$ ($\dim \mathbb{M}_B = B(B-3)/2$)*

- *$\mathbb{L}_B$ has eigenvalue $\frac{(B-2)(u-2v)+2w}{B}$ ($\dim \mathbb{L}_B = B-1$)*

*Proof.* The case of $\mathbb{M}_B$ is obvious. The case for $\mathbb{R}G$ follows from Lemma E.70. The case for $\mathbb{L}_B$ follows from Lemma E.67. By dimensionality considerations these are all of the eigenspaces. □

# F FORWARD DYNAMICS

In this section we will be interested in studying the dynamics on PSD matrices of the form

$$\Sigma^l = V_{\mathcal{B}_\phi}(\Sigma^{l-1}) = \mathbb{E}[\phi(\sqrt{B}Gh/\|Gh\|)^{\otimes 2} : h \sim \mathcal{N}(0, \Sigma^{l-1})] \tag{43}$$

where $\Sigma^l \in \mathcal{S}_B$ and $\phi : \mathbb{R} \to \mathbb{R}$.

## F.1 GLOBAL CONVERGENCE

Basic questions regarding the dynamics Eq. (43) are 1) does it converge? 2) What are the limit points? 3) How fast does it converge? Here we answer these questions definitively when $\phi = \text{id}$.

The following is the key lemma in our analysis.

**Lemma F.1.** *Consider the dynamics $\Sigma^l = \mathbb{E}[(h/\|h\|)^{\otimes 2} : h \sim \mathcal{N}(0, \Sigma^{l-1})]$ on $\Sigma^l \in \mathcal{S}_A$. Suppose $\Sigma^0$ is full rank. Then*

1. *$\lim_{l\to\infty} \Sigma^l = \frac{1}{A}I$.*

2. *This convergence is exponential in the sense that, for any full rank $\Sigma^0$, there is a constant $K < 1$ such that $\lambda_1(\Sigma^l) - \lambda_A(\Sigma^l) < K(\lambda_1(\Sigma^{l-1}) - \lambda_A(\Sigma^{l-1}))$ for all $l \geq 2$. Here $\lambda_1$ (resp. $\lambda_A$) denotes that largest (resp. smallest) eigenvalue.*

3. *Asymptotically, $\lambda_1(\Sigma^l) - \lambda_A(\Sigma^l) = O((1 - \frac{2}{A+2})^l)$.*

*Proof.* Let $\lambda_1{}^l \geq \lambda_2{}^l \geq \cdots \geq \lambda_A{}^l$ be the eigenvalues of $\Sigma^l$. It's easy to see that $\sum_i \lambda_i{}^l = 1$ for all $l \geq 1$. So WLOG we assume $l \geq 1$ and this equality holds.

We will show the "exponential convergence" statement; that $\lim_{l \to \infty} \Sigma^l = \frac{1}{A}I$ then follows from the trace condition above.

**Proof of Item 2.** For brevity, we write suppress the superscript index $l$, and use $\bar{\cdot}$ to denote $\cdot^l$. We will now compute the eigenvalues $\bar{\lambda}_1 \geq \cdots \geq \bar{\lambda}_A$ of $\bar{\Sigma}$.

First, notice that $\bar{\Sigma}$ and $\Sigma$ can be simultaneously diagonalized, and by induction all of $\{\Sigma^l\}_{l \geq 0}$ can be simultaneous diagonalized. Thus we will WLOG assume that $\Sigma$ is diagonal, $\Sigma = \mathrm{Diag}(\lambda_1, \ldots, \lambda_A)$, so that $\bar{\Sigma} = \mathrm{Diag}(\gamma_1, \ldots, \gamma_A)$ for some $\{\gamma_i\}_i$. These $\{\gamma_i\}_i$ form the eigenvalues of $\bar{\Sigma}$ but a priori we don't know whether they fall in decreasing order; in fact we will soon see that they do, and $\gamma_i = \bar{\lambda}_i$.

We have

$$
\begin{aligned}
\gamma_i &= \left(\frac{\pi}{2}\right)^{-A/2} \int_{0^A}^{\infty^A} \frac{\lambda_i x_i^2}{\sum_j \lambda_j x_j^2} e^{-\|x\|^2/2} \, \mathrm{d}x \\
&= \left(\frac{\pi}{2}\right)^{-A/2} \int_{0^A}^{\infty^A} \lambda_i x_i^2 e^{-\|x\|^2/2} \, \mathrm{d}x \int_0^\infty e^{-s \sum_j \lambda_j x_j^2} \, \mathrm{d}s \\
&= \left(\frac{\pi}{2}\right)^{-A/2} \int_{0^{A+1}}^{\infty^{A+1}} \lambda_i x_i^2 e^{-\frac{1}{2} \sum_j (1+2s\lambda_j) x_j^2} \, \mathrm{d}x \, \mathrm{d}s \\
&= \left(\frac{\pi}{2}\right)^{-A/2} \int_{0^2}^{\infty^2} \lambda_i x_i^2 e^{-\frac{1}{2}(1+2s\lambda_i) x_i^2} \cdot \left(\frac{\pi}{2}\right)^{\frac{A-1}{2}} \prod_{j \neq i} (1 + 2s\lambda_j)^{-1/2} \, \mathrm{d}x_i \, \mathrm{d}s \\
&= \left(\frac{\pi}{2}\right)^{-1/2} \int_{0^2}^{\infty^2} \lambda_i x_i^2 e^{-\frac{1}{2}(1+2s\lambda_i) x_i^2} \, \mathrm{d}x_i \prod_{j \neq i} (1 + 2s\lambda_j)^{-1/2} \, \mathrm{d}s \\
&= \int_0^\infty \lambda_i (1 + 2s\lambda_i)^{-3/2} \prod_{j \neq i} (1 + 2s\lambda_j)^{-1/2} \, \mathrm{d}s \\
&= \int_0^\infty \prod_{j=1}^A (1 + 2s\lambda_j)^{-1/2} \lambda_i (1 + 2s\lambda_i)^{-1} \, \mathrm{d}s
\end{aligned}
\tag{44}
$$

Therefore,

$$
\begin{aligned}
\gamma_i - \gamma_k &= \int_0^\infty \prod_{j=1}^A (1 + 2s\lambda_j)^{-1/2} \left(\lambda_i (1 + 2s\lambda_i)^{-1} - \lambda_k (1 + 2s\lambda_k)^{-1}\right) \mathrm{d}s \\
&= (\lambda_i - \lambda_k) \int_0^\infty \prod_{j=1}^A (1 + 2s\lambda_j)^{-1/2} (1 + 2s\lambda_i)^{-1} (1 + 2s\lambda_k)^{-1} \, \mathrm{d}s
\end{aligned}
$$

Since the RHS integral is always positive, $\lambda_i \geq \lambda_k \implies \gamma_i \geq \gamma_k$ and thus $\gamma_i = \bar{\lambda}_i$ for each $i$.

Define $T(\lambda) := \int_0^\infty \prod_{j=1}^A (1 + 2s\lambda_1)^{-1/2} (1 + 2s\lambda_1)^{-1} (1 + 2s\lambda_A)^{-1} \, \mathrm{d}s$, so that $\bar{\lambda}_1 - \bar{\lambda}_A = (\lambda_1 - \lambda_A) T(\lambda)$.

Note first that $\prod_{j=1}^A (1 + 2s\lambda_j)^{-1/2} (1 + 2s\lambda_i)^{-1} (1 + 2s\lambda_k)^{-1}$ is (strictly) log-convex and hence (strictly) convex in $\lambda$. Furthermore, $T(\lambda)$ is (strictly) convex because it is an integral of (strictly) convex functions. Thus $T$ is maximized over any convex region by its extremal points, and only by its extremal points because of strict convexity.

The convex region we are interested in is given by

$$
\mathcal{A} := \{(\lambda_i)_i : \lambda_1 \geq \lambda_2 \geq \cdots \geq \lambda_A \geq 0 \ \& \ \sum_i \lambda_i = 1\}.
$$

The unit sum condition follows from the normalization $h/\|h\|$ of the iteration map. The extremal points of $\mathcal{A}$ are $\{\omega^k := (\overbrace{1/k, 1/k, \cdots, 1/k}^{k \text{ times}}, 0, 0, \cdots, 0)\}_{k=1}^A$. For $k = 1, \ldots, A-1$, we have

$$
\begin{aligned}
T(\omega^k) &= \int_0^\infty (1 + 2s/k)^{-k/2-1}\, \mathrm{d}s \\
&= \left. \frac{2/k}{-k/2}(1 + 2s/k)^{-k/2} \right|_0^\infty \\
&= 1.
\end{aligned}
$$

But for $k = A$,

$$
\begin{aligned}
T(\omega^A) &= \int_0^\infty (1 + 2s/A)^{-A/2-2}\, \mathrm{d}s \\
&= \left. \frac{2/A}{-A/2-1}(1 + 2s/A)^{-k/2} \right|_0^\infty \\
&= \frac{A}{A+2}
\end{aligned}
$$

This shows that $T(\lambda) \leq 1$, with equality iff $\lambda = \omega^k$ for $k = 1, \ldots, A-1$. In fact, because every point $\lambda \in \mathcal{A}$ is a convex combination of $\omega^k$ for $k = 1, \ldots, A$, $\lambda = \sum_{k=1}^A a_k \omega^k$, by convexity of $T$, we must have

$$
\begin{aligned}
T(\lambda) &\leq \sum_{k=1}^A a_k T(\omega^k) \\
&= 1 - a_A \frac{2}{A+2} \\
&= 1 - \lambda_A \frac{2}{A+2}
\end{aligned}
\tag{45}
$$

where the last line follows because $\omega^A$ is the only point with last coordinate nonzero so that $a_A = \lambda_A$.

We now show that the gap $\lambda_1{}^l - \lambda_A{}^l \to 0$ as $l \to \infty$. There are two cases: if $\lambda_A{}^l$ is bounded away from 0 infinitely often, then $T(\lambda) < 1 - \epsilon$ infinitely often for a fixed $\epsilon > 0$ so that the gap indeed vanishes with $l$. Now suppose otherwise, that $\lambda_A{}^l$ converges to 0; we will show this leads to a contradiction. Notice that

$$
\overline{\lambda}_A = \int_0^\infty \prod_{j=1}^A (1 + 2s\lambda_j)^{-1/2} \lambda_A (1 + 2s\lambda_A)^{-1}\, \mathrm{d}s
$$

$$
\overline{\lambda}_A/\lambda_A \geq \int_0^\infty \prod_{j=1}^A (1 + 2s(1-\lambda_A)/(A-1))^{-(A-1)/2}(1 + 2s\lambda_A)^{-3/2}\, \mathrm{d}s
$$

where the first lines is Eq. (44) and the 2nd line follows from the convexity of $\prod_{j=1}^{A-1}(1 + 2s\lambda_j)^{-1/2}$ as a function of $(\lambda_1, \ldots, \lambda_{A-1})$. By a simple application of dominated convergence, as $\lambda_A \to 0$, this integral converges to a particular simple form,

$$
\begin{aligned}
\overline{\lambda}_A/\lambda_A &\geq \int_0^\infty \prod_{j=1}^A (1 + 2s/(A-1))^{-(A-1)/2}\, \mathrm{d}s \\
&= 1 + \frac{2}{A-3}
\end{aligned}
$$

Thus for large enough $l$, $\lambda_A{}^{l+1}/\lambda_A{}^l$ is at least $1 + \epsilon$ for some $\epsilon > 0$, but this contradicts the convergence of $\lambda_A$ to 0.

Altogether, this proves that $\lambda_1{}^l - \lambda_A{}^l \to 0$ and therefore $\lambda^l \to \omega^A$ as $l \to \infty$. Consequently, $\lambda_A{}^l$ is bounded from below, say by $K'$ (where $K' > 0$ because by Eq. (44), $\lambda_A{}^l$ is never 0), for all $l$,

and by Eq. (45), we prove Item 2 by taking $K = 1 - K'\frac{2}{A+2}$. In addition, asymptotically, the gap decreases exponentially as $T(\omega^A)^l = \left(\frac{A}{A+2}\right)^l$, proving Item 3. $\qquad\square$

**Theorem F.2.** *Consider the dynamics $\Sigma^l = \mathbb{E}[(h/\|h\|)^{\otimes 2} : h \sim \mathcal{N}(0, \Sigma^{l-1})]$ on $\Sigma^l \in \mathcal{S}_A$. Suppose $\Sigma^0 = M^T D M$ where $M$ is orthogonal and $D$ is a diagonal matrix. If $D$ has rank $C$ and $D_{ii} \neq 0, \forall 1 \leq i \leq C$, then*

1. *$\lim_{l\to\infty} \Sigma^l = \frac{1}{C} M^T D' M$ where $D'$ is the diagonal matrix with $D'_{ii} = \mathbb{I}(D_{ii} \neq 0)$.*

2. *This convergence is exponential in the sense that, for any $\Sigma^0$ of rank $C$, there is a constant $K < 1$ such that $\lambda_1(\Sigma^l) - \lambda_C(\Sigma^l) < K(\lambda_1(\Sigma^{l-1}) - \lambda_C(\Sigma^{l-1}))$ for all $l \geq 2$. Here $\lambda_i$ denotes the $i$th largest eigenvalue.*

3. *Asymptotically, $\lambda_1(\Sigma^l) - \lambda_C(\Sigma^l) = O((1 - \frac{2}{C+2})^l)$.*

*Proof.* Note that $\Sigma^l$ can always be simultaneously diagonalized with $\Sigma^0$, so that $\Sigma^l = M^T D^l M$ for some diagonal $D^l$ which has $D^l_{ii} = 0, \forall 0 \leq i \leq C$. Then we have $(D^l)' = \mathbb{E}[(h/\|h\|)^{\otimes 2} : h \sim \mathcal{N}(0, (D^{l-1})')]$, where $(D^l)'$ means the diagonal matrix obtained from $D^l$ by deleting the dimensions with zero eigenvalues. The proof then finishes by Lemma F.1. $\qquad\square$

From this it easily follows the following characterization of the convergence behavior.

**Corollary F.3.** *Consider the dynamics of Eq. (43) for $\phi = \mathrm{id}$: $\Sigma^l = B\,\mathbb{E}[(Gh/\|Gh\|)^{\otimes 2} : h \sim \mathcal{N}(0, \Sigma^{l-1})]$ on $\Sigma^l \in \mathcal{S}_B$. Suppose $G\Sigma G$ has rank $C < B$ and factors as $\hat{\mathrm{e}} D \hat{\mathrm{e}}^T$ where $D \in \mathbb{R}^{C\times C}$ is a diagonal matrix with no zero diagonal entries and $\hat{\mathrm{e}}$ is an $B \times C$ matrix whose columns form an orthonormal basis of a subspace of $\mathrm{im}\, G$. Then*

1. *$\lim_{l\to\infty} \Sigma^l = \frac{B}{C} \hat{\mathrm{e}} I_C \hat{\mathrm{e}}^T$.*

2. *This convergence is exponential in the sense that, for any $G\Sigma^0 G$ of rank $C$, there is a constant $K < 1$ such that $\lambda_1(\Sigma^l) - \lambda_C(\Sigma^l) < K(\lambda_1(\Sigma^{l-1}) - \lambda_C(\Sigma^{l-1}))$ for all $l \geq 2$. Here $\lambda_i$ denotes the $i$th largest eigenvalue.*

3. *Asymptotically, $\lambda_1(\Sigma^l) - \lambda_C(\Sigma^l) = O((1 - \frac{2}{C+2})^l)$.*

**General nonlinearity.** We don't (currently) have a proof of any characterization of the basin of attraction for Eq. (43) for general $\phi$. Thus we are forced to resort to finding its fixed points manually and characterize their local convergence properties.

## F.2 LIMIT POINTS

Batchnorm $\mathcal{B}_\phi$ is permutation-equivariant, in the sense that $\mathcal{B}_\phi(\pi h) = \pi \mathcal{B}_\phi(h)$ for any permutation matrix $\pi$. Along with the case of $\phi = \mathrm{id}$ studied above, this suggests that we look into BSB1 fixed points $\Sigma^*$. What are the BSB1 fixed points of Eq. (43)?

### F.2.1 SPHERICAL INTEGRATION

The main result (Thm F.5) of this section is an expression of the BSB1 fixed point diagonal and off-diagonal entries in terms of 1- and 2-dimensional integrals. This allows one to numerically compute such fixed points.

By a simple symmetry argument, we have the following

**Lemma F.4.** *Suppose $X$ is a random vector in $\mathbb{R}^B$, symmetric in the sense that for any permutation matrix $\pi$ and any subset $U$ of $\mathbb{R}^B$ measurable with respect to the distribution of $X$, $P(X \in U) = P(X \in \pi(U))$. Let $\Phi : \mathbb{R}^B \to \mathbb{R}^B$ be a symmetric function in the sense that $\Phi(\pi x) = \pi\Phi(x)$ for any permutation matrix $\pi$. Then $\mathbb{E}\,\Phi(X)\Phi(X)^T = \mu I + \nu \mathbf{1}\mathbf{1}^T$ for some $\mu$ and $\nu$.*

This lemma implies that $V_{\mathcal{B}_\phi}(\Sigma)$ is BSB1 whenever $\Sigma$ is BSB1. In the following, We in fact show that for any BSB1 $\Sigma$, $V_{\mathcal{B}_\phi}(\Sigma)$ is the same, and this is the unique BSB1 fixed point of Eq. (43).

**Theorem F.5.** *The BSB1 fixed point $\Sigma^* = \mu^* I + \nu^* \mathbf{1}\mathbf{1}^T$ to Eq. (43) is unique and satisfies*

$$\mu^* + \nu^* = \frac{\Gamma(\frac{B-1}{2})}{\Gamma(\frac{B-2}{2})\sqrt{\pi}} \int_0^\pi d\theta_1 \sin^{B-3}\theta_1 \phi(\sqrt{B}\zeta_1(\theta_1))^2$$

$$\nu^* = \begin{cases} \frac{B-3}{2\pi}\int_0^\pi d\theta_1 \int_0^\pi d\theta_2 \sin^{B-3}\theta_1 \sin^{B-4}\theta_2 \phi(\sqrt{B}\zeta_1(\theta_1))\phi(\sqrt{B}\zeta_2(\theta_1,\theta_2)), & \text{if } B \geq 4 \\ \frac{1}{2\pi}\int_0^{2\pi} d\theta \phi(-\sqrt{2}\sin\theta)\phi(\sin\theta/\sqrt{2} - \cos\theta\sqrt{3}/\sqrt{2}), & \text{if } B = 3 \end{cases}$$

*where*

$$\zeta_1(\theta) = -\sqrt{\frac{B-1}{B}}\cos\theta$$

$$\zeta_2(\theta_1,\theta_2) = \frac{1}{\sqrt{B(B-1)}}\cos\theta_1 - \sqrt{\frac{B-2}{B-1}}\sin\theta_1\cos\theta_2$$

*Proof.* For any BSB1 $\Sigma$, $\mathbb{e}^T \Sigma \mathbb{e} = \mu I_{B-1}$ for some $\mu$. Thus we can apply Proposition E.33 to $G\Sigma G$. We observe that $\mathcal{K}(v; G\Sigma G) = 1$, so that

$$V_{\mathcal{B}_\phi}(\Sigma) = V_{\mathcal{B}_\phi}(G\Sigma G) = \underset{v\sim S^{B-2}}{\mathbb{E}} \phi(\sqrt{B}\mathbb{e}v)^{\otimes 2}.$$

Note that this is now independent of the specific values of $\Sigma$. Now, using the specific form of $\mathbb{e}$ given by Eq. (33), we see that $V_{\mathcal{B}_\phi}(\Sigma)_{11}$ only depends on $\theta_1$ and $V_{\mathcal{B}_\phi}(\Sigma)_{12}$ only depends on $\theta_1$ and $\theta_2$. Then applying Proposition E.33 followed by Lemma E.36 and Lemma E.37 yield the desired result. □

### F.2.2 LAPLACE METHOD

In the case that $\phi$ is positive-homogeneous, we can apply Laplace's method Appendix E.1 and obtain the BSB1 fixed point in a much nicer form.

**Definition F.6.** For any $\alpha \geq 0$, define $K_{\alpha,B} := \mathsf{c}_\alpha P\left(\frac{B-1}{2},\alpha\right)^{-1}\left(\frac{B-1}{2}\right)^\alpha$ where $P(a,b) := \Gamma(a+b)/\Gamma(a)$ is the Pochhammer symbol.

Note that

**Proposition F.7.** $\lim_{B\to\infty} K_{\alpha,B} = \mathsf{c}_\alpha$.

We give a closed form description of the BSB1 fixed point for a positive homogeneous $\phi$ in terms of its J function.

**Theorem F.8.** *Suppose $\phi : \mathbb{R} \to \mathbb{R}$ is degree $\alpha$ positive-homogeneous. For any BSB1 $\Sigma \in \mathcal{S}_B$, $V_{\mathcal{B}_\phi}(\Sigma)$ is BSB1. The diagonal entries are $K_{\alpha,B}J_\phi(1)$ and the off-diagonal entries are $K_{\alpha,B}J_\phi\left(\frac{-1}{B-1}\right)$. Here $\mathsf{c}_\alpha$ is as defined in Defn E.7 and $J_\phi$ is as defined in Defn E.20. Thus a BSB1 fixed point of Eq. (43) exists and is unique.*

*Proof.* Let $\mathbb{e}$ be an $B \times (B-1)$ matrix whose columns form an orthonormal basis of $\text{im } G := \{Gv : v \in \mathbb{R}^B\} = \{w \in \mathbb{R}^B : \sum_i w_i = 0\}$. Let $\text{BSB1}(a,b)$ denote a matrix with $a$ on the diagonal and $b$ on the off-diagnals. Note that $V_\phi$ is positive homogeneous of degree $\alpha$, so for any $\mu$,

$$V_\phi(\mathbb{e}\mu I_{B-1}(I_{B-1} + 2s\mu I_{B-1})^{-1}\mathbb{e}^T)$$

$$= V_\phi(\frac{\mu}{1+2s\mu}\mathbb{e}\mathbb{e}^T)$$

$$= \left(\frac{\mu}{1+2s\mu}\right)^\alpha V_\phi(\mathbb{e}\mathbb{e}^T) \qquad \text{by Proposition E.24}$$

$$= \left(\frac{\mu}{1+2s\mu}\right)^\alpha V_\phi(G)$$

$$= \left(\frac{\mu}{1+2s\mu}\frac{B-1}{B}\right)^\alpha V_\phi\left(\text{BSB1}\left(1,\frac{-1}{B-1}\right)\right)$$

$$= \left(\frac{\mu}{1+2s\mu}\frac{B-1}{B}\right)^\alpha \mathsf{c}_\alpha J_\phi\left(\text{BSB1}\left(1,\frac{-1}{B-1}\right)\right) \qquad \text{by Corollary E.19}$$

So by Eq. (26),

$$
\begin{aligned}
V_{\mathcal{B}_\phi}(\mathbb{e}\mu I \mathbb{e}^T) &= B^\alpha \Gamma(\alpha)^{-1} \int_0^\infty \mathrm{d}s\, s^{\alpha-1}(1+2s\mu)^{-(B-1)/2} \left(\frac{\mu}{1+2s\mu}\frac{B-1}{B}\right)^\alpha \mathsf{c}_\alpha \mathrm{J}_\phi\left(\mathrm{BSB1}\left(1, \frac{-1}{B-1}\right)\right) \\
&= \Gamma(\alpha)^{-1}(B-1)^\alpha \mathsf{c}_\alpha \mathrm{J}_\phi\left(\mathrm{BSB1}\left(1, \frac{-1}{B-1}\right)\right) \int_0^\infty \mathrm{d}s\, s^{\alpha-1}(1+2s\mu)^{-(B-1)/2-\alpha}\mu^\alpha \\
&= \Gamma(\alpha)^{-1}(B-1)^\alpha \mathsf{c}_\alpha \mathrm{J}_\phi\left(\mathrm{BSB1}\left(1, \frac{-1}{B-1}\right)\right) \mathrm{Beta}(\alpha, (B-1)/2)2^{-\alpha} \\
&= \mathsf{c}_\alpha \frac{\Gamma((B-1)/2)}{\Gamma(\alpha+(B-1)/2)}\left(\frac{B-1}{2}\right)^\alpha \mathrm{J}_\phi\left(\mathrm{BSB1}\left(1, \frac{-1}{B-1}\right)\right)
\end{aligned}
$$

$\square$

**Corollary F.9.** *Suppose $\phi : \mathbb{R} \to \mathbb{R}$ is degree $\alpha$ positive-homogeneous. If $\Sigma^*$ is the BSB1 fixed point of Eq. (43) as given by Thm F.8, then $G\Sigma^* G = \mu^* I_{B-1}$ where $\mu^* = K_{\alpha,B}\big(\mathrm{J}_\phi(1) - \mathrm{J}_\phi(-1/(B-1))\big)$*

By setting $\alpha = 1$ and $\phi = \mathrm{relu}$, we get

**Corollary F.10.** *For any BSB1 $\Sigma \in \mathcal{S}_B$, $V_{\mathcal{B}_{\mathrm{relu}}}(\Sigma)$ is BSB1 with diagonal entries $\frac{1}{2}$ and off-diagonal entries $\frac{1}{2}\mathrm{J}_1\left(\frac{-1}{B-1}\right)$, so that $G^{\otimes 2}\{V_{\mathcal{B}_{\mathrm{relu}}}(\Sigma)\} = G(V_{\mathcal{B}_{\mathrm{relu}}}(\Sigma))G = \left(\frac{1}{2} - \frac{1}{2}\mathrm{J}_1\left(\frac{-1}{B-1}\right)\right)G.$*

By setting $\alpha = 1$ and $\phi = \mathrm{id} = x \mapsto \mathrm{relu}(x) - \mathrm{relu}(-x)$, we get

**Corollary F.11.** *For any BSB1 $\Sigma \in \mathcal{S}_B$, $V_{\mathcal{B}_{\mathrm{id}}}(\Sigma)$ is BSB1 with diagonal entries $1$ and off-diagonal entries $\frac{-1}{B-1}$, so that $G^{\otimes 2}\{V_{\mathcal{B}_{\mathrm{id}}}(\Sigma)\} = B/(B-1)G.$*

*Remark* F.12. One might hope to tweak the Laplace method for computing the fixed point to work for the Fourier method, but because there is no nice relation between $V_\phi(c\Sigma)$ and $V_\phi(\Sigma)$ in general, we cannot simplify Eq. (32) as we can Eq. (26) and Eq. (27).

### F.2.3 GEGENBAUER EXPANSION

It turns out that the BSB1 fixed point can be described very cleanly using the Gegenbauer coefficients of $\phi$.

When $\Sigma$ is BSB1, $\mathcal{K}(v, \Sigma) = 1$ for all $v \in S^{B-2}$, so that by Proposition E.34,

$$
\Sigma^* = \mathop{\mathbb{E}}_{v \sim S^{B-2}} \phi(\sqrt{B}\mathbb{e}v)^{\otimes 2}
$$
$$
\Sigma^*_{ab} = \mathop{\mathbb{E}}_{v \sim S^{B-2}} \phi(\sqrt{B}\mathbb{e}_{a,:}v)\phi(\sqrt{B}\mathbb{e}_{b,:}v)
$$

for any $a, b \in [B]$, independent of the actual values of $\Sigma$. Here $\mathbb{e}_{a,:}$ is the $a$th row of $\mathbb{e}$.

**Theorem F.13.** *If $\phi(\sqrt{B-1}x) \in L^2((1-x^2)^{\frac{B-3}{2}})$ has Gegenbauer expansion $\sum_{l=0}^\infty a_l \frac{1}{c_{B-1,l}} C_l^{(\frac{B-3}{2})}(x)$, then $\Sigma^* = \mu^* I + \nu^* \mathbf{1}\mathbf{1}^T$ with*

$$
\begin{aligned}
\mu^* &= \sum_{l=0}^\infty a_l^2 \frac{1}{c_{B-1,l}} \left(C_l^{(\frac{B-3}{2})}(1) - C_l^{(\frac{B-3}{2})}\left(\frac{-1}{B-1}\right)\right) \\
&= \sum_{l=0}^\infty a_l^2 \frac{1}{c_{B-1,l}} \left(\binom{B-4+l}{l} - C_l^{(\frac{B-3}{2})}\left(\frac{-1}{B-1}\right)\right) \\
\nu^* &= \sum_{l=0}^\infty a_l^2 \frac{1}{c_{B-1,l}} C_l^{(\frac{B-3}{2})}\left(\frac{-1}{B-1}\right)
\end{aligned}
$$

*Proof.* By the reproducing property Fact E.46, we have

$$
\begin{aligned}
\Sigma_{ab}^* &= \langle \phi(\sqrt{B}\mathbb{e}_{a,:}\cdot), \phi(\sqrt{B}\mathbb{e}_{b,:}\cdot) \rangle_{S^{B-2}} \\
&= \langle \phi(\sqrt{B-1}\hat{\mathbb{e}}_{a,:}\cdot), \phi(\sqrt{B-1}\hat{\mathbb{e}}_{b,:}\cdot) \rangle_{S^{B-2}} \\
&= \left\langle \sum_{l=0}^{\infty} a_l \frac{1}{c_{B-1,l}} C_l^{(\frac{B-3}{2})}(\hat{\mathbb{e}}_{a,:}\cdot), \sum_{l=0}^{\infty} a_l \frac{1}{c_{B-1,l}} C_l^{(\frac{B-3}{2})}(\hat{\mathbb{e}}_{b,:}\cdot) \right\rangle_{S^{B-2}} \\
&= \left\langle \sum_{l=0}^{\infty} a_l Z_{\hat{\mathbb{e}}_{a,:}}^{B-2,(l)}, \sum_{l=0}^{\infty} a_l Z_{\hat{\mathbb{e}}_{b,:}}^{B-2,(l)} \right\rangle_{S^{B-2}} \\
&= \sum_{l=0}^{\infty} a_l^2 Z_{\hat{\mathbb{e}}_{a,:}}^{B-2,(l)}(\hat{\mathbb{e}}_{b,:}) \\
&= \sum_{l=0}^{\infty} a_l^2 \frac{1}{c_{B-1,l}} C_l^{(\frac{B-3}{2})}(\langle \hat{\mathbb{e}}_{a,:}, \hat{\mathbb{e}}_{b,:} \rangle) \\
&= \sum_{l=0}^{\infty} a_l^2 \frac{1}{c_{B-1,l}} C_l^{(\frac{B-3}{2})} \left( \begin{cases} 1 & \text{if } a = b \\ -1/(B-1) & \text{else} \end{cases} \right)
\end{aligned}
$$

This gives the third equation. Since $\mu^* = \Sigma_{11}^* - \Sigma_{12}^*$, straightforward arithmetic yields the first equation. The second follows from Fact E.41. $\square$

We will see later that the rate of gradient explosion also decomposes nicely in terms of Gegenbauer coefficients (Thm G.5). However, this is not quite true for the eigenvalues of the forward convergence (Thms F.22 and F.24).

## F.3 LOCAL CONVERGENCE

In this section we consider linearization of the dynamics given in Eq. (43). Thus we must consider linear operators on the space of PSD linear operators $\mathcal{S}_B$. To avoid confusion, we use the following notation: If $\mathcal{T} : \mathcal{S}_B \to \mathcal{S}_B$ (for example the Jacobian of $V_{\mathcal{B}_\phi}$) and $\Sigma \in \mathcal{S}_B$, then write $\mathcal{T}\{\Sigma\}$ for the image of $\Sigma$ under $\mathcal{T}$.

*A priori*, the Jacobian of $V_{\mathcal{B}_\phi}$ at its BSB1 fixed point may seem like a very daunting object. But a moment of thought shows that

**Proposition F.14.** *The Jacobian* $\frac{dV_{\mathcal{B}_\phi}}{d\Sigma}\big|_\Sigma : \mathcal{H}_B \to \mathcal{H}_B$ *is ultrasymmetric for any BSB1* $\Sigma$.

Now we prepare several helper reults in order to make progress understanding batchnorm.

**Definition F.15.** Define $\mathsf{n}(x) = \sqrt{B}x/\|x\|$, i.e. division by sample standard deviation.

Batchnorm $\mathcal{B}_\phi$ can be decomposed as the composition of three steps, $\phi \circ \mathsf{n} \circ G$, where $G$ is mean-centering, $\mathsf{n}$ is division by standard deviation, and $\phi$ is coordinate-wise application of nonlinearity. We have, as operators $\mathcal{H} \to \mathcal{H}$,

$$
\begin{aligned}
\frac{dV_{\mathcal{B}_\phi}(\Sigma)}{d\Sigma} &= \frac{d\,\mathbb{E}[\mathcal{B}_\phi(z)^{\otimes 2} : z \in \mathcal{N}(0, \Sigma)]}{d\Sigma} \\
&= \frac{d\,\mathbb{E}[(\phi \circ \mathsf{n})(x)^{\otimes 2} : x \in \mathcal{N}(0, G\Sigma G)]}{d\Sigma} \\
&= \frac{dV_{(\phi \circ \mathsf{n})}(\Sigma^G)}{d\Sigma^G} \circ \frac{d\Sigma^G}{d\Sigma} \\
&= \frac{dV_{(\phi \circ \mathsf{n})}(\Sigma^G)}{d\Sigma^G} \circ G^{\otimes 2}
\end{aligned}
$$

**Definition F.16.** With $\Sigma^*$ being the unique BSB1 fixed point, write $\mathcal{U} := G^{\otimes 2} \circ \frac{dV_{(\phi \circ \mathsf{n})}(\Sigma^G)}{d\Sigma^G}\big|_{\Sigma^G = G\Sigma^* G} : \mathcal{H}^G \to \mathcal{H}^G$.

It turns out to be advantageous to study $\mathcal{U}$ first and relate its eigendecomposition back to that of $\frac{dV_{\mathcal{B}_\phi}(\Sigma)}{d\Sigma}\big|_{\Sigma=\Sigma^*}$, where $\Sigma^*$ is the BSB1 fixed point, by applying Lemma F.17.

**Lemma F.17.** *Let $X$ and $Y$ be two vector spaces. Consider linear operators $A : X \to Y, B : Y \to X$. Then*

1. $\operatorname{rank} AB = \operatorname{rank} BA$

2. *If $v \in Y$ is an eigenvector of $AB$ with nonzero eigenvalue, then $X \ni Bv \neq 0$ and $Bv$ is an eigenvector of $BA$ of the same eigenvalue*

3. *Suppose $AB$ has $k = \operatorname{rank} AB$ linearly independent eigenvectors $\{v_i\}_{i=1}^k$ of nonzero eigenvalues $\{\lambda_i\}_{i=1}^k$. Then $BA$ has $k$ linearly independent eigenvectors $\{Bv_i\}_{i=1}^k$ with the same eigenvalues $\{\lambda_i\}_{i=1}^k$ which are all eigenvectors of $BA$ with nonzero eigenvalues, up to linear combinations within eigenvectors with the same eigenvalue.*

With $A = G^{\otimes 2}$ and $B = \frac{dV_{(\phi\circ n)}(\Sigma^G)}{d\Sigma^G}$, Thm E.61 implies that $AB$ and $BA$ can both be diagonalized, and this lemma implies that all nonzero eigenvalues of $\frac{dV_{\mathcal{B}_\phi}(\Sigma)}{d\Sigma}$ can be recovered from those of $\mathcal{U}$.

*Proof.* (Item 1) Observe $\operatorname{rank} AB = \operatorname{rank} ABAB \leq \operatorname{rank} BA$. By symmetry the two sides are in fact equal.

(Item 2) $Bv$ cannot be zero or otherwise $ABv = A0 = 0$, contradicting the fact that $v$ is an eigenvector with nonzero eigenvalue. Suppose $\lambda$ is the eigenvalue associated to $v$. Then $BA(Bv) = B(ABv) = B(\lambda v) = \lambda Bv$, so $Bv$ is an eigenvector of $BA$ with the same eigenvalue.

(Item 3) Item 2 shows that $\{Bv_i\}_{i=1}^k$ are eigenvectors $BA$ with the same eigenvalues $\{\lambda_i\}_{i=1}^k$. The eigenspaces with different eigenvalues are linearly independent, so it suffices to show that if $\{Bv_{i_j}\}_j$ are eigenvectors of the same eigenvalue $\lambda_s$, then they are linearly independent. But $\sum_j a_j Bv_{i_j} = 0 \implies \sum_j a_j v_{i_j} = 0$ because $B$ is injective on eigenvectors by Item 2, so that $a_j = 0$ identically. Hence $\{Bv_j\}_j$ is linearly independent.

Since $\operatorname{rank} BA = k$, these are all of the eigenvectors with nonzero eigenvalues of $BA$ up to linear combinations.

$\square$

**Lemma F.18.** *Let $f : \mathbb{R}^B \to \mathbb{R}^A$ be measurable, and $\Sigma \in \mathcal{S}_B$ be invertible. Then for any $\Lambda \in \mathbb{R}^{B \times B}$, with $\langle \cdot, \cdot \rangle$ denoting trace inner product,*

$$\frac{d}{d\Sigma} \mathbb{E}[f(z) : z \sim \mathcal{N}(0, \Sigma)]\{\Lambda\} = \frac{1}{2} \mathbb{E}[f(z) \langle \Sigma^{-1} z z^T \Sigma^{-1} - \Sigma^{-1}, \Lambda \rangle : z \sim \mathcal{N}(0, \Sigma)]$$

*If $f$ is in addition twice-differentiable, then*

$$\frac{d}{d\Sigma} \mathbb{E}[f(z) : z \sim \mathcal{N}(0, \Sigma)] = \frac{1}{2} \mathbb{E}\left[ \frac{d^2 f(z)}{dz^2} : z \sim \mathcal{N}(0, \Sigma) \right]$$

*whenever both sides exist.*

*Proof.* Let $\Sigma_t, t \in (-\epsilon, \epsilon)$ be a smooth path in $\mathcal{S}_B$, with $\Sigma_0 = \Sigma$. Write $\frac{d}{dt}\Sigma_t = \dot{\Sigma}_t$. Then

$$
\frac{d}{dt}\mathbb{E}[f(z) : z \sim \mathcal{N}(0, \Sigma_t)]
$$

$$
= \frac{d}{dt}(2\pi)^{-B/2} \det \Sigma_t^{-1/2} \int dz\, e^{-\frac{1}{2}z^T \Sigma_t^{-1} z} f(z)\Big|_{t=0}
$$

$$
= (2\pi)^{-B/2}\frac{-1}{2} \det \Sigma_0^{-1/2} \operatorname{tr}\left(\Sigma_0^{-1}\dot{\Sigma}_0\right) \int dz\, e^{-\frac{1}{2}z^T \Sigma_0^{-1} z/2} f(z)
$$

$$
\quad + (2\pi)^{-B/2} \det \Sigma_0^{-1/2} \int dz\, e^{-\frac{1}{2}z^T \Sigma_0^{-1} z}\frac{-1}{2}z^T(-\Sigma_0^{-1}\dot{\Sigma}_0\Sigma_0^{-1})z f(z)
$$

$$
= \frac{-1}{2}(2\pi)^{-B/2} \det \Sigma_0^{-1/2} \int dz\, e^{-\frac{1}{2}z^T \Sigma_0^{-1} z} f(z)(\operatorname{tr}\Sigma_0^{-1}\dot{\Sigma}_0 - z^T \Sigma_0^{-1}\dot{\Sigma}_0\Sigma_0^{-1}z)
$$

$$
= \frac{1}{2}(2\pi)^{-B/2} \det \Sigma^{-1/2} \int dz\, e^{-\frac{1}{2}z^T \Sigma^{-1} z} f(z) \left\langle \Sigma^{-1}zz^T\Sigma^{-1} - \Sigma^{-1}, \dot{\Sigma}_0 \right\rangle
$$

$$
= \frac{-1}{2}\mathbb{E}[f(z) \left\langle \Sigma^{-1} - \Sigma^{-1}zz^T\Sigma^{-1}, \dot{\Sigma}_0 \right\rangle : z \sim \mathcal{N}(0, \Sigma)]
$$

Note that

$$
v^T \frac{d}{dz}e^{\frac{-1}{2}z^T\Sigma^{-1}z} = -v^T\Sigma^{-1}z e^{\frac{-1}{2}z^T\Sigma^{-1}z}
$$

$$
w^T\left(\frac{d^2}{dz^2}e^{\frac{-1}{2}z^T\Sigma^{-1}z}\right)v = \left(w^T\Sigma^{-1}zz^T\Sigma^{-1}v - w^T\Sigma^{-1}v\right)e^{\frac{-1}{2}z^T\Sigma^{-1}z}
$$

$$
= \left\langle \Sigma^{-1}zz^T\Sigma^{-1} - \Sigma^{-1}, vw^T \right\rangle e^{\frac{-1}{2}z^T\Sigma^{-1}z}
$$

so that as a cotensor,

$$
\frac{d^2}{dz^2}e^{\frac{-1}{2}z^T\Sigma^{-1}z}\{\Lambda\} = \left\langle \Sigma^{-1}zz^T\Sigma^{-1} - \Sigma^{-1}, \Lambda \right\rangle e^{\frac{-1}{2}z^T\Sigma^{-1}z}
$$

for any $\Lambda \in \mathbb{R}^{B \times B}$.

Therefore,

$$
\frac{d\,\mathbb{E}[f(z) : z \sim \mathcal{N}(0, \Sigma)]}{d\Sigma}\{\Lambda\} = \frac{1}{2}(2\pi)^{-B/2} \det \Sigma^{-1/2} \int dz\, f(z)\frac{d^2 e^{-\frac{1}{2}z^T\Sigma^{-1}z}}{d^2 z}\{\Lambda\}
$$

$$
= \frac{1}{2}(2\pi)^{-B/2} \det \Sigma^{-1/2} \int dz\, e^{-\frac{1}{2}z^T\Sigma^{-1}z}\frac{d^2 f(z)}{d^2 z}\{\Lambda\}
$$

(by integration by parts)

$$
= \frac{1}{2}\mathbb{E}\left[\frac{d^2 f(z)}{d^2 z} : z \sim \mathcal{N}(0, \Sigma)\right]\{\Lambda\}
$$

$\square$

Note that for any $\Lambda \in \mathcal{H}_B$ with $\|\Lambda\|_{op} < \mu^*$,

$$
\mathbb{E}[\phi \circ \mathsf{n}(z)^{\otimes 2} : z \in \mathcal{N}(0, \mu^* G + \Lambda)] = \mathbb{E}[\phi \circ \mathsf{n}(Gz)^{\otimes 2} : z \in \mathcal{N}(0, \mu^* I + \Lambda)]
$$

$$
= \mathbb{E}[\mathcal{B}_\phi(z)^{\otimes 2} : z \in \mathcal{N}(0, \mu^* I + \Lambda)]
$$

so that we have, for any $\Lambda \in \mathcal{H}_B$,

$$
\begin{aligned}
\mathcal{U}\{\Lambda\} &= \left.\frac{\mathrm{d}V_{\mathcal{B}_\phi}(\Sigma)}{\mathrm{d}\Sigma}\right|_{\Sigma=\mu^* I} \{\Lambda\} \\
&= \frac{1}{2}\, \mathbb{E}[\mathcal{B}_\phi(z)^{\otimes 2}\langle \mu^{*-1}zz^T - \mu^{*-1}I, \Lambda\rangle : z \sim \mathcal{N}(0, \mu^* I)] \\
&\hspace{6cm} \text{(by Lemma F.18)} \\
&= \frac{1}{2}\mu^{*-2}\, \mathbb{E}[\mathcal{B}_\phi(z)^{\otimes 2}\langle zz^T, \Lambda\rangle : z \sim \mathcal{N}(0, \mu^* I)] - \frac{1}{2}\mu^{*-1}\Sigma^*\langle I, \Lambda\rangle \\
&\hspace{6cm} \text{(by Thm F.8)} \\
&= (2\mu^*)^{-1}\left(\mathbb{E}[\mathcal{B}_\phi(z)^{\otimes 2}\langle zz^T, \Lambda\rangle : z \sim \mathcal{N}(0, I)] - \Sigma^*\langle I, \Lambda\rangle\right) \\
&\hspace{6cm} (\mathcal{B}_\phi \text{ is scale-invariant}) \\
&= (2\mu^*)^{-1}\left(\mathbb{E}[\mathcal{B}_\phi(z)^{\otimes 2}\langle Gzz^T G, \Lambda\rangle : z \sim \mathcal{N}(0, I)] - \Sigma^*\langle I, \Lambda\rangle\right) \\
&\hspace{6cm} (\Lambda = G\Lambda G)
\end{aligned}
$$

Let's extend to all matrices by this formula:

**Definition F.19.** Define

$$
\tilde{\mathcal{U}} : \mathbb{R}^{B\times B} \to \mathcal{S}_B,
$$
$$
\Lambda \mapsto (2\mu^*)^{-1}\left(\mathbb{E}[\mathcal{B}_\phi(z)^{\otimes 2}\langle Gzz^T G, \Lambda\rangle : z \sim \mathcal{N}(0, I)] - \Sigma^*\langle I, \Lambda\rangle\right)
$$

### F.3.1 Spherical Integration

So $\tilde{\mathcal{U}} \upharpoonright \mathcal{H}_B = \mathcal{U} \upharpoonright \mathcal{H}_B$. Ultimately we will apply Thm E.61 to $G^{\otimes 2} \circ \tilde{\mathcal{U}} \upharpoonright \mathcal{H}_B = G^{\otimes 2} \circ \mathcal{U}$

**Definition F.20.** Write $\tau_{ij} = \frac{1}{2}\left(\delta_i\delta_j^T + \delta_j\delta_i^T\right)$.

Then

$$
\begin{aligned}
\tilde{\mathcal{U}}\{\tau_{ij}\}_{kl} &= (2\mu^*)^{-1}\left(\mathbb{E}[\mathcal{B}_\phi(z)_k\mathcal{B}_\phi(z)_l(Gz)_i(Gz)_j : z \sim \mathcal{N}(0, I)] - \Sigma_{kl}^*\mathbb{I}(i = j)\right) \\
&= (2\mu^*)^{-1}\left(\mathbb{E}[\phi(\mathsf{n}(y))_k\phi(\mathsf{n}(y))_l y_i y_j : y \sim \mathcal{N}(0, G)] - \Sigma_{kl}^*\mathbb{I}(i = j)\right) \\
&= (2\mu^*)^{-1}\left(\mathbb{E}[\phi(\mathsf{n}(y))_k\phi(\mathsf{n}(y))_l y_i y_j : y \sim \mathcal{N}(0, \mathbb{e}\mathbb{e}^T)] - \Sigma_{kl}^*\mathbb{I}(i = j)\right) \\
&\hspace{4cm} \text{(See Defn E.3 for defn of } \mathbb{e}) \\
&= (2\mu^*)^{-1}\left(\mathbb{E}[\phi(\mathsf{n}(\mathbb{e}x))_k\phi(\mathsf{n}(\mathbb{e}x))_l(\mathbb{e}x)_i(\mathbb{e}x)_j : x \sim \mathcal{N}(0, I_{B-1})] - \Sigma_{kl}^*\mathbb{I}(i = j)\right)
\end{aligned}
$$

Here we will realize $\mathbb{e}$ as the matrix in Eq. (33).

**Definition F.21.** Define $W_{ij|kl} := \mathbb{E}[\phi(\mathsf{n}(\mathbb{e}x))_k\phi(\mathsf{n}(\mathbb{e}x))_l(\mathbb{e}x)_i(\mathbb{e}x)_j : x \sim \mathcal{N}(0, I_{B-1})]$.

Then $\tilde{\mathcal{U}}\{\tau_{ij}\}_{kl} = (2\mu^*)^{-1}(W_{ij|kl} - \Sigma_{kl}^*\mathbb{I}(i = j))$. If we can evaluate $W_{ij|kl}$ then we can use Thm E.61 to compute the eigenvalues of $G^{\otimes 2} \circ \mathcal{U}$. It's easy to see that $W_{ij|kl}$ is ultrasymmetric. Thus WLOG we can take $i, j, k, l$ from $\{1, 2, 3, 4\}$.

By Lemma E.35, and the fact that $x \mapsto \mathbb{e}x$ is an isometry,

$$
\begin{aligned}
W_{ij|kl} &= \mathbb{E}[r^2\phi(\sqrt{B}\mathbb{e}v)_k\phi(\sqrt{B}\mathbb{e}v)_l(\mathbb{e}v)_i(\mathbb{e}v)_j : x \sim \mathcal{N}(0, I_{B-1})] \\
&= (B - 5)(B - 3)(B - 1)(2\pi)^{-2}\times \\
&\qquad \int_0^\pi \mathrm{d}\theta_1 \cdots \int_0^\pi \mathrm{d}\theta_4\, \phi(\sqrt{B}\mathbb{e}v)_k\phi(\sqrt{B}\mathbb{e}v)_l(\mathbb{e}v)_i(\mathbb{e}v)_j \sin^{B-3}\theta_1 \cdots \sin^{B-6}\theta_4
\end{aligned}
$$

If WLOG we further assume that $k, l \in \{1, 2\}$ (by ultrasymmetry), then there is no dependence on $\theta_3$ and $\theta_4$ inside $\phi$. So we can expand $(\mathbb{e}v)_i$ and $(\mathbb{e}v)_j$ in trigonometric expressions as in Eq. (34) and integrate out $\theta_3$ and $\theta_4$ via Lemma E.29. We will not write out this integral explicitly but instead focus on other techniques for evaluating the eigenvalues.

### F.3.2   GEGENBAUER EXPANSION

Now let's compute the local convergence rate via Gegenbauer expansion. By differentiating Proposition E.34 through a path $\Sigma_t \in \mathcal{S}_B, t \in (-\epsilon, \epsilon)$, we get

$$
\frac{d}{dt}V_{\mathcal{B}_\phi}(\Sigma_t) = \mathop{\mathbb{E}}_{v \sim S^{B-2}} \phi(\sqrt{B}\mathbb{e}v)^{\otimes 2}\frac{d}{dt}\mathcal{K}(v; \Sigma_t)
$$

$$
= \mathop{\mathbb{E}}_{v \sim S^{B-2}} \phi(\sqrt{B}\mathbb{e}v)^{\otimes 2}\mathcal{K}(v; \Sigma_t)\left(\frac{B-1}{2}(v^T\Sigma_\square^{-1}v)^{-1}v^T\Sigma_\square^{-1}\frac{d\Sigma_\square}{dt}\Sigma_\square^{-1}v - \frac{1}{2}\text{tr}(\Sigma_\square^{-1}\frac{d\Sigma_\square}{dt})\right)
$$

where $\Sigma_\square = \Sigma_{\square t} = \mathbb{e}^T\Sigma_t\mathbb{e} \in \mathcal{S}_{B-1}$ (as introduced below Defn E.3). At $\Sigma_0 = \Sigma^*$, the BSB1 fixed point, we have $\Sigma_{\square 0} = \mu^*I$, and

$$
\frac{d}{dt}V_{\mathcal{B}_\phi}(\Sigma_t)\Big|_{t=0} = \mathop{\mathbb{E}}_{v \sim S^{B-2}} \phi(\sqrt{B}\mathbb{e}v)^{\otimes 2}\mathcal{K}(v; \Sigma^*)\mu^{*-1}\left(\frac{B-1}{2}v^T\frac{d\Sigma_\square}{dt}v - \frac{1}{2}\text{tr}(\frac{d\Sigma_\square}{dt})\right)
$$

$$
= \mu^{*-1}\mathop{\mathbb{E}}_{v \sim S^{B-2}}\phi(\sqrt{B}\mathbb{e}v)^{\otimes 2}\left(\frac{B-1}{2}v^T\frac{d\Sigma_\square}{dt}v - \frac{1}{2}\text{tr}(\frac{d\Sigma_\square}{dt})\right) \tag{46}
$$

If $\frac{d}{dt}\Sigma|_{t=0} = G$, then the term in the parenthesis vanishes. This shows that $\frac{dV_{\mathcal{B}_\phi}}{d\Sigma}\Big|_{\Sigma=\Sigma^*}\{G\} = 0$.

**Eigenvalue for** $\mathbb{L}$. If $\frac{d}{dt}\Sigma|_{t=0} = L(B-2,1) = BG(\delta_1^{\otimes 2} - \delta_2^{\otimes 2})G$ where $\delta_1 = (1,0,\ldots,0), \delta_2 = (0,1,0,\ldots,0)$, both in $\mathbb{R}^B$ (Proposition E.59), then we know by Thm E.62 that $G^{\otimes 2} \circ \frac{d}{dt}V_{\mathcal{B}_\phi}(\Sigma_t)\big|_{t=0}$ is a multiple of $L(B-2,1)$. We compute

$$
\left(\frac{d}{dt}V_{\mathcal{B}_\phi}(\Sigma_t)\Big|_{t=0}\right)_{ab} = B\frac{B-1}{2}\mu^{*-1}\mathop{\mathbb{E}}_{v \sim S^{B-2}}\phi(\sqrt{B}\mathbb{e}_{a,:}v)\phi(\sqrt{B}\mathbb{e}_{b,:}v)((\mathbb{e}_{1,:}v)^2 - (\mathbb{e}_{2,:}v)^2).
$$

Clearly, $\frac{d}{dt}V_{\mathcal{B}_\phi}(\Sigma_t)\big|_{t=0}$ is L-shaped.

Now define the quantity $\mathcal{A}(a,b;c) := \mathbb{E}_{v \sim S^{B-2}}\phi(\sqrt{B}\mathbb{e}_{a,:}v)\phi(\sqrt{B}\mathbb{e}_{b,:}v)(\mathbb{e}_{c,:}v)^2$, so that $\left(\frac{d}{dt}V_{\mathcal{B}_\phi}(\Sigma_t)\big|_{t=0}\right)_{ab} = \frac{B(B-1)}{2}\mu^{*-1}(\mathcal{A}(a,b;1) - \mathcal{A}(a,b;2))$. Because $\mathbb{e}$ is an isometry, $\sum_b(\mathbb{e}_{b,:}v)^2 = \|v\|^2 = 1$ for all $v \in S^{B-2}$. Thus

$$
\sum_{c=1}^{B}\mathcal{A}(a,b;c) = \mathop{\mathbb{E}}_{v \sim S^{B-2}}\phi(\sqrt{B}\mathbb{e}_{a,:}v)\phi(\sqrt{B}\mathbb{e}_{b,:}v) = \Sigma_{ab}^*. \tag{47}
$$

By symmetry, $\mathcal{A}(a,b;a) = \mathcal{A}(a,b;b)$ and for any $c,c' \notin \{a,b\}, \mathcal{A}(a,b;c) = \mathcal{A}(a,b;c')$. So we have, for any $a,b,c$ not equal,

$$
\mathcal{A}(a,a;a) + (B-1)\mathcal{A}(a,a;c) = \Sigma_{aa}^*
$$
$$
2\mathcal{A}(a,b;b) + (B-2)\mathcal{A}(a,b;c) = \Sigma_{ab}^*
$$

So the eigenvalue associated to $L(B-2,1)$ is, So by Lemma E.60, $G^{\otimes 2} \circ \mathcal{U}\{L(B-2,1)\} = \lambda_{\mathbb{L}}^\uparrow L(B-2,1)$, where

$$
\lambda_{\mathbb{L}}^\uparrow = B\frac{B-1}{2}\mu^{*-1}\frac{\left(\frac{d}{dt}V_{\mathcal{B}_\phi}(\Sigma_t)\big|_{t=0}\right)_{11} - 2\left(\frac{d}{dt}V_{\mathcal{B}_\phi}(\Sigma_t)\big|_{t=0}\right)_{13}}{B}
$$

$$
= B\frac{B-1}{2}\mu^{*-1}\frac{\mathcal{A}(1,1;1) - \mathcal{A}(1,1;2) - 2(\mathcal{A}(1,3;1) - \mathcal{A}(1,3;2))}{B}
$$

$$
= \frac{B-1}{2}\mu^{*-1}(\mathcal{A}(1,1;1) - \frac{1}{B-1}(\Sigma_{aa}^* - \mathcal{A}(1,1;1)) - 2(\mathcal{A}(1,3;1) - \frac{1}{B-2}(\Sigma_{ab}^* - 2\mathcal{A}(1,3;1))))
$$

$$
= \frac{B-1}{2}\mu^{*-1}(\frac{B}{B-1}\mathcal{A}(1,1;1) - \frac{1}{B-1}\Sigma_{aa}^* - 2\frac{B}{B-2}\mathcal{A}(1,3;1) + \frac{2}{B-2}\Sigma_{ab}^*) \tag{48}
$$

Thus, if $\phi(\sqrt{B-1}x) \in L^2((1-x^2)^{\frac{B-3}{2}})$ has Gegenbauer expansion $\sum_{l=0}^{\infty} a_l \frac{1}{c_{B-1,l}} C_l^{(\frac{B-3}{2})}(x)$, then by Proposition E.43,

$$x^2 \phi(\sqrt{B-1}x)$$

$$= \sum_{l=0}^{\infty} a_l \frac{1}{c_{B-1,l}} x^2 C_l^{(\frac{B-3}{2})}(x)$$

$$= \sum_{l=0}^{\infty} a_l \frac{1}{c_{B-1,l}} \left( \kappa_2(l, \frac{B-3}{2}) C_{l+2}^{(\frac{B-3}{2})}(x) \right.$$

$$\left. + \kappa_0(l, \frac{B-3}{2}) C_l^{(\frac{B-3}{2})}(x) + \kappa_{-2}(l, \frac{B-3}{2}) C_{l-2}^{(\frac{B-3}{2})}(x) \right)$$

where $C_l^{(\frac{B-3}{2})}(x)$ is understood to be 0 for $l < 0$

$$= \sum_{l=0}^{\infty} C_l^{(\frac{B-3}{2})}(x)$$

$$\left( \frac{a_{l-2}}{c_{B-1,l-2}} \kappa_2(l-2, \frac{B-3}{2}) + \frac{a_l}{c_{B-1,l}} \kappa_0(l, \frac{B-3}{2}) + \frac{a_{l+2}}{c_{B-1,l+2}} \kappa_{-2}(l+2, \frac{B-3}{2}) \right)$$

where $\kappa_i(l, \frac{B-3}{2})$ is understood to be 0 for $l < 0$

$$= \sum_{l=0}^{\infty} c_{B-1,l}^{-1} C_l^{(\frac{B-3}{2})}(x)$$

$$\left( \frac{a_{l-2} c_{B-1,l}}{c_{B-1,l-2}} \kappa_2(l-2, \frac{B-3}{2}) + a_l \kappa_0(l, \frac{B-3}{2}) + \frac{a_{l+2} c_{B-1,l}}{c_{B-1,l+2}} \kappa_{-2}(l+2, \frac{B-3}{2}) \right)$$

We can evaluate, for any $a, b$ (possibly equal),

$$\mathcal{A}(a,b;b) = \mathop{\mathbb{E}}_{v \sim S^{B-2}} \phi(\sqrt{B} \mathbb{e}_{a,:} v) \phi(\sqrt{B} \mathbb{e}_{b,:} v) (\mathbb{e}_{b,:} v)^2$$

$$= \mathop{\mathbb{E}}_{v \sim S^{B-2}} \phi(\sqrt{B-1} \hat{\mathbb{e}}_{a,:} v) \phi(\sqrt{B-1} \hat{\mathbb{e}}_{b,:} v) (\hat{\mathbb{e}}_{b,:} v)^2 \frac{B-1}{B}$$

$$= \frac{B-1}{B} \sum_{l=0}^{\infty} c_{B-1,l}^{-1} C_l^{(\frac{B-3}{2})}(\langle \hat{\mathbb{e}}_{a,:}, \hat{\mathbb{e}}_{b,:} \rangle)$$

$$a_l \left( \frac{a_{l-2} c_{B-1,l}}{c_{B-1,l-2}} \kappa_2(l-2, \frac{B-3}{2}) + a_l \kappa_0(l, \frac{B-3}{2}) + \frac{a_{l+2} c_{B-1,l}}{c_{B-1,l+2}} \kappa_{-2}(l+2, \frac{B-3}{2}) \right)$$

(49)

By Eq. (48), the eigenvalue associated with $L(B-2, 1)$ is

$$\frac{B-1}{2} \mu^{*-1} (\frac{B}{B-1} \mathcal{A}(1,1;1) - \frac{1}{B-1} \Sigma_{aa}^* - 2\frac{B}{B-2} \mathcal{A}(1,3;1) + \frac{2}{B-2} \Sigma_{ab}^*)$$

$$= \frac{B-1}{2} \mu^{*-1} \sum_{l=0}^{\infty} c_{B-1,l}^{-1} (\gamma_l C_l^{(\frac{B-3}{2})}(1) - \tau_l C_l^{(\frac{B-3}{2})}(\frac{-1}{B-1}))$$

where

$$\gamma_l := a_l \left( \frac{a_{l-2} c_{B-1,l}}{c_{B-1,l-2}} \kappa_2(l-2, \frac{B-3}{2}) + a_l \kappa_0(l, \frac{B-3}{2}) + \frac{a_{l+2} c_{B-1,l}}{c_{B-1,l+2}} \kappa_{-2}(l+2, \frac{B-3}{2}) \right) - \frac{1}{B-1} a_l^2$$

$$\tau_l := \frac{2(B-1)}{B-2} \gamma_l.$$

Thus,

**Theorem F.22.** *Suppose $\phi(\sqrt{B-1}x)$ has Gegenbauer expansion $\sum_{l=0}^{\infty} a_l \frac{1}{c_{B-1,l}} C_l^{(\frac{B-3}{2})}(x)$. Then the eigenvalue of $\mathcal{U}$ with respect to the eigenspace $\mathbb{L}$ is a ratio of quadratic forms*

$$\lambda_{\mathbb{L}}^{\uparrow} = \frac{\sum_{l=0}^{\infty} a_l^2 w_{B-1,l} + a_l a_{l+2} u_{B-1,l}}{\sum_{l=0}^{\infty} a_l^2 v_{B-1,l}}$$

*where*

$$v_{B-1,l} := c_{B-1,l}^{-1} \left( C_l^{(\frac{B-3}{2})}(1) - C_l^{(\frac{B-3}{2})}(\frac{-1}{B-1}) \right)$$

$$w_{B-1,l} := \frac{B-1}{2} \left( \kappa_0 \left( l, \frac{B-3}{2} \right) - \frac{1}{B-1} \right) c_{B-1,l}^{-1} \left( C_l^{(\frac{B-3}{2})}(1) - \frac{2(B-1)}{B-2} C_l^{(\frac{B-3}{2})}(\frac{-1}{B-1}) \right)$$

$$= \frac{l(B-3+l)(B-3)}{(B-5+2l)(B-1+2l)} c_{B-1,l}^{-1} \left( C_l^{(\frac{B-3}{2})}(1) - \frac{2(B-1)}{B-2} C_l^{(\frac{B-3}{2})}(\frac{-1}{B-1}) \right)$$

$$= \frac{l(B-3+l)(B-3+2l)}{(B-5+2l)(B-1+2l)} \left( C_l^{(\frac{B-3}{2})}(1) - \frac{2(B-1)}{B-2} C_l^{(\frac{B-3}{2})}(\frac{-1}{B-1}) \right)$$

$$u_{B-1,l} := \frac{B-1}{2} \left( c_{B-1,l+2}^{-1} \kappa_{-2} \left( l+2, \frac{B-3}{2} \right) \left( C_l^{(\frac{B-3}{2})}(1) - \frac{2(B-1)}{B-2} C_l^{(\frac{B-3}{2})}(\frac{-1}{B-1}) \right) \right.$$

$$\left. + c_{B-1,l}^{-1} \kappa_2 \left( l, \frac{B-3}{2} \right) \left( C_{l+2}^{(\frac{B-3}{2})}(1) - \frac{2(B-1)}{B-2} C_{l+2}^{(\frac{B-3}{2})}(\frac{-1}{B-1}) \right) \right)$$

$$= \frac{B-1}{2} \left( \frac{(B-3+l)(B-2+l)}{(B-3)(B-1+2l)} \left( C_l^{(\frac{B-3}{2})}(1) - \frac{2(B-1)}{B-2} C_l^{(\frac{B-3}{2})}(\frac{-1}{B-1}) \right) \right.$$

$$\left. + \frac{(l+1)(l+2)}{(B-3)(B-1+2l)} \left( C_{l+2}^{(\frac{B-3}{2})}(1) - \frac{2(B-1)}{B-2} C_{l+2}^{(\frac{B-3}{2})}(\frac{-1}{B-1}) \right) \right).$$

Note that $v_{B-1,0} = w_{B-1,0} = u_{B-1,0} = 0$, so that there is in fact no dependence on $a_0$, as expected since batchnorm is invariant under additive shifts of $\phi$.

We see that $\lambda_{\mathbb{L}}^{\uparrow} \geq 1$ iff

$$0 \leq \sum_{l=1}^{\infty} a_l^2 (w_{B-1,l} - v_{B-1,l}) + a_l a_{l+2} u_{B-1,l}.$$

This is a quadratic form on the coefficients $\{a_l\}_l$. We now analyze it heuristically and argue that the eigenvalue is $\geq 1$ typically when $\phi(\sqrt{B-1}x)$ explodes sufficiently as $x \to 1$ or $x \to -1$; in other words, the more explosive $\phi$ is, the less likely it is to induce Eq. (43) to converge to a BSB1 fixed point.

Heuristically, $C_{l+2}^{(\frac{B-3}{2})}(\frac{-1}{B-1})$ is negligible compared to $C_{l+2}^{(\frac{B-3}{2})}(1)$ for sufficiently large $B$ and $l$, so that $w_{B-1,l} - v_{B-1,l} \approx \left( \frac{l(B-3+l)(B-3)}{(B-5+2l)(B-1+2l)} - 1 \right) c_{B-1,l}^{-1} C_{l+2}^{(\frac{B-3}{2})}(1) = \left( \frac{B-7}{4} + O(B^{-2}) + O(l^{-2}) \right) c_{B-1,l}^{-1} C_{l+2}^{(\frac{B-3}{2})}(1)$ and is positive. For small $l$, we can calculate

$$w_{B-1,1} - v_{B-1,1} = \frac{-2B}{1+B} < 0$$

$$w_{B-1,2} - v_{B-1,2} = \frac{B(B+1)(B^2 - 9B + 12)}{2(B-1)(B+3)} \geq 0, \forall B \geq 10$$

$$w_{B-1,3} - v_{B-1,3} = \frac{(B-3)(B+3)B^2(2B^3 - 19B^2 + 16B + 13)}{6(B-1)^2(B+1)(B+5)} \geq 0, \forall B \geq 10.$$

In fact, plotting $w_{B-1,l} - v_{B-1,l}$ for various values of $l$ suggests that for $B \geq 10$, $w_{B-1,l} - v_{B-1,l} \geq 0$ for all $l \geq 1$ (Fig. 10).

Thus, the more "linear" $\phi$ is, the larger $a_1^2$ is compared to the rest of $\{a_l\}_l$, and the more likely that the eigenvalue is $< 1$. In the case that $a_l a_{l+2} = 0 \; \forall l$, then indeed the eigenvalue is $< 1$ precisely when $a_1^2$ is sufficiently large. Because higher degree Gegenbauer polynomials explodes more violently as $x \to \pm 1$, this is consistent with our claim.

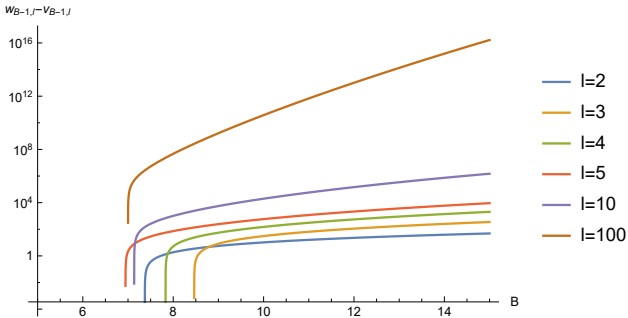

Figure 10: Plots of $w_{B-1,l} - v_{B-1,l}$ over $B$ for various $l$

**Eigenvalue for $\mathbb{M}$**  We use a similar technique shows the Gegenbauer expansion of the eigenvalue for $\mathbb{M}$.

Define $\tilde{\mathcal{A}}_{ab}(c,d) := \mathbb{E}_{v \sim S^{B-2}} \phi(\sqrt{B}\mathbb{e}_{a,:}v)\phi(\sqrt{B}\mathbb{e}_{b,:}v)(\mathbb{e}_{c,:}v)(\mathbb{e}_{d,:}v)$. Then $\tilde{\mathcal{A}}_{ab}(c,c) = \mathcal{A}(a,b;c)$. Note the ultrasymmetries $\tilde{\mathcal{A}}_{ab}(c,d) = \tilde{\mathcal{A}}_{\pi(a)\pi(b)}(\pi(c),\pi(d))$ for any permutation $\pi$, and $\tilde{\mathcal{A}}_{ab}(c,d) = \tilde{\mathcal{A}}_{ab}(d,c) = \tilde{\mathcal{A}}_{ba}(c,d) = \tilde{\mathcal{A}}_{ba}(d,c)$.

By Eq. (46), we have

$$\tilde{\mathcal{U}}\{\delta_c\delta_d^T\}_{ab}$$
$$= \mu^{*-1} \mathbb{E}_{v \sim S^{B-2}} \phi(\sqrt{B}\mathbb{e}_{a,:}v)\phi(\sqrt{B}\mathbb{e}_{b,:}v) \left( \frac{B-1}{2}(\mathbb{e}_{c,:}v)(\mathbb{e}_{d,:}v) - \frac{1}{2}\left( \mathbb{I}(c=d) - \frac{1}{B} \right) \right)$$
$$= \mu^{*-1}\frac{B-1}{2}\tilde{\mathcal{A}}(a,b;c,d) - \mu^{*-1}\frac{1}{2}\Sigma_{ab}^* \left( \mathbb{I}(c=d) - \frac{1}{B} \right).$$

By Thm E.62, $G^{\otimes 2} \circ \mathcal{U}$ has eigenspace $\mathbb{M}$ with eigenvalue

$$\lambda_{\mathbb{M}}^{\uparrow} = \tilde{\mathcal{U}}\{\delta_1\delta_2^T\}_{12} + \tilde{\mathcal{U}}\{\delta_2\delta_1^T\}_{12} - 4\tilde{\mathcal{U}}\{\delta_1\delta_3^T\}_{12} + 2\tilde{\mathcal{U}}\{\delta_3\delta_4^T\}_{12}$$
$$= 2(\tilde{\mathcal{U}}\{\delta_1\delta_2^T\}_{12} - 2\tilde{\mathcal{U}}\{\delta_1\delta_3^T\}_{12} + \tilde{\mathcal{U}}\{\delta_3\delta_4^T\}_{12})$$
$$= \frac{B-1}{\mu^*}(\tilde{\mathcal{A}}_{12}(12) - \tilde{\mathcal{A}}_{12}(13) + \tilde{\mathcal{A}}_{12}(34)).$$

Thus we need to evaluate $\tilde{\mathcal{A}}_{12}(12), \tilde{\mathcal{A}}_{12}(13), \tilde{\mathcal{A}}_{12}(34)$. We will do so by exploiting linear dependences between different values of $\tilde{\mathcal{A}}_{ab}(c,d)$, the computed values of $\tilde{\mathcal{A}}_{ab}(c,c) = \mathcal{A}(a,b;c)$, and the value of $\tilde{\mathcal{A}}_{ab}(a,b)$ computed below in Lemma F.23. Indeed, we have

$$\sum_{d=1}^{B} \tilde{\mathcal{A}}_{ab}(c,d) = \mathbb{E}_{v \sim S^{B-2}} \phi(\sqrt{B}\mathbb{e}_{a,:}v)\phi(\sqrt{B}\mathbb{e}_{b,:}v)(\mathbb{e}_{c,:}v)\sum_{d=1}^{B}(\mathbb{e}_{d,:}v) = 0.$$

Leveraging the symmmetries of $\tilde{\mathcal{A}}$, we get

$$(B-3)\tilde{\mathcal{A}}_{12}(34) + 2\tilde{\mathcal{A}}_{12}(13) + \tilde{\mathcal{A}}_{12}(33) = 0$$
$$(B-2)\tilde{\mathcal{A}}_{12}(13) + \tilde{\mathcal{A}}_{12}(11) + \tilde{\mathcal{A}}_{12}(12) = 0.$$

Expressing in terms of $\tilde{\mathcal{A}}_{12}(11)$ and $\tilde{\mathcal{A}}_{12}(12)$, we get

$$\tilde{\mathcal{A}}_{12}(13) = \frac{-1}{B-2}\left( \tilde{\mathcal{A}}_{12}(11) + \tilde{\mathcal{A}}_{12}(12) \right)$$

$$\tilde{\mathcal{A}}_{12}(34) = \frac{-1}{B-3}\left( 2\tilde{\mathcal{A}}_{12}(13) + \tilde{\mathcal{A}}_{12}(33) \right)$$
$$= \frac{-1}{B-3}\left( \frac{-2}{B-2}\left( \tilde{\mathcal{A}}_{12}(11) + \tilde{\mathcal{A}}_{12}(12) \right) + \frac{1}{B-2}\left( \Sigma_{12}^* - 2\tilde{\mathcal{A}}_{12}(22) \right) \right)$$
By Eq. (47)
$$= \frac{-1}{(B-3)(B-2)}\left( \Sigma_{12}^* - 4\tilde{\mathcal{A}}_{12}(11) - 2\tilde{\mathcal{A}}_{12}(12) \right).$$

These relations allow us to simplify

$$\lambda_{\mathbb{M}}^{\uparrow} = \frac{B-1}{(B-3)\mu^*} \left( \frac{-1}{B-2} \Sigma_{12}^* + \frac{2(B-1)}{B-2} \tilde{\mathcal{A}}_{12}(11) + (B-1)\tilde{\mathcal{A}}_{12}(12) \right).$$

By Eq. (49), we know the Gegenbauer expansion of $\tilde{\mathcal{A}}_{12}(11)$. The following shows the Gegenbauer expansion of $\tilde{\mathcal{A}}_{12}(12)$.

**Lemma F.23.** *Suppose* $\phi(\sqrt{B-1}x), x\phi(\sqrt{B-1}x) \in L^2((1-x^2)^{\frac{B-3}{2}})$, *and* $\phi(\sqrt{B-1}x)$ *has Gegenbauer expansion* $\sum_{l=0}^{\infty} a_l \frac{1}{c_{B-1,l}} C_l^{(\frac{B-3}{2})}(x)$. *Then*

$$\tilde{\mathcal{A}}_{12}(12) = \mathbb{E}_{v \sim S^{B-2}} \phi(\sqrt{B}\mathbb{e}_{1,:}v)\phi(\sqrt{B}\mathbb{e}_{2,:}v)(\mathbb{e}_{1,:}v)(\mathbb{e}_{2,:}v)$$

$$= \sum_{l=0}^{\infty} \alpha_l a_l^2 + \beta_l a_l a_{l+2}$$

*where*

$$\alpha_l = \frac{B-1}{B} \times \begin{cases} \left(\frac{l+1}{B-1+2l}\right)^2 c_{B-1,l+1}^{-1} C_{l+1}^{(\frac{B-3}{2})}\left(\frac{-1}{B-1}\right) + \left(\frac{B+l-4}{B+2l-5}\right)^2 c_{B-1,l-1}^{-1} C_{l-1}^{(\frac{B-3}{2})}\left(\frac{-1}{B-1}\right) & \text{if } l \geq 1 \\ \left(\frac{1}{B-1}\right)^2 c_{B-1,1}^{-1} C_1^{(\frac{B-3}{2})}\left(\frac{-1}{B-1}\right) & \text{otherwise} \end{cases}$$

$$\beta_l = \frac{B-1}{B} \times \frac{2(l+1)(B+l-2)}{(B+2l-1)^2} c_{B-1,l+1}^{-1} C_{l+1}^{(\frac{B-3}{2})}\left(\frac{-1}{B-1}\right).$$

*Proof.* Let $\psi(x) = x\phi(\sqrt{B-1}x)$. Then by Proposition E.43, we have

$$\psi(x) = \sum_{l=0}^{\infty} a_l c_{B-1,l}^{-1} \frac{1}{B-3+2l} \left( (l+1)C_{l+1}^{(\frac{B-3}{2})}(x) + (B+l-4)C_{l-1}^{(\frac{B-3}{2})}(x) \right)$$

$$= \sum_{l=0}^{\infty} \left( a_{l-1} \frac{l}{B+2l-3} + a_{l+1} \frac{B+l-3}{B+2l-3} \right) c_{B-1,l}^{-1} C_l^{(\frac{B-3}{2})}(x).$$

Then

$$\tilde{\mathcal{A}}_{12}(12) = \frac{B-1}{B} \mathbb{E}_{v \sim S^{B-2}} \psi(\hat{\mathbb{e}}_{1,:}v)\psi(\hat{\mathbb{e}}_{2,:}v)$$

$$= \frac{B-1}{B} \sum_{l=0}^{\infty} \left( a_{l-1} \frac{l}{B+2l-3} + a_{l+1} \frac{B+l-3}{B+2l-3} \right)^2 c_{B-1,l}^{-1} C_l^{(\frac{B-3}{2})}\left(\frac{-1}{B-1}\right).$$

Rearranging the sum in terms of $a_l$ gives the result. $\qquad\square$

Combining all of the above, we obtain the following

**Theorem F.24.** *Suppose* $\phi(\sqrt{B-1}x)$ *has Gegenbauer expansion* $\sum_{l=0}^{\infty} a_l \frac{1}{c_{B-1,l}} C_l^{(\frac{B-3}{2})}(x)$. *Then the eigenvalue of* $\mathcal{U}$ *with respect to the eigenspace* $\mathbb{M}$ *is a ratio of quadratic forms*

$$\lambda_{\mathbb{M}}^{\uparrow} = \frac{\sum_{l=0}^{\infty} a_l^2 \tilde{w}_{B-1,l} + a_l a_{l+2} \tilde{u}_{B-1,l}}{\sum_{l=0}^{\infty} a_l^2 v_{B-1,l}}$$

*where*

$$
\tilde{u}_{B-1,l} := \frac{2(B-1)^3}{(B-3)^2(B-2)B} \left( \frac{(B-3+l)(B-2+l)}{B-1+2l} C_l^{(\frac{B-3}{2})}\left(\frac{-1}{B-1}\right) \right.
$$
$$
+ \frac{(B-2)(l+1)(B-2+l)}{B-1+2l} C_{l+1}^{(\frac{B-3}{2})}\left(\frac{-1}{B-1}\right)
$$
$$
\left. + \frac{(l+3)(l+4)(B-3+2l)}{(B+2l+1)(B+2l+3)} C_{l+2}^{(\frac{B-3}{2})}\left(\frac{-1}{B-1}\right) \right)
$$
$$
\tilde{w}_{B-1,l} := \frac{B-1}{(B-3)^2(B-2)} \left[ \rule{0pt}{2.2em} \right.
$$
$$
\left( \frac{2(B-1)^2(B+2l-3)\left(2Bl+B+2l^2-6l-5\right)}{B(B+2l-5)(B+2l-1)} - (B-3+2l) \right) C_l^{(\frac{B-3}{2})}\left(\frac{-1}{B-1}\right)
$$
$$
+ \frac{(B-2)(B-1)^2(l+1)^2}{B(B+2l-1)} C_{l+1}^{(\frac{B-3}{2})}\left(\frac{-1}{B-1}\right)
$$
$$
\left. + \frac{(B-2)(B-1)^2(B-4+l)^2}{B(B+2l-5)} C_{l-1}^{(\frac{B-3}{2})}\left(\frac{-1}{B-1}\right) \right]
$$

*where $C_{-1}^{(\frac{B-3}{2})}(x) = 0$ by convention.*

Note that $\tilde{u}_{B-1,l}$ and $\tilde{w}_{B-1,l}$ depends only on $C_\bullet^{(\frac{B-3}{2})}\left(\frac{-1}{B-1}\right)$ which is much smaller than $C_\bullet^{(\frac{B-3}{2})}(1)$ for degree larger than 0. Thus the eigenvalue for $\mathbb{M}$ is typically much smaller than the eigenvalue for $\mathbb{L}$ (though there are counterexamples, like sin).

### F.3.3 LAPLACE METHOD

**Differentiating Eq. (26)** In what follows, let $\phi$ be a positive-homogeneous function of degree $\alpha$. We begin by studying $\frac{\mathrm{dV}_{(\phi \circ n)}(\Sigma^G)}{\mathrm{d}\Sigma^G}$. We will differentiate Eq. (26) directly at $G^{\otimes 2}\{\Sigma^*\} = G\Sigma^*G$ where $\Sigma^*$ is the BSB1 fixed point given by Thm F.8. To that end, consider a smooth path $\Sigma_t \in \mathcal{S}_B^G, t \in (-\epsilon, \epsilon)$ for some $\epsilon > 0$, with $\Sigma_0 = G\Sigma^*G$. Set $\Sigma_{\square t} = \mathbb{e}^T \Sigma_t \mathbb{e} \in \mathcal{S}_{B-1}$, so that $\Sigma_t = \mathbb{e}\Sigma_{\square t}\mathbb{e}^T$ and $\Sigma_{\square 0} = \mu^* I_{B-1}$ where $\mu^* = K_{\alpha,B}(\mathrm{J}_\phi(1) - \mathrm{J}_\phi(\frac{-1}{B-1}))$ as in Thm F.8. If we write $\dot{\Sigma}_\square$ and $\dot{\Sigma}$ for the time derivatives, we have

$$B^{-\alpha}\Gamma(\alpha)\frac{d}{dt}\mathrm{V}_{(\phi\mathrm{on})}(\mathbb{e}\Sigma_\square t\mathbb{e}^T)\bigg|_{t=0}$$

$$= B^{-\alpha}\Gamma(\alpha)\frac{d}{dt}\mathrm{V}_{\mathcal{B}_\phi}(\mathbb{e}\Sigma_\square t\mathbb{e}^T)\bigg|_{t=0}$$

$$= \int_0^\infty ds\, s^{\alpha-1}\left[\frac{d}{dt}\det(I+2s\Sigma_{\square t})^{-1/2}\bigg|_{t=0} \times \mathrm{V}_\phi\left(\mathbb{e}\Sigma_{\square 0}(I+2s\Sigma_{\square 0})^{-1}\mathbb{e}^T\right)\right.$$

$$\left.+\det(I+2s\Sigma_{\square 0})^{-1/2}\frac{d}{dt}\mathrm{V}_\phi(\mathbb{e}\Sigma_{\square t}(I+2s\Sigma_{\square t})^{-1}\mathbb{e}^T)\bigg|_{t=0}\right]$$

$$= \int_0^\infty ds\, s^{\alpha-1}\left[-\frac{s}{1+2s\mu^*}\det(I+2s\Sigma_{\square 0})^{-1/2}\mathrm{tr}\left(\dot\Sigma_{\square 0}\right)\times\left(\frac{\mu^*}{1+2s\mu^*}\right)^\alpha\mathrm{V}_\phi(G)\right.$$

$$+\det(I+2s\Sigma_{\square 0})^{-1/2}\frac{d\mathrm{V}_\phi(\Sigma)}{d\Sigma}\bigg|_{\Sigma=\mathbb{e}\Sigma_{\square 0}(I+2s\Sigma_{\square 0})^{-1}\mathbb{e}^T}$$

$$\left.\left\{\mathbb{e}(I+2s\Sigma_{\square 0})^{-1}\dot\Sigma_{\square 0}(I+2s\Sigma_{\square 0})^{-1}\mathbb{e}^T\right\}\right]$$

(apply Lemma F.26 and Lemma F.27)

$$= \int_0^\infty ds\, s^{\alpha-1}\left[-\frac{s}{1+2s\mu^*}(1+2s\mu^*)^{-(B-1)/2}\mathrm{tr}\left(\dot\Sigma_{\square 0}\right)\times\left(\frac{\mu^*}{1+2s\mu^*}\right)^\alpha\mathrm{V}_\phi(G)\right.$$

$$\left.+(1+2s\mu^*)^{-(B-1)/2}\left(\frac{\mu^*}{1+2s\mu^*}\right)^{\alpha-1}\frac{d\mathrm{V}_\phi(\Sigma)}{d\Sigma}\bigg|_{\Sigma=G}\left\{(1+2s\mu^*)^{-2}\mathbb{e}\dot\Sigma_{\square 0}\mathbb{e}^T\right\}\right]$$

(using fact that $d\mathrm{V}_\phi/d\Sigma$ is degree $(\alpha-1)$-positive homogeneous)

$$= -\left(\int_0^\infty(1+2s\mu^*)^{-\frac{B-1}{2}-1-\alpha}\mu^{*\alpha}s^\alpha\right)\mathrm{tr}\left(\dot\Sigma_{\square 0}\right)\mathrm{V}_\phi(G)$$

$$+\left(\int_0^\infty(1+2s\mu^*)^{-\frac{B-1}{2}-1-\alpha}\mu^{*\alpha-1}s^{\alpha-1}\right)\frac{d\mathrm{V}_\phi(\Sigma)}{d\Sigma}\bigg|_{\Sigma=G}\left\{\mathbb{e}\dot\Sigma_{\square 0}\mathbb{e}^T\right\}$$

$$= -\mu^{*-1}2^{-1-\alpha}\mathrm{Beta}\left(\frac{B-1}{2},\alpha+1\right)\mathrm{tr}\left(\dot\Sigma_{\square 0}\right)\mathrm{V}_\phi(G)$$

$$+\mu^{*-1}2^{-\alpha}\mathrm{Beta}\left(\frac{B+1}{2},\alpha\right)\frac{d\mathrm{V}_\phi(\Sigma)}{d\Sigma}\bigg|_{\Sigma=G}\left\{\dot\Sigma_0\right\}$$

(apply Lemma F.28)

if $\frac{B-1}{2}+2>\alpha$ (precondition for Lemma F.28).

With some trivial simplifications, we obtain the following

**Lemma F.25.** *Let $\phi$ be positive-homogeneous of degree $\alpha$. Consider a smooth path $\Sigma_t\in\mathcal{S}_B^G$ with $\Sigma_0=\mu^*G$. If $\frac{B-1}{2}+2>\alpha$, then*

$$\frac{d}{dt}\mathrm{V}_{(\phi\mathrm{on})}(\Sigma_t)\bigg|_{t=0}$$

$$= \frac{d\mathrm{V}_{(\phi\mathrm{on})}(\Sigma)}{d\Sigma}\bigg|_{\Sigma=\mu^*G}\left\{\dot\Sigma_0\right\}$$

$$= -\alpha B^\alpha\mu^{*-1}2^{-1-\alpha}\mathrm{P}\left(\frac{B-1}{2},\alpha+1\right)^{-1}\mathrm{tr}\left(\dot\Sigma_{\square 0}\right)\mathrm{V}_\phi(G)$$

$$+B^\alpha\mu^{*-1}2^{-\alpha}\mathrm{P}\left(\frac{B+1}{2},\alpha\right)^{-1}\frac{d\mathrm{V}_\phi(\Sigma)}{d\Sigma}\bigg|_{\Sigma=G}\left\{\dot\Sigma_0\right\} \tag{50}$$

*where $\mathrm{P}(a,b)=\Gamma(a+b)/\Gamma(a)$ is the Pochhammer symbol.*

**Lemma F.26.** *For any $s\in\mathbb{R}$, $\frac{d}{dt}\det(I+2s\Sigma_\square)^{-1/2} = -s\det(I+2s\Sigma_\square)^{-1/2}\mathrm{tr}((I+2s\Sigma_\square)^{-1}d\Sigma_\square/dt)$.*

*For $\Sigma_{\square 0} = \mu I$, this is also equal to $-\frac{s}{1+2s\mu} \det(I + 2s\Sigma_{\square})^{-1/2} \operatorname{tr}(\frac{d}{dt}\Sigma_{\square})$ at $t = 0$.*

*Proof.* Straightforward computation. $\qquad\square$

**Lemma F.27.** *For any $s \in \mathbb{R}$, $\frac{d}{dt}\Sigma_{\square}(I + 2s\Sigma_{\square})^{-1} = (I + 2s\Sigma_{\square})\frac{d\Sigma_{\square}}{dt}(I + 2s\Sigma_{\square})^{-1}$.*

*Proof.* Straightforward computation. $\qquad\square$

**Lemma F.28.** *For $a > b + 1$, $\int_0^\infty (1 + 2s\mu)^{-a} s^b \, ds = (2\mu)^{-1-b} \operatorname{Beta}(b + 1, a - 1 - b)$.*

*Proof.* Apply change of variables $x = \frac{2\mu s}{1+2\mu s}$. $\qquad\square$

This immediately gives the following consequence.

**Theorem F.29.** *Let $\phi : \mathbb{R} \to \mathbb{R}$ be any function with finite first and second Gaussian moments. Then $G^{\otimes 2} \circ \frac{dV_\phi}{d\Sigma}\big|_{\Sigma = \mathrm{BSB1}(a,b)} : \mathcal{H}_B^G \to \mathcal{H}_B^G$ has the following eigendecomposition:*

- *$\mathbb{R}G$ has eigenvalue $\frac{(B-1)(u-2v)+w}{B}$ ($\dim \mathbb{R}G = 1$)*

- *$\mathbb{M}_B$ has eigenvalue $w$ ($\dim \mathbb{M}_B = B(B-3)/2$)*

- *$\mathbb{L}_B$ has eigenvalue $\frac{(B-2)(u-2v)+2w}{B}$ ($\dim \mathbb{L}_B = B - 1$)*

*where*

$$
u = \frac{\partial V_\phi(\Sigma)_{11}}{\partial \Sigma_{11}}\bigg|_{\Sigma = \mathrm{BSB1}(a,b)}, \quad v = \frac{\partial V_\phi(\Sigma)_{12}}{\partial \Sigma_{11}}\bigg|_{\Sigma = \mathrm{BSB1}(a,b)}, \quad w = \frac{\partial V_\phi(\Sigma)_{12}}{\partial \Sigma_{12}}\bigg|_{\Sigma = \mathrm{BSB1}(a,b)}.
$$

**Theorem F.30.** *Let $\phi : \mathbb{R} \to \mathbb{R}$ be positive-homogeneous of degree $\alpha$. Then for any $p \neq 0, c \in \mathbb{R}$, $G^{\otimes 2} \circ \frac{dV_\phi}{d\Sigma}\big|_{\Sigma = \mathrm{BSB1}(p,cp)} : \mathcal{H}_B^G \to \mathcal{H}_B^G$ has the following eigendecomposition:*

- *$\mathbb{M}_B$ has eigenvalue $\mathsf{c}_\alpha p^{\alpha-1} \mathrm{J}'_\phi(c)$*

- *$\mathbb{L}_B$ has eigenvalue $\mathsf{c}_\alpha p^{\alpha-1} \frac{1}{B}\left((B-2)\alpha \left[\mathrm{J}_\phi(1) - \mathrm{J}_\phi(c)\right] + (2 + c(B-2))\mathrm{J}'_\phi(c)\right)$*

- *$\mathbb{R}G$ has eigenvalue $\mathsf{c}_\alpha p^{\alpha-1} \frac{1}{B}\left((B-1)\alpha \left[\mathrm{J}_\phi(1) - \mathrm{J}_\phi(c)\right] + (1 + c(B-1))\mathrm{J}'_\phi(c)\right)$*

*Proof.* By Proposition E.22, $\frac{dV_\phi}{d\Sigma}\big|_{\Sigma = \mathrm{BSB1}(p,cp)}$ is $\mathrm{DOS}(u,v,w)$ with

$$
\begin{aligned}
u &= \frac{\partial V_\phi(\Sigma)_{11}}{\partial \Sigma_{11}}\bigg|_{\Sigma = \mathrm{BSB1}(p,cp)} \\
&= \mathsf{c}_\alpha \alpha p^{\alpha-1} \mathrm{J}_\phi(1) \\
v &= \frac{\partial V_\phi(\Sigma)_{12}}{\partial \Sigma_{11}}\bigg|_{\Sigma = \mathrm{BSB1}(p,cp)} \\
&= \frac{1}{2}\mathsf{c}_\alpha \Sigma_{ii}^{\frac{1}{2}(\alpha-2)} \Sigma_{jj}^{\frac{1}{2}\alpha} (\alpha \mathrm{J}_\phi(c_{ij}) - c_{ij}\mathrm{J}'_\phi(c_{ij})) \\
&= \frac{1}{2}\mathsf{c}_\alpha p^{\alpha-1}\left(\alpha \mathrm{J}_\phi(c) - c\mathrm{J}'_\phi(c)\right) \\
w &= \frac{\partial V_\phi(\Sigma)_{12}}{\partial \Sigma_{12}}\bigg|_{\Sigma = \mathrm{BSB1}(p,cp)} \\
&= \mathsf{c}_\alpha \Sigma_{ii}^{(\alpha-1)/2} \Sigma_{jj}^{(\alpha-1)/2} \mathrm{J}'_\phi(c_{ij}) \\
&= \mathsf{c}_\alpha p^{\alpha-1} \mathrm{J}'_\phi(c)
\end{aligned}
$$

With Thm E.73, we can do the computation:

- $\mathbb{M}_B$ has eigenvalue $w = \mathsf{c}_\alpha p^{\alpha-1}\mathrm{J}'_\phi(c)$

- $\mathbb{L}_B$ has eigenvalue

$$\frac{(u-2v)(B-2)+2w}{B}$$

$$= \frac{1}{B}\left((B-2)\left(\mathsf{c}_\alpha\alpha p^{\alpha-1}\mathrm{J}_\phi(1) - \mathsf{c}_\alpha p^{\alpha-1}\left(\alpha\mathrm{J}_\phi(c) - c\mathrm{J}'_\phi(c)\right)\right) + 2\mathsf{c}_\alpha p^{\alpha-1}\mathrm{J}'_\phi(c)\right)$$

$$= \mathsf{c}_\alpha p^{\alpha-1}\frac{1}{B}\left((B-2)\alpha\left[\mathrm{J}_\phi(1) - \mathrm{J}_\phi(c)\right] + (2 + c(B-2))\mathrm{J}'_\phi(c)\right)$$

- $\mathbb{R}G$ has eigenvalue

$$\frac{(B-1)(u-2v)+w}{B}$$

$$= \frac{1}{B}\left((B-1)\left(\mathsf{c}_\alpha\alpha p^{\alpha-1}\mathrm{J}_\phi(1) - \mathsf{c}_\alpha p^{\alpha-1}\left(\alpha\mathrm{J}_\phi(c) - c\mathrm{J}'_\phi(c)\right)\right) + \mathsf{c}_\alpha p^{\alpha-1}\mathrm{J}'_\phi(c)\right)$$

$$= \mathsf{c}_\alpha p^{\alpha-1}\frac{1}{B}\left((B-1)\alpha\left[\mathrm{J}_\phi(1) - \mathrm{J}_\phi(c)\right] + (1 + c(B-1))\mathrm{J}'_\phi(c)\right)$$

$\square$

We record the following consequence which will be used frequently in the sequel.

**Theorem F.31.** *Let $\phi : \mathbb{R} \to \mathbb{R}$ be positive-homogeneous of degree $\alpha$. Then $G^{\otimes 2} \circ \frac{\mathrm{dV}_\phi}{\mathrm{d\Sigma}}\big|_{\Sigma=G}$ : $\mathcal{H}_B^G \to \mathcal{H}_B^G$ has the following eigendecomposition:*

- $\mathbb{M}_B$ *has eigenvalue* $\lambda_{\mathbb{M}}^{G,\phi}(B,\alpha) := \mathsf{c}_\alpha\left(\frac{B-1}{B}\right)^{\alpha-1}\mathrm{J}'_\phi\left(\frac{-1}{B-1}\right)$

- $\mathbb{L}_B$ *has eigenvalue* $\lambda_{\mathbb{L}}^{G,\phi}(B,\alpha) := \mathsf{c}_\alpha\left(\frac{B-1}{B}\right)^{\alpha-1}\frac{1}{B}\left((B-2)\alpha\left[\mathrm{J}_\phi(1) - \mathrm{J}_\phi\left(\frac{-1}{B-1}\right)\right] + \frac{B}{B-1}\mathrm{J}'_\phi\left(\frac{-1}{B-1}\right)\right)$

- $\mathbb{R}G$ *has eigenvalue* $\lambda_G^{G,\phi}(B,\alpha) := \mathsf{c}_\alpha\left(\frac{B-1}{B}\right)^{\alpha}\alpha\left[\mathrm{J}_\phi(1) - \mathrm{J}_\phi\left(\frac{-1}{B-1}\right)\right]$

*Proof.* Plug in $p = \frac{B-1}{B}$ and $c = -1/(B-1)$ for Thm F.30. $\square$

**Theorem F.32.** *Let $\phi : \mathbb{R} \to \mathbb{R}$ be a degree $\alpha$ positive-homogeneous function. Then for $p \neq 0, c \in \mathbb{R}$, $\frac{\mathrm{dV}_\phi}{\mathrm{d\Sigma}}\big|_{\Sigma=\mathrm{BSB1}(p,cp)} : \mathcal{H}_B \to \mathcal{H}_B$ has the following eigendecomposition:*

- $\overline{\mathbb{M}}_B$ *has eigenvalue* $\mathsf{c}_\alpha p^{\alpha-1}\mathrm{J}'_\phi(c)$

- $\overline{\mathbb{L}}_B(a,b)$ *has eigenvalue* $\mathsf{c}_\alpha\alpha p^{\alpha-1}\mathrm{J}_\phi(1)$ *where* $a = 2\left(\mathrm{J}'_\phi(c) - \alpha\mathrm{J}_\phi(1)\right)$ *and* $b = \alpha\mathrm{J}_\phi(c) - c\mathrm{J}'_\phi(c)$

*Proof.* Use Thm E.72 with the computations from the proof of Thm F.30 as well as the following computation

$$w - u = \mathsf{c}_\alpha p^{\alpha-1}\mathrm{J}'_\phi(c) - \mathsf{c}_\alpha\alpha p^{\alpha-1}\mathrm{J}_\phi(1)$$

$$= \mathsf{c}_\alpha p^{\alpha-1}\left(\mathrm{J}'_\phi(c) - \alpha\mathrm{J}_\phi(1)\right)$$

$$v = \frac{1}{2}\mathsf{c}_\alpha p^{\alpha-1}\left(\alpha\mathrm{J}_\phi(c) - c\mathrm{J}'_\phi(c)\right)$$

$\square$

**Theorem F.33.** *Let $\phi$ be positive-homogeneous with degree $\alpha$. Assume $\frac{B-1}{2} > \alpha$. The operator $\mathcal{U} = G^{\otimes 2} \circ \frac{\mathrm{dV}_{(\phi \circ n)}(\Sigma^G)}{\mathrm{d\Sigma^G}}\big|_{\Sigma^G=\mu^* G} : \mathcal{H}_B^G \to \mathcal{H}_B^G$ has 3 distinct eigenvalues. They are as follows:*

1. $\lambda_G^\uparrow(B,\alpha) := 0$ *with dimension 1 eigenspace* $\mathbb{R}G$.

2.

$$\lambda_{\mathbb{M}}^\uparrow(B,\alpha) := B^\alpha \mu^{*-1} 2^{-\alpha} \mathrm{P}\left(\frac{B+1}{2},\alpha\right)^{-1} \lambda_{\mathbb{M}}^{G,\phi}(B,\alpha)$$

$$= \mathsf{c}_\alpha B(B-1)^{\alpha-1} \mu^{*-1} 2^{-\alpha} \mathrm{P}\left(\frac{B+1}{2},\alpha\right)^{-1} \mathrm{J}_\phi'\left(\frac{-1}{B-1}\right)$$

*with dimension* $\frac{B(B-3)}{2}$ *eigenspace* $\mathbb{M}$.

3.

$$\lambda_{\mathbb{L}}^\uparrow(B,\alpha) := B^\alpha \mu^{*-1} 2^{-\alpha} \mathrm{P}\left(\frac{B+1}{2},\alpha\right)^{-1} \lambda_{\mathbb{L}}^{G,\phi}(B,\alpha)$$

$$= \mathsf{c}_\alpha (B-1)^{\alpha-1} \mu^{*-1} 2^{-\alpha} \mathrm{P}\left(\frac{B+1}{2},\alpha\right)^{-1}$$

$$\left((B-2)\alpha\left[\mathrm{J}_\phi(1) - \mathrm{J}_\phi\left(\frac{-1}{B-1}\right)\right] + \frac{B}{B-1}\mathrm{J}_\phi'\left(\frac{-1}{B-1}\right)\right)$$

*with dimension* $B-1$ *eigenspace* $\mathbb{L}$.

*Proof.* **Item 1.** **The case of** $\lambda_G^\uparrow(B,\alpha)$. Since $(\phi \circ \mathsf{n})(\mu^*G) = (\phi \circ \mathsf{n})(\mu^*G + \omega G)$ for any $\omega$, the operator $\frac{\mathrm{dV}_{(\phi \circ \mathsf{n})}(\Sigma^G)}{\mathrm{d}\Sigma^G}\Big|_{\Sigma^G = \mu^*G}$ sends $G$ to 0.

**Item 2.** **The case of** $\lambda_{\mathbb{M}}^\uparrow(B,\alpha)$. Assume $\dot\Sigma_0 \in \mathbb{M}$, Thm F.31 gives (with $\odot$ denoting Hadamard product, i.e. entrywise multiplication)

$$\frac{\mathrm{dV}_\phi(\Sigma)}{\mathrm{d}\Sigma}\bigg|_{\Sigma=G} \{\dot\Sigma_0\} = \lambda_{\mathbb{M}}^{G,\phi}(B,\alpha)\dot\Sigma_0$$

$$= \mathsf{c}_\alpha \left(\frac{B-1}{B}\right)^{\alpha-1} \mathrm{J}_\phi'\left(\frac{-1}{B-1}\right)\dot\Sigma_0$$

Since $\mathrm{tr}(\dot\Sigma_0) = 0$, Eq. (50) gives

$$G^{\otimes 2} \circ \frac{\mathrm{dV}_{(\phi \circ \mathsf{n})}(\Sigma^G)}{\mathrm{d}\Sigma^G}\bigg|_{\Sigma^G = \mu^*G} \{\dot\Sigma_0\}$$

$$= G^{\otimes 2} \circ \frac{d}{dt}\mathrm{V}_{(\phi \circ \mathsf{n})}(\Sigma_t)\bigg|_{t=0} \{\dot\Sigma_0\}$$

$$= B^\alpha \mu^{*-1} 2^{-\alpha} \mathrm{P}\left(\frac{B+1}{2},\alpha\right)^{-1} G^{\otimes 2} \circ \frac{\mathrm{dV}_\phi(\Sigma)}{\mathrm{d}\Sigma}\bigg|_{\Sigma=G}\{\dot\Sigma_0\}$$

$$= B^\alpha \mu^{*-1} 2^{-\alpha} \mathrm{P}\left(\frac{B+1}{2},\alpha\right)^{-1} \mathsf{c}_\alpha\left(\frac{B-1}{B}\right)^{\alpha-1} \mathrm{J}_\phi'\left(\frac{-1}{B-1}\right) G^{\otimes 2}\{\dot\Sigma_0\}$$

$$= \mathsf{c}_\alpha B(B-1)^{\alpha-1} \mu^{*-1} 2^{-\alpha} \mathrm{P}\left(\frac{B+1}{2},\alpha\right)^{-1} \mathrm{J}_\phi'\left(\frac{-1}{B-1}\right)\dot\Sigma_0$$

So the eigenvalue for $\mathbb{M}$ is $\lambda_{\mathbb{L}}^\uparrow = \mathsf{c}_\alpha B(B-1)^{\alpha-1} \mu^{*-1} 2^{-\alpha} \mathrm{P}\left(\frac{B+1}{2},\alpha\right)^{-1} \mathrm{J}_\phi'\left(\frac{-1}{B-1}\right)$.

**Item 3.** **The case of** $\lambda_{\mathbb{L}}^\uparrow(B,\alpha)$. Let $\dot\Sigma_0 = \mathrm{L}(B-2,1)$. Thm F.31 gives

$$G^{\otimes 2} \circ \frac{\mathrm{dV}_\phi(\Sigma)}{\mathrm{d}\Sigma}\bigg|_{\Sigma=G}\{\dot\Sigma_0\} = \lambda_{\mathbb{L}}^{G,\phi}(B,\alpha)\dot\Sigma_0$$

where

$$\lambda_{\mathbb{L}}^{G,\phi}(B,\alpha) := \frac{1}{B}\mathsf{c}_\alpha \left(\frac{B-1}{B}\right)^{\alpha-1} \left((B-2)\alpha \left[\mathrm{J}_\phi(1) - \mathrm{J}_\phi\left(\frac{-1}{B-1}\right)\right] + \frac{B}{B-1}\mathrm{J}_\phi'\left(\frac{-1}{B-1}\right)\right)$$

Since $\mathrm{tr}(\dot\Sigma_0) = 0$, Eq. (50) gives

$$G^{\otimes 2} \circ \left.\frac{\mathrm{d}V_{(\phi\mathrm{on})}(\Sigma^G)}{\mathrm{d}\Sigma^G}\right|_{\Sigma^G = \mu^* G} \{\dot\Sigma_0\}$$

$$= G^{\otimes 2} \circ \left.\frac{d}{dt}V_{(\phi\mathrm{on})}(\Sigma_t)\right|_{t=0}$$

$$= B^\alpha \mu^{*-1} 2^{-\alpha} \mathrm{P}\left(\frac{B+1}{2},\alpha\right)^{-1} G^{\otimes 2} \circ \left.\frac{\mathrm{d}V_\phi(\Sigma)}{\mathrm{d}\Sigma}\right|_{\Sigma=G} \{\dot\Sigma_0\}$$

$$= B^\alpha \mu^{*-1} 2^{-\alpha} \mathrm{P}\left(\frac{B+1}{2},\alpha\right)^{-1} \tilde\lambda \dot\Sigma_0$$

$$= (B-1)^{\alpha-1}\mathsf{c}_\alpha\mu^{*-1}2^{-\alpha}\mathrm{P}\left(\frac{B+1}{2},\alpha\right)^{-1}$$

$$\times \left((B-2)\alpha\left[\mathrm{J}_\phi(1) - \mathrm{J}_\phi\left(\frac{-1}{B-1}\right)\right] + \frac{B}{B-1}\mathrm{J}_\phi'\left(\frac{-1}{B-1}\right)\right)$$

$$\times \dot\Sigma_0$$

So $\lambda_{\mathbb{L}}^{\uparrow} = (B-1)^{\alpha-1}\mathsf{c}_\alpha\mu^{*-1}2^{-\alpha}\mathrm{P}\left(\frac{B+1}{2},\alpha\right)^{-1}\left((B-2)\alpha\left[\mathrm{J}_\phi(1) - \mathrm{J}_\phi\left(\frac{-1}{B-1}\right)\right] + \frac{B}{B-1}\mathrm{J}_\phi'\left(\frac{-1}{B-1}\right)\right)$

$\square$

With some routine computation, we obtain

**Proposition F.34.** *With $\phi$ and $\alpha$ as above, as $B \to \infty$,*

$$\lambda_{\mathbb{L}}^{\uparrow}(B,\alpha) \sim \alpha + B^{-1}(2\alpha^3 - 4\alpha^2 - \alpha + \frac{\mathrm{J}_\phi'(0)}{\mathrm{J}_\phi(1) - \mathrm{J}_\phi(0)}) + O(B^{-2})$$

$$\lambda_{\mathbb{M}}^{\uparrow}(B,\alpha) \sim \frac{\mathrm{J}_\phi'(0)}{\mathrm{J}_\phi(1) - \mathrm{J}_\phi(0)}$$

$$+ B^{-1}\left((2\alpha^2 - 4\alpha + 1)\frac{\mathrm{J}_\phi'(0)}{\mathrm{J}_\phi(1) - \mathrm{J}_\phi(0)} - \frac{\mathrm{J}_\phi''(0)}{\mathrm{J}_\phi(1) - \mathrm{J}_\phi(0)} - \left(\frac{\mathrm{J}_\phi'(0)}{\mathrm{J}_\phi(1) - \mathrm{J}_\phi(0)}\right)^2\right)$$

$$+ O\left(B^{-2}\right)$$

*We can compute, by Proposition E.23,*

$$\frac{\mathrm{J}_\phi'(0)}{\mathrm{J}_\phi(1) - \mathrm{J}_\phi(0)} = \frac{(a+b)^2\frac{1}{\sqrt\pi}\frac{\Gamma(\frac{\alpha}{2}+1)^2}{\Gamma(\alpha+\frac{1}{2})}}{(a^2+b^2) - (a-b)^2\frac{1}{2\sqrt\pi}\frac{\Gamma(\frac{\alpha}{2}+\frac{1}{2})^2}{\Gamma(\alpha+\frac{1}{2})}}$$

$$\leq \frac{2\Gamma\left(\frac{\alpha}{2}+1\right)^2}{\sqrt\pi\Gamma\left(\alpha+\frac{1}{2}\right)}$$

*where the last part is obtained by optimizing over $a$ and $b$. On $\alpha \in (-1/2,\infty)$, this is greater than $\alpha$ iff $\alpha < 1$. Thus for $\alpha \geq 1$, the maximum eigenvalue of $\mathcal{U}$ is always achieved by eigenspace $\mathbb{L}$ for large enough $B$.*

## G  BACKWARD DYNAMICS

Let's extend the definition of the V operator:

**Definition G.1.** Suppose $T_x : \mathbb{R}^n \to \mathbb{R}^m$ is a linear operator parametrized by $x \in \mathbb{R}^k$. Then for a PSD matrix $\Sigma \in \mathcal{S}_k$, define $\mathrm{V}_T(\Sigma) := \mathbb{E}[T_x \otimes T_x : x \in \mathcal{N}(0, \Sigma)] : \mathbb{R}^{n \times n} \to \mathbb{R}^{m \times m}$, which acts on $n \times n$ matrices $\Pi$ by

$$\mathrm{V}_T(\Sigma)\{\Pi\} = \mathbb{E}[T_x \Pi T_x^T : x \in \mathcal{N}(0, \Sigma)] \in \mathbb{R}^{m \times m}.$$

Under this definition, $\mathrm{V}_{\mathcal{B}'_\phi}(\Sigma)$ is a linear operator $\mathbb{R}^{B \times B} \to \mathbb{R}^{B \times B}$. Recall the notion of adjoint:

**Definition G.2.** If $V$ is a (real) vector space, then $V^\dagger$, its *dual space*, is defined as the space of linear functionals $f : V \to \mathbb{R}$ on $V$. If $T : V \to W$ is a linear operator, then $T^\dagger$, its *adjoint*, is a linear operator $W^\dagger \to V^\dagger$, defined by $T^\dagger(f) = v \mapsto f(T(v))$.

If a linear operator is represented by a matrix, with function application represented by matrix-vector multiplication (matrix on the left, vector on the right), then the adjoint is represented by matrix transpose.

**The backward equation.** In this section we are interested in the *backward dynamics*, given by the following equation

$$\Pi^l = \mathrm{V}_{\mathcal{B}'_\phi}(\Sigma^l)^\dagger\{\Pi^{l+1}\}$$

where $\Sigma^l$ is given by the forward dynamics. Particularly, we are interested in the specific case when we have exponential convergence of $\Sigma^l$ to a BSB1 fixed point. Thus we will study the asymptotic approximation of the above, namely the linear system.

$$\Pi^l = \mathrm{V}_{\mathcal{B}'_\phi}(\Sigma^*)^\dagger\{\Pi^{l+1}\} \tag{51}$$

where $\Sigma^*$ is the BSB1 fixed point. Note that after one step of backprop, $\Pi^{L-1} = \mathrm{V}_{\mathcal{B}'_\phi}(\Sigma^*)^\dagger\{\Pi^L\}$ is in $\mathcal{H}_B^G$. Thus the large $L$ dynamics of Eq. (51) is given by the eigendecomposition of $\mathrm{V}_{\mathcal{B}'_\phi}(\Sigma^*)^\dagger \upharpoonright \mathcal{H}_B^G : \mathcal{H}_B^G \to \mathcal{H}_B^G$. It turns out to be much more convenient to study its adjoint $G^{\otimes 2} \circ \mathrm{V}_{\mathcal{B}'_\phi}(\Sigma^*)$, which has the same eigenvalues (in fact, it will turn out that it is self-adjoint).

## G.1 Eigendecomposition of $G^{\otimes 2} \circ \mathrm{V}_{\mathcal{B}'_\phi}(\Sigma^*)$.

By chain rule, $\mathcal{B}'_\phi(x) = \frac{\mathrm{d}\phi \circ \mathsf{n}(z)}{\mathrm{d}z}\big|_{z=Gx} \frac{\mathrm{d}Gx}{\mathrm{d}x} = \frac{\mathrm{d}\phi \circ \mathsf{n}(z)}{\mathrm{d}z}\big|_{z=Gx} G, \mathcal{B}'_\phi(x)\Delta x = \frac{\mathrm{d}\phi \circ \mathsf{n}(z)}{\mathrm{d}z}\big|_{z=Gx} G\Delta x$
Thus

$$\begin{aligned}
\mathrm{V}_{\mathcal{B}'_\phi}(\Sigma) &= \mathbb{E}\left[\mathcal{B}'_\phi(x)^{\otimes 2} : x \sim \mathcal{N}(0, \Sigma)\right] \\
&= \mathbb{E}\left[\left(\frac{\mathrm{d}\phi \circ \mathsf{n}(z)}{\mathrm{d}z}\bigg|_{z=Gx} G\right)^{\otimes 2} : x \sim \mathcal{N}(0, \Sigma)\right] \\
&= \mathbb{E}\left[\frac{\mathrm{d}\phi \circ \mathsf{n}(z)}{\mathrm{d}z}\bigg|_{z=Gx}^{\otimes 2} \circ G^{\otimes 2} : x \sim \mathcal{N}(0, \Sigma)\right] \\
&= \mathbb{E}\left[\frac{\mathrm{d}\phi \circ \mathsf{n}(z)}{\mathrm{d}z}\bigg|_{z=Gx}^{\otimes 2} : x \sim \mathcal{N}(0, \Sigma)\right] \circ G^{\otimes 2} \\
&= \mathrm{V}_{\left[x \mapsto \frac{\mathrm{d}\phi \circ \mathsf{n}(z)}{\mathrm{d}z}\big|_{z=Gx}\right]}(\Sigma) \circ G^{\otimes 2}
\end{aligned}$$

If we let $\mathcal{F}(\Sigma) := \mathrm{V}_{\left[x \mapsto \frac{\mathrm{d}\phi \circ \mathsf{n}(z)}{\mathrm{d}z}\big|_{z=Gx}\right]}(\Sigma)$, then $\mathrm{V}_{\mathcal{B}'_\phi}(\Sigma) = \mathcal{F}(\Sigma) \circ G^{\otimes 2}$. As discussed above, we will seek the eigendecomposition of $G^{\otimes 2} \circ \mathcal{F}(\Sigma) \upharpoonright \mathcal{H}_B^G : \mathcal{H}_B^G \to \mathcal{H}_B^G$.

We first make some basic calculations.
**Proposition G.3.**

$$\frac{\mathrm{d}\mathsf{n}(z)}{\mathrm{d}z}\bigg|_{z=y} = \sqrt{B}r^{-1}(I - vv^T),$$

*where* $r = \|y\|, v = y/\|y\| = \mathsf{n}(y)/\sqrt{B}$.

*Proof.* We have $\frac{d\mathsf{n}(z)}{dz}\Big|_{z=y} = \sqrt{B}\,\frac{dz/\|z\|}{dz}\Big|_{z=y}$, and

$$
\frac{\partial(y_i/\|y\|)}{\partial y_j} = \frac{\delta_{ij}\|y\| - (\partial\|y\|/\partial y_j)y_i}{\|y\|^2}
$$

$$
= \frac{\delta_{ij}}{\|y\|} - \frac{y_i}{\|y\|^2}\frac{\partial\sqrt{y\cdot y}}{\partial y_j}
$$

$$
= \frac{\delta_{ij}}{\|y\|} - \frac{y_i}{\|y\|^2}\frac{y_j}{\|y\|}
$$

$$
= \frac{\delta_{ij}}{\|y\|} - \frac{y_i y_j}{\|y\|^3}
$$

so that

$$
\frac{dz/\|z\|}{dz}\Big|_{z=y} = r^{-1}(I - vv^T).
$$

$\square$

By chain rule, this easily gives

**Proposition G.4.**

$$
\frac{d\phi\circ\mathsf{n}(z)}{dz}\Big|_{z=y} = \sqrt{B}Dr^{-1}(I - vv^T),
$$

*where* $D = \mathrm{Diag}(\phi'(\mathsf{n}(y))), r = \|y\|, v = y/\|y\| = \mathsf{n}(y)/\sqrt{B}$.

With $v = y/\|y\|$, $r = \|y\|$, and $\tilde{D} = \mathrm{Diag}(\phi'(\mathsf{n}(y)))$, we then have

$$
\mathcal{F}(\Sigma) = \mathbb{E}\left[\tilde{D}^{\otimes 2}\circ\left(\sqrt{B}\,\frac{dz/\|z\|}{dz}\Big|_{z=Gx}\right)^{\otimes 2} : x\sim\mathcal{N}(0,\Sigma)\right]
$$

$$
= \mathbb{E}\left[\tilde{D}^{\otimes 2}\circ\left(\sqrt{B}\,\frac{dz/\|z\|}{dz}\Big|_{z=y}\right)^{\otimes 2} : y\sim\mathcal{N}(0,G\Sigma G)\right]
$$

$$
= B\,\mathbb{E}\left[r^{-2}\left(\tilde{D}^{\otimes 2} + (\tilde{D}vv^T)^{\otimes 2} - \tilde{D}\otimes(\tilde{D}vv^T) - (\tilde{D}vv^T)\otimes\tilde{D}\right) : y\sim\mathcal{N}(0,G\Sigma G)\right]
$$

(52)

$$
\mathcal{F}(\Sigma)\{\Lambda\} = B\,\mathbb{E}\left[r^{-2}\left(\tilde{D}\Lambda\tilde{D} + \tilde{D}vv^T\Lambda vv^T\tilde{D} - \tilde{D}\Lambda vv^T\tilde{D} - \tilde{D}vv^T\Lambda\tilde{D}\right) : y\sim\mathcal{N}(0,G\Sigma G)\right]
$$

### G.1.1 SPHERICAL INTEGRATION

When $\Sigma = \Sigma^*$ is the BSB1 fixed point, one can again easily see that $\mathcal{F}(\Sigma^*)$ is ultrasymmetric. With $\tau_{ij} = \frac{1}{2}\left(\delta_i\delta_j^T + \delta_j\delta_i^T\right)$ (Defn F.20), we have

$$
\mathcal{F}(\Sigma^*)\{\Lambda\} = B\,\mathbb{E}\left[r^{-2}\left(\tilde{D}\Lambda\tilde{D} + \tilde{D}vv^T\Lambda vv^T\tilde{D} - \tilde{D}\Lambda vv^T\tilde{D} - \tilde{D}vv^T\Lambda\tilde{D}\right) : y\sim\mathcal{N}(0,\mu^*G)\right]
$$

$$
= \mu^{*-1}B\,\mathbb{E}\left[r^{-2}\left(\tilde{D}\Lambda\tilde{D} + \tilde{D}vv^T\Lambda vv^T\tilde{D} - \tilde{D}\Lambda vv^T\tilde{D} - \tilde{D}vv^T\Lambda\tilde{D}\right) : y\sim\mathcal{N}(0,G)\right]
$$

$$
\mathcal{F}(\Sigma^*)\{\tau_{ij}\}_{kl} = \mu^{*-1}B\mathop{\mathbb{E}}_{rv\sim\mathcal{N}(0,G)}\left[r^{-2}\left(\phi'(\sqrt{B}v_k)\phi'(\sqrt{B}v_l)\frac{\mathbb{I}(k=i\ \&\ l=j) + \mathbb{I}(k=j\ \&\ l=i)}{2}\right.\right.
$$

$$
+ v_i v_j v_k\phi'(\sqrt{B}v_k)v_l\phi'(\sqrt{B}v_l)
$$

$$
- \frac{1}{2}\left(\mathbb{I}(i=k)\phi'(\sqrt{B}v_k)v_j v_l\phi'(\sqrt{B}v_l) + \mathbb{I}(j=k)\phi'(\sqrt{B}v_k)v_i v_l\phi'(\sqrt{B}v_l)\right.
$$

$$
\left.\left.\left. + \mathbb{I}(i=l)\phi'(\sqrt{B}v_l)v_j v_k\phi'(\sqrt{B}v_k) + \mathbb{I}(j=l)\phi'(\sqrt{B}v_l)v_i v_k\phi'(\sqrt{B}v_k)\right)\right)\right]
$$

$$
= \mu^{*-1}B\frac{\Gamma((B-3)/2)}{\Gamma((B-5)/2)}2^{-2/2}\pi^{-2}\int_0^\pi d\theta_1\cdots\int_0^\pi d\theta_4\, f(v_1^\theta,\ldots,v_4^\theta)\sin^{B-3}\theta_1\cdots\sin^{B-6}\theta_4
$$

$$
= \mu^{*-1}B(B-5)(2\pi)^{-2}\int_0^\pi d\theta_1\cdots\int_0^\pi d\theta_4\, f(v_1^\theta,\ldots,v_4^\theta)\sin^{B-3}\theta_1\cdots\sin^{B-6}\theta_4
$$

by Lemma E.35, where we assume, WLOG by ultrasymmetry, $k, l \in \{1, 2\}; i, j \in \{1, \ldots, 4\}$, and

$$
\begin{aligned}
f(v_1, \ldots, v_4) := \frac{1}{2} & \phi'(\sqrt{B}\mathbb{e}v)_k \phi'(\sqrt{B}\mathbb{e}v)_l \Big( \\
& \mathbb{I}(k = i \ \& \ l = j) + \mathbb{I}(k = j \ \& \ l = i) + 2(\mathbb{e}v)_i (\mathbb{e}v)_j (\mathbb{e}v)_k (\mathbb{e}v)_l \\
& - (\mathbb{I}(i = k)(\mathbb{e}v)_j (\mathbb{e}v)_l + \mathbb{I}(j = k)(\mathbb{e}v)_i (\mathbb{e}v)_l + \mathbb{I}(i = l)(\mathbb{e}v)_j (\mathbb{e}v)_k + \mathbb{I}(j = l)(\mathbb{e}v)_i (\mathbb{e}v)_k) \\
& ) \\
v_1^\theta &= \cos \theta_1 \\
v_2^\theta &= \sin \theta_1 \cos \theta_2 \\
v_3^\theta &= \sin \theta_1 \sin \theta_2 \cos \theta_3 \\
v_4^\theta &= \sin \theta_1 \sin \theta_2 \sin \theta_3 \cos \theta_4.
\end{aligned}
$$

and $\mathbb{e}$ as in Eq. (33).

We can integrate out $\theta_3$ and $\theta_4$ symbolically by Lemma E.29 since their dependence only appear outside of $\phi'$. This reduces each entry of $\mathcal{F}(\Sigma^*)\{\tau_{ij}\}_{kl}$ to 2-dimensional integrals to evaluate numerically. The eigenvalues of $G^{\otimes 2} \circ \mathcal{F}(\Sigma^*)$ can then be obtained from Thm E.62. We omit details here and instead focus on results from other methods.

### G.1.2 GEGENBAUER EXPANSION

Notice that when $\Lambda = G$, the expression for $\mathcal{F}(\Sigma^*)\{\Lambda\}$ significantly simplifies because $G$ acts as identity on $v$:

$$
\begin{aligned}
\mathcal{F}(\Sigma^*)\{G\} &= \mu^{*-1} B \, \mathbb{E} \left[ r^{-2} \left( \tilde{D}\Lambda\tilde{D} + \tilde{D}vv^T Gvv^T \tilde{D} - \tilde{D}Gvv^T \tilde{D} - \tilde{D}vv^T G\tilde{D} \right) : y \sim \mathcal{N}(0, G) \right] \\
&= \mu^{*-1} B \, \mathbb{E} \left[ r^{-2} \left( \tilde{D}G\tilde{D} - \tilde{D}vv^T \tilde{D} \right) : y \sim \mathcal{N}(0, G) \right]
\end{aligned}
$$

Then the diagonal of the image satisfies

$$
\begin{aligned}
& \mathcal{F}(\Sigma^*)\{G\}_{aa} \\
&= \mu^{*-1} B \, \mathbb{E} \left[ r^{-2} \left( \frac{B-1}{B} \phi'(\sqrt{B}v_a)^2 - v_a^2 \phi'(\sqrt{B}v_a)^2 \right) : rv \sim \mathcal{N}(0, G) \right] \\
&= \mu^{*-1} B(B-3)^{-1} \underset{v \sim S^{B-2}}{\mathbb{E}} \left( \frac{B-1}{B} \phi'(\sqrt{B}\mathbb{e}_{a,:}v)^2 - (\mathbb{e}_{a,:}v)^2 \phi'(\sqrt{B}\mathbb{e}_{a,:}v)^2 \right) \mathcal{K}(v; G, 2) \\
& \hspace{10cm} \text{(by Proposition E.33)} \\
&= \mu^{*-1} B(B-3)^{-1} \underset{v \sim S^{B-2}}{\mathbb{E}} \left( 1 - (\hat{\mathbb{e}}_{a,:}v)^2 \right) \frac{B-1}{B} \phi'(\sqrt{B-1}\hat{\mathbb{e}}_{a,:}v)^2 \\
& \hspace{10cm} (\mathcal{K}(v; G, 2) = 1) \\
&= \mu^{*-1}(B-1)(B-3)^{-1} \underset{v \sim S^{B-2}}{\mathbb{E}} \left( 1 - (\hat{\mathbb{e}}_{a,:}v)^2 \right) \phi'(\sqrt{B-1}\hat{\mathbb{e}}_{a,:}v)^2
\end{aligned}
$$

And the off-diagonal entries satisfy

$$
\begin{aligned}
&\mathcal{F}(\Sigma^*)\{G\}_{ab} \\
&= \mu^{*-1} B \, \mathbb{E}\left[ r^{-2} \left( \frac{-1}{B} \phi'(\sqrt{B} \mathbb{e}_{a,:} v) \phi'(\sqrt{B} \mathbb{e}_{b,:} v) \right. \right. \\
&\qquad\qquad \left. \left. - (\mathbb{e}_{a,:} v)(\mathbb{e}_{b,:} v) \phi'(\sqrt{B} \mathbb{e}_{a,:} v) \phi'(\mathbb{e}_{a,:} v) \right] : rv \sim \mathcal{N}(0, G) \right] \\
&= \mu^{*-1} B (B-3)^{-1} \mathop{\mathbb{E}}_{v \sim S^{B-2}} \left( \frac{-1}{B} \phi'(\sqrt{B-1} \hat{\mathbb{e}}_{a,:} v) \phi'(\sqrt{B-1} \hat{\mathbb{e}}_{b,:} v) \right. \\
&\qquad\qquad \left. - \frac{B-1}{B} (\hat{\mathbb{e}}_{a,:} v)(\hat{\mathbb{e}}_{b,:} v) \phi'(\sqrt{B-1} \hat{\mathbb{e}}_{a,:} v) \phi'(\sqrt{B-1} \hat{\mathbb{e}}_{b,:} v) \right) \mathcal{K}(v; G, 2) \\
&\qquad\qquad\qquad\qquad\qquad\qquad\qquad\qquad\qquad\qquad\qquad \text{(by \textcolor{red}{Proposition E.33})} \\
&= \mu^{*-1} B (B-3)^{-1} \mathop{\mathbb{E}}_{v \sim S^{B-2}} \frac{-1}{B} \left( 1 + (B-1)(\hat{\mathbb{e}}_{a,:} v)(\hat{\mathbb{e}}_{b,:} v) \right) \\
&\qquad\qquad\qquad \phi'(\sqrt{B-1} \hat{\mathbb{e}}_{a,:} v) \phi'(\sqrt{B-1} \hat{\mathbb{e}}_{b,:} v) \\
&\qquad\qquad\qquad\qquad\qquad\qquad\qquad\qquad\qquad\qquad\qquad (\mathcal{K}(v; G, 2) = 1) \\
&= -\mu^{*-1} (B-3)^{-1} \mathop{\mathbb{E}}_{v \sim S^{B-2}} \left( 1 + (B-1)(\hat{\mathbb{e}}_{a,:} v)(\hat{\mathbb{e}}_{b,:} v) \right) \phi'(\sqrt{B-1} \hat{\mathbb{e}}_{a,:} v) \phi'(\sqrt{B-1} \hat{\mathbb{e}}_{b,:} v)
\end{aligned}
$$

**Lifting Spherical Expectation.** We first show a way of expressing $\mathcal{F}(\Sigma^*)_{aa}$ in terms of Gegenbauer basis by lifting the spherical integral over $S^{B-2}$ to spherical integral over $S^B$. While this technique cannot be extended to $\mathcal{F}(\Sigma^*)_{ab}, a \neq b$, we believe it could be of use to future related problems.

If $\phi(\sqrt{B-1}x)$ has Gegenbauer expansion $\sum_{l=0}^{\infty} a_l \frac{1}{c_{B-1,l}} C_l^{(\frac{B-3}{2})}(x)$, then $\phi'(\sqrt{B-1}x) = \frac{1}{\sqrt{B-1}} \sum_{l=0}^{\infty} a_l \frac{1}{c_{B-1,l}} C_l^{(\frac{B-3}{2})\prime}(x) = \frac{1}{\sqrt{B-1}} \sum_{l=0}^{\infty} a_l \frac{B-3}{c_{B-1,l}} C_{l-1}^{(\frac{B-1}{2})}(x)$, so that $\phi'(\sqrt{B} \mathbb{e}_{a,:} v) = \frac{1}{\sqrt{B-1}} \sum_{l=0}^{\infty} a_l \frac{B-3}{c_{B-1,l}} C_{l-1}^{(\frac{B-1}{2})}(\hat{\mathbb{e}}_{a,:} v) = \frac{1}{\sqrt{B-1}} \sum_{l=0}^{\infty} a_l (B-3) \frac{c_{B+1,l-1}}{c_{B-1,l}} Z_{\hat{\mathbb{e}}_{a,:}}^{B,(l-1)}(v)$. Here $\hat{\mathbb{e}}_{a,:}$ and $v$ are treated as points on the $B$-dimensional sphere with last 2 coordinates 0.

Thus by Lemmas E.38 and E.39,

$$\mathcal{F}(\Sigma^*)\{G\}_{aa}$$

$$= \mu^{*-1}\frac{\omega_{B-3}}{\omega_{B-2}}(B-1)(B-3)^{-1}\int_{-1}^{1}\mathrm{d}h\,(1-h^2)^{\frac{B-4}{2}}(1-h^2)\phi'(\sqrt{B-1}h)^2$$

$$= \mu^{*-1}\frac{\omega_{B-3}}{\omega_{B-2}}(B-1)(B-3)^{-1}\frac{\omega_B}{\omega_{B-1}}\frac{\omega_{B-1}}{\omega_B}\int_{-1}^{1}\mathrm{d}h\,(1-h^2)^{\frac{B-2}{2}}\phi'(\sqrt{B-1}h)^2$$

$$= \mu^{*-1}\frac{\omega_{B-3}\omega_B}{\omega_{B-2}\omega_{B-1}}(B-1)(B-3)^{-1}\mathop{\mathbb{E}}_{w\sim S^B}\phi'(\sqrt{B-1}\widetilde{\hat{e}_{a,:}},w)^2$$

$$= \mu^{*-1}\frac{\omega_{B-3}\omega_B}{\omega_{B-2}\omega_{B-1}}(B-1)(B-3)^{-1}\mathop{\mathbb{E}}_{w\sim S^B}\left(\frac{1}{\sqrt{B-1}}\sum_{l=0}^{\infty}a_l(B-3)\frac{c_{B+1,l-1}}{c_{B-1,l}}Z^{B,(l-1)}_{\widehat{\hat{e}_{a,:}}}(w)\right)^2$$

$$= \mu^{*-1}\frac{\omega_{B-3}\omega_B}{\omega_{B-2}\omega_{B-1}}(B-3)\sum_{l=0}^{\infty}a_l^2\left(\frac{c_{B+1,l-1}}{c_{B-1,l}}\right)^2 c_{B+1,l-1}^{-1}C_{l-1}^{(\frac{B-1}{2})}(1)$$

$$= \mu^{*-1}\frac{\omega_{B-3}\omega_B}{\omega_{B-2}\omega_{B-1}}(B-1)\sum_{l=0}^{\infty}a_l^2\frac{1}{c_{B-1,l}}C_{l-1}^{(\frac{B-1}{2})}(1)$$

by Lemma E.45

$$= \mu^{*-1}(B-2)\sum_{l=0}^{\infty}a_l^2\frac{1}{c_{B-1,l}}\binom{l+B-3}{l-1}$$

$$= \mu^{*-1}(B-2)\sum_{l=0}^{\infty}a_l^2\frac{1}{c_{B-1,l}}\binom{l+B-3}{B-2}$$

$$= \mu^{*-1}\sum_{l=0}^{\infty}a_l^2\frac{l+B-3}{c_{B-1,l}}\binom{l+B-4}{B-3}$$

$$= \mu^{*-1}\sum_{l=0}^{\infty}a_l^2\frac{l+B-3}{c_{B-1,l}}\frac{l}{B-3}\binom{l+B-4}{B-4}$$

**Dirichlet-like form of $\phi$.** For $\mathcal{F}(\Sigma^*)\{G\}_{ab}$, there's no obvious way of lifting the spherical expectation to a higher dimension with the weight $(1+(B-1)(\hat{e}_{a,:}v)(\hat{e}_{b,:}v))$. We will instead leverage Thm E.47. In Thm E.47, setting $u_1=\hat{e}_{a,:}=u_2$, we get

$$\mathcal{F}(\Sigma^*)\{G\}_{aa} = \mu^{*-1}\frac{B-1}{B-3}\mathop{\mathbb{E}}_{v\sim S^{B-2}}((\hat{e}_{a,:}\cdot\hat{e}_{a,:})-(\hat{e}_{a,:}v)^2)\phi'(\sqrt{B-1}\hat{e}_{a,:}v)^2$$

$$= \mu^{*-1}\sum_{l=0}^{\infty}a_l^2\frac{l+B-3}{B-3}lc_{B-1,l}^{-1}C_l^{(\frac{B-3}{2})}(1).$$

Likewise, setting $u_1=\hat{e}_{a,:}, u_2=\hat{e}_{b,:}$, we get

$$\mathcal{F}(\Sigma^*)\{G\}_{ab} = \mu^{*-1}\frac{B-1}{B-3}\mathop{\mathbb{E}}_{v\sim S^{B-2}}((\hat{e}_{a,:}\cdot\hat{e}_{b,:})-(\hat{e}_{a,:}v)(\hat{e}_{b,:}v))\phi'(\sqrt{B-1}\hat{e}_{a,:}v)\phi'(\sqrt{B-1}\hat{e}_{b,:}v)$$

$$= \mu^{*-1}\sum_{l=0}^{\infty}a_l^2\frac{l+B-3}{B-3}lc_{B-1,l}^{-1}C_l^{(\frac{B-3}{2})}\left(\frac{-1}{B-1}\right)$$

Thus,

**Theorem G.5.** *Suppose $\phi(\sqrt{B-1}x),\phi'(\sqrt{B-1}x)\in L^2((1-x^2)^{\frac{B-3}{2}})$ and $\phi(\sqrt{B-1}x)$ has Gegenbauer expansion $\sum_{l=0}^{\infty}a_l\frac{1}{c_{B-1,l}}C_l^{(\frac{B-3}{2})}(x)$. Then for $\Sigma^*$ being the unique BSB1 fixed point*

*of Eq. (43), the eigenvalue of $G^{\otimes 2} \circ \mathcal{F}(\Sigma^*)$ corresponding to eigenspace $\mathbb{R}G$ is*

$$\lambda_G^{\downarrow} = \mu^{*-1} \sum_{l=0}^{\infty} a_l^2 \frac{l+B-3}{B-3} l c_{B-1,l}^{-1} \left( C_l^{(\frac{B-3}{2})}(1) - C_l^{(\frac{B-3}{2})} \left( \frac{-1}{B-1} \right) \right)$$

$$= \frac{\sum_{l=0}^{\infty} a_l^2 \frac{l+B-3}{B-3} l c_{B-1,l}^{-1} \left( C_l^{(\frac{B-3}{2})}(1) - C_l^{(\frac{B-3}{2})} \left( \frac{-1}{B-1} \right) \right)}{\sum_{l=0}^{\infty} a_l^2 c_{B-1,l}^{-1} \left( C_l^{(\frac{B-3}{2})}(1) - C_l^{(\frac{B-3}{2})} \left( \frac{-1}{B-1} \right) \right)}$$

$$> 1$$

*This is minimized (over choices of $\phi$) iff $\phi$ is linear, in which case $\lambda_G^{\downarrow} = \frac{B-2}{B-3}$.*

### G.2 LAPLACE METHOD

In this section suppose that $\phi$ is degree $\alpha$ positive-homogeneous. Set $D = \text{Diag}(\phi'(y))$ (and recall $v = y/\|y\|, r = \|y\|$). Then $D$ is degree $\alpha - 1$ positive-homogeneous in $x$ (because $\phi'$ is). Consequently we can rewrite Eq. (52) as follows,

$$\mathcal{F}(\Sigma^*)$$

$$= B^{\alpha} \mathop{\mathbb{E}}_{y \sim \mathcal{N}(0, G\Sigma^* G)} \left[ r^{-2} \left( r^{-2(\alpha-1)} D^{\otimes 2} + r^{-2(\alpha+1)} (Dvv^T)^{\otimes 2} \right. \right.$$

$$\left. \left. - r^{-2\alpha} D \otimes (Dvv^T) - r^{-2\alpha} (Dvv^T) \otimes D \right) \right]$$

$$= B^{\alpha} \mathop{\mathbb{E}}_{y \sim \mathcal{N}(0, \mu^* G)} \left[ r^{-2\alpha} D^{\otimes 2} + r^{-2(\alpha+2)} (Dyy^T)^{\otimes 2} - r^{-2(\alpha+1)} \left( D \otimes (Dyy^T) + (Dyy^T) \otimes D \right) \right]$$

$$= B^{\alpha} (\mathsf{A} + \mathsf{B} - \mathsf{C}) \tag{53}$$

where $\Sigma^*$ is the BSB1 fixed point of Eq. (43), $G\Sigma^* G = \mu^* G$, and

$$\mathsf{A} = \mathbb{E} \left[ r^{-2\alpha} D^{\otimes 2} : y \sim \mathcal{N}(0, \mu^* G) \right]$$

$$\mathsf{B} = \mathbb{E} \left[ r^{-2(\alpha+2)} (Dyy^T)^{\otimes 2} : y \sim \mathcal{N}(0, \mu^* G) \right]$$

$$\mathsf{C} = \mathbb{E} \left[ r^{-2(\alpha+1)} \left( D \otimes (Dyy^T) + (Dyy^T) \otimes D \right) : y \sim \mathcal{N}(0, \mu^* G) \right]$$

Each term in the sum above is ultrasymmetric, so has the same eigenspaces $\mathbb{R}G, \mathbb{M}, \mathbb{L}$ (this will also become apparent in the computations below without resorting to Thm E.61). We can thus compute the eigenvalues for each of them in order.

In all computation below, we apply Lemma E.2 first to relate the quantity in question to $\mathsf{V}_{\phi}$.

**Computing $\mathsf{A}$.** For two matrices $\Sigma, \Lambda$, write $\Sigma \odot \Lambda$ for entrywise multiplication. We have by Lemma E.2

$$\mathsf{A}\{\Lambda\} = \mathbb{E} \left[ r^{-2\alpha} D\Lambda D : y \sim \mathcal{N}(0, \mu^* G) \right]$$

$$= \mathbb{E} \left[ r^{-2\alpha} \Lambda \odot \phi'(y) \phi'(y)^T : y \sim \mathcal{N}(0, \mu^* G) \right]$$

$$= \Gamma(\alpha)^{-1} \int_0^{\infty} ds \, s^{\alpha-1} \det(I + 2s\mu^* G)^{-1/2} \Lambda \odot \mathsf{V}_{\phi'} \left( \frac{\mu^*}{1+2s\mu^*} G \right)$$

$$= \Gamma(\alpha)^{-1} \int_0^{\infty} ds \, s^{\alpha-1} (1+2s\mu^*)^{-(B-1)/2} \Lambda \odot \left( \frac{\mu^*}{1+2s\mu^*} \right)^{\alpha-1} \mathsf{V}_{\phi'}(G)$$

$$= \Gamma(\alpha)^{-1} \Lambda \odot \mathsf{V}_{\phi'}(G) \int_0^{\infty} ds \, (\mu^* s)^{\alpha-1} (1+2s\mu^*)^{-(B-1)/2 - \alpha + 1}$$

$$= \Gamma(\alpha)^{-1} \Lambda \odot \mathsf{V}_{\phi'}(G) \, \text{Beta} \left( \frac{B-3}{2}, \alpha \right) \mu^{*-1} 2^{-\alpha}$$

$$= \Lambda \odot \mathsf{c}_{\alpha-1} \left( \frac{B-1}{B} \right)^{\alpha-1} \text{BSB1} \left( \mathsf{J}_{\phi'}(1), \mathsf{J}_{\phi'} \left( \frac{-1}{B-1} \right) \right) \mathsf{P} \left( \frac{B-3}{2}, \alpha \right)^{-1} \mu^{*-1} 2^{-\alpha}$$

Then Thm E.73 gives the eigendecomposition for $B^{\alpha} G^{\otimes 2} \circ \mathsf{A} \upharpoonright \mathcal{H}_B^G$

**Theorem G.6.** *Let* $R_{B,\alpha} := (2\alpha-1)\mathsf{c}_{\alpha-1}B(B-1)^{\alpha-1}\mathrm{P}\left(\frac{B-3}{2},\alpha\right)^{-1}2^{-\alpha}$. *Then* $B^\alpha G^{\otimes 2}\circ\mathsf{A}\upharpoonright\mathcal{H}_B^G$ *has the following eigendecomposition.*

$$\lambda_G^{\mathsf{A}} = R_{B,\alpha}\mu^{*-1}\left(\frac{B-1}{B}\mathrm{J}_\phi'(1) + \frac{1}{B}\mathrm{J}_\phi'\left(\frac{-1}{B-1}\right)\right)$$

$$= \frac{(2\alpha + B - 3)\left(\alpha'^2(B-1)\mathrm{J}_\phi(1) + (2\alpha-1)\mathrm{J}_\phi\left(\frac{1}{1-B}\right)\right)}{(2\alpha-1)(B-3)(B-1)\left(\mathrm{J}(1) - \mathrm{J}\left(\frac{1}{1-B}\right)\right)}$$

$$\lambda_{\mathbb{L}}^{\mathsf{A}} = R_{B,\alpha}\mu^{*-1}\left(\frac{B-2}{B}\mathrm{J}_\phi'(1) + \frac{2}{B}\mathrm{J}_\phi'\left(\frac{-1}{B-1}\right)\right)$$

$$= \frac{(2\alpha + B - 3)\left(\alpha^2(B-2)\mathrm{J}_\phi(1) + 2(2\alpha-1)\mathrm{J}_\phi'\left(\frac{1}{1-B}\right)\right)}{(2\alpha-1)(B-3)(B-1)\left(\mathrm{J}(1) - \mathrm{J}\left(\frac{1}{1-B}\right)\right)}$$

$$\lambda_{\mathbb{M}}^{\mathsf{A}} = R_{B,\alpha}\mu^{*-1}\mathrm{J}_\phi'\left(\frac{-1}{B-1}\right)$$

$$= \frac{B(2\alpha + B - 3)\mathrm{J}_\phi'\left(\frac{1}{1-B}\right)}{(B-3)(B-1)\left(\mathrm{J}_\phi(1) - \mathrm{J}_\phi\left(\frac{1}{1-B}\right)\right)}$$

*Proof.* We have $B^\alpha\mathsf{A} = (2\alpha-1)^{-1}R_{B,\alpha}\mu^{*-1}\mathrm{DOS}\left(\mathrm{J}_{\phi'}(1), 0, \mathrm{J}_{\phi'}\left(\frac{-1}{B-1}\right)\right)$. This allows us to apply Thm E.73. We make a further simplification $\mathrm{J}_{\phi'}(c) = (2\alpha-1)\mathrm{J}_\phi'(c)$ via Proposition E.22. $\square$

**Computing** $\mathsf{B}$. We simplify

$$\mathsf{B}\{\Lambda\} = \mathbb{E}\left[r^{-2(\alpha+2)}Dyy^T\Lambda yy^T D : y \sim \mathcal{N}(0, \mu^* G)\right]$$

$$= \mathbb{E}\left[r^{-2(\alpha+2)}\psi(y)y^T\Lambda y\psi(y)^T : y \sim \mathcal{N}(0, \mu^* G)\right]$$

where $\psi(y) := y\phi'(y)$, which, in this case of $\phi$ being degree $\alpha$ positive-homogeneous, is also equal to $\alpha\phi(y)$.

By Lemma E.2, $\mathsf{B}\{\Lambda\}$ equals

$$\Gamma(\alpha+2)^{-1}\int_0^\infty \mathrm{d}s\, s^{\alpha+1}\det(I + 2s\mu^* G)^{-1/2}\,\mathbb{E}[\psi(y)y^T\Lambda y\psi(y)^T : y \sim \mathcal{N}(0, \frac{\mu^*}{1 + 2s\mu^*}G)]$$

This expression naturally leads us to consider the following definition.

**Definition G.7.** Let $\psi : \mathbb{R} \to \mathbb{R}$ be measurable and $\Sigma \in \mathcal{S}_B$. Define $\mathrm{V}_\psi^{(4)}(\Sigma) : \mathcal{H}_B \to \mathcal{H}_B$ by

$$\mathrm{V}_\psi^{(4)}(\Sigma)\{\Lambda\} = \mathbb{E}[\psi(y)y^T\Lambda y\psi(y)^T : y \sim \mathcal{N}(0, \Sigma)]$$

For two matrices $\Sigma, \Lambda$, write $\langle\Sigma, \Lambda\rangle := \mathrm{tr}(\Sigma^T\Lambda)$. We have the following identity.

**Proposition G.8.** $\mathrm{V}_\phi^{(4)}(\Sigma)\{\Lambda\} = \mathrm{V}_\phi(\Sigma)\langle\Sigma, \Lambda\rangle + 2\frac{\mathrm{d}\mathrm{V}_\phi(\Sigma)}{\mathrm{d}\Sigma}\{\Sigma\Lambda\Sigma\}$.

*Proof.* We have

$$\frac{\mathrm{d}\mathrm{V}_\phi(\Sigma)}{\mathrm{d}\Sigma}\{\Lambda\} = (2\pi)^{-B/2}\int \mathrm{d}z\, \phi(z)^{\otimes 2}\left(\frac{\mathrm{d}}{\mathrm{d}\Sigma}\det(\Sigma)^{-1/2}e^{-\frac{1}{2}z^T\Sigma^{-1}z}\right)\{\Lambda\}$$

$$= (2\pi)^{-B/2}\int \mathrm{d}z\, \phi(z)^{\otimes 2}\left\langle\left[\frac{-1}{2}\det\Sigma^{-1/2}\Sigma^{-1} + \det\Sigma^{-1/2}\frac{1}{2}\Sigma^{-1}zz^T\Sigma^{-1}\right]e^{-\frac{1}{2}z^T\Sigma^{-1}z}, \Lambda\right\rangle$$

$$= \frac{1}{2}(2\pi)^{-B/2}\det\Sigma^{-1/2}\int \mathrm{d}z\, \phi(z)^{\otimes 2}e^{-\frac{1}{2}z^T\Sigma^{-1}z}\left[\langle zz^T, \Sigma^{-1}\Lambda\Sigma^{-1}\rangle - \langle\Sigma^{-1}, \Lambda\rangle\right]$$

$$= \frac{1}{2}\mathrm{V}_\phi^{(4)}(\Sigma)\{\Sigma^{-1}\Lambda\Sigma^{-1}\} - \frac{1}{2}\mathrm{V}_\phi(\Sigma)\langle\Sigma^{-1}, \Lambda\rangle.$$

Making the substitution $\Lambda \to \Sigma\Lambda\Sigma$, we get the desired result. $\square$

If $\varphi$ is degree $\alpha$ positive-homogeneous, then $V_\varphi^{(4)}$ is degree $\alpha + 1$ positive-homogeneous. Thus,

$$
\begin{aligned}
\mathsf{B}\{\Lambda\} &= \Gamma(\alpha+2)^{-1} \int_0^\infty ds\, s^{\alpha+1} \det(I + 2s\mu^* G)^{-1/2} V_\psi^{(4)}\left(\frac{\mu^*}{1+2s\mu^*}G\right)\{\Lambda\} \\
&= \Gamma(\alpha+2)^{-1} \int_0^\infty ds\, s^{\alpha+1}(1+2s\mu^*)^{-(B-1)/2}\left(\frac{\mu^*}{1+2s\mu^*}\right)^{\alpha+1} V_\psi^{(4)}(G)\{\Lambda\} \\
&= \Gamma(\alpha+2)^{-1} V_\psi^{(4)}(G)\{\Lambda\} \int_0^\infty ds\,(\mu^* s)^{\alpha+1}(1+2s\mu^*)^{-(B-1)/2-\alpha-1} \\
&= \Gamma(\alpha+2)^{-1} V_\psi^{(4)}(G)\{\Lambda\} \operatorname{Beta}(\frac{B-3}{2},\alpha+2)2^{-\alpha-2}\mu^{*-1} \\
&= \mathsf{P}\left(\frac{B-3}{2},\alpha+2\right)^{-1} 2^{-\alpha-2}\mu^{*-1} V_\psi^{(4)}(G)\{\Lambda\}.
\end{aligned}
$$

By [Proposition G.8](#), $V_\psi^{(4)}(G)\{\Lambda\} = V_\psi(G)\operatorname{tr}\Lambda + 2\frac{dV_\psi(\Sigma)}{d\Sigma}\{\Lambda\}$ if $\Lambda \in \mathcal{H}_B^G$. Thus

**Theorem G.9.** $B^\alpha G^{\otimes 2} \circ \mathsf{B} \upharpoonright \mathcal{H}_B^G$ *has the following eigendecomposition (note that here $\mu^*$ is still with respect to $\phi$, not $\psi = \alpha\phi$)*

1. *Eigenspace $\mathbb{R}G$ with eigenvalue*

$$
\lambda_G^{\mathsf{B}} := \frac{\alpha^2}{B-3}
$$

2. *Eigenspace $\mathbb{L}$ with eigenvalue*

$$
\lambda_{\mathbb{L}}^{\mathsf{B}} := B^\alpha \mathsf{P}\left(\frac{B-3}{2},\alpha+2\right)^{-1} 2^{-\alpha-2}\mu^{*-1} 2\lambda_{\mathbb{L}}^{G,\psi}(B,\alpha)
$$

$$
= \frac{2\alpha^3(B-2)}{(B-3)(B-1)(B-1+2\alpha)} + \frac{2\alpha^2 B \mathsf{J}'_\phi\left(\frac{-1}{B-1}\right)}{(B-3)(B-1)^2(B-1+2\alpha)(\mathsf{J}_\phi(1) - \mathsf{J}_\phi\left(\frac{-1}{B-1}\right))}
$$

3. *Eigenspace $\mathbb{M}$ with eigenvalue*

$$
\lambda_{\mathbb{M}}^{\mathsf{B}} := B^\alpha \mathsf{P}\left(\frac{B-3}{2},\alpha+2\right)^{-1} 2^{-\alpha-2}\mu^{*-1} 2\lambda_{\mathbb{M}}^{G,\psi}(B,\alpha)
$$

$$
= \frac{2\alpha^2 B \mathsf{J}'_\phi\left(\frac{-1}{B-1}\right)}{(B-3)(B-1)(B-1+2\alpha)(\mathsf{J}_\phi(1) - \mathsf{J}_\phi\left(\frac{-1}{B-1}\right))}
$$

*Proof.* The only thing to justify is the value of $\lambda_G^{\mathsf{B}}$. We have

$$
\lambda_G^{\mathsf{B}} = B^\alpha \mathrm{P}\left(\frac{B-3}{2}, \alpha+2\right)^{-1} 2^{-\alpha-2}\mu^{*-1}
$$

$$
\times \left((B-1)\mathsf{c}_\alpha\left(\frac{B-1}{B}\right)^\alpha\left(\mathrm{J}_\psi(1) - \mathrm{J}_\psi\left(\frac{-1}{B-1}\right)\right) + 2\lambda_{\mathsf{M}}^{G,\psi}(B,\alpha)\right)
$$

$$
= B^\alpha \mathrm{P}\left(\frac{B-3}{2}, \alpha+2\right)^{-1} 2^{-\alpha-2}\mu^{*-1}(B-1+2\alpha)\mathsf{c}_\alpha\left(\frac{B-1}{B}\right)^\alpha\left(\mathrm{J}_\psi(1) - \mathrm{J}_\psi\left(\frac{-1}{B-1}\right)\right)
$$

$$
= \mathsf{c}_\alpha(B-1)^\alpha(B-1+2\alpha)\mathrm{P}\left(\frac{B-3}{2}, \alpha+2\right)^{-1} 2^{-\alpha-2}\mu^{*-1}\left(\mathrm{J}_\psi(1) - \mathrm{J}_\psi\left(\frac{-1}{B-1}\right)\right)
$$

$$
= \mathsf{c}_\alpha(B-1)^\alpha\mathrm{P}\left(\frac{B-3}{2}, \alpha+1\right)^{-1} 2^{-\alpha-1}\mu^{*-1}\left(\mathrm{J}_\psi(1) - \mathrm{J}_\psi\left(\frac{-1}{B-1}\right)\right)
$$

$$
= \mathsf{c}_\alpha(B-1)^\alpha\mathrm{P}\left(\frac{B-3}{2}, \alpha+1\right)^{-1} 2^{-\alpha-1}K_{\alpha,B}^{-1}\alpha^2
$$

$$
\text{(since } \mathrm{J}_\psi = \alpha^2\mathrm{J}_\phi)
$$

$$
= \frac{\alpha^2\mathsf{c}_\alpha(B-1)^\alpha\mathrm{P}\left(\frac{B-3}{2}, \alpha+1\right)^{-1} 2^{-\alpha-1}}{\mathsf{c}_\alpha\mathrm{P}\left(\frac{B-1}{2}, \alpha\right)^{-1}\left(\frac{B-1}{2}\right)^\alpha}
$$

$$
\text{(Defn F.6)}
$$

$$
= \frac{\alpha^2 2^{-1}}{(B-3)/2}
$$

$$
= \frac{\alpha^2}{B-3}
$$

$\square$

**Computing $\mathsf{C}$.** By Lemma E.2,

$$
\mathsf{C}\{\Lambda\} = \mathbb{E}\left[r^{-2(\alpha+1)}\left(D\Lambda yy^T D + Dyy^T\Lambda D\right) : y \sim \mathcal{N}(0, \mu^*G)\right]
$$

$$
= \Gamma(\alpha+1)^{-1}\int \mathrm{d}s\, s^\alpha \det(I + 2s\mu^*G)^{-1/2}\,\mathbb{E}[D\Lambda yy^T D + Dyy^T\Lambda D : y \sim \mathcal{N}(0, \frac{\mu^*}{1+2s\mu^*}G)]
$$

$$
= \Gamma(\alpha+1)^{-1}\int \mathrm{d}s\, s^\alpha(1+2s\mu^*)^{-(B-1)/2}\left(\frac{\mu^*}{1+2s\mu^*}\right)^\alpha\,\mathbb{E}[D\Lambda yy^T D + Dyy^T\Lambda D : y \sim \mathcal{N}(0, G)]
$$

$$
= \Gamma(\alpha+1)^{-1}\int \mathrm{d}s\, (\mu^*s)^\alpha(1+2s\mu^*)^{-(B-1)/2-\alpha}\,\mathbb{E}[D\Lambda yy^T D + Dyy^T\Lambda D : y \sim \mathcal{N}(0, G)]
$$

$$
= \Gamma(\alpha+1)^{-1}\mathrm{Beta}\left(\frac{B-3}{2}, \alpha+1\right) 2^{-\alpha-1}\mu^{*-1}\,\mathbb{E}[D\Lambda yy^T D + Dyy^T\Lambda D : y \sim \mathcal{N}(0, G)]
$$

$$
= \mathrm{P}\left(\frac{B-3}{2}, \alpha+1\right)^{-1} 2^{-\alpha-1}\mu^{*-1}\,\mathbb{E}[D\Lambda yy^T D + Dyy^T\Lambda D : y \sim \mathcal{N}(0, G)]
$$

**Lemma G.10.** *Suppose $\phi$ is degree $\alpha$ positive-homogeneous. Then for $\Lambda \in \mathcal{H}_B$,*

$$
\mathbb{E}[D \otimes (Dyy^T) + Dyy^T \otimes D : y \sim \mathcal{N}(0, \Sigma)] = \alpha\left.\frac{\mathrm{dV}_\phi(\Pi\Sigma\Pi)}{\mathrm{d}\Pi}\right|_{\Pi=I}
$$

$$
\mathbb{E}[D\Lambda yy^T D + Dyy^T\Lambda D : y \sim \mathcal{N}(0, \Sigma)] = \alpha\frac{\mathrm{dV}_\phi(\Sigma)}{\mathrm{d}\Sigma}\{\Sigma\Lambda + \Lambda\Sigma\}
$$

*where $D = \mathrm{Diag}(\phi'(y))$.*

*Proof.* Let $\Pi_t, t \in (-\epsilon, \epsilon)$ be a smooth path in $\mathcal{H}_B$ with $\Pi_0 = I$. Write $D_t = \mathrm{Diag}(\phi'(\Pi_t y))$, so that $D_0 = D$. Then, using $\dot{\Pi}_t$ to denote $t$ derivative,

$$\frac{d}{dt}\mathrm{V}_\phi(\Pi_t \Sigma \Pi_t) = \frac{d}{dt}\mathbb{E}[\phi(\Pi_t y)\phi(\Pi_t y)^T : y \sim \mathcal{N}(0, \Sigma)]$$

$$= \mathbb{E}[D_t \dot{\Pi}_t y \phi(\Pi_t y)^T + \phi(\Pi_t y)y^T \dot{\Pi}_t D_t : y \sim \mathcal{N}(0, \Sigma)]$$

$$\frac{d}{dt}\mathrm{V}_\phi(\Pi_t \Sigma \Pi_t)\bigg|_{t=0} = \mathbb{E}[D\dot{\Pi}_0 y \phi(y)^T + \phi(y)y^T \dot{\Pi}_0 D : y \sim \mathcal{N}(0, \Sigma)]$$

Because for $x \in \mathbb{R}$, $\alpha\phi(x) = x\phi'(x)$, for $y \in \mathbb{R}^B$ we can write $\phi(y) = \alpha^{-1}\mathrm{Diag}(\phi'(y))y$. Then

$$\alpha\frac{\mathrm{dV}_\phi(\Sigma)}{\mathrm{d}\Sigma}\{\Sigma\dot{\Pi}_0 + \dot{\Pi}_0\Sigma\} = \alpha\frac{\mathrm{dV}_\phi(\Sigma)}{\mathrm{d}\Sigma}\left\{\frac{d\Pi_t\Sigma\Pi_t}{dt}\right\}$$

$$= \alpha\frac{d}{dt}\mathrm{V}_\phi(\Pi_t\Sigma\Pi_t)\bigg|_{t=0}$$

$$= \mathbb{E}[D\dot{\Pi}_0 yy^T D + Dyy^T \dot{\Pi}_0 D : y \sim \mathcal{N}(0, \Sigma)]$$

$\square$

Therefore, for $\Lambda \in \mathcal{H}_B^G$,

$$\mathsf{C}\{\Lambda\} = \mathrm{P}\left(\frac{B-3}{2}, \alpha+1\right)^{-1} 2^{-\alpha-1}\mu^{*-1}\alpha\frac{\mathrm{dV}_\phi(\Sigma)}{\mathrm{d}\Sigma}\bigg|_{\Sigma=G}\{G\Lambda + \Lambda G\}$$

$$= 2\alpha\mathrm{P}\left(\frac{B-3}{2}, \alpha+1\right)^{-1} 2^{-\alpha-1}\mu^{*-1}\frac{\mathrm{dV}_\phi(\Sigma)}{\mathrm{d}\Sigma}\bigg|_{\Sigma=G}\{\Lambda\}$$

So Thm F.31 gives

**Theorem G.11.** $B^\alpha G^{\otimes 2} \circ \mathsf{C}$ *has the following eigendecomposition*

1. *eigenspace $\mathbb{R}G$ with eigenvalue*

$$\lambda_G^\mathsf{C} := 2B^\alpha\alpha\mathrm{P}\left(\frac{B-3}{2}, \alpha+1\right)^{-1} 2^{-\alpha-1}\mu^{*-1}\lambda_G^{G,\phi}(B, \alpha)$$

$$= \frac{2\alpha^2}{B-3}$$

2. *eigenspace $\mathbb{L}$ with eigenvalue*

$$\lambda_\mathbb{L}^\mathsf{C} := 2B^\alpha\alpha\mathrm{P}\left(\frac{B-3}{2}, \alpha+1\right)^{-1} 2^{-\alpha-1}\mu^{*-1}\lambda_\mathbb{L}^{G,\phi}(B, \alpha)$$

$$= \frac{2\alpha^2(B-2)}{(B-3)(B-1)} + \frac{2\alpha B\mathrm{J}_\phi'\left(\frac{-1}{B-1}\right)}{(B-3)(B-1)^2(\mathrm{J}_\phi(1) - \mathrm{J}_\phi\left(\frac{-1}{B-1}\right))}$$

3. *eigenspace $\mathbb{M}$ with eigenvalue*

$$\lambda_\mathbb{M}^\mathsf{C} := 2B^\alpha\alpha\mathrm{P}\left(\frac{B-3}{2}, \alpha+1\right)^{-1} 2^{-\alpha-1}\mu^{*-1}\lambda_\mathbb{M}^{G,\phi}(B, \alpha)$$

$$= \frac{2\alpha B\mathrm{J}_\phi'\left(\frac{-1}{B-1}\right)}{(B-3)(B-1)(\mathrm{J}_\phi(1) - \mathrm{J}_\phi\left(\frac{-1}{B-1}\right))}$$

Altogether, by Eq. (53) and Thms G.6, G.9 and G.11, this implies

**Theorem G.12.** $G^{\otimes 2} \circ \mathcal{F}(\Sigma^*) : \mathcal{H}_B^G \to \mathcal{H}_B^G$ *has eigenspaces* $\mathbb{R}G, \mathbb{M}, \mathbb{L}$ *respectively with the following eigenvalues*

$$\lambda_G^{\downarrow} = \frac{(B - 3 + 2\alpha)\left((2\alpha - 1)J'_\phi\left(\frac{-1}{B-1}\right) + \alpha^2(B-1)J_\phi(1)\right)}{(2\alpha - 1)(B - 3)(B - 1)(J_\phi(1) - J_\phi\left(\frac{-1}{B-1}\right))} - \frac{\alpha^2}{B - 3}$$

$$= \frac{\alpha^2(B - 3 + 2\alpha)\left(J_\phi(1) + \frac{1}{B-1}J_{\alpha^{-1}\phi'}\left(\frac{-1}{B-1}\right)\right)}{(2\alpha - 1)(B - 3)(J_\phi(1) - J_\phi\left(\frac{-1}{B-1}\right))} - \frac{\alpha^2}{B - 3}$$

$$= \frac{\alpha^2\left((B - 2)J_\phi(1) + (B - 3 + 2\alpha)\frac{1}{B-1}J_{\alpha^{-1}\phi'}\left(\frac{-1}{B-1}\right) + (2\alpha - 1)(J_\phi(1) - J_\phi\left(\frac{-1}{B-1}\right))\right)}{(2\alpha - 1)(B - 3)(J_\phi(1) - J_\phi\left(\frac{-1}{B-1}\right))}$$

$$\lambda_\mathbb{L}^{\downarrow} = -\frac{2\alpha^2(B - 2)(B - 1 + \alpha)}{(B - 3)(B - 1)(B - 1 + 2\alpha)}$$

$$+ 2\frac{\alpha^2(3B - 4) + \alpha\left(3B^2 - 11B + 8\right) + (B - 3)(B - 1)^2}{(B - 3)(B - 1)^2(B - 1 + 2\alpha)} \frac{J'_\phi\left(\frac{-1}{B-1}\right)}{J_\phi(1) - J_\phi\left(\frac{-1}{B-1}\right)}$$

$$+ \frac{\alpha^2(B - 2)(B - 3 + 2\alpha)}{(2\alpha - 1)(B - 3)(B - 1)} \frac{J_\phi(1)}{J_\phi(1) - J_\phi\left(\frac{-1}{B-1}\right)}$$

$$\lambda_\mathbb{M}^{\downarrow} = \frac{B(B^2 + 2(\alpha - 2)B + 2(\alpha - 3)\alpha + 3)}{(B - 3)(B - 1)(B - 1 + 2\alpha)} \frac{J'_\phi\left(\frac{-1}{B-1}\right)}{J_\phi(1) - J_\phi\left(\frac{-1}{B-1}\right)}.$$

## H  CROSS BATCH: FORWARD DYNAMICS

In this section, we study the generalization of Eq. (43) to multiple batches.

**Definition H.1.** For linear operators $\mathcal{T}_i : \mathcal{X}_i \to \mathcal{Y}_i, i = 1, \ldots, k$, we write $\bigoplus_i \mathcal{T}_i : \bigoplus_i \mathcal{X}_i \to \bigoplus_i \mathcal{Y}_i$ for the operator $(x_1, \ldots, x_k) \mapsto (\mathcal{T}_1(x_1), \ldots, \mathcal{T}_k(x_k))$. We also write $\mathcal{T}_1^{\oplus n} := \bigoplus_{j=1}^n \mathcal{T}_1$ for the direct sum of $n$ copies of $\mathcal{T}_1$.

For $k \geq 2$, now consider the extended ("$k$-batch") dynamics on $\widetilde{\Sigma} \in \mathcal{S}_{kB}$ defined by

$$\widetilde{\Sigma}^l = V_{\mathcal{B}_\phi^{\oplus k}}(\widetilde{\Sigma}^{l-1}) \tag{54}$$

$$= \mathbb{E}[(\mathcal{B}_\phi(h_{1:B}), \mathcal{B}_\phi(h_{B+1:2B}), \ldots, \mathcal{B}_\phi(h_{(k-1)B+1:kB}))^{\otimes 2} : h \sim \mathcal{N}(0, \widetilde{\Sigma}^{l-1})]$$

If we restrict the dynamics to just the upper left $B \times B$ submatrix (as well as any of the diagonal $B \times B$ blocks) of $\widetilde{\Sigma}^l$, then we recover Eq. (43).

### H.1  LIMIT POINTS

In general, like in the case of Eq. (43), it is difficult to prove global convergence behavior. Thus we manually look for fixed points and the local convergence behaviors around them. A very natural extension of the notion of BSB1 matrices is the following

**Definition H.2.** We say a matrix $\widetilde{\Sigma} \in \mathcal{S}_{kB}$ is *CBSB1* (short for "1-Step Cross-Batch Symmetry Breaking") if $\widetilde{\Sigma}$ in block form ($k \times k$ blocks, each of size $B \times B$) has one common BSB1 block on the diagonal and one common constant block on the off-diagonal, i.e.

$$\begin{pmatrix} \mathrm{BSB1}(a, b) & c\mathbf{1}\mathbf{1}^T & c\mathbf{1}\mathbf{1}^T & \cdots \\ c\mathbf{1}\mathbf{1}^T & \mathrm{BSB1}(a, b) & c\mathbf{1}\mathbf{1}^T & \cdots \\ c\mathbf{1}\mathbf{1}^T & c\mathbf{1}\mathbf{1}^T & \mathrm{BSB1}(a, b) & \cdots \\ \vdots & \vdots & \vdots & \ddots \end{pmatrix}$$

We will study the fixed point $\widetilde{\Sigma}^*$ to Eq. (54) of CBSB1 form.

### H.1.1 SPHERICAL INTEGRATION AND GEGENBAUER EXPANSION

**Theorem H.3.** *Let* $\widetilde{\Sigma} \in \mathcal{S}_{kB}$. *If* $\widetilde{\Sigma}$ *is CBSB1 then* $\mathrm{V}_{\mathcal{B}_\phi^{\oplus k}}(\widetilde{\Sigma})$ *is CBSB1 and equals* $\widetilde{\Sigma}^* :=$ $\begin{pmatrix} \Sigma^* & c^* \\ c^* & \Sigma^* \end{pmatrix}$ *where* $\Sigma^*$ *is the BSB1 fixed point of* [Thm F.5](#) *and* $c^* \geq 0$ *with* $\sqrt{c^*} = \frac{\Gamma((B-1)/2)}{\Gamma((B-2)/2)} \pi^{-1/2} \int_0^\pi \mathrm{d}\theta\ \phi(-\sqrt{B-1}\cos\theta)\sin^{B-3}\theta$. *If* $\phi(\sqrt{B-1}x)$ *has Gegenbauer expansion* $\sum_{l=0}^\infty a_l \frac{1}{c_{B-1,l}} C_l^{(\frac{B-3}{2})}(x)$, *then* $c^* = a_0^2$.

*Proof.* We will prove for $k = 2$. The general $k$ cases follow from the same reasoning.

Let $\widetilde{\Sigma} = \begin{pmatrix} \Sigma & c\mathbf{1}\mathbf{1}^T \\ c\mathbf{1}\mathbf{1}^T & \Sigma \end{pmatrix}$ where $\Sigma = \mathrm{BSB1}(a,b)$. As remarked below [Eq. (54)](#), restricting to any diagonal blocks just recovers the dynamics of [Eq. (43)](#), which gives the claim about the diagonal blocks being $\Sigma^*$ through [Thm F.5](#).

We now look at the off-diagonal blocks.

$$\mathrm{V}_{\mathcal{B}_\phi^{\oplus 2}}(\widetilde{\Sigma})_{1:B,B+1:2B} = \mathbb{E}[\mathcal{B}_\phi(z_{1:B}) \otimes \mathcal{B}_\phi(z_{B+1:2B}) : z \sim \mathcal{N}(0,\widetilde{\Sigma})]$$

$$= \mathbb{E}[\phi \circ \mathsf{n}(y_{1:B}) \otimes \phi \circ \mathsf{n}(y_{B+1:2B}) : y \sim \mathcal{N}(0,\widetilde{\Sigma}^G)]$$

where $\widetilde{\Sigma}^G := G^{\oplus 2}\widetilde{\Sigma}G^{\oplus 2} = \begin{pmatrix} G & 0 \\ 0 & G \end{pmatrix} \widetilde{\Sigma} \begin{pmatrix} G & 0 \\ 0 & G \end{pmatrix} = \begin{pmatrix} G\Sigma G & cG\mathbf{1}\mathbf{1}^T G \\ cG\mathbf{1}\mathbf{1}^T G & G\Sigma G \end{pmatrix} = \begin{pmatrix} (a-b)G & 0 \\ 0 & (a-b)G \end{pmatrix}$ (the last step follows from [Lemma E.52](#)). Thus $y_{1:B}$ is independent from $y_{B+1:2B}$, and

$$\mathrm{V}_{\mathcal{B}_\phi^{\oplus 2}}(\widetilde{\Sigma})_{1:B,B+1:2B} = \mathbb{E}[\phi \circ \mathsf{n}(x) : x \sim \mathcal{N}(0,(a-b)G)]^{\otimes 2}$$

By symmetry,

$$\mathbb{E}[\phi \circ \mathsf{n}(x) : x \sim \mathcal{N}(0,(a-b)G)] = \sqrt{c^*}\mathbf{1}$$

$$\mathbb{E}[\phi \circ \mathsf{n}(x) : x \sim \mathcal{N}(0,(a-b)G)]^{\otimes 2} = c^*\mathbf{1}\mathbf{1}^T$$

where $\sqrt{c^*} := \mathbb{E}[\phi(\mathsf{n}(x)_1) : x \sim \mathcal{N}(0,(a-b)G)]$. We can compute

$$\sqrt{c^*} = \mathbb{E}[\phi(\mathsf{n}(x)_1) : x \sim \mathcal{N}(0,G)]$$

$$\text{because } \mathsf{n} \text{ is scale-invariant}$$

$$= \mathbb{E}[\phi(\mathsf{n}(\mathrm{e}x)_1) : x \sim \mathcal{N}(0,I_{B-1})]$$

$$\text{where } \mathrm{e} \text{ is as in } \text{Eq. (33)}$$

$$= \mathbb{E}[\phi((\mathrm{e}\mathsf{n}(x))_1) : x \sim \mathcal{N}(0,I_{B-1})]$$

$$\text{because } \mathrm{e} \text{ is an isometry}$$

$$= \mathbb{E}[\phi(\sqrt{B}(\mathrm{e}v)_1) : x \sim \mathcal{N}(0,I_{B-1})]$$

$$\text{with } v = x/\|x\|$$

$$= \frac{\Gamma((B-1)/2)}{\Gamma((B-2)/2)} \pi^{-1/2} \int_0^\pi \mathrm{d}\theta\ \phi(-\sqrt{B-1}\cos\theta)\sin^{B-3}\theta$$

by [Lemma E.37](#). At the same time, by [Proposition E.33](#), we can obtain the Gegenbauer expansion

$$\sqrt{c^*} = \mathop{\mathbb{E}}_{v\sim S^{B-2}} \phi(\sqrt{B-1}\hat{\mathrm{e}}_{1,:}v)$$

$$= \mathop{\mathbb{E}}_{v\sim S^{B-2}} \sum_{l=0}^\infty a_l \frac{1}{c_{B-1,l}} C_l^{(\frac{B-3}{2})}(\mathrm{e}_{1,:}v)$$

$$= \left\langle \sum_{l=0}^\infty a_l Z_{\hat{\mathrm{e}}_{1,:}}^{B-2,(l)}, 1 \right\rangle$$

$$= a_0 Z_{\hat{\mathrm{e}}_{1,:}}^{B-2,(0)}(\hat{\mathrm{e}}_{1,:})$$

$$= a_0.$$

$\square$

**Corollary H.4.** *With the notation as in* Thm H.3, *if $\phi$ is positive-homogeneous of degree $\alpha$ and* $\phi(x) = a\rho_\alpha(x) - b\rho_\alpha(-x)$, *then* $\sqrt{c^*} = (a-b)\frac{1}{2\sqrt{\pi}}(B-1)^{\alpha/2}\frac{\Gamma((B-1)/2)\Gamma((\alpha+1)/2)}{\Gamma((\alpha+B-1)/2)}$.

*Proof.* We compute

$$\int_0^{\pi/2} d\theta \, (\cos\theta)^\alpha \sin^{B-3}\theta = \frac{1}{2}\operatorname{Beta}\left(0,1;\frac{\alpha+1}{2},\frac{B-2}{2}\right)$$

$$\text{by Lemma E.28}$$

$$= \frac{1}{2}\operatorname{Beta}\left(\frac{\alpha+1}{2},\frac{B-2}{2}\right)$$

$$\int_{\pi/2}^{\pi} d\theta \, (-\cos\theta)^\alpha \sin^{B-3}\theta = \int_0^{\pi/2} d\theta \, (\cos\theta)^\alpha \sin^{B-3}\theta$$

$$= \frac{1}{2}\operatorname{Beta}\left(\frac{\alpha+1}{2},\frac{B-2}{2}\right)$$

So for a positive homogeneous function $\phi(x) = a\rho_\alpha(x) - b\rho_\alpha(-x)$,

$$\sqrt{c^*} = \frac{\Gamma((B-1)/2)}{\Gamma((B-2)/2)}\pi^{-1/2}\left(a\int_{\pi/2}^{\pi} d\theta \, (-\sqrt{B-1}\cos\theta)^\alpha \sin^{B-3}\theta\right.$$

$$\left. - b\int_0^{\pi/2} d\theta \, (\sqrt{B-1}\cos\theta)^\alpha \sin^{B-3}\theta\right)$$

$$= (a-b)\frac{\Gamma((B-1)/2)}{\Gamma((B-2)/2)}\pi^{-1/2}(B-1)^{\alpha/2}\frac{1}{2}\operatorname{Beta}\left(\frac{\alpha+1}{2},\frac{B-2}{2}\right)$$

$$= (a-b)\frac{1}{2\sqrt{\pi}}(B-1)^{\alpha/2}\frac{\Gamma((B-1)/2)\Gamma((\alpha+1)/2)}{\Gamma((\alpha+B-1)/2)}$$

Expanding the beta function and combining with Thm H.3 gives the desired result. $\square$

### H.1.2 LAPLACE METHOD

While we don't need to use the Laplace method to compute $c^*$ for positive homogeneous functions (since it is already given by Corollary H.4), going through the computation is instructive for the machinery for computing the eigenvalues in a later section.

**Lemma H.5** (The Laplace Method Cross Batch Master Equation). *For $A, B, C \in \mathbb{N}$, let $f : \mathbb{R}^{A+B} \to \mathbb{R}^C$ and let $a, b \geq 0$. Suppose for any $y \in \mathbb{R}^A, z \in \mathbb{R}^B$, $\|f(y,z)\| \leq h(\sqrt{\|y\|^2 + \|z\|^2})$ for some nondecreasing function $h : \mathbb{R}^{\geq 0} \to \mathbb{R}^{\geq 0}$ such that $\mathbb{E}[h(r\|z\|) : z \in \mathcal{N}(0, I_{A+B})]$ exists for every $r \geq 0$. Define $\wp(\Sigma) := \mathbb{E}[\|y\|^{-2a}\|z\|^{-2b}f(y,z) : (y,z) \sim \mathcal{N}(0,\Sigma)]$. Then on $\{\Sigma \in \mathcal{S}_{A+B} : \operatorname{rank}\Sigma > 2(a+b)\}$, $\wp(\Sigma)$ is well-defined and continuous, and furthermore satisfies*

$$\wp(\Sigma) = \Gamma(a)^{-1}\Gamma(b)^{-1}\int_0^\infty ds \int_0^\infty dt \, s^{a-1}t^{b-1}\det(I_{A+B}+2\Omega)^{-1/2}\mathbb{E}_{(y,z)\sim\mathcal{N}(0,\Pi)} f(y,z) \quad (55)$$

*where $D = \begin{pmatrix} \sqrt{s}I_A & 0 \\ 0 & \sqrt{t}I_B \end{pmatrix}$, $\Omega = D\Sigma D$, and $\Pi = D^{-1}\Omega(I+2\Omega)^{-1}D^{-1}$.*

*Proof.* If $\Sigma$ is full rank, we can show Fubini-Tonelli theorem is valid in the following computation by the same arguments of the proof of Lemma E.2.

$$\mathbb{E}[\|y\|^{-2a}\|z\|^{-2b}f(y,z) : (y,z) \sim \mathcal{N}(0,\Sigma)]$$

$$= \mathbb{E}[f(y,z) \int_0^\infty ds\, \Gamma(a)^{-1}s^{a-1}e^{-\|y\|^2 s} \int_0^\infty dt\, \Gamma(b)^{-1}t^{b-1}e^{-\|y\|^2 t} : (y,z) \sim \mathcal{N}(0,\Sigma)]$$

$$= (2\pi)^{-\frac{A+B}{2}}\Gamma(a)^{-1}\Gamma(b)^{-1}\det\Sigma^{-1/2}$$

$$\times \int_0^\infty ds \int_0^\infty dt\, s^{a-1}t^{b-1} \int_{\mathbb{R}^{A+B}} dy\, dz\, f(y,z)e^{-\frac{1}{2}(y,z)(\Sigma^{-1}+2D^2)(y,z)^T}$$

$$\text{(Fubini-Tonelli)}$$

$$= \Gamma(a)^{-1}\Gamma(b)^{-1} \int_0^\infty ds \int_0^\infty dt\, s^{a-1}t^{b-1}\det(\Sigma(\Sigma^{-1}+2D^2))^{-1/2}\,\mathbb{E}\,f(y,z)$$

where in the last line, $(y,z) \sim \mathcal{N}(0,(\Sigma^{-1}+2D^2)^{-1})$. We recover the equation in question with the following simplifications.

$$(\Sigma^{-1}+2D^2)^{-1} = \Sigma(I_{A+B}+2D^2\Sigma)^{-1}$$
$$= \Sigma(D^{-1}+2D\Sigma)^{-1}D^{-1}$$
$$= \Sigma D(I_{A+B}+2D\Sigma D)^{-1}D^{-1}$$
$$= D^{-1}\Omega(I_{A+B}+2\Omega)^{-1}D^{-1}$$
$$\det(\Sigma(\Sigma^{-1}+2D^2)) = \det(I_{A+B}+2\Sigma D^2)$$
$$= \det(I_{A+B}+2D\Sigma D)$$
$$= \det(I_{A+B}+2\Omega)$$

The case of general $\Sigma$ with $\operatorname{rank}\Sigma > 2(a+b)$ is given by the same continuity arguments as in Lemma E.2. $\qquad\square$

Let $\widetilde{\Sigma} \in \mathcal{S}_{2B}$, $\widetilde{\Sigma} = \begin{pmatrix} \Sigma & \Xi \\ \Xi^T & \Sigma' \end{pmatrix}$ where $\Sigma, \Sigma' \in \mathcal{S}_B$ and $\Xi \in \mathbb{R}^{B\times B}$. Consider the off-diagonal block of $V_{\mathcal{B}_\phi^{\oplus 2}}(\widetilde{\Sigma})$.

$$\mathbb{E}[\mathcal{B}_\phi(z) \otimes \mathcal{B}_\phi(z') : (z,z') \sim \mathcal{N}(0,\widetilde{\Sigma})]$$

$$= \mathbb{E}[\phi(\mathsf{n}(y)) \otimes \phi(\mathsf{n}(y')) : (y,y') \sim \mathcal{N}(0,\widetilde{\Sigma}^G)]$$

$$= B^\alpha\,\mathbb{E}[\|y\|^{-\alpha}\|y'\|^{-\alpha}\phi(y) \otimes \phi(y') : (y,y') \sim \mathcal{N}(0,\widetilde{\Sigma}^G)]$$

$$= B^\alpha\Gamma(\alpha/2)^{-2} \int_0^\infty ds \int_0^\infty dt\, (st)^{\alpha/2-1}\det(I_{2B}+2\Omega)^{-1/2} \mathop{\mathbb{E}}_{(y,y')\sim\mathcal{N}(0,\Pi)} \phi(y) \otimes \phi(y')$$

$$\text{(by Lemma H.5)}$$

$$= B^\alpha\Gamma(\alpha/2)^{-2} \int_0^\infty ds \int_0^\infty dt\, (st)^{\alpha/2-1}\det(I_{2B}+2\Omega)^{-1/2}V_\phi(\Pi)_{1:B,B+1:2B} \qquad (56)$$

where $\Omega = \begin{pmatrix} s\Sigma^G & \sqrt{st}\Xi^G \\ \sqrt{st}(\Xi^G)^T & t\Sigma'^G \end{pmatrix}$ and $\Pi = D^{-1}\Omega(I+2\Omega)^{-1}D^{-1}$ with $D = \sqrt{s}I_B \oplus \sqrt{t}I_B = \begin{pmatrix} \sqrt{s}I_B & 0 \\ 0 & \sqrt{t}I_B \end{pmatrix}$.

**Theorem H.6** (Rephrasing and adding to Corollary H.4). *Let $\widetilde{\Sigma} \in \mathcal{S}_{kB}$ and $\phi$ be positive homogeneous of degree $\alpha$. If $\widetilde{\Sigma}$ is CBSB1 then $V_{\mathcal{B}_\phi^{\oplus k}}(\widetilde{\Sigma})$ is CBSB1 and equals $\widetilde{\Sigma}^* = \begin{pmatrix} \Sigma^* & c^*\mathbf{1}\mathbf{1}^T \\ c^*\mathbf{1}\mathbf{1}^T & \Sigma^* \end{pmatrix}$ where $\Sigma^*$ is the BSB1 fixed point of Thm F.5 and*

$$c^* = \mathsf{c}_\alpha\left(\frac{B-1}{2}\right)^\alpha \mathrm{P}\left(\frac{B-1}{2},\frac{\alpha}{2}\right)^{-2}\mathrm{J}_\phi(0)$$

$$= (a-b)^2\frac{1}{4\pi}(B-1)^\alpha\left(\frac{\Gamma((B-1)/2)\Gamma((\alpha+1)/2)}{\Gamma((\alpha+B-1)/2)}\right)^2$$

*Proof.* Like in the proof of Corollary H.4, we only need to compute the cross-batch block, which is given above by Eq. (56). Recall that $\widetilde{\Sigma} = \begin{pmatrix} \Sigma & c\mathbf{1}\mathbf{1}^T \\ c\mathbf{1}\mathbf{1}^T & \Sigma \end{pmatrix}$ where $\Sigma = \mathrm{BSB1}(a,b)$. Set $\mu := a - b$. Note that because the off-diagonal block $\Xi = c\mathbf{1}\mathbf{1}^T$ by assumption, $\Xi^G = 0$, so that

$$\Omega = s\Sigma^G \oplus t\Sigma^G$$
$$= s\mu G \oplus t\mu G$$
$$\Omega(I + 2\Omega)^{-1} = \frac{s\mu}{1 + 2s\mu}G \oplus \frac{t\mu}{1 + 2t\mu}G$$
$$\Pi = \frac{\mu}{1 + 2s\mu}G \oplus \frac{\mu}{1 + 2t\mu}G$$

Therefore,

$$\det(I + 2\Omega) = (1 + 2s\mu)^{B-1}(1 + 2t\mu)^{B-1}$$
$$\mathrm{V}_\phi(\Pi) = \mu^\alpha \left(\frac{B-1}{B}\right)^\alpha \Delta \mathrm{V}_\phi \begin{pmatrix} \mathrm{BSB1}(1, \frac{-1}{B-1}) & 0 \\ 0 & \mathrm{BSB1}(1, \frac{-1}{B-1}) \end{pmatrix} \Delta$$

where $\Delta = \begin{pmatrix} (1 + 2s\mu)^{-\alpha/2}I_B & 0 \\ 0 & (1 + 2t\mu)^{-\alpha/2}I_B \end{pmatrix}$. In particular, the off-diagonal block is constant with entry $\mathsf{c}_\alpha \mu^\alpha \left(\frac{B-1}{B}\right)^\alpha (1 + 2s\mu)^{-\alpha/2}(1 + 2t\mu)^{-\alpha/2}\mathrm{J}_\phi(0)$.

Finally, by Eq. (56),

$$\mathbb{E}[\mathcal{B}_\phi(z) \otimes \mathcal{B}_\phi(z') : (z, z') \sim \mathcal{N}(0, \widetilde{\Sigma})]$$
$$= B^\alpha \Gamma(\alpha/2)^{-2} \int_0^\infty \mathrm{d}s \int_0^\infty \mathrm{d}t\, (st)^{\alpha/2-1} \det(I_{2B} + 2\Omega)^{-1/2}\mathrm{V}_\phi(\Pi)_{1:B, B+1:2B}$$
$$= B^\alpha \Gamma(\alpha/2)^{-2} \int_0^\infty \mathrm{d}s \int_0^\infty \mathrm{d}t\, (st)^{\alpha/2-1}[(1 + 2s\mu)(1 + 2t\mu)]^{-\frac{B-1}{2}}$$
$$\times \mathsf{c}_\alpha \mu^\alpha \left(\frac{B-1}{B}\right)^\alpha [(1 + 2s\mu)(1 + 2t\mu)]^{-\alpha/2}\mathrm{J}_\phi(0)\mathbf{1}\mathbf{1}^T$$
$$= \mathsf{c}_\alpha(B-1)^\alpha \Gamma(\alpha/2)^{-2}\mathrm{J}_\phi(0)\mathbf{1}\mathbf{1}^T \int_0^\infty \mathrm{d}\sigma \int_0^\infty \mathrm{d}\tau\, (\sigma\tau)^{\alpha/2-1}(1 + 2\sigma)^{-\frac{B-1+\alpha}{2}}(1 + 2\tau)^{-\frac{B-1+\alpha}{2}}$$
$$\text{(with } \sigma = \mu s, \tau = \mu t\text{)}$$
$$= \mathsf{c}_\alpha(B-1)^\alpha \Gamma(\alpha/2)^{-2}\mathrm{J}_\phi(0)\mathbf{1}\mathbf{1}^T \left(2^{-\alpha/2}\,\mathrm{Beta}\left(\frac{B-1}{2}, \frac{\alpha}{2}\right)\right)^2$$
$$= \mathsf{c}_\alpha \left(\frac{B-1}{2}\right)^\alpha \mathrm{P}\left(\frac{B-1}{2}, \frac{\alpha}{2}\right)^{-2}\mathrm{J}_\phi(0)\mathbf{1}\mathbf{1}^T$$

Simplification with Defn E.7 and Proposition E.21 gives the result. $\square$

## H.2 LOCAL CONVERGENCE

We now look at the linearized dynamics around CBSB1 fixed points of Eq. (54), and investigate the eigen-properties of $\left.\frac{\mathrm{dV}_\phi^{\mathcal{B}\oplus 2}}{\mathrm{d}\widetilde{\Sigma}}\right|_{\widetilde{\Sigma}=\widetilde{\Sigma}^*}$. By Lemma F.17, its nonzero eigenvalues are exactly those of $\tilde{\mathcal{F}} := (G^{\oplus 2})^{\otimes 2} \circ \left.\frac{\mathrm{dV}_{(\phi \circ n)\oplus 2}(\widetilde{\Sigma}^G)}{\mathrm{d}\widetilde{\Sigma}^G}\right|_{\widetilde{\Sigma}^G = \mu^* G^{\oplus 2}}$, $\tilde{\mathcal{F}}\{\Lambda\} = G^{\oplus 2}\left(\left.\frac{\mathrm{dV}_{(\phi \circ n)\oplus 2}}{\mathrm{d}\widetilde{\Sigma}^G}\right|_{\mu^* G^{\oplus 2}}\{\Lambda\}\right)G^{\oplus 2}$. Here $\widetilde{\Sigma}^G = G^{\oplus 2}\widetilde{\Sigma}G^{\oplus 2}$. Thus it suffices to obtain the eigendecomposition of $\tilde{\mathcal{F}}$.

First, notice that $\tilde{\mathcal{F}}$ acts on the diagonal blocks as $G^{\otimes 2} \circ \frac{\mathrm{dV}_{\phi \circ n}}{\mathrm{d}\Sigma^G}$, and the off-diagonal components of $\Lambda$ has no effect on the diagonal blocks of $\tilde{\mathcal{F}}\{\Lambda\}$. We formalize this notion as follows

**Definition H.7.** Let $\mathcal{T} : \mathcal{H}_{kB} \to \mathcal{H}_{kB}$ be a linear operator such that for any block matrix $\widetilde{\Sigma} \in \mathcal{H}_{kB}$ with $k \times k$ blocks $\widetilde{\Sigma}^{ij}, i, j \in [k]$, of size $B$, we have

$$\mathcal{T}\{\widetilde{\Sigma}\}^{ii} = \mathcal{U}\{\widetilde{\Sigma}^{ii}\}$$

$$\mathcal{T}\{\widetilde{\Sigma}\}^{ij} = \mathcal{V}\{\widetilde{\Sigma}^{ii}\} + \mathcal{V}\{\widetilde{\Sigma}^{jj}\}^T + \mathcal{W}\{\widetilde{\Sigma}^{ij}\}, \forall i \neq j,$$

where $\mathcal{U} : \mathcal{H}_B \to \mathcal{H}_B, \mathcal{V} : \mathcal{H}_B \to \mathbb{R}^{B \times B}$, and $\mathcal{W} : \mathbb{R}^{B \times B} \to \mathbb{R}^{B \times B}$ are linear operators, and $\mathcal{T}\{\widetilde{\Sigma}\}^{ij}$ denotes the $(i, j)$th block of $\mathcal{T}\{\widetilde{\Sigma}\}$. We say that $\mathcal{T}$ is *blockwise diagonal-off-diagonal semidirect*, or BDOS for short. We write more specifically $\mathcal{T} = \mathrm{BDOS}(\mathcal{U}, \mathcal{V}, \mathcal{W})$.

We have $\tilde{\mathcal{F}} = \mathrm{BDOS}(\mathcal{U}, \mathcal{V}, \mathcal{W})$ where

$$\mathcal{U} = G^{\otimes 2} \circ \left.\frac{\mathrm{d}V_{\phi \mathrm{on}}}{\mathrm{d}\Sigma^G}\right|_{\Sigma^G = \mu^* G}$$

$$\mathcal{V} = G^{\otimes 2} \circ \frac{\mathrm{d}}{\mathrm{d}(\widetilde{\Sigma}^G)^{11}} \left.\underset{(h,h') \sim \mathcal{N}(0, \widetilde{\Sigma}^G)}{\mathbb{E}} \phi(\mathsf{n}(h)) \otimes \phi(\mathsf{n}(h'))\right|_{\widetilde{\Sigma}^G = \mu^* G^{\oplus 2}}$$

$$\mathcal{W} = 2G^{\otimes 2} \circ \frac{\mathrm{d}}{\mathrm{d}(\widetilde{\Sigma}^G)^{12}} \left.\underset{(h,h') \sim \mathcal{N}(0, \widetilde{\Sigma}^G)}{\mathbb{E}} \phi(\mathsf{n}(h)) \otimes \phi(\mathsf{n}(h'))\right|_{\widetilde{\Sigma}^G = \mu^* G^{\oplus 2}}. \tag{57}$$

Note that the factor 2 in $\mathcal{W}$ is due to the contribution of $(\widetilde{\Sigma}^G)^{21}$ by symmetry. We will first show that $\mathcal{W}$ is multiplication by a constant.

### H.2.1 SPHERICAL INTEGRATION AND GEGENBAUER EXPANSION

**Theorem H.8.** *Suppose $\phi(\sqrt{B-1}x)$ has Gegenbauer expansion $\sum_{l=0}^{\infty} a_l \frac{1}{c_{B-1,l}} C_l^{(\frac{B-3}{2})}(x)$. Then for any $\Lambda \in \mathbb{R}^{B \times B}$ satisfying $G\Lambda G = \Lambda$,*

$$\mathcal{W}\{\Lambda\} = 2a_1^2 \mu^{*-1} \mathrm{P}\left(\frac{B-1}{2}, \frac{1}{2}\right)^2 \frac{B}{B-1}\Lambda$$

$$= \frac{a_1^2 \frac{1}{2} \mathrm{P}\left(\frac{B}{2}, \frac{1}{2}\right)^{-2} B(B-1)}{\sum_{l=0}^{\infty} a_l^2 \frac{1}{c_{B-1,l}} \left(C_l^{(\frac{B-3}{2})}(1) - C_l^{(\frac{B-3}{2})}\left(\frac{-1}{B-1}\right)\right)}\Lambda$$

*Proof.* Let $J = \frac{\mathrm{d}}{\mathrm{d}(\widetilde{\Sigma}^G)^{12}} \left.\mathbb{E}_{(h,h') \sim \mathcal{N}(0, \widetilde{\Sigma}^G)} \phi(\mathsf{n}(h)) \otimes \phi(\mathsf{n}(h'))\right|_{\widetilde{\Sigma}^G = \mu^* G^{\oplus 2}}$. Combining Lemma F.18 with projection by $\mathbb{e}$, we get

$$J\{\Lambda\} = \frac{1}{2} \underset{y,y' \sim \mathcal{N}(0, \mu^* I_{B-1})}{\mathbb{E}} \phi\left(\mathsf{n}(\mathbb{e}y)\right) \otimes \phi\left(\mathsf{n}(\mathbb{e}y')\right) \langle \mu^{*-2} yy'^T - \mu^{*-1} I_{B-1}, \mathbb{e}^T \Lambda \mathbb{e} \rangle$$

$$= \frac{1}{2} \mu^{*-1} \underset{y,y' \sim \mathcal{N}(0, I_{B-1})}{\mathbb{E}} \phi\left(\mathsf{n}(\mathbb{e}y)\right) \otimes \phi\left(\mathsf{n}(\mathbb{e}y')\right) (y'^T \mathbb{e}^T \Lambda \mathbb{e}y - \mathrm{tr}\,\Lambda)$$

$$= \frac{1}{2} \mu^{*-1} \left(\left(\sqrt{2}\mathrm{P}\left(\frac{B-1}{2}, \frac{1}{2}\right)\right)^2 \underset{v,v' \sim S^{B-2}}{\mathbb{E}} \phi\left(\sqrt{B}\mathbb{e}v\right) \otimes \phi\left(\sqrt{B}\mathbb{e}v'\right) (v'^T \mathbb{e}^T \Lambda \mathbb{e}v)\right.$$

$$\left. - (\mathrm{tr}\,\Lambda) \underset{v,v' \sim S^{B-2}}{\mathbb{E}} \phi\left(\sqrt{B}\mathbb{e}v\right) \otimes \phi\left(\sqrt{B}\mathbb{e}v'\right)\right)$$

by Proposition E.33.

Note that $\mathbb{E}_{v,v' \sim S^{B-2}} \phi\left(\sqrt{B}\mathbb{e}v\right) \otimes \phi\left(\sqrt{B}\mathbb{e}v'\right) = \left(\mathbb{E}_{v \sim S^{B-2}} \phi\left(\sqrt{B}\mathbb{e}v\right)\right)^{\otimes 2}$ is a constant matrix, because of the independence of $v$ from $v'$ and the spherical symmetry of the integrand. Thus this term drops out in $G^{\otimes 2} \circ J\{\Lambda\}$.

Next, we analyze $H\{\Lambda\} = \mathbb{E}_{v,v' \sim S^{B-2}} \phi\left(\sqrt{B}\mathbb{e}v\right) \otimes \phi\left(\sqrt{B}\mathbb{e}v'\right) (v'^T \mathbb{e}^T \Lambda \mathbb{e}v)$. This is a linear function in $\Lambda$ with coefficients

$$H_{ab|cd} = \frac{B-1}{B} \left(\underset{v \sim S^{B-2}}{\mathbb{E}} \phi(\sqrt{B-1}\hat{\mathbb{e}}_{b,:}v)(\hat{\mathbb{e}}_{c,:}v)\right) \left(\underset{v \sim S^{B-2}}{\mathbb{E}} \phi(\sqrt{B-1}\hat{\mathbb{e}}_{a,:}v)(\hat{\mathbb{e}}_{d,:}v)\right).$$

By the usual zonal spherical harmonics argument,

$$\mathbb{E}_{v\sim S^{B-2}} \phi(\sqrt{B-1}\hat{e}_{b,:}v)(\hat{e}_{c,:}v)$$

$$= \left\langle \sum_{l=0}^{\infty} a_l \frac{1}{c_{B-1,l}} C_l^{(\frac{B-3}{2})}(\langle \hat{e}_{b,:}, \cdot \rangle), \langle \hat{e}_{c,:}, \cdot \rangle \right\rangle_{S^{B-2}}$$

$$= \left\langle \sum_{l=0}^{\infty} a_l Z_{\hat{e}_{b,:}}^{B-2,(l)}, \frac{1}{B-1} Z_{\hat{e}_{c,:}}^{B-2,(1)} \right\rangle_{S^{B-2}}$$

$$= \frac{a_1}{B-1} Z_{\hat{e}_{c,:}}^{B-2,(1)}(\hat{e}_{b,:})$$

$$= \frac{a_1}{B-1} \frac{1}{c_{B-1,1}} C_1^{(\frac{B-3}{2})}(\langle \hat{e}_{b,:}, \hat{e}_{c,:}\rangle)$$

$$= \begin{cases} a_1 & \text{if } b = c \\ \frac{-a_1}{B-1} & \text{otherwise.} \end{cases}$$

Thus,

$$H\{\Lambda\}_{ab} = \frac{B-1}{B} \left( \left( \frac{-a_1}{B-1} \right)^2 \sum_{c,d\in[B]} \Lambda_{cd} \right.$$

$$+ \left( a_1 - \frac{-a_1}{B-1} \right) \left( \frac{-a_1}{B-1} \right) \sum_{c=b \text{ XOR } d=a} \Lambda_{cd}$$

$$+ \left. \left( a_1^2 - \frac{a_1^2}{(B-1)^2} \right) \Lambda_{ab} \right)$$

$$= \frac{B-1}{B} \left( -2\left( a_1 - \frac{-a_1}{B-1} \right) \left( \frac{-a_1}{B-1} \right) \Lambda_{ab} + \left( a_1^2 - \frac{a_1^2}{(B-1)^2} \right) \Lambda_{ab} \right)$$

because rows and columns of $\Lambda$ sum to 0

$$= a_1^2 \frac{B}{B-1} \Lambda_{ab}.$$

Therefore,

$$\mathcal{W}\{\Lambda\} = 2G^{\otimes 2} \circ J\{\Lambda\}$$

$$= 2a_1^2 \mu^{*-1} \mathrm{P}\left( \frac{B-1}{2}, \frac{1}{2} \right)^2 \frac{B}{B-1} \Lambda$$

$$= \frac{a_1^2 \frac{1}{2} \mathrm{P}\left( \frac{B}{2}, \frac{1}{2} \right)^{-2} B(B-1)}{\sum_{l=0}^{\infty} a_l^2 \frac{1}{c_{B-1,l}} \left( C_l^{(\frac{B-3}{2})}(1) - C_l^{(\frac{B-3}{2})}\left( \frac{-1}{B-1} \right) \right)} \Lambda$$

where we used the identity $\mathrm{P}\left( \frac{B-1}{2}, \frac{1}{2} \right) \left( \frac{2}{B-1} \right) = \mathrm{P}\left( \frac{B}{2}, \frac{1}{2} \right)^{-1}$. $\qquad \square$

Next, we step back a bit to give a full treatment of the eigen-properties of BDOS operators. Then we specialize back to our case and give the result for $\tilde{\mathcal{F}}$.

**Definition H.9.** Let $\mathcal{H}_{kB}^G := \{\widetilde{\Sigma} \in \mathcal{H}_{kB} : G^{\oplus k}\widetilde{\Sigma}G^{\oplus k} = \widetilde{\Sigma}\}$ be the subspace of $\mathcal{H}_{kB}$ stable under conjugation by $G^{\oplus k}$.

**Definition H.10.** Let $\widetilde{\mathbb{M}}_{kB} \in \mathcal{H}_{kB}^G$ be the space of block matrices $\widetilde{\Sigma} \in \mathcal{H}_{kB}^G$ where the diagonal blocks $\widetilde{\Sigma}^{ii}$ are all zero. Likewise, let $\overline{\widetilde{\mathbb{M}}}_{kB} \in \mathcal{H}_{kB}$ be the space of block matrices $\widetilde{\Sigma} \in \mathcal{H}_{kB}$ where the diagonal blocks $\widetilde{\Sigma}^{ii}$ are all zero. We suppress the subscript $kB$ when it is clear from the context.

**Definition H.11.** Define the *block L-shaped matrix*

$$\overline{\mathrm{L}}_{kB}(\Sigma, \Lambda) := \begin{pmatrix} \Sigma & -\Lambda & \cdots \\ -\Lambda^T & 0 & \cdots \\ \vdots & \vdots & \ddots \end{pmatrix}$$

BDOS operators of the form $\mathrm{BDOS}(\mathcal{U}, \mathcal{V}, w)$, where $w$ stands for multiplication by a constant $w$, like $\tilde{\mathcal{F}}$, have simple eigendecomposition, just like DOS operators.

**Theorem H.12.** *If $\mathcal{V} - wI_B$ is not singular, then $\mathrm{BDOS}(\mathcal{V}, \mathcal{U}, w) : \mathcal{H}_{kB} \to \mathcal{H}_{kB}$ has the following eigenspaces and eigenvectors*

1. *$\overline{\widetilde{\mathbb{M}}}$ with associated eigenvalue $w$.*

2. *For each eigenvalue $\lambda$ and eigenvector $\Sigma \in \mathcal{H}_B$ of $\mathcal{U}$, the block matrix*

$$\overline{\mathrm{L}}_{kB}(\Sigma, -(\lambda - w)^{-1}\mathcal{V}\{\Sigma\}) = \begin{pmatrix} \Sigma & (\lambda - w)^{-1}\mathcal{V}\{\Sigma\} & \cdots \\ (\lambda - w)^{-1}\mathcal{V}\{\Sigma\}^T & 0 & \cdots \\ \vdots & \vdots & \ddots \end{pmatrix}$$

   *along with any matrix obtained from permutating its block columns and rows simultaneously. The associated eigenvalue is $\lambda$.*

*If $\mathrm{BDOS}(\mathcal{V}, \mathcal{U}, w)$ also restricts to $\mathcal{H}_{kB}^G \to \mathcal{H}_{kB}^G$, then with respect to this signature, the eigendecomposition is obtained by replacing $\mathcal{H}_{kB}$ with $\mathcal{H}_{kB}^G$ in the above:*

1. *Block diagonal matrices $\widetilde{\Sigma} \in \mathcal{H}_{kB}^G$ where the diagonal blocks $\widetilde{\Sigma}^{ii}$ are all zero. The associated eigenvalue is $w$.*

2. *For each eigenvalue $\lambda$ and eigenvector $\Sigma \in \mathcal{H}_B^G$ of $\mathcal{U}$, the matrix $\overline{\mathrm{L}}_{kB}(\Sigma, -(\lambda - w)^{-1}\mathcal{V}\{\Sigma\})$ along with any matrix obtained from permutating its block columns and rows simultaneously. The associated eigenvalue is $\lambda$.*

*Proof.* The proofs are similar in both cases: One can easily verify that these are eigenvectors and eigenvalues. Then by dimensionality considerations, these must be all of them. □

By Thms H.8 and H.12, we easily obtain

**Theorem H.13.** *Suppose $\phi(\sqrt{B-1}x)$ has Gegenbauer expansion $\sum_{l=0}^{\infty} a_l \frac{1}{c_{B-1,l}} C_l^{(\frac{B-3}{2})}(x)$. The eigendecomposition of $\tilde{\mathcal{F}} : \mathcal{H}_{2B}^G \to \mathcal{H}_{2B}^G$ is given below, as long as $\lambda_{\widetilde{\mathbb{M}}}^{\uparrow} \neq \lambda_{\mathbb{L}}^{\uparrow}, \lambda_{\mathbb{M}}^{\uparrow}$.*

1. *$\widetilde{\mathbb{M}}$ with associated eigenvalue*

$$\lambda_{\widetilde{\mathbb{M}}}^{\uparrow} := 2a_1^2 \mu^{*-1} \mathrm{P}\left(\frac{B-1}{2}, \frac{1}{2}\right)^2 \frac{B}{B-1}$$

$$= \frac{a_1^2 \frac{1}{2} \mathrm{P}\left(\frac{B}{2}, \frac{1}{2}\right)^{-2} B(B-1)}{\sum_{l=0}^{\infty} a_l^2 \frac{1}{c_{B-1,l}} \left(C_l^{(\frac{B-3}{2})}(1) - C_l^{(\frac{B-3}{2})}\left(\frac{-1}{B-1}\right)\right)}.$$

2. *The space generated by $\overline{\mathrm{L}}_{2B}(G, -(\lambda_{\mathbb{L}}^{\uparrow} - \lambda_{\widetilde{\mathbb{M}}}^{\uparrow})^{-1}\mathcal{V}\{G\})$ along with any matrix obtained from permutating its block columns and rows simultaneously. The associated eigenvalue is 0.*

3. *The space generated by $\overline{\mathrm{L}}_{2B}(L, -(\lambda_{\mathbb{L}}^{\uparrow} - \lambda_{\widetilde{\mathbb{M}}}^{\uparrow})^{-1}\mathcal{V}\{L\})$, where $L = \mathrm{L}_B(B-2, 1)$, along with any matrix obtained from permutating its block columns and rows simultaneously. The associated eigenvalue is $\lambda_{\mathbb{L}}^{\uparrow}$.*

4. *The space generated by $\overline{\mathrm{L}}_{2B}(M, -(\lambda_{\mathbb{M}}^{\uparrow} - \lambda_{\widetilde{\mathbb{M}}}^{\uparrow})^{-1}\mathcal{V}\{M\})$, for any $M \in \mathbb{M}$, along with any matrix obtained from permutating its block columns and rows simultaneously. The associated eigenvalue is $\lambda_{\mathbb{M}}^{\uparrow}$.*

*Here, $\mathcal{V}$ is as in Eq. (57). The local convergence dynamics of Eq. (54) to the CBSB1 fixed point $\widetilde{\Sigma}^*$ has the same eigenvalues $\lambda_{\widetilde{\mathbb{M}}}^{\uparrow}, \lambda_{\mathbb{L}}^{\uparrow}, \lambda_{\mathbb{M}}^{\uparrow}$, and 0.*

Note that, to reason about $\tilde{\mathcal{F}}$ extended over $k$ batches, one can replace $2B$ with $kB$ everywhere in Thm H.13 and the theorem will still hold. As we will see below, $\lambda_{\widetilde{\mathbb{M}}}^{\uparrow}$ is the same as the eigenvalue governing the dynamics of the cross batch correlations in the gradient propagation.

### H.2.2 LAPLACE METHOD

As with the single batch case, with positive homogeneous $\phi$, we can obtain a more direct answer of the new eigenvalue $\lambda_{\widetilde{\mathbb{M}}}^{\uparrow}$ in terms of the J function of $\phi$.

**Theorem H.14.** *Let $\phi$ be degree $\alpha$ positive homogeneous. Then the eigenvalue of $\tilde{\mathcal{F}}$ on $\widetilde{\mathbb{M}}$ is*

$$\lambda_{\widetilde{\mathbb{M}}}^{\uparrow} = \frac{B}{2}\left(\frac{B-1}{2}\right)^{\alpha-1} \mathrm{P}\left(\frac{B}{2}, \frac{\alpha}{2}\right)^{-2} \mu^{*-1} \mathsf{c}_\alpha \mathrm{J}_\phi'(0)$$

$$= \frac{B}{B-1} \frac{\mathrm{P}\left(\frac{B-1}{2}, \alpha\right)}{\mathrm{P}\left(\frac{B}{2}, \frac{\alpha}{2}\right)^2} \frac{\mathrm{J}_\phi'(0)}{\mathrm{J}_\phi(1) - \mathrm{J}_\phi\left(\frac{-1}{B-1}\right)}$$

*Proof.* Let $\widetilde{\Sigma}^* = \begin{pmatrix} \Sigma^* & c^*\mathbf{11}^T \\ c^*\mathbf{11}^T & \Sigma^* \end{pmatrix}$ be the CBSB1 fixed point in Thm H.6. Take a smooth path $\widetilde{\Sigma}_\tau = \begin{pmatrix} \Sigma_\tau & \Xi_\tau \\ \Xi_\tau^T & \Sigma_\tau' \end{pmatrix} \in \mathcal{S}_{2B}, \tau \in (-\epsilon, \epsilon)$ such that $\widetilde{\Sigma}_0 = \widetilde{\Sigma}^*, \dot{\widetilde{\Sigma}}_0 := \frac{d}{d\tau}\widetilde{\Sigma} = \begin{pmatrix} 0 & \Upsilon \\ \Upsilon^T & 0 \end{pmatrix}$ for some $\Upsilon \in \mathbb{R}^{B \times B}$. Then with $\Omega_\tau = \begin{pmatrix} s\Sigma_\tau^G & \sqrt{st}\Xi_\tau^G \\ \sqrt{st}(\Xi_\tau^G)^T & t\Sigma_\tau'^G \end{pmatrix}$ and $\Pi_\tau = D^{-1}\Omega_\tau(I + 2\Omega_\tau)^{-1}D^{-1}$ with $D = \sqrt{s}I_B \oplus \sqrt{t}I_B = \begin{pmatrix} \sqrt{s}I_B & 0 \\ 0 & \sqrt{t}I_B \end{pmatrix}$,

$$\frac{d}{d\tau}\mathbb{E}[\mathcal{B}_\phi(z) \otimes \mathcal{B}_\phi(z') : (z, z') \sim \mathcal{N}(0, \widetilde{\Sigma})]\Big|_{\tau=0}$$

$$= \frac{d}{d\tau}B^\alpha \Gamma(\alpha/2)^{-2} \int_0^\infty \mathrm{d}s \int_0^\infty \mathrm{d}t\, (st)^{\alpha/2-1} \det(I_{2B} + 2\Omega)^{-1/2} \mathrm{V}_\phi(\Pi)_{1:B,B+1:2B}\Big|_{\tau=0}$$

$$= B^\alpha \Gamma(\alpha/2)^{-2} \int_0^\infty \mathrm{d}s \int_0^\infty \mathrm{d}t\, (st)^{\alpha/2-1} \left(\frac{d}{d\tau}\det(I_{2B} + 2\Omega)^{-1/2}\Big|_{\tau=0} \mathrm{V}_\phi(\Pi)_{1:B,B+1:2B}\right.$$

$$\left. + \det(I_{2B} + 2\Omega)^{-1/2}\frac{d}{d\tau}\mathrm{V}_\phi(\Pi)_{1:B,B+1:2B}\Big|_{\tau=0}\right)$$

$$= B^\alpha \Gamma(\alpha/2)^{-2} \int_0^\infty \mathrm{d}s \int_0^\infty \mathrm{d}t\, (st)^{\alpha/2-1} \left(-\det(I + 2\Omega_0)^{-1/2} \mathrm{tr}\left((I + 2\Omega_0)^{-1}\dot{\Omega}_0\right)\mathrm{V}_\phi(\Pi)\right.$$

$$\left. + \det(I + 2\Omega_0)^{-1/2}(D^{-\alpha})^{\otimes 2} \circ \mathcal{J}\left\{\frac{d}{d\tau}\Omega(I + 2\Omega)^{-1}\Big|_{\tau=0}\right\}\right)_{1:B,B+1:2B} \tag{58}$$

by chain rule and Lemma F.26, where $\mathcal{J} = \frac{\mathrm{dV}_\phi}{\mathrm{d}\Sigma}\Big|_{\Sigma=\Omega_0(I+2\Omega_0)^{-1}}$. We compute

$$\Omega_0 = \mu^*(sG \oplus tG)$$

$$\dot{\Omega}_0 = \sqrt{st}\begin{pmatrix} 0 & \Upsilon^G \\ (\Upsilon^G)^T & 0 \end{pmatrix}$$

$$(I + 2\Omega_0)^{-1} = ((1 + 2s\mu^*)^{-1}G + \frac{1}{B}\mathbf{11}^T) \oplus ((1 + 2t\mu^*)^{-1}G + \frac{1}{B}\mathbf{11}^T)$$

$$\det(I + 2\Omega_0) = (1 + 2s\mu^*)^{B-1}(1 + 2t\mu^*)^{B-1}$$

$$\Omega_0(I + 2\Omega_0)^{-1} = \frac{s\mu^*}{1 + 2s\mu^*}G \oplus \frac{t\mu^*}{1 + 2t\mu^*}G.$$

So then

$$\mathcal{J} = \left( \left( \frac{s\mu^*}{1+2s\mu^*} \right)^{(\alpha-1)/2} I_B \oplus \left( \frac{t\mu^*}{1+2t\mu^*} \right)^{(\alpha-1)/2} I_B \right)^{\otimes 2} \circ \left( \frac{B-1}{B} \right)^{\alpha-1} \frac{dV_\phi}{d\Sigma} \Bigg|_{\Sigma=\mathrm{BSB1}(1,\frac{-1}{B-1})^{\oplus 2}}$$

With

$$\frac{d}{d\tau} \Omega(I+2\Omega)^{-1} \bigg|_{\tau=0} = (I+2\Omega_0)^{-1}\dot\Omega_0(I+2\Omega_0)^{-1}$$

$$\text{(by Lemma F.27)}$$

$$= \frac{\sqrt{st}}{(1+2s\mu^*)(1+2t\mu^*)} \begin{pmatrix} 0 & \Upsilon^G \\ (\Upsilon^G)^T & 0 \end{pmatrix}$$

we have

$$\mathcal{J} \left\{ \frac{d}{d\tau} \Omega(I+2\Omega)^{-1} \bigg|_{\tau=0} \right\}$$

$$= \left( \frac{B-1}{B} \right)^{\alpha-1} \frac{\sqrt{st}}{(1+2s\mu^*)(1+2t\mu^*)} \mathsf{c}_\alpha \mathrm{J}'_\phi(0)$$

$$\times \left( \frac{s\mu^*}{1+2s\mu^*} \right)^{(\alpha-1)/2} \left( \frac{t\mu^*}{1+2t\mu^*} \right)^{(\alpha-1)/2} \begin{pmatrix} 0 & \Upsilon^G \\ (\Upsilon^G)^T & 0 \end{pmatrix}$$

$$= \left( \frac{B-1}{B} \right)^{\alpha-1} \frac{(st)^{\alpha/2}\mu^{*\alpha-1}\mathsf{c}_\alpha \mathrm{J}'_\phi(0)}{(1+2s\mu^*)^{(\alpha+1)/2}(1+2t\mu^*)^{(\alpha+1)/2}} \begin{pmatrix} 0 & \Upsilon^G \\ (\Upsilon^G)^T & 0 \end{pmatrix}.$$

Thus the product $(I+2\Omega_0)^{-1}\dot\Omega_0$ has zero diagonal blocks so that its trace is 0. Therefore, only the second term in the sum in Eq. (58) above survives, and we have

$$\frac{d}{d\tau} \mathbb{E}[\mathcal{B}_\phi(z) \otimes \mathcal{B}_\phi(z') : (z,z') \sim \mathcal{N}(0,\widetilde\Sigma)] \bigg|_{\tau=0}$$

$$= B^\alpha \Gamma(\alpha/2)^{-2} \int_0^\infty \mathrm{d}s \int_0^\infty \mathrm{d}t \, (st)^{\alpha/2-1}(1+2s\mu^*)^{-\frac{B-1}{2}}(1+2t\mu^*)^{-\frac{B-1}{2}} \times$$

$$(st)^{-\alpha/2} \left( \frac{B-1}{B} \right)^{\alpha-1} \frac{(st)^{\alpha/2}\mu^{*\alpha-1}\mathsf{c}_\alpha \mathrm{J}'_\phi(0)}{(1+2s\mu^*)^{(\alpha+1)/2}(1+2t\mu^*)^{(\alpha+1)/2}} \Upsilon^G$$

$$= B(B-1)^{\alpha-1}\Gamma(\alpha/2)^{-2} \int_0^\infty \mathrm{d}s \int_0^\infty \mathrm{d}t \, (st)^{\alpha/2-1} \frac{\mu^{*\alpha-1}\mathsf{c}_\alpha \mathrm{J}'_\phi(0)}{(1+2s\mu^*)^{(B+\alpha)/2}(1+2t\mu^*)^{(B+\alpha)/2}} \Upsilon^G$$

$$= B(B-1)^{\alpha-1}\Gamma(\alpha/2)^{-2}\mu^{*-1}\mathsf{c}_\alpha \mathrm{J}'_\phi(0) \left( \int_0^\infty \mathrm{d}(\mu^*s) \frac{(\mu^*s)^{\alpha/2-1}}{(1+2s\mu^*)^{(B+\alpha)/2}} \right)^2 \Upsilon^G$$

$$= B(B-1)^{\alpha-1}\Gamma(\alpha/2)^{-2}\mu^{*-1}\mathsf{c}_\alpha \mathrm{J}'_\phi(0) \left( 2^{-\alpha/2}\mathrm{Beta}\left( \frac{B}{2}, \frac{\alpha}{2} \right) \right)^2 \Upsilon^G$$

$$= \frac{B}{2} \left( \frac{B-1}{2} \right)^{\alpha-1} \mathrm{P}\left( \frac{B}{2}, \frac{\alpha}{2} \right)^{-2} \mu^{*-1}\mathsf{c}_\alpha \mathrm{J}'_\phi(0)\Upsilon^G$$

$$\square$$

## I  CROSS BATCH: BACKWARD DYNAMICS

For $k \geq 2$, we wish to study the following generalization of Eq. (51),

$$\tilde\Pi^l = \mathrm{V}_{\mathcal{B}_\phi^{\oplus k'}}(\widetilde\Sigma^l)^\dagger\{\tilde\Pi^{l+1}\}.$$

As in the single batch case, we approximate it by taking $\widetilde\Sigma^l$ to its CBSB1 limit $\widetilde\Sigma^*$, so that we analyze

$$\tilde\Pi^l = \mathrm{V}_{\mathcal{B}_\phi^{\oplus k'}}(\widetilde\Sigma^*)^\dagger\{\tilde\Pi^{l+1}\} \tag{59}$$

Its dynamics is then given by the eigendecomposition of $V_{\mathcal{B}_\phi^{\oplus k'}}(\widetilde{\Sigma}^*)$, in parallel to the single batch scenario.

WLOG, we only need to consider the case $k = 2$.

$$V_{\mathcal{B}_\phi^{\oplus 2'}}(\widetilde{\Sigma}^*) = \underset{(x,y)\sim\mathcal{N}(0,\widetilde{\Sigma}^*)}{\mathbb{E}}[(\mathcal{B}'_\phi(x) \oplus \mathcal{B}'_\phi(y))^{\otimes 2}]$$

$$V_{\mathcal{B}_\phi^{\oplus 2'}}(\widetilde{\Sigma}^*)\left\{\begin{pmatrix}\Sigma & \Xi \\ \Xi^T & \Lambda\end{pmatrix}\right\} = \underset{(x,y)\sim\mathcal{N}(0,\widetilde{\Sigma}^*)}{\mathbb{E}}\left[\begin{pmatrix}\mathcal{B}'_\phi(x)\Sigma\mathcal{B}'_\phi(x)^T & \mathcal{B}'_\phi(x)\Xi\mathcal{B}'_\phi(y)^T \\ \mathcal{B}'_\phi(y)\Xi^T\mathcal{B}'_\phi(x)^T & \mathcal{B}'_\phi(y)\Lambda\mathcal{B}'_\phi(y)^T\end{pmatrix}\right]$$

From this one sees that $V_{[(\mathcal{B}_\phi^{\oplus 2})']}$ acts independently on each block, and consequently so does its adjoint. The diagonal blocks of Eq. (59) evolves according to Eq. (51) which we studied in Appendix G. In this section we will study the evolution of the off-diagonal blocks, which is given by

$$\Xi^l = \underset{(x,y)\sim\mathcal{N}(0,\widetilde{\Sigma}^*)}{\mathbb{E}}[\mathcal{B}'_\phi(x)\otimes\mathcal{B}'_\phi(y)]^\dagger\{\Xi^{l+1}\} = \underset{(x,y)\sim\mathcal{N}(0,\widetilde{\Sigma}^*)}{\mathbb{E}}[\mathcal{B}'_\phi(y)^T\otimes\mathcal{B}'_\phi(x)^T]\{\Xi^{l+1}\} \quad (60)$$

$$= \underset{(x,y)\sim\mathcal{N}(0,\widetilde{\Sigma}^*)}{\mathbb{E}}[\mathcal{B}'_\phi(y)^T\Xi^{l+1}\mathcal{B}'_\phi(x)^T]$$

**Definition I.1.** Let $\mathcal{M}_B := \mathbb{R}^{B\times B}$ be the set of all $B \times B$ real matrices, and let $\mathcal{M}_B^G := \{\Xi \in \mathcal{M}_B : G\Xi G = \Xi\}$.

Define $\mathcal{T} := \mathbb{E}_{(x,y)\sim\mathcal{N}(0,\widetilde{\Sigma}^*)}[\mathcal{B}'_\phi(x)\otimes\mathcal{B}'_\phi(y)]$. As in the single batch case, after one step of backprop, $\Xi^{L-1} = \mathcal{V}^\dagger\{\Xi^L\}$ is in $\mathcal{M}_B^G$. Thus it suffices to study the eigendecomposition of $G^{\otimes 2}\circ\mathcal{T}^\dagger\upharpoonright\mathcal{M}_B^G : \mathcal{M}_B^G \to \mathcal{M}_B^G$. Below, we will show that its adjoint $\mathcal{T}\circ G^{\otimes 2}$ acting on $\mathcal{M}_B^G$ is in fact multiplication by a constant, and thus so is it.

We first compute

$$\mathcal{T} = \underset{(\xi,\eta)\sim\mathcal{N}(0,G^{\oplus 2}\widetilde{\Sigma}^* G^{\oplus 2})}{\mathbb{E}}\left[\left.\frac{\mathrm{d}\phi\circ\mathsf{n}(z)}{\mathrm{d}z}\right|_{z=\xi}\otimes\left.\frac{\mathrm{d}\phi\circ\mathsf{n}(z)}{\mathrm{d}z}\right|_{z=\eta}\right]\circ G^{\otimes 2}$$

$$= \underset{(\xi,\eta)\sim\mathcal{N}(0,\mu^*\left(\begin{smallmatrix}G & 0 \\ 0 & G\end{smallmatrix}\right))}{\mathbb{E}}\left[\left.\frac{\mathrm{d}\phi\circ\mathsf{n}(z)}{\mathrm{d}z}\right|_{z=\xi}\otimes\left.\frac{\mathrm{d}\phi\circ\mathsf{n}(z)}{\mathrm{d}z}\right|_{z=\eta}\right]\circ G^{\otimes 2}$$

since $\widetilde{\Sigma}^*$ is CBSB1 with diagonal blocks $\Sigma^*$ (Corollary H.4). Then $\xi$ is independent from $\eta$, so this is just

$$\underset{(x,y)\sim\mathcal{N}(0,\widetilde{\Sigma}^*)}{\mathbb{E}}[\mathcal{B}'_\phi(x)\otimes\mathcal{B}'_\phi(y)] = \underset{\xi\sim\mathcal{N}(0,\mu^*G)}{\mathbb{E}}\left[\left.\frac{\mathrm{d}\phi\circ\mathsf{n}(z)}{\mathrm{d}z}\right|_{z=\xi}\right]^{\otimes 2}\circ G^{\otimes 2}$$

$$= \underset{\xi\sim\mathcal{N}(0,\mu^*G)}{\mathbb{E}}\left[\sqrt{B}Dr^{-1}(I - vv^T)\right]^{\otimes 2}\circ G^{\otimes 2}$$

(by Proposition G.4)

where $D = \mathrm{Diag}(\phi'(\sqrt{B}v)), r = \|\xi\|, v = \xi/\|\xi\|$. But notice that $T := \mathbb{E}_{\xi\sim\mathcal{N}(0,\mu^*G)}[Dr^{-1}(I - vv^T)]$ is actually BSB1, with diagonal

$$\underset{rv\sim\mathcal{N}(0,\mu^*G)}{\mathbb{E}}r^{-1}(1-v_i^2)\phi'(\sqrt{B}v_i) = \underset{rv\sim\mathcal{N}(0,\mu^*G)}{\mathbb{E}}r^{-1}(1-v_1^2)\phi'(\sqrt{B}v_1), \forall i \in [B]$$

and off-diagonal

$$\underset{rv\sim\mathcal{N}(0,\mu^*G)}{\mathbb{E}}r^{-1}(-v_iv_j)\phi'(\sqrt{B}v_i) = \underset{rv\sim\mathcal{N}(0,\mu^*G)}{\mathbb{E}}r^{-1}(-v_1v_2)\phi'(\sqrt{B}v_1), \forall i \neq j \in [B].$$

Thus by Lemma E.51, $T$ has two eigenspaces, $\{x : Gx = x\}$ and $\mathbb{R}\mathbf{1}$, with respective eigenvalues $T_{11} - T_{12}$ and $T_{11} + (B-1)T_{12}$. However, because of the composition with $G^{\otimes 2}$, only the former eigenspace survives with nonzero eigenvalue in $G^{\otimes 2}\circ\mathcal{T}$.

**Theorem I.2.** $\mathcal{T}$ on $\mathcal{M}_B^G$ is just multiplication by $\lambda_{\mathbb{M}}^{\downarrow} :=$ $\frac{1}{2\pi} B\mu^{*-1} \left( \int_0^\pi \mathrm{d}\theta \ \sin^{B-1}\theta \phi'(-\sqrt{B-1}\cos\theta) \right)^2$.

*Proof.* By the above reasoning, the sole eigenspace with nonzero eigenvalue is $\{x : Gx = x\}^{\otimes 2} = \mathcal{M}_B^G$. It has eigenvalue

$$B \left( \mathbb{E}[r^{-1}(1 - v_1^2 + v_1 v_2)\phi'(\sqrt{B}v_1) : rv \sim \mathcal{N}(0, \mu^* G)] \right)^2$$

$$= B\mu^{*-1} \left( \mathbb{E}[r^{-1}(1 - v_1^2 + v_1 v_2)\phi'(\sqrt{B}v_1) : rv \sim \mathcal{N}(0, G)] \right)^2$$

$$= B\mu^{*-1} \left( \mathbb{E}[r^{-1}(1 - (\mathbb{e}w)_1^2 + (\mathbb{e}w)_1(\mathbb{e}w)_2)\phi'(\sqrt{B}(\mathbb{e}w)_1) : rw \sim \mathcal{N}(0, I_{B-1})] \right)^2$$

$$= B\mu^{*-1} \left( \frac{\Gamma((B-2)/2)}{\Gamma((B-3)/2)} 2^{-1/2} \pi^{-1} \right.$$

$$\left. \int_0^\pi \mathrm{d}\theta_1 \int_0^\pi \mathrm{d}\theta_2 \ (1 - \zeta(\theta_1)^2 + \zeta(\theta_1)\omega(\theta_1, \theta_2))\phi'(\sqrt{B}\zeta(\theta_1)) \sin^{B-3}\theta_1 \sin^{B-4}\theta_2 \right)^2$$

where we applied Lemma E.36 with $\zeta(\theta) = -\sqrt{\frac{B-1}{B}}\cos\theta$ and $\omega(\theta_1, \theta_2) = \frac{1}{\sqrt{B(B-1)}}\cos\theta_1 - \sqrt{\frac{B-2}{B-1}}\sin\theta_1\cos\theta_2$.

We can further simplify

$$\int_0^\pi \mathrm{d}\theta_1 \int_0^\pi \mathrm{d}\theta_2 \ (1 - \zeta(\theta_1)^2 + \zeta(\theta_1)\omega(\theta_1, \theta_2))\phi'(\sqrt{B}\zeta(\theta_1)) \sin^{B-3}\theta_1 \sin^{B-4}\theta_2$$

$$= \int_0^\pi \mathrm{d}\theta_1 \ \sin^{B-3}\theta_1 \phi'(\sqrt{B}\zeta(\theta_1)) \left( \left(1 - \zeta(\theta_1)^2 + \zeta(\theta_1)\frac{1}{\sqrt{B(B-1)}}\cos\theta_1\right) \int_0^\pi \mathrm{d}\theta_2 \ \sin^{B-4}\theta_2 \right.$$

$$\left. - \sqrt{\frac{B-2}{B-1}}\sin\theta_1\zeta(\theta_1) \int_0^\pi \mathrm{d}\theta_2 \ \sin^{B-4}\theta_2\cos\theta_2 \right)$$

$$= \int_0^\pi \mathrm{d}\theta_1 \ \sin^{B-3}\theta_1 \phi'(\sqrt{B}\zeta(\theta_1)) \left( \left(1 - \zeta(\theta_1)^2 + \zeta(\theta_1)\frac{1}{\sqrt{B(B-1)}}\cos\theta_1\right) \mathrm{Beta}\left(\frac{B-3}{2}, \frac{1}{2}\right) \right)$$

$$\text{(by Lemma E.29)}$$

$$= \int_0^\pi \mathrm{d}\theta_1 \ \sin^{B-3}\theta_1 \phi'(\sqrt{B}\zeta(\theta_1)) \left( \sin^2\theta_1 \mathrm{Beta}\left(\frac{B-3}{2}, \frac{1}{2}\right) \right)$$

$$= \mathrm{Beta}\left(\frac{B-3}{2}, \frac{1}{2}\right) \int_0^\pi \mathrm{d}\theta_1 \ \sin^{B-1}\theta_1 \phi'(-\sqrt{B-1}\cos\theta_1)$$

so the cross-batch backward off-diagonal eigenvalue is

$$B\mu^{*-1} \left( \frac{1}{\sqrt{2\pi}} \int_0^\pi \mathrm{d}\theta_1 \ \sin^{B-1}\theta_1 \phi'(-\sqrt{B-1}\cos\theta_1) \right)^2$$

$$= \frac{1}{2\pi} B\mu^{*-1} \left( \int_0^\pi \mathrm{d}\theta_1 \ \sin^{B-1}\theta_1 \phi'(-\sqrt{B-1}\cos\theta_1) \right)^2$$

$\square$

For positive homogeneous functions we can evaluate the eigenvalues explicitly.

**Theorem I.3.** *If $\phi$ is positive-homogeneous of degree $\alpha$ with $\phi(c) = a\rho_\alpha(c) - b\rho_\alpha(-c)$, then $\mathcal{T} : \mathcal{M}_B^G \to \mathcal{M}_B^G$ is just multiplication by*

$$\lambda_{\mathbb{M}}^{\downarrow} = \frac{1}{8\pi} B(B-1)^{\alpha-1}\mu^{*-1}\alpha^2(a+b)^2 \mathrm{Beta}\left(\frac{\alpha}{2}, \frac{B}{2}\right)^2.$$

*Proof.* As in the proof of Corollary H.4,

$$\int_0^{\pi/2} d\theta \, (\cos\theta)^{\alpha-1} \sin^{B-1}\theta = \int_{\pi/2}^{\pi} d\theta \, (-\cos\theta)^{\alpha-1} \sin^{B-1}\theta = \frac{1}{2}\mathrm{Beta}\left(\frac{\alpha}{2}, \frac{B}{2}\right)$$

So

$$\frac{1}{2\pi} B\mu^{*-1}\left(\int_0^\pi d\theta_1 \, \sin^{B-1}\theta_1 \phi'(-\sqrt{B-1}\cos\theta_1)\right)^2$$

$$= \frac{1}{2\pi} B\mu^{*-1}(B-1)^{\alpha-1}\alpha^2\left(a\int_{\pi/2}^\pi d\theta \, (-\cos\theta)^{\alpha-1}\sin^{B-1}\theta + b\int_0^{\pi/2} d\theta \, (\cos\theta)^{\alpha-1}\sin^{B-1}\theta\right)^2$$

$$= \frac{1}{2\pi} B\mu^{*-1}(B-1)^{\alpha-1}\alpha^2(a+b)^2 2^{-2}\mathrm{Beta}\left(\frac{\alpha}{2}, \frac{B}{2}\right)^2$$

$$= \frac{1}{8\pi} B(B-1)^{\alpha-1}\mu^{*-1}\alpha^2(a+b)^2\mathrm{Beta}\left(\frac{\alpha}{2}, \frac{B}{2}\right)^2$$

$\square$

It is also straightforward to obtain an expression of the eigenvalue in terms of Gegenbauer coefficients.

**Theorem I.4.** *If $\phi(\sqrt{B-1}x)$ has Gegenbauer expansion $\sum_{l=0}^\infty a_l \frac{1}{c_{B-1,l}} C_l^{(\frac{B-3}{2})}(x)$, then $\mathcal{T}$ : $\mathcal{M}_B^G \to \mathcal{M}_B^G$ is just multiplication by*

$$\lambda_{\widetilde{\mathbb{M}}}^\downarrow = \frac{1}{2\pi} B(B-1)\mu^{*-1} a_1^2 \mathrm{Beta}\left(\frac{B}{2}, \frac{1}{2}\right)^2$$

$$= \frac{\frac{1}{2\pi} B(B-1) a_1^2 \mathrm{Beta}\left(\frac{B}{2}, \frac{1}{2}\right)^2}{\sum_{l=0}^\infty a_l^2 \frac{1}{c_{B-1,l}}\left(C_l^{(\frac{B-3}{2})}(1) - C_l^{(\frac{B-3}{2})}\left(\frac{-1}{B-1}\right)\right)}$$

$$= \lambda_{\widetilde{\mathbb{M}}}^\uparrow.$$

*Proof.* It suffices to express the integral in Thm I.2 in Gegenbauer coefficients. Changing coordinates $x = -\cos\theta$, so that $dx = \sin\theta \, d\theta$, we get

$$\int_0^\pi d\theta \, \sin^{B-1}\theta\phi'(-\sqrt{B-1}\cos\theta) = \int_{-1}^1 dx(1-x^2)^{\frac{B-2}{2}}\phi'(\sqrt{B-1}x).$$

By Eq. (35),

$$\phi'(\sqrt{B-1}x) = (B-1)^{-1/2}\frac{d}{dx}\phi(\sqrt{B-1}x)$$

$$= (B-1)^{-1/2}\sum_{l=0}^\infty a_l \frac{1}{c_{B-1,l}} C_l^{(\frac{B-3}{2})'}(x)$$

$$= (B-1)^{-1/2}\sum_{l=1}^\infty a_l \frac{B-3}{c_{B-1,l}} C_{l-1}^{(\frac{B-1}{2})}(x).$$

Then by the orthogonality relations among $\{C_{l-1}^{(\frac{B-1}{2})}\}_{l\geq 1}$,

$$\int_0^\pi d\theta \, \sin^{B-1}\theta\phi'(-\sqrt{B-1}\cos\theta) = \int_{-1}^1 dx(1-x^2)^{\frac{B-2}{2}}\phi'(\sqrt{B-1}x)$$

$$= (B-1)^{-1/2} a_1 \frac{B-3}{c_{B-1,1}}\int_{-1}^1\left[C_0^{(\frac{B-1}{2})}(x)\right]^2(1-x^2)^{\frac{B-2}{2}} dx$$

$$= (B-1)^{-1/2} a_1 \frac{B-3}{c_{B-1,1}}\frac{\sqrt{\pi}\Gamma\left(\frac{B}{2}\right)}{\Gamma\left(\frac{B+1}{2}\right)}$$

$$= a_1\sqrt{B-1}\,\mathrm{Beta}\left(\frac{B}{2}, \frac{1}{2}\right).$$

□

To summarize,

**Theorem I.5.** *The backward cross batch dynamics Eq. (59) over $\mathcal{H}_{kB}^G$ is separable over each individual $B \times B$ block. The diagonal blocks evolve as in Eq. (51), and this linear dynamics has eigenvalues $\lambda_G^\downarrow, \lambda_\mathbb{L}^\downarrow, \lambda_\mathbb{M}^\downarrow$ as described by Appendix G. The off-diagonal blocks dynamics is multiplication by $\lambda_{\widetilde{\mathbb{M}}}^\downarrow = \lambda_{\widetilde{\mathbb{M}}}^\uparrow$.*

Note that

$$\lambda_{\widetilde{\mathbb{M}}}^\downarrow = \lambda_{\widetilde{\mathbb{M}}}^\uparrow = \frac{\frac{1}{2\pi}B(B-1)a_1^2 \operatorname{Beta}\left(\frac{B}{2}, \frac{1}{2}\right)^2}{\sum_{l=0}^{\infty} a_l^2 \frac{1}{c_{B-1,l}}\left(C_l^{\left(\frac{B-3}{2}\right)}(1) - C_l^{\left(\frac{B-3}{2}\right)}\left(\frac{-1}{B-1}\right)\right)}$$

$$\leq \frac{\frac{1}{2\pi}B(B-1)\operatorname{Beta}\left(\frac{B}{2}, \frac{1}{2}\right)^2}{\frac{1}{c_{B-1,1}}\left(C_1^{\left(\frac{B-3}{2}\right)}(1) - C_1^{\left(\frac{B-3}{2}\right)}\left(\frac{-1}{B-1}\right)\right)}$$

$$= \frac{B-1}{2}\operatorname{P}\left(\frac{B}{2}, \frac{1}{2}\right)^{-2}$$

which increases to 1 from below as $B \to \infty$. Thus,

**Corollary I.6.** *For any $\phi$ inducing Eq. (54) to converge to a CBSB1 fixed point, $\lambda_{\widetilde{\mathbb{M}}}^\downarrow = \lambda_{\widetilde{\mathbb{M}}}^\uparrow < 1$. For any fixed batch size $B$, $\lambda_{\widetilde{\mathbb{M}}}^\downarrow$ is maximized by $\phi = \operatorname{id}$. Furthermore, for $\phi = \operatorname{id}$, $\lambda_{\widetilde{\mathbb{M}}}^\downarrow = \frac{B-1}{2}\operatorname{P}\left(\frac{B}{2}, \frac{1}{2}\right)^{-2}$ and $\lim_{B\to\infty} \lambda_{\widetilde{\mathbb{M}}}^\downarrow = 1$, increasing to the limit from below.*

This corollary indicates that stacking batchnorm in a deep neural network will always cause chaotic behavior, in the sense that cross batch correlation, both forward and backward, decreases to 0 exponentially with depth. The $\phi$ that can maximally ameliorate the exponential loss of information is linear.

## J   UNCOMMON REGIMES

In the above exposition of the mean field theory of batchnorm, we have assumed $B \geq 4$ and that $\phi$ induces BSB1 fixed points in Eq. (8).

**Small Batch Size**   What happens when $B < 4$? It is clear that for $B = 1$, batchnorm is not well-defined. For $B = 2$, $\mathcal{B}_\phi(h) = (\pm 1, \mp 1)$ depending on the signs of $h$. Thus, the gradient of a batchnorm network with $B = 2$ is 0. Therefore, we see an abrupt phase transition from the immediate gradient vanishing of $B = 2$ to the gradient explosion of $B \geq 4$. We empirically see that $B = 3$ suffers similar gradient explosion as $B = 4$ and conjecture that a generalization of Thm 3.9 holds for $B = 3$.

**Batch Symmetry Breaking**   What about other nonlinearities? Empirically, we observe that if the fixed point is not BSB1, then it is BSB2, like in Fig. 1, where a submatrix of $\Sigma$ (the dominant block) is much larger in magnitude than everything else (see Defn K.1). If the initial $\Sigma^0$ is permutation-invariant, then convergence to this fixed point requires spontaneous symmetry breaking, as the dominant block can appear in any part of $\Sigma$ along the diagonal. This symmetry breaking is lost when we take the mean field limit, but in real networks, the symmetry is broken by the network weight randomness. Because small fluctuation in the input can also direct the dynamics toward one BSB1 fixed point against others, causing large change in the output, the gradient is intuitively large as well. Additionally, at the BSB2 fixed point, we expect the dominant block goes through a similar dynamics as if it were a BSB1 fixed point for a smaller $B$, thus suffering from similar gradient explosion. Appendix K discusses several results on our current understanding of BSB2 fixed points. A specific form of BSB2 fixed point with a $1 \times 1$ dominant block can be analyzed much further, and this is done in Appendix K.1.

**Finite width effect** For certain nonlinearities, the favored fixed point can be different between the large width limit and small width. For example, for $\phi = C_5^{\left(\frac{B-3}{2}\right)}(x/\sqrt{B-1}) - C_7^{\left(\frac{B-3}{2}\right)}(x/\sqrt{B-1})$ where $B = 10$, one can observe that for width 100, Eq. (43) favors BSB2 fixed points, but for width 1000 and more, Eq. (43) favors BSB1 fixed points.

## K    TOWARD UNDERSTANDING BSB2 FIXED POINTS

**Definition K.1.** For $B_1, B_2 \geq 2$, a BSB2 matrix is one of the form

$$\text{BSB2}_{B_1,B_2}(d_1, f_1, d_2, f_2, c) := \begin{pmatrix} \text{BSB1}_{B_1}(d_1, f_1) & c \\ c & \text{BSB1}_{B_2}(d_2, f_2) \end{pmatrix}$$

up to simultaneous permutation of rows and columns. A $\widehat{\text{BSB2}}$ matrix is a specific kind of BSB2 matrix $\widehat{\text{BSB2}}_B^{B'}(d, f, c, b) = \text{BSB2}_{B',B-B'}(d, f, c, b, b)$, where $B' \in [2, B-1]$ (where for $B' = B-1$ the lower right hand block is a scalar $b$). We call the upper left hand block of $\widehat{\text{BSB2}}_B^{B'}(d, f, c, b)$ its *main block*.

When $\phi$ grows quickly, Eq. (43) can converge to BSB2 fixed points of the form $\widehat{\text{BSB2}}_B^{B'}(d, f, c, b)$, up to simultaneous permutation of rows and columns. In this section we present several results on BSB2 fixed points that sheds light on their structure. We leave the study of the forward Eq. (43) and backward dynamics Eq. (51) to future work.

First, we give the eigendecomposition of BSB2 matrices.

**Theorem K.2.** *Let* $\Sigma = \text{BSB2}_{B_1,B_2}(d_1, f_1, d_2, f_2, c)$, *where* $B_1, B_2 \geq 2$. *Then* $G^{\otimes 2}\{\Sigma\} = G\Sigma G = \Sigma^G$ *has the following eigenspaces and eigenvalues.*

1. $(B_1 - 1)$-*dimensional eigenspace* $\mathbb{Z}_1 := \{(x_1, \ldots, x_{B_1}, 0, \ldots, 0) : \sum_{i=1}^{B_1} x_i = 0\}$, *with eigenvalue* $d_1 - f_1$.

2. $(B_2 - 1)$-*dimensional eigenspace* $\mathbb{Z}_2 := \{(0, \ldots, 0, y_1, \ldots, y_{B_2}) : \sum_{j=1}^{B_2} y_j = 0\}$, *with eigenvalue* $d_2 - f_2$.

3. *1-dimensional* $\mathbb{R}\boldsymbol{q}$ *with eigenvalue* $\frac{(d_2-f_2)B_1 + (d_1-f_1)B_2 + (f_1+f_2-2c)B_1B_2}{B_1+B_2}$, *where* $\boldsymbol{q} = (\overbrace{B_1^{-1}, \ldots, B_1^{-1}}^{B_1}, \overbrace{-B_2^{-1}, \ldots, -B_2^{-1}}^{B_2})$.

4. *1-dimensional* $\mathbb{R}\boldsymbol{1}$ *with eigenvalue 0.*

*Proof.* We can verify all eigenspaces and their eigenvalues in a straightforward manner. These then must be all of them by a dimensionality argument. $\square$

Specializing to $\widehat{\text{BSB2}}$ matrices, we get

**Corollary K.3.** *Let* $\Sigma = \widehat{\text{BSB2}}_B^{B'}(d, f, c, b)$ *with* $B' \geq 2$. *Then* $G^{\otimes 2}\{\Sigma\} = G\Sigma G = \Sigma^G$ *has the following eigenspaces and eigenvalues*

- $(B' - 1)$-*dimensional eigenspace* $\mathbb{Z}_1 := \{(x_1, \ldots, x_{B'}, 0, \ldots, 0) : \sum_{i=1}^{B'} x_i = 0\}$, *with eigenvalue* $d - f$.

- *1-dimensional* $\mathbb{R}\boldsymbol{q}$ *with eigenvalue* $\frac{(d-f)(B-B') + (f-c)B(B-B')}{B}$, *where* $\boldsymbol{q} = (\overbrace{B'^{-1}, \ldots, B'^{-1}}^{B'}, \overbrace{-B_2^{-1}, \ldots, -B_2^{-1}}^{B_2})$ *where* $B_2 = B - B'$.

- $(B - B')$-*dimensional eigenspace* $\{(\overbrace{-\mu, \ldots, -\mu}^{B'}, y_1, \ldots, y_{B-B'} : \mu = \frac{1}{B-B'}\sum_{j=1}^{B-B'} y_j\}$ *with eigenvalue 0.*

Note also that when $B' = B$, then $\mathbb{Z}_1$ and $\mathbb{R}q$ become the usual eigenspaces of $\mathrm{BSB1}(d, f)$.

For fixed $B' < B$, specializing the forward dynamics Eq. (43) to $\widehat{\mathrm{BSB2}}_B^{B'}$ fixed points yields a 2-dimensional dynamics on the eigenvalues for the eigenspaces $\mathbb{Z}_1$ and $\mathbb{R}q$. This dynamics in general is not degenerate, so that the fixed point seems difficult to obtain analytically. Moreoever, the Gegenbauer method, essential for proving that gradient explosion is unavoidable when Eq. (43) has a BSB1 fixed point, does not immediately generalize to the BSB2 case, since $\mathbb{Z}_1$ and $\mathbb{R}q$ in general do not have the same eigenvalues so that we cannot reduce the integrals to that on a sphere. For this reason, at present we do not have any rigorous result on the BSB2 case.

However, we expect that the main block of the $\widehat{\mathrm{BSB2}}$ fixed point should undergo a similar dynamics to that of a BSB1 fixed point, leading to similar gradient explosion.

## K.1  $\widehat{\mathrm{BSB2}}_B^1$ FIXED POINTS

When $\phi$ grows extremely rapidly, for example $\phi(x) = \mathrm{relu}(x)^{30}$, and the width of the network is not too large, then we sometimes observe $\widehat{\mathrm{BSB2}}_B^1$ fixed points

$$\widehat{\mathrm{BSB2}}_B^1(d, c, b) = \begin{pmatrix} d & c\mathbf{1}^T \\ c\mathbf{1} & b\mathbf{1}\mathbf{1}^T \end{pmatrix}$$

where $\mathbf{1} \in \mathbb{R}^{B-1}$ is the all 1s column vector. In these situations we can in fact see pathological gradient vanishing, in a way reminiscent of the gradient vanishing for $B = 2$ batchnorm, as we shall see shortly.

We can extend Corollary K.3 to the $B' = 1$ case.

**Theorem K.4.** $\widehat{\mathrm{BSB2}}_B^1(d, c, b)^G$ is rank 1 and equal to $\lambda\hat{q}^{\otimes 2}$, where $\lambda = \frac{B-1}{B}(d - 2c + b)$ and $\hat{q} = (1, \frac{-1}{B-1}, \ldots, \frac{-1}{B-1})\sqrt{\frac{B-1}{B}}$.

*Proof.* Straightforward verification. $\qquad\square$

Because $\widehat{\mathrm{BSB2}}_B^1$ under $G$-projection is rank 1, such a fixed point of Eq. (43) can be determined (unlike the general $\widehat{\mathrm{BSB2}}_B^{B'}$ case).

**Theorem K.5.** *There is a unique $\widehat{\mathrm{BSB2}}_B^1$ fixed point of Eq. (43), $\widehat{\mathrm{BSB2}}_B^1(d^*, c^*, b^*)$ where*

$$d^* = \frac{1}{2}\left(\phi(\sqrt{B-1})^2 + \phi(-\sqrt{B-1})^2\right)$$

$$c^* = \frac{1}{2}\left(\phi(\sqrt{B-1})\phi(\frac{-1}{\sqrt{B-1}}) + \phi(-\sqrt{B-1})\phi(\frac{1}{\sqrt{B-1}})\right)$$

$$b^* = \frac{1}{2}\left(\phi(\frac{-1}{\sqrt{B-1}})^2 + \phi(\frac{1}{\sqrt{B-1}})^2\right).$$

*Additionally, $\widehat{\mathrm{BSB2}}_B^1(d^*, c^*, b^*)^G = \lambda^*\hat{q}^{\otimes 2}$, where*

$$\lambda^* = \frac{B-1}{2B}\left[\left(\phi(\sqrt{B-1}) - \phi(\frac{-1}{\sqrt{B-1}})\right)^2 + \left(\phi(-\sqrt{B-1}) - \phi(\frac{1}{\sqrt{B-1}})\right)^2\right]$$

*Proof.* If $\Sigma = \widehat{\mathrm{BSB2}}_B^1(d, c, b)$, then

$$\mathrm{V}_{\mathcal{B}_\phi}(\Sigma) = \mathop{\mathbb{E}}_{y \sim \mathcal{N}(0, G\Sigma G)} \phi(\mathsf{n}(y))^{\otimes 2}$$

$$= \frac{1}{2}\left[\phi(\sqrt{B}\hat{q})^{\otimes 2} + \phi(-\sqrt{B}\hat{q})^{\otimes 2}\right]$$

since $G\Sigma G = \lambda\hat{q}^{\otimes 2}$ for some $\lambda$ by Thm K.4. The rest then follows from straightforward computation and another application of Thm K.4. $\qquad\square$

The rank-1 nature of the $\widehat{\mathrm{BSB2}}_B^1$ fixed point under $G$-projection is reminiscent of the case of $B = 2$ batchnorm. As we will see, gradients of a certain form will vanish similarly.

Let $\delta$ be a gradient vector. Then by Proposition G.4, for $\Sigma^*$ being the $\widehat{\mathrm{BSB2}}_B^1$ fixed point, one step of backpropagation yields

$$\left.\frac{\mathrm{d}\mathcal{B}_\phi}{\mathrm{d}\Sigma}\right|_{\Sigma=\Sigma^*}^{\dagger} \delta = \sqrt{B}\, r^{-1} G(I - vv^T) D\delta$$

where $D = \mathrm{Diag}(\phi'(\mathsf{n}(y)))$, $r = \|y\|$, $v = y/\|y\|$, and $y \sim \mathcal{N}(0, G\Sigma^*G)$. Because $G\Sigma^*G = \lambda^* \hat{q}^{\otimes 2}$ by Thm K.5, $\mathsf{n}(y) = \sqrt{B} v = \pm\sqrt{B}\hat{q}$ with equal probability and $r \sim |\mathcal{N}(0, \lambda^*)|$ (the absolute value of a Gaussian). Furthermore, $G(I - \hat{q}^{\otimes 2}) = G - \hat{q}^{\otimes 2}$ because $G\hat{q} = \hat{q}$. Thus

$$\left.\frac{\mathrm{d}\mathcal{B}_\phi}{\mathrm{d}\Sigma}\right|_{\Sigma=\Sigma^*}^{\dagger} \delta = \sqrt{B}|r|^{-1}(G - \hat{q}^{\otimes 2})(\phi'(\sqrt{B}\,\mathrm{sgn}(r)\hat{q}) \odot \delta), r \sim \mathcal{N}(0, \lambda^*)$$

Now notice the following

**Lemma K.6.** $G_B - \hat{q}^{\otimes 2} = \begin{pmatrix} 0 & 0 \\ 0 & G_{B-1} \end{pmatrix} = \begin{pmatrix} 0 & 0 & 0 & \cdots & 0 \\ 0 & 1 - \frac{1}{B-1} & \frac{-1}{B-1} & \cdots & \frac{-1}{B-1} \\ 0 & \frac{-1}{B-1} & 1 - \frac{1}{B-1} & \cdots & \frac{-1}{B-1} \\ 0 & \frac{-1}{B-1} & \frac{-1}{B-1} & \cdots & 1 - \frac{1}{B-1} \end{pmatrix}.$

*Proof.* Straightforward computation. $\qquad\square$

Thus if $\delta = (\delta_1, \delta_2)^T \in \mathbb{R}^1 \times \mathbb{R}^{B-1}$, then

$$\left.\frac{\mathrm{d}\mathcal{B}_\phi}{\mathrm{d}\Sigma}\right|_{\Sigma=\Sigma^*}^{\dagger} \delta = \sqrt{B}\phi'\left(\frac{-\,\mathrm{sgn}(r)}{\sqrt{B-1}}\right)|r|^{-1}(0, G_{B-1}\delta_2)^T, r \sim \mathcal{N}(0, \lambda^*)$$

If $\delta_2 = 0$, then this is deterministically 0. Otherwise, this random variable does not have finite mean because of the $|r|^{-1}$ term, and this is solely due to $\Sigma^*$ being rank 1. Thus as long as the gradient $\delta$ is not constant in $\delta_2$, then we expect severe gradient explosion; otherwise we expect severe gradient vanishing.

*Remark* K.7. We would like to note that, in pytorch, with either Float Tensors or Double Tensors, the gradient empirically vanishes rapidly with generic gradient $\delta$, when $\phi$ is rapidly increasing and leads to a $\widehat{\mathrm{BSB2}}_B^1$ fixed point, e.g. $\phi(x) = \mathrm{relu}(x)^{30}$. This is due to numerical issues rather than mathematical issues, where the computation of $(G - \hat{q}^{\otimes 2})\delta$ in backprop does not kill $\delta_1$ completely.

## L  WEIGHT GRADIENTS

Recall weight gradient and the backpropagation equation Eq. (16) of the main text.

$$\frac{\partial\mathcal{L}}{\partial W_{\alpha\beta}^l} = \sum_i \delta_{\alpha i}^l x_{\beta i}^{l-1}, \qquad \delta_{\alpha i}^l = \sum_{\beta j} \frac{\partial x_{\alpha j}^l}{\partial h_{\alpha i}^l} W_{\beta\alpha}^{l+1} \delta_{\beta j}^{l+1}$$

where we have used $x_{\beta i}^l$ to denote $\mathcal{B}_\phi(h_\beta^l)_i$, $\delta_i^l = \nabla_{h_i^l}\mathcal{L}$, and subscript $i$ denotes slice over sample $i$ in the batch, and subscript $\alpha$ denotes slice over neuron $\alpha$ in the layer. Then

$$\mathbb{E}\left[\left(\frac{\partial\mathcal{L}}{\partial W_{\alpha\beta}^l}\right)^2\right] = \mathbb{E}\left[\left(\sum_i \delta_{\alpha i}^l x_{\beta i}^{l-1}\right)^2\right]$$

$$= \mathbb{E}\left[\sum_{i,j} \delta_{\alpha i}^l x_{\beta i}^{l-1} \delta_{\alpha j}^l x_{\beta j}^{l-1}\right].$$

By gradient independence assumption (Appendix B),

$$\mathbb{E}\,\delta^l_{\alpha i}x^{l-1}_{\beta i}\delta^l_{\alpha j}x^{l-1}_{\beta j} = \mathbb{E}[\delta^l_{\alpha i}\delta^l_{\alpha j}]\,\mathbb{E}[x^{l-1}_{\beta i}x^{l-1}_{\beta j}]$$
$$= \Pi^l_{ij}\Sigma^{l-1}_{ij}.$$

Thus

$$\mathbb{E}\left[\left(\frac{\partial\mathcal{L}}{\partial W^l_{\alpha\beta}}\right)^2\right] = \sum_{i,j}\Pi^l_{ij}\Sigma^{l-1}_{ij} = \langle\Pi^l,\Sigma^{l-1}\rangle.$$

If $l$ is large and $\phi$ is well-behaved, then we expect $\Sigma^{l-1}\approx\Sigma^*$, the unique BSB1 fixed point of Eq. (43). Assuming equality, we then have

$$\mathbb{E}\left[\left(\frac{\partial\mathcal{L}}{\partial W^l_{\alpha\beta}}\right)^2\right] = \langle\Pi^l,\Sigma^*\rangle = \langle G^{\otimes 2}\{\Pi^l\},\Sigma^*\rangle$$
$$= \langle\Pi^l,G^{\otimes 2}\{\Sigma^*\}\rangle = \langle\Pi^l,\mu^*G\rangle$$
$$= \mu^*\operatorname{tr}\Pi^l.$$

Thus in general, we expect the magnitude of weight gradients to follow the magnitude of hidden unit gradient $\delta^l_i$.

