# OpenReview forum: "A Mean Field Theory of Batch Normalization"
_ICLR.cc/2019/Conference_

### Official Review · AnonReviewer3 · 2018-10-31
**Interesting work, strong results**

**Rating:** 7
**Confidence:** 3

**Review:**

This paper provides a new dynamic perspective on deep neural network. Based on Gaussian weights and biases, the paper investigates the evolution of the covariance matrix along with the layers. Eventually the matrices achieve a stationary point, i.e., fixed point of the dynamic system. Local performance around the fixed point is explored. Extensions are provided to include the batch normalization. I believe this paper may stimulate some interesting ideas for other researchers.

Two technical questions:

1. When the layers tends to infinity, the covariance matrix reaches stationary (fixed) point. How to understand this phenomenon? Does this mean that the distribution of the layer outputs will not change too much if the layer is deep enough? This somewhat conflicts the commonsense of "the deeper the better?"

2. Typos: the weight matrix in the end of page 2 should be N_l times N_{l-1}. Also, the x_i's in the first line of page 3 should be bold.

---

> ### Author Response · Authors · 2018-11-12
> **Reply**
>
> Thank you for the review and careful reading of our paper! We’re glad that you found it of interest. On revision we will fix the typos that you identified.
>
> Regarding the first point, your intuition is exactly correct and a slightly simpler discussion of this phenomenon can be found in [1]. When the network is deep enough that the covariance matrix has reached its fixed point, the distribution of the outputs of the network will be independent of the inputs. At this point the network becomes untrainable. To reconcile this with the commonsense intuition that “deeper is better”, our answer is twofold.
>
> 1) As in [1] and [2] it is often possible to find configurations or architectural modifications where the covariance matrix doesn’t approach its fixed point over depths often considered in machine learning. When this is the case one can safely increase the depth without sacrificing accuracy.
>
> 2) It seems that the role of depth in performance is more subtle than standard intuition would dictate. For example, in [3] note that although the authors were able to train a 10k hidden layer network, they did not observe any improvement in accuracy.
>
> In the next version of the manuscript (both in response to your review and that of referee 1) we will add a more intuitive discussion of these results which we agree are somewhat technical.
>
> [1] S. S. Schoenholz, J. Gilmer, S. Ganguli, J. Sohl-Dickstein. Deep Information Propagation (https://arxiv.org/abs/1611.01232)
> [2] G. Yang and S. S. Schoenholz. Mean Field Residual Networks (https://arxiv.org/abs/1712.08969)
> [3] L. Xiao, Y. Bahri, J. Sohl-Dickstein, S. S. Schoenholz, J. Pennington. Dynamical Isometry and a Mean Field Theory of CNNs: How to Train 10,000-Layer Vanilla Convolutional Neural Networks (https://arxiv.org/abs/1806.05393)

---

### Official Review · AnonReviewer2 · 2018-11-02
**The detailed analysis of the training of DNN with the batch normalization is quite interesting.**

**Rating:** 6
**Confidence:** 1

**Review:**


This paper investigates the effect of the batch normalization in DNN learning.
The mean field theory in statistical mechanics was employed to analyze the
progress of variance matrices between layers.
As the results, the batch normalization itself is found to be the cause of gradient explosion.
Moreover, the authors pointed out that near-linear activation function can improve such gradient explosion.
Some numerical studies were reported to confirm theoretical findings.

The detailed analysis of the training of DNN with the batch normalization is quite interesting.
There are some minor comments below.

- in page 3, 2line above eq(2): what is delta in the variance of the multivariate normal distribution?
- the notation q appeared in the middle part of page 3 before the definition of q is shown in the last paragraph of p.3.
- The randomized weight is not very practical. Though it may be the standard approach of mean field,
some comments would be helpful to the readers.

---

> ### Author Response · Authors · 2018-11-12
> **Reply**
>
> Thank you for your review and very useful comments! We’re happy you found our manuscript interesting.
>
> To address your comments:
>
> 1) Thank you for pointing out that we had not defined the delta. Here delta is the Kronecker delta defined so that \delta_{a,b} = 1 if a = b and 0 if a != b. In the context of the variance of the multivariate normal distribution, the delta function indicates that the different neurons in each layer have zero covariance. We’ll add an explicit discussion of this fact to the manuscript.
>
> 2) Thanks for pointing this out, we’ll correct it in the next revision.
>
> 3) It is true that the extent to which randomized weights describe trained networks is unclear. However, it is true that most commonly used weight initialization schemes are random. For example, He initialization [1] and Xavier initialization [2] strategies are both special cases of the setup considered here. We therefore view our theory as a theory of neural networks at initialization. (There are, however, initialization schemes that are not random and that are not described by our theory).
>
> [1] K. He, X. Zhang, S. Ren, J. Sun. Delving deep into rectifiers: Surpassing human-level performance on imagenet classification. (http://www.cv-foundation.org/openaccess/content_iccv_2015/html/He_Delving_Deep_into_ICCV_2015_paper.html)
> [2] X. Glorot, Y. Bengio, Y. W. Teh, M. Titterington. Understanding the difficulty of training deep feedforward neural networks. (http://proceedings.mlr.press/v9/glorot10a.html)

---

### Official Review · AnonReviewer1 · 2018-11-08
**Interesting and counter-intuitive results about batch-normalization**

**Rating:** 7
**Confidence:** 3

**Review:**

This paper develops a mean field theory for batch normalization (BN) in fully-connected networks with randomly initialized weights. There are a number of interesting predictions made in this paper on the basis of this analysis. The main technical results of the paper are Theorems 5-8 which compute the statistics of the covariance of the activations and the gradients.

Comments:

1. The observation that gradients explode in spite of BN is quite counter-intuitive. Can you give an intuitive explanation of why this occurs?

2. In a similar vein, there a number of highly technical results in the paper and it would be great if the authors provide an intuitive explanation of their theorems.

3. Can the statistics of activations be controlled using activation functions or operations which break the symmetry? For instance, are BSB1 fixed points good for training neural networks?

4. Mean field analysis, although it lends an insight into the statistics of the activations, needs to connected with empirical observations. For instance, when the authors observe that the structure of the fixed point is such that activations are of identical norm equally spread apart in terms of angle, this is quite far from practice. It would be good to mention this in the introduction or the conclusions.

---

> ### Author Response · Authors · 2018-11-12
> **Reply**
>
> Thank you for your careful review and useful comments! Overall, in response to your review and that of referee 3 we will include a more intuitive discussion of our results in the next revision of our text.
>
> To reply to your other specific comments,
>
> 1) The intuition for batchnorm can be put in a more general setting. If a function f: X -> Y tends to spread out small clusters in the input space almost evenly in the output space, then one can expect that its gradients will be large typically. In our case, a batchnorm network can be understood as a function that sends a batch of inputs to a batch of outputs. In the appendix, we showed that the correlation between two different batches tend to a constant value independent of the input batches. No matter how close two input batches are, the output batches will have the same “distance” from each other -- small movements in the input space leads to large movements in the output space. Thus we can expect the gradients to be large as well. We have added a new figure to the Appendix to further support this intuition. In it, we pass through a linear batchnorm network 2 minibatches. Both minibatches contain points on the same circle and 1 point off the circle that is unique to each minibatch. While the circle in each minibatch will remain an ellipse as they are propagated through the network, the angle between the planes spanned by them increasingly becomes chaotic with depth.
>
> 3) As observed in [1] and [2], depthwise convergence to covariance fixed points is bad for training, and the best networks are either moderately deep or initialized such that the depthwise convergence rate to the fixed point is as slow as possible. We observe that deep networks whose activation statistics resemble a non-BSB1 fixed point typically feature worse gradient explosion than BSB1 networks. This seems to be because the nonlinearities that induce these fixed points increase rapidly (for example, polynomials with high degrees), so that the corresponding derivatives are also large, causing gradient explosion.
>
> (The reason that rapidly increasing nonlinearities don’t converge to BSB1 fixed points is that, after a spontaneous symmetry-breaking, begins a “winner-take-all” covariance dynamics, in which the activations of a few examples in the batch suddenly dominates those of the others in the batch, and this dominance persists across each layer.)
>
> 4) We were a bit confused by what was meant by “practice” here. We have thoroughly verified that for realistic input distributions (MNIST and CIFAR10) and common initialization strategies (weights that are randomly distributed) our theory makes accurate prediction. Moreover, we have shown that these predictions can be connected to practice in the sense that they predict whether or not the network can be trained.
>
> Having said this, if by practice you meant that the neural network is accurately described by our theory during training then we do not expect this to be true. We are happy to emphasize this in the camera ready.
>
> If this did not properly address your question, please feel free to let us to know and we will improve this response!
>
> [1] S. S. Schoenholz, J. Gilmer, S. Ganguli, J. Sohl-Dickstein. Deep Information Propagation (https://arxiv.org/abs/1611.01232)
> [2] L. Xiao, Y. Bahri, J. Sohl-Dickstein, S. S. Schoenholz, J. Pennington. Dynamical Isometry and a Mean Field Theory of CNNs: How to Train 10,000-Layer Vanilla Convolutional Neural Networks (https://arxiv.org/abs/1806.05393)

---

### Public Comment · ~Angus_Galloway1 · 2018-10-04
**Clarification re Fig. 3**

Interesting work, I have a few questions about Fig. 3. Were the numerator and denominator of the y-axes flipped accidentally? The current labeling seems to imply that the gradient wrt each of the quantities in a,b,c is monotonically decreasing with depth L at t=0, which I wouldn't expect for layers with constant width. Also, what was gamma set to in this case (e.g. fixed, or learnable with initialization X)? And any reason as to why layer 10 was chosen as normalization? This confused me initially because I only saw in-text references to "10" in the context of training steps.

---

> ### Author Response · Authors · 2018-10-10
> **Clarification**
>
> Thanks for your interest!
>
> 1) The numerator and denominator are correct, but I think the meaning might be slightly unclear. When we write |\nabla_{W^l}L|^2, we mean the norm of the gradient of the loss with respect to the weights in layer l. Thus, in fig. 3 we plot the ratio of the size of the gradient of the loss with respect to the weights in layer 10 compared with layer L for a network of depth L (note that both gradients are for a network of depth L). When gradients explode, the norm of the gradient increases during backprop so that the magnitude is larger nearer to the "input" to the network than the "output". Thus, this is essentially saying that the gradient in the 10'th layer of the network is exponentially larger for deeper networks than for shallower ones.
>
> 2) \gamma = 1, \beta = 0 at initialization for figure 3. In all experiments, \gamma and \beta were free parameters that could be learned during training.
>
> 3) The choice of 10 was arbitrary and the results don't depend strongly on the choice. The only thing that one has to be careful of is that our theoretical results are "asymptotic" in the depth. So if you look at layers that are too close to the inputs you can get transient effects that change the clean exponential scaling.
>
> Please let us know if you have any further questions or if this response was unclear.

---

### Public Comment · (anonymous) · 2018-11-07
**comments on the work**

I finally found the time to read this paper, and was glad that I did so.

The conclusion that at initialization batch norm actually harms at large depth is quite surprising, and this cannot be reached without precise analysis. This is the unique strength of the mean-field framework, in contrast with many other theoretical studies.

I have a question. The authors plot the depth scale in Fig 2. The depth scale, which is based on calculations at initialization, looks very predictive as in Fig 2, but as Fig 3 suggested, the behavior at t=10 is much different from the behavior at initialization. Yet interestingly the limit depth in Fig 2 (about L=50) coincides so well with the pictures in Fig 3, where we see there is a sort of phase transition around L=50. Are they related somehow?

A suggestion. The back-propagation calculation seem to assume that the "back-prop weight" is independent of the "forward-prop weight". If the assumption is used, I think it should be mentioned, since this assumption is highly non-trivial.

It would be interesting to carry out the calculations for residual architectures.

---

> ### Author Response · Authors · 2018-11-12
> **Reply**
>
> Thank you for your thoughtful comments about our paper, we’re very happy that you found it interesting! We absolutely agree about the need for precise analysis to disentangle the many complexities that compete in deep learning.
>
> Regarding your question, great observation! Indeed, they are manifestations of the same underlying phenomenon. Below around L=50, the amount of gradient explosion is small enough that it doesn’t significantly deteriorate performance --- this is shown in figure 2. At the same time, the gradients are small compared to the corresponding weights, so that after the first few steps, the weights themselves don’t change much --- this is shown in figure 3a. If the weights don’t change much, then the gradient dynamics remain roughly the same --- this is shown in figure 3b and 3c. Conversely, for L > 50, gradient explosion dominates the weight matrices, so much that |W| at time 1 is roughly the same as the norm of the corresponding gradient. After 1 step, the exponentially decreasing norms (with depth) of the weights attenuate the gradient explosion. This is because of the batchnorm property dBN(ax)/d(ax) = 1/a dBN(x)/dx. Thus in figure 3b and 3c we see gradient vanishing for L > 50 after 1 step of SGD.
>
> We indeed used the “back-prop weight” assumption, and we will be more explicit about its usage in the next revision.
>
> We are also extremely interested in carrying out the calculation for residual networks!

---

> > ### Public Comment · (anonymous) · 2018-11-13
> > **thanks**
> >
> > Thanks for the detailed reply!

---

### Meta-Review · Area_Chair1 · 2018-12-17
**Interesting and surprising findings with a mean-field-theory analysis of batch normalization**

**Confidence:** 4
**Recommendation:** Accept (Poster)

**Metareview:**

This paper provides a mean-field-theory analysis of batch normalization. First there is a negative result as to the necessity of gradient explosion when using batch normalization in a fully connected network. They then provide further insights as to what can be done about this, along with experiments to confirm their theoretical predictions.

The reviewers (and random commenters) found this paper very interesting. The reviewers were unanimous in their vote to accept.